# Linearly Converging Error Compensated SGD

**Eduard Gorbunov**
MIPT, Yandex and Sirius, Russia
KAUST, Saudi Arabia

**Dmitry Kovalev**
KAUST, Saudi Arabia

**Dmitry Makarenko**
MIPT, Russia

**Peter Richtárik**
KAUST, Saudi Arabia

## Abstract

In this paper, we propose a unified analysis of variants of distributed `SGD` with arbitrary compressions and delayed updates. Our framework is general enough to cover different variants of quantized `SGD`, Error-Compensated `SGD` (`EC-SGD`) and `SGD` with delayed updates (`D-SGD`). Via a single theorem, we derive the complexity results for all the methods that fit our framework. For the existing methods, this theorem gives the best-known complexity results. Moreover, using our general scheme, we develop new variants of `SGD` that combine variance reduction or arbitrary sampling with error feedback and quantization and derive the convergence rates for these methods beating the state-of-the-art results. In order to illustrate the strength of our framework, we develop 16 new methods that fit this. In particular, we propose the first method called `EC-SGD-DIANA` that is based on error-feedback for biased compression operator and quantization of gradient differences and prove the convergence guarantees showing that `EC-SGD-DIANA` converges to the exact optimum asymptotically in expectation with constant learning rate for both convex and strongly convex objectives when workers compute full gradients of their loss functions. Moreover, for the case when the loss function of the worker has the form of finite sum, we modified the method and got a new one called `EC-LSVRG-DIANA` which is the first distributed stochastic method with error feedback and variance reduction that converges to the exact optimum asymptotically in expectation with a constant learning rate.

## 1 Introduction

We consider distributed optimization problems of the form

$$\min_{x\in\mathbb{R}^d}\left\{f(x) = \tfrac{1}{n}\sum_{i=1}^{n} f_i(x)\right\}, \tag{1}$$

where $n$ is the number of workers/devices/clients/nodes. The information about function $f_i$ is stored on the $i$-th worker only. Problems of this form appear in the distributed or federated training of supervised machine learning models [42, 30]. In such applications, $x \in \mathbb{R}^d$ describes the parameters identifying a statistical model we wish to train, and $f_i$ is the (generalization or empirical) loss of model $x$ on the data accessible by worker $i$. If worker $i$ has access to data with distribution $\mathcal{D}_i$, then $f_i$ is assumed to have the structure

$$f_i(x) = \mathbf{E}_{\xi_i \sim \mathcal{D}_i}\left[f_{\xi_i}(x)\right]. \tag{2}$$

Dataset $\mathcal{D}_i$ may or may not be available to worker $i$ in its entirety. Typically, we assume that worker $i$ has only access to samples from $\mathcal{D}_i$. If the dataset is fully available, it is typically

finite, in which case we can assume that $f_i$ has the finite-sum form[1]:

$$f_i(x) = \tfrac{1}{m} \sum_{j=1}^{m} f_{ij}(x). \tag{3}$$

**Communication bottleneck.** The key bottleneck in practical distributed [14] and federated [30, 21] systems comes from the high cost of communication of messages among the clients needed to find a solution of sufficient qualities. Several approaches to addressing this communication bottleneck have been proposed in the literature.

In the very rare situation when it is possible to adjust the network architecture connecting the clients, one may consider a fully decentralized setup [6], and allow each client in each iteration to communicate to their neighbors only. One can argue that in some circumstances and in a certain sense, decentralized architecture may be preferable to centralized architectures [34]. Another natural way to address the communication bottleneck is to do more meaningful (which typically means more expensive) work on each client before each communication round. This is done in the hope that such extra work will produce more valuable messages to be communicated, which hopefully results in the need for fewer communication rounds. A popular technique of this type which is particularly relevant to Federated Learning is based in applying multiple *local updates* instead of a single update only. This is the main idea behind `Local-SGD` [43]; see also [4, 15, 22, 24, 29, 46, 50]. However, in this paper, we contribute to the line work which aims to resolve the communication bottleneck issue via *communication compression*. That is, the information that is normally exchanged—be it iterates, gradients or some more sophisticated vectors/tensors—is compressed in a lossy manner before communication. By applying compression, fewer bits are transmitted in each communication round, and one hopes that the increase in the number of communication rounds necessary to solve the problem, if any, is compensated by the savings, leading to a more efficient method overall.

**Error-feedback framework.** In order to address these issues, in this paper we study a broad class of distributed stochastic first order methods for solving problem (1) described by the iterative framework

$$\begin{aligned} x^{k+1} &= x^k - \tfrac{1}{n} \sum_{i=1}^{n} v_i^k, \tag{4} \\ e_i^{k+1} &= e_i^k + \gamma g_i^k - v_i^k, \qquad i = 1, 2, \ldots, n. \tag{5} \end{aligned}$$

In this scheme, $x^k$ represents the key iterate, $v_i^k$ is the contribution of worker $i$ towards the update in iteration $k$, $g_i^k$ is an unbiased estimator of $\nabla f_i(x^k)$ computed by worker $i$, $\gamma > 0$ is a fixed stepsize and $e_i^k$ is the error accumulated at node $i$ prior to iteration $k$ (we set to $e_i^0 = 0$ for all $i$). In order to better understand the role of the vectors $v_i^k$ and $e_i^k$, first consider the simple special case with $v_i^k \equiv \gamma g_i^k$. In this case, $e_i^k = 0$ for all $i$ and $k$, and method (4)–(5) reduces to distributed `SGD`:

$$x^{k+1} = x^k - \tfrac{\gamma}{n} \sum_{i=1}^{n} g_i^k. \tag{6}$$

However, by allowing to chose the vectors $v_i^k$ in a different manner, we obtain a more general update rule than what the `SGD` update (6) can offer. Stich and Karimireddy [46], who studied (4)–(5) in the $n = 1$ regime, show that this flexibility allows to capture several types of methods, including those employing i) compressed communication, ii) delayed gradients, and iii) minibatch gradient updates. While our general results apply to all these special cases and more, in order to not dilute the focus of the paper, in the main body of this paper we concentrate on the first use case—compressed communication—which we now describe.

**Error-compensated compressed gradient methods.** Note that in distributed `SGD` (6), each worker needs to know the aggregate gradient $g^k = \tfrac{1}{n} \sum_{i=1}^{n} g_i^k$ to form $x^{k+1}$, which is needed before the next iteration can start. This can be achieved, for example, by each

worker $i$ communicating their gradient $g_i^k$ to all other workers. Alternatively, in a parameter server setup, a dedicated master node collects the gradients from all workers, and broadcasts their average $g^k$ to all workers. Instead of communicating the gradient vectors $g_i^k$, which is expensive in distributed learning in general and in federated learning in particular, and especially if $d$ is large, we wish to communicate other but closely related vectors which can be represented with fewer bits. To this effect, each worker $i$ sends the vector

$$v_i^k = \mathcal{C}(e_i^k + \gamma g_i^k), \qquad \forall i \in [n] \tag{7}$$

instead, where $\mathcal{C} : \mathbb{R}^d \to \mathbb{R}^d$ is a (possibly randomized, and in such a case, drawn independently of all else in iteration $k$) compression operator used to reduce communication. We assume throughout that there exists $\delta \in (0, 1]$ such that the following inequality holds for all $x \in \mathbb{R}^d$

$$\mathbf{E}\left[\|\mathcal{C}(x) - x\|^2\right] \le (1 - \delta)\|x\|^2. \tag{8}$$

For any $k \ge 0$, the vector $e_i^{k+1} = \sum_{t=0}^{k} \gamma g_i^t - v_i^t$ captures the *error* accumulated by worker $i$ up to iteration $k$. This is the difference between the ideal `SGD` update $\sum_{t=0}^{k} \gamma g_i^t$ and the applied update $\sum_{t=0}^{k} v_i^t$. As we see in (7), at iteration $k$ the current error $e_i^k$ is added to the gradient update $\gamma g_i^k$—this is referred to as *error feedback*—and subsequently compressed, which defines the update vector $v_i^k$. Compression introduces additional error, which is added to $e_i^k$, and the process is repeated.

**Compression operators.** For a rich collection of specific operators satisfying (8), we refer the reader to Stich and Karimireddy [46] and Beznosikov et al. [7]. These include various unbiased or contractive sparsification operators such as RandK and TopK, respectively, and quantization operators such as natural compression and natural dithering [18]. Several additional comments related to compression operators are included in Section B.

## 2 Summary of Contributions

We now summarize the key contributions of this paper.

⋄ **General theoretical framework.** In this work we propose a *general theoretical framework* for analyzing a wide class of methods that can be written in the the error-feedback form (4)-(5). We perform *complexity analysis* under $\mu$-strong quasi convexity (Assumption 3.1) and $L$-smoothness (Assumption 3.2) assumptions on the functions $f$ and $\{f_i\}$, respectively. Our analysis is based on an additional *parametric assumption* (using parameters $A$, $A'$, $B_1$, $B_1'$, $B_2$, $B_2'$, $C_1$, $C_2$, $D_1$, $D_1'$, $D_2$, $D_3$, $\eta$, $\rho_1$, $\rho_2$, $F_1$, $F_2$, $G$) on the relationship between the iterates $x^k$, stochastic gradients $g^k$, errors $e^k$ and a few other quantities (see Assumption 3.4, and the stronger Assumption 3.3). We prove a single theorem (Theorem 3.1) from which all our complexity results follow as special cases. That is, for each existing or new specific method, we *prove* that one (or both) of our parametric assumptions holds, and specify the parameters for which it holds. These parameters have direct impact on the theoretical rate of the method. A summary of the values of the parameters for all methods developed in this paper is provided in Table 5 in the appendix. We remark that the values of the parameters $A, A', B_1, B_1', B_2, B_2', C_1, C_2$ and $\rho_1, \rho_2$ influence the theoretical stepsize.

⋄ **Sharp rates.** For existing methods covered by our general framework, our main convergence result (Theorem 3.1) recovers the best known rates for these methods up to constant factors.

⋄ **Eight new error-compensated (EC) methods.** We study several specific EC methods for solving problem (1). First, we recover the `EC-SGD` method first analyzed in the $n = 1$ case by Stich and Karimireddy [46] and later in the general $n \ge 1$ case by Beznosikov et al. [7]. More importantly, we develop *eight new methods*: `EC-SGDsr`, `EC-GDstar`, `EC-SGD-DIANA`[2],

Table 1: Complexity of Error-Compensated SGD methods established in this paper. Symbols: $\varepsilon$ = error tolerance; $\delta$ = contraction factor of compressor $\mathcal{C}$; $\omega$ = variance parameter of compressor $\mathcal{Q}$; $\kappa = {}^L/_{\mu}$; $\mathcal{L}$ = expected smoothness constant; $\sigma_*^2$ = variance of the stochastic gradients in the solution; $\zeta_*^2$ = average of $\|\nabla f_i(x^*)\|^2$; $\sigma^2$ = average of the uniform bounds for the variances of stochastic gradients of workers. `EC-GDstar`, `EC-LSVRGstar` and `EC-LSVRG-DIANA` are the first EC methods with a linear convergence rate without assuming that $\nabla f_i(x^*) = 0$ for all $i$. `EC-LSVRGstar` and `EC-LSVRG-DIANA` are the first EC methods with a linear convergence rate which do not require the computation of the full gradient $\nabla f_i(x^k)$ by all workers in each iteration. Out of these three methods, only `EC-LSVRG-DIANA` is practical. $^{\dagger}$`EC-GD-DIANA` is a special case of `EC-SGD-DIANA` where each worker $i$ computes the full gradient $\nabla f_i(x^k)$.

| Problem | Method | Alg # | Citation | Sec # | Rate (constants ignored) |
|---|---|---|---|---|---|
| (1)+(3) | EC-SGDsr | Alg 3 | new | J.1 | $\widetilde{\mathcal{O}}\left( \frac{\mathcal{L}}{\mu} + \frac{L+\sqrt{\delta L \mathcal{L}}}{\delta \mu} + \frac{\sigma_*^2}{n\mu\varepsilon} + \frac{\sqrt{L(\sigma_*^2+\zeta_*^2/\delta)}}{\mu\sqrt{\delta\varepsilon}} \right)$ |
| (1)+(2) | EC-SGD | Alg 4 | [46] | J.2 | $\widetilde{\mathcal{O}}\left( \frac{\kappa}{\delta} + \frac{\sigma_*^2}{n\mu\varepsilon} + \frac{\sqrt{L(\sigma_*^2+\zeta_*^2/\delta)}}{\delta\mu\sqrt{\varepsilon}} \right)$ |
| (1)+(3) | EC-GDstar | Alg 5 | new | J.3 | $\mathcal{O}\left( \frac{\kappa}{\delta} \log\frac{1}{\varepsilon} \right)$ |
| (1)+(2) | EC-SGD-DIANA | Alg 6 | new | J.4 | Opt. I: $\widetilde{\mathcal{O}}\left( \omega + \frac{\kappa}{\delta} + \frac{\sigma^2}{n\mu\varepsilon} + \frac{\sqrt{L\sigma^2}}{\delta\mu\sqrt{\varepsilon}} \right)$ <br> Opt. II: $\widetilde{\mathcal{O}}\left( \frac{1+\omega}{\delta} + \frac{\kappa}{\delta} + \frac{\sigma^2}{n\mu\varepsilon} + \frac{\sqrt{L\sigma^2}}{\mu\sqrt{\delta\varepsilon}} \right)$ |
| (1)+(3) | EC-SGDsr-DIANA | Alg 7 | new | J.5 | Opt. I: $\widetilde{\mathcal{O}}\left( \omega + \frac{\mathcal{L}}{\mu} + \frac{\sqrt{L\mathcal{L}}}{\delta\mu} + \frac{\sigma_*^2}{n\mu\varepsilon} + \frac{\sqrt{L\sigma_*^2}}{\delta\mu\sqrt{\varepsilon}} \right)$ <br> Opt. II: $\widetilde{\mathcal{O}}\left( \frac{1+\omega}{\delta} + \frac{\mathcal{L}}{\mu} + \frac{\sqrt{L\mathcal{L}}}{\delta\mu} + \frac{\sigma_*^2}{n\mu\varepsilon} + \frac{\sqrt{L\sigma_*^2}}{\mu\sqrt{\delta\varepsilon}} \right)$ |
| (1)+(2) | EC-GD-DIANA$^{\dagger}$ | Alg 6 | new | J.4 | $\mathcal{O}\left( \left(\omega + \frac{\kappa}{\delta}\right) \log\frac{1}{\varepsilon} \right)$ |
| (1)+(3) | EC-LSVRG | Alg 8 | new | J.6 | $\widetilde{\mathcal{O}}\left( m + \frac{\kappa}{\delta} + \frac{\sqrt{L\zeta_*^2}}{\delta\mu\sqrt{\varepsilon}} \right)$ |
| (1)+(3) | EC-LSVRGstar | Alg 9 | new | J.7 | $\mathcal{O}\left( \left(m + \frac{\kappa}{\delta}\right) \log\frac{1}{\varepsilon} \right)$ |
| (1)+(3) | EC-LSVRG-DIANA | Alg 10 | new | J.8 | $\mathcal{O}\left( \left(\omega + m + \frac{\kappa}{\delta}\right) \log\frac{1}{\varepsilon} \right)$ |

`EC-SGDsr-DIANA`, `EC-GD-DIANA`, `EC-LSVRG`, `EC-LSVRGstar` and `EC-LSVRG-DIANA`. Some of these methods are designed to work with the expectation structure of the local functions $f_i$ given in (2), and others are specifically designed to exploit the finite-sum structure (3). All these methods follow the error-feedback framework (4)–(5), with $v_i^k$ chosen as in (7). They differ in how the gradient estimator $g_i^k$ is *constructed* (see Table 2 for a compact description of all these methods; formal descriptions can be found in the appendix). As we shall see, the existing `EC-SGD` method uses a rather naive gradient estimator, which renders it less efficient in theory and practice when compared to the best of our new methods. A key property of our parametric assumption described above is that it allows for the construction and modeling of more elaborate gradient estimators, which leads to new EC methods with superior theoretical and practical properties when compared to prior state of the art.

$\diamond$ **First linearly converging EC methods.** The key theoretical consequence of our general framework is the development of the *first linearly converging* error-compensated SGD-type methods for distributed training with biased communication compression. In particular, we design four such methods: two simple but impractical methods, `EC-GDstar` and `EC-LSVRGstar`, with rates $\mathcal{O}\left(\frac{\kappa}{\delta}\ln\frac{1}{\varepsilon}\right)$ and $\mathcal{O}\left(\left(m+\frac{\kappa}{\delta}\right)\ln\frac{1}{\varepsilon}\right)$, respectively, and two practical but more elaborate methods, `EC-GD-DIANA`, with rate $\mathcal{O}\left(\left(\omega+\frac{\kappa}{\delta}\right)\ln\frac{1}{\varepsilon}\right)$, and `EC-LSVRG-DIANA`, with rate $\mathcal{O}\left(\left(\omega+m+\frac{\kappa}{\delta}\right)\ln\frac{1}{\varepsilon}\right)$. In these rates, $\kappa = {}^L/_{\mu}$ is the condition number, $0 < \delta \leq 1$ is the contraction parameter associated with the compressor $\mathcal{C}$ used in (7), and $\omega$ is the variance parameter associated with a *secondary unbiased compressor*[3] $\mathcal{Q}$ which plays a key role in the construction of the gradient estimator $g_i^k$. The complexity of the first and third

Table 2: Error compensated methods developed in this paper. In all cases, $v_i^k = \mathcal{C}(e_i^k + \gamma g_i^k)$. The full descriptions of the algorithms are included in the appendix.

| Problem | Method | $g_i^k$ | Comment |
|---|---|---|---|
| (1) + (3) | `EC-SGDsr` | $\frac{1}{m}\sum_{j=1}^{m}\xi_{ij}\nabla f_{ij}(x^k)$ | $\mathbf{E}\left[\xi_{ij}\right]=1$<br>$\mathbf{E}_{\mathcal{D}_i}\left[\|\nabla f_{\xi_i}(x)-\nabla f_{\xi_i}(x^*)\|^2\right]$<br>$\leq 2\mathcal{L}D_{f_i}(x,x^*)$ |
| (1) + (2) | `EC-SGD` | $\nabla f_{\xi_i}(x^k)$ | |
| (1) | `EC-GDstar` | $\nabla f_i(x^k) - \nabla f_i(x^*)$ | known $\nabla f_i(x^*)\ \forall i$ |
| (1) + (2) | `EC-SGD-DIANA` | $\hat{g}_i^k - h_i^k + h^k$ | $\mathbf{E}\left[\hat{g}_i^k\right]=\nabla f_i(x^k)$<br>$\mathbf{E}_k\left[\|\hat{g}_i^k - \nabla f_i(x^k)\|^2\right]\leq D_{1,i}$<br>$h_i^{k+1}=h_i^k+\alpha\mathcal{Q}(\hat{g}_i^k - h_i^k)$<br>$h^k=\frac{1}{n}\sum_{i=1}^{n}h_i^k$ |
| (1) + (3) | `EC-SGDsr-DIANA` | $\nabla f_{\xi_i^k}(x^k) - h_i^k + h^k$ | $\mathbf{E}\left[\nabla f_{\xi_i^k}(x^k)\right]=\nabla f_i(x^k)$<br>$\mathbf{E}_{\mathcal{D}_i}\left[\|\nabla f_{\xi_i}(x)-\nabla f_{\xi_i}(x^*)\|^2\right]$<br>$\leq 2\mathcal{L}D_{f_i}(x,x^*)$<br>$h_i^{k+1}=h_i^k+\alpha\mathcal{Q}(\nabla f_{\xi_i^k}(x^k) - h_i^k)$<br>$h^k=\frac{1}{n}\sum_{i=1}^{n}h_i^k$ |
| (1) + (3) | `EC-LSVRG` | $\nabla f_{il}(x^k) - \nabla f_{il}(w_i^k)$<br>$+ \nabla f_i(w_i^k)$ | $l$ chosen uniformly from $[m]$<br>$w_i^{k+1}=\begin{cases}x^k, & \text{with prob. } p,\\ w_i^k, & \text{with prob. } 1-p\end{cases}$ |
| (1) + (3) | `EC-LSVRGstar` | $\nabla f_{il}(x^k) - \nabla f_{il}(w_i^k)$<br>$+ \nabla f_i(w_i^k) - \nabla f_i(x^*)$ | $l$ chosen uniformly from $[m]$<br>$w_i^{k+1}=\begin{cases}x^k, & \text{with prob. } p,\\ w_i^k, & \text{with prob. } 1-p\end{cases}$ |
| (1) + (3) | `EC-LSVRG-DIANA` | $\hat{g}_i^k - h_i^k + h^k$<br>where<br>$\hat{g}_i^k = \nabla f_{il}(x^k)$<br>$- \nabla f_{il}(w_i^k) + \nabla f_i(w_i^k)$ | $h_i^{k+1}=h_i^k+\alpha\mathcal{Q}(\hat{g}_i^k - h_i^k)$<br>$h^k=\frac{1}{n}\sum_{i=1}^{n}h_i^k$<br>$l$ chosen uniformly from $[m]$<br>$w_i^{k+1}=\begin{cases}x^k, & \text{with prob. } p,\\ w_i^k, & \text{with prob. } 1-p\end{cases}$ |

methods does not depend on $m$ as they require the computation of the full gradient $\nabla f_i(x^k)$ for each $i$. The remaining two methods only need to compute $\mathcal{O}(1)$ stochastic gradients $\nabla f_{ij}(x^k)$ on each worker $i$.

The first two methods, while impractical, provided us with the intuition which enabled us to develop the practical variant. We include them in this paper due to their simplicity, because of the added insights they offer, and to showcase the flexibility of our general theoretical framework, which is able to describe them. `EC-GDstar` and `EC-LSVRGstar` are impractical since they require the knowledge of the gradients $\{\nabla f_i(x^*)\}$, where $x^*$ is an optimal solution of (1), which are obviously not known since $x^*$ is not known.

The only known linear convergence result for an error compensated `SGD` method is due to Beznosikov et al. [7], who require the computation of the full gradient of $f_i$ by each machine $i$ (i.e., $m$ stochastic gradients), and the additional assumption that $\nabla f_i(x^*) = 0$ for all $i$. We do not need such assumptions, thereby resolving a major theoretical issue with EC methods.

◇ **Results in the convex case.** Our theoretical analysis goes beyond distributed optimization and recovers the results from Gorbunov et al. [11], Khaled et al. [25] (without regularization) in the special case when $v_i^k \equiv \gamma g_i^k$. As we have seen, in this case $e_i^k \equiv 0$ for all $i$ and $k$, and the error-feedback framework (4)–(5) reduces to distributed `SGD` (6). In this regime, the relation (19) in Assumption 3.4 becomes void, while relations (15) and (16) with $\sigma_{2,k}^2 \equiv 0$ are precisely those used by Gorbunov et al. [11] to analyze a wide array of `SGD` methods, including vanilla `SGD` [41], `SGD` with arbitrary sampling [13], as well as variance

reduced methods such as `SAGA` [9], `SVRG` [20], `LSVRG` [17, 31], `JacSketch` [12], `SEGA` [16] and `DIANA` [37, 19]. Our theorem recovers the rates of all the methods just listed in both the convex case $\mu = 0$ Khaled et al. [25] and the strongly-convex case $\mu > 0$ Gorbunov et al. [11] under the more general Assumption 3.4.

◇ `DIANA` **with bi-directional quantization.** To illustrate how our framework can be used even in the case when $v_i^k \equiv \gamma g_i^k$, $e_i^k \equiv 0$, we develop analyze a new version of `DIANA` called `DIANAsr-DQ` that uses arbitrary sampling on every node and double quantization[4], i.e., unbiased compression not only on the workers' side but also on the master's one.

◇ **Methods with delayed updates.** Following Stich [44], we also show that our approach covers `SGD` with delayed updates [1, 3, 10] (`D-SGD`), and our analysis shows the best-known rate for this method. Due to the flexibility of our framework, we are able develop several new variants of `D-SGD` with and without quantization, variance reduction, and arbitrary sampling. Again, due to space limitations, we put these methods together with their convergence analyses in the appendix.

## 3  Main Result

In this section we present the main theoretical result of our paper. First, we introduce our assumption on $f$, which is a relaxation of $\mu$-strong convexity.

**Assumption 3.1** ($\mu$-strong quasi-convexity)**.** *Assume that function $f$ has a unique minimizer $x^*$. We say that function $f$ is strongly quasi-convex with parameter $\mu \geq 0$ if for all $x \in \mathbb{R}^d$*

$$f(x^*) \geq f(x) + \langle \nabla f(x), x^* - x \rangle + \frac{\mu}{2}\|x - x^*\|^2. \tag{9}$$

We allow $\mu$ to be zero, in which case $f$ is sometimes called *weakly quasi-convex* (see [44] and references therein). Second, we introduce the classical *L*-smoothness assumption.

**Assumption 3.2.** *L-smoothness We say that function $f$ is L-smooth if it is differentiable and its gradient is L-Lipschitz continuous, i.e., for all $x, y \in \mathbb{R}^d$*

$$\|\nabla f(x) - \nabla f(y)\| \leq L\|x - y\|. \tag{10}$$

It is a well-known fact [38] that *L*-smoothness of convex function $f$ implies that

$$\|\nabla f(x) - \nabla f(y)\|^2 \leq 2L(f(x) - f(y) - \langle \nabla f(y), x - y \rangle) \stackrel{\text{def}}{=} 2LD_f(x, y). \tag{11}$$

We now introduce our key parametric assumption on the stochastic gradient $g^k$. This is a generalization of the assumption introduced by Gorbunov et al. [11] for the particular class of methods described covered by the EF framework (4)–(5).

**Assumption 3.3.** *For all $k \geq 0$, the stochastic gradient $g^k$ is an average of stochastic gradients $g_i^k$ such that*

$$g^k = \frac{1}{n}\sum_{i=1}^n g_i^k, \qquad \mathbf{E}\left[g^k \mid x^k\right] = \nabla f(x^k). \tag{12}$$

*Moreover, there exist constants $A, \widetilde{A}, A', B_1, B_2, \widetilde{B}_1, \widetilde{B}_2, B_1', B_2', C_1, C_2, G, D_1, \widetilde{D}_1, D_1', D_2, D_3 \geq 0$, and $\rho_1, \rho_2 \in [0, 1]$ and two sequences of (probably random) variables $\{\sigma_{1,k}\}_{k \geq 0}$ and $\{\sigma_{2,k}\}_{k \geq 0}$, such that the following recursions hold:*

$$\frac{1}{n}\sum_{i=1}^n \left\|\bar{g}_i^k\right\|^2 \leq 2A(f(x^k) - f(x^*)) + B_1\sigma_{1,k}^2 + B_2\sigma_{2,k}^2 + D_1, \tag{13}$$

$$\frac{1}{n}\sum_{i=1}^n \mathbf{E}\left[\left\|g_i^k - \bar{g}_i^k\right\|^2 \mid x^k\right] \leq 2\widetilde{A}(f(x^k) - f(x^*)) + \widetilde{B}_1\sigma_{1,k}^2 + \widetilde{B}_2\sigma_{2,k}^2 + \widetilde{D}_1, \tag{14}$$

$$\mathbf{E}\left[\|g^k\|^2 \mid x^k\right] \leq 2A'(f(x^k) - f(x^*)) + B_1'\sigma_{1,k}^2 + B_2'\sigma_{2,k}^2 + D_1', \tag{15}$$

$$\mathbf{E}\left[\sigma_{1,k+1}^2 \mid \sigma_{1,k}^2, \sigma_{2,k}^2\right] \leq (1 - \rho_1)\sigma_{1,k}^2 + 2C_1\left(f(x^k) - f(x^*)\right) + G\rho_1\sigma_{2,k}^2 + D_2, \tag{16}$$

$$\mathbf{E}\left[\sigma_{2,k+1}^2 \mid \sigma_{2,k}^2\right] \leq (1 - \rho_2)\sigma_{2,k}^2 + 2C_2\left(f(x^k) - f(x^*)\right), \tag{17}$$

*where $\bar{g}_i^k = \mathbf{E}\left[g_i^k \mid x^k\right]$.*

Let us briefly explain the intuition behind the assumption and the meaning of the introduced parameters. First of all, we assume that the stochastic gradient at iteration $k$ is conditionally unbiased estimator of $\nabla f(x^k)$, which is a natural and commonly used assumption on the stochastic gradient in the literature. However, we explicitly do *not* require unbiasedness of $g_i^k$, which is very useful in some special cases. Secondly, let us consider the simplest special case when $g^k \equiv \nabla f(x^k)$ and $f_1 = \ldots = f_n = f$, i.e., there is no stochasticity/randomness in the method and the workers have the same functions. Then due to $\nabla f(x^*) = 0$, we have that

$$\|\nabla f(x^k)\|^2 \overset{(11)}{\leq} 2L(f(x^k) - f(x^*)),$$

which implies that Assumption 3.3 holds in this case with $A = A' = L$, $\widetilde{A} = 0$ and $B_1 = B_2 = \widetilde{B}_1 = \widetilde{B}_2 = B_1' = B_2' = C_1 = C_2 = D_1 = \widetilde{D}_1 = D_1' = D_2 = 0$, $\rho = 1$, $\sigma_{1,k}^2 \equiv \sigma_{2,k}^2 \equiv 0$.

In general, if $g^k$ satisfies Assumption 3.4, then parameters $A$, $\widetilde{A}$ and $A'$ are usually connected with the smoothness properties of $f$ and typically they are just multiples of $L$, whereas terms $B_1\sigma_{1,k}^2$, $B_2\sigma_{2,k}^2$, $\widetilde{B}_1\sigma_{1,k}^2$, $\widetilde{B}_2\sigma_{2,k}^2$, $B_1'\sigma_{1,k}^2$, $B_2'\sigma_{2,k}^2$ and $D_1$, $\widetilde{D}_1$, $D_1'$ appear due to the stochastic nature of $g_i^k$. Moreover, $\{\sigma_{1,k}^2\}_{k\geq0}$ and $\{\sigma_{2,k}^2\}_{k\geq0}$ are sequences connected with variance reduction processes and for the methods; without any kind of variance reduction these sequences contains only zeros. Parameters $B_1$ and $B_2$ are often 0 or small positive constants, e.g., $B_1 = B_2 = 2$, and $D_1$ characterizes the remaining variance in the estimator $g^k$ that is not included in the first two terms.

Inequalities (16) and (17) describe the variance reduction processes: one can interpret $\rho_1$ and $\rho_2$ as the *rates* of the variance reduction processes, $2C_1(f(x^k) - f(x^*))$ and $2C_2(f(x^k) - f(x^*))$ are "optimization" terms and, similarly to $D_1$, $D_2$ represents the remaining variance that is not included in the first two terms. Typically, $\sigma_{1,k}^2$ controls the variance coming from compression and $\sigma_{2,k}^2$ controls the variance taking its origin in finite-sum type randomization (i.e., subsampling) by each worker. In the case $\rho_1 = 1$ we assume that $B_1 = B_1' = C_1 = G = 0, D_2 = 0$ (for $\rho_2 = 1$ analogously), since inequality (16) becomes superfluous.

However, in our main result we need a slightly different assumption.

**Assumption 3.4.** *For all $k \geq 0$, the stochastic gradient $g^k$ is an unbiased estimator of $\nabla f(x^k)$:*

$$\mathbf{E}\left[g^k \mid x^k\right] = \nabla f(x^k). \tag{18}$$

*Moreover, there exist non-negative constants $A', B_1', B_2', C_1, C_2, F_1, F_2, G, D_1', D_2, D_3 \geq 0, \rho_1, \rho_2 \in [0,1]$ and two sequences of (probably random) variables $\{\sigma_{1,k}\}_{k\geq0}$ and $\{\sigma_{2,k}\}_{k\geq0}$ such that inequalities (15), (16) and (17) hold and*

$$3L \sum_{k=0}^{K} w_k \mathbf{E}\|e^k\|^2 \leq \tfrac{1}{4} \sum_{k=0}^{K} w_k \mathbf{E}\left[f(x^k) - f(x^*)\right] + F_1\sigma_{1,0}^2 + F_2\sigma_{2,0}^2 + \gamma D_3 W_K \tag{19}$$

*for all $k, K \geq 0$, where $e^k = \tfrac{1}{n}\sum_{i=1}^{n} e_i^k$ and $\{W_K\}_{K\geq0}$ and $\{w_k\}_{k\geq0}$ are defined as*

$$W_K = \sum_{k=0}^{K} w_k, \quad w_k = (1-\eta)^{-(k+1)}, \quad \eta = \min\left\{\tfrac{\gamma\mu}{2}, \tfrac{\rho_1}{4}, \tfrac{\rho_2}{4}\right\}. \tag{20}$$

This assumption is more flexible than Assumption 3.3 and helps us to obtain a unified analysis of all methods falling in the error-feedback framework. We emphasize that in this assumption we do not assume that (13) and (14) hold *explicitly*. Instead of this, we introduce inequality (19), which is the key tool that helps us to analyze the effect of error-feedback and comes from the analysis from [46] with needed adaptations connected with the first three inequalities. As we show in the appendix, this inequality can be derived for `SGD` with error compensation and delayed updates under Assumption 3.3 and, in particular, using (13) and (14). As before, $D_3$ hides a variance that is not handled by variance reduction processes and $F_1$ and $F_2$ are some constants that typically depend on $L, B_1, B_2, \rho_1, \rho_2$ and $\gamma$.

We now proceed to stating our main theorem.

**Theorem 3.1.** *Let Assumptions 3.1, 3.2 and 3.4 be satisfied and $\gamma \leq 1/4(A' + C_1 M_1 + C_2 M_2)$. Then for all $K \geq 0$ we have*

$$\mathbf{E}\left[f(\bar{x}^K) - f(x^*)\right] \leq (1-\eta)^K \frac{4(T^0 + \gamma F_1 \sigma_{1,0}^2 + \gamma F_2 \sigma_{2,0}^2)}{\gamma} + 4\gamma \left(D_1' + M_1 D_2 + D_3\right) \qquad (21)$$

*when $\mu > 0$ and*

$$\mathbf{E}\left[f(\bar{x}^K) - f(x^*)\right] \leq \frac{4(T^0 + \gamma F_1 \sigma_{1,0}^2 + \gamma F_2 \sigma_{2,0}^2)}{\gamma K} + 4\gamma \left(D_1' + M_1 D_2 + D_3\right) \qquad (22)$$

*when $\mu = 0$, where $\eta = \min\left\{\gamma\mu/2, \rho_1/4, \rho_2/4\right\}$, $T^k \stackrel{def}{=} \|\tilde{x}^k - x^*\|^2 + M_1 \gamma^2 \sigma_{1,k}^2 + M_2 \gamma^2 \sigma_{2,k}^2$ and $M_1 = \frac{4B_1'}{3\rho_1}$, $M_2 = \frac{4\left(B_2' + \frac{4}{3}G\right)}{3\rho_2}$.*

All the complexity results summarized in Table 1 follow from this theorem; the detailed proofs are included in the appendix. Furthermore, in the appendix we include similar results but for methods employing *delayed* updates. The methods, and all associated theory is included there, too.

# 4 Numerical Experiments

To justify our theory, we conduct several numerical experimentson logistic regression problem with $\ell_2$-regularization:

$$\min_{x \in \mathbb{R}^d} \left\{ f(x) = \frac{1}{N} \sum_{i=1}^{N} \log\left(1 + \exp\left(-y_i \cdot (Ax)_i\right)\right) + \frac{\mu}{2}\|x\|^2 \right\}, \qquad (23)$$

where $N$ is a number of features, $x \in \mathbb{R}^d$ represents the weights of the model, $A \in \mathbb{R}^{N \times d}$ is a feature matrix, vector $y \in \{-1, 1\}^N$ is a vector of labels and $(Ax)_i$ denotes the $i$-th component of vector $Ax$. Clearly, this problem is $L$-smooth and $\mu$-strongly convex with $L = \mu + \lambda_{\max}(A^\top A)/4N$, where $\lambda_{\max}(A^\top A)$ is a largest eigenvalue of $A^\top A$. The datasets were taken from LIBSVM library [8], and the code was written in Python 3.7 using standard libraries. Our code is available at `https://github.com/eduardgorbunov/ef_sigma_k`.

We simulate parameter-server architecture using one machine with Intel(R) Core(TM) i7-9750 CPU 2.60 GHz in the following way. First of all, we always use such $N$ that $N = n \cdot m$ and consider $n = 20$ and $n = 100$ workers. The choice of $N$ for each dataset that we consider is stated in Table 3. Next, we shuffle the data and split in $n$ groups of size $m$. To emulate

Table 3: Summary of datasets: $N$ = total # of data samples; $d$ = # of features.

|   | a9a | w8a | gisette | mushrooms | madelon | phishing |
|---|---|---|---|---|---|---|
| $N$ | $32,000$ | $49,700$ | $6,000$ | $8,000$ | $2,000$ | $11,000$ |
| $d$ | $123$ | $300$ | $5,000$ | $112$ | $500$ | $68$ |

the work of workers, we use a single machine and run the methods with the parallel loop in series. Since in these experiments we study sample complexity and number of bits used for communication, this setup is identical to the real parameter-server setup in this sense.

In all experiments we use the stepsize $\gamma = 1/L$ and $\ell_2$-regularization parameter $\mu = 10^{-4}\lambda_{\max}(A^\top A)/4N$. The starting point $x^0$ for each dataset was chosen so that $f(x^0) - f(x^*) \sim 10$. In experiments with stochastic methods we used batches of size 1 and uniform sampling for simplicity. For `LSVRG`-type methods we choose $p = 1/m$.

**Compressing stochastic gradients.** The results for `a9a`, `madelon` and `phishing` can be found in Figure 1 (included here) and for `w8a`, `mushrooms` and `gisette` in Figure 3 (in the Appendix). We choose number of components for TopK operator of the order $\max\{1, d/100\}$. Clearly, in these experiments we see two levels of noise. For some datasets, like `a9a`, `phishing` or `mushrooms`, the noise that comes from the stochasticity of the gradients dominates the noise coming from compression. Therefore, methods such as `EC-SGD` and `EC-SGD-DIANA` start to oscillate around a larger value of the loss function than other methods we consider. `EC-LSVRG` reduces the largest source of noise and, as a result, finds a better approximation of

the solution. However, at some point, it reaches another level of the loss function and starts to oscillate there due to the noise coming from compression. Finally, `EC-LSVRG-DIANA` reduces the variance of both types, and as a result, finds an even better approximation of the solution. In contrast, for the `madelon` dataset, both noises are of the same order, and therefore, `EC-LSVRG` and `EC-SGD-DIANA` behave similarly to `EC-SGD`. However, `EC-LSVRG-DIANA` again reduces both types of noise effectively and finds a better approximation of the solution after a given number of epochs. In the experiments with `w8a` and `gisette` datasets, the noise produced by compression is dominated by the noise coming from the stochastic gradients. As a result, we see that the `DIANA`-trick is not needed here.

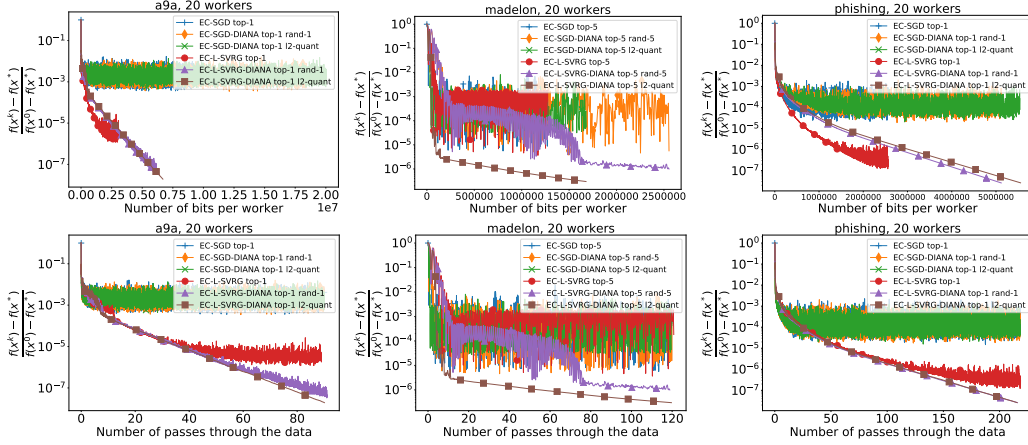

Figure 1: Trajectories of `EC-SGD`, `EC-SGD-DIANA`, `EC-LSVRG` and `EC-LSVRG-DIANA` applied to solve logistic regression problem with 20 workers.

**Compressing full gradients.** In order to show the effect of `DIANA`-type variance reduction itself, we consider the case when all workers compute the full gradients of their functions, see Figure 2 (included here) and Figures 4–7 (in the Appendix). Clearly, for all datasets except `mushrooms`, `EC-GD` with constant stepsize converges to a neighborhood of the solution only, while `EC-GDstar` and `EC-GD-DIANA` converge with linear rate asymptotically to the exact solution. `EC-GDstar` always show the best performance, however, it is impractical: we used a very good approximation of the solution to apply this method. In contrast, `EC-DIANA` converges slightly slower and requires more bits for communication; but it is practical and shows better performance than `EC-GD`. On the `mushrooms` datasets, `EC-GD` does not reach the oscillation region after the given number of epochs, therefore, it is preferable there.

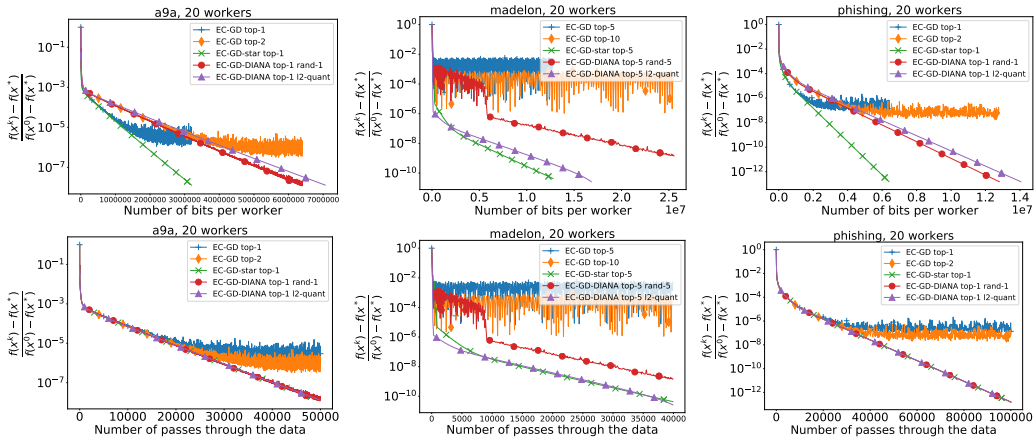

Figure 2: Trajectories of `EC-GD`, `EC-GD-star` and `EC-DIANA` applied to solve logistic regression problem with 20 workers.

## Broader Impact

Our contribution is primarily theoretical. Therefore, a broader impact discussion is not applicable.

## Acknowledgments and Disclosure of Funding

The work of Peter Richtárik, Eduard Gorbunov and Dmitry Kovalev was supported by KAUST Baseline Research Fund. Part of this work was done while E. Gorbunov was a research intern at KAUST. The work of E. Gorbunov in Sections 4, A–D, H, and K was also partially supported by the Ministry of Science and Higher Education of the Russian Federation (Goszadaniye) 075-00337-20-03, project No. 0714-2020-0005, and in Sections 3, E–G, I, J – by RFBR, project number 19-31-51001.

## Footnotes

[1]The implicit assumption that each worker contains exactly $m$ data points is for simplicity only; all our results have direct analogues in the general setting with $m_i$ data points on worker $i$.

[2]Inspired by personal communication with D. Kovalev in November 2019 who shared a key algorithm and preliminary results of our paper, Stich [45] studied almost the same algorithm and also other related methods and independently derived convergence rates. Our work was finalized and submitted to NeurIPS 2020 in June 2020, while the results in [45] were obtained in Summer 2020 and appeared on arXiv in September 2020. Moreover, in our work, we obtain tighter rates (see Table 1 for the details).

[3]We assume that $\mathbf{E}\mathcal{Q}(x) = x$ and $\mathbf{E}\|\mathcal{Q}(x) - x\|^2 \leq \omega\|x\|^2$ for all $x \in \mathbb{R}^d$.

[4]In the concurrent work (which appeared on arXiv after we have submitted our paper to NeurIPS) a similar method was independently proposed under the name of `Artemis` [40]. However, our analysis is more general, see all the details on this method in the appendix. This footnote was added to the paper during the preparation of the camera-ready version of our paper.

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
