[Supplementary Material]

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

[5]When $\rho_1 = 1$ and $\rho_2 = 1$ one can always set the parameters in such a way that $B_1 = \widetilde{B}_1 = B_2 = \widetilde{B}_2 = C_1 = C_2 = 0$, $D_2 = 0$. In this case we assume that $\frac{2}{1-\rho_1}\left(\frac{C_1}{\rho_1} + \frac{2GC_2}{\rho_2(1-\rho_2)}\right)\left(\frac{2B_1}{\delta} + \widetilde{B}_1\right) + \frac{2C_2\left(\frac{2B_2}{\delta} + \widetilde{B}_2\right)}{\rho_2(1-\rho_2)} = 0$.

[6]When $\rho_1 = 1$ and $\rho_2 = 1$ one can always set the parameters in such a way that $B_1 = B_1' = B_2 = B_2' = C_1 = C_2 = 0$, $D_2 = 0$. In this case we assume that $\frac{2B_1'C_1}{\rho_1(1-\rho_1)} = \frac{2B_2'C_2}{\rho_2(1-\rho_2)} = 0$.

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

## A.2  Compressing full gradients

Figure 4: Trajectories of `EC-GD`, `EC-GD-star` and `EC-DIANA` applied to solve logistic regression problem with 20 workers.

Figure 5: Trajectories of `EC-GD`, `EC-GD-star` and `EC-DIANA` applied to solve logistic regression problem with 100 workers.

Figure 6: Trajectories of `EC-GD`, `EC-GD-star`, `EC-DIANA` and `GD` applied to solve logistic regression problem with 20 workers.

Figure 7: Trajectories of `EC-GD`, `EC-GD-star`, `EC-DIANA` and `GD` applied to solve logistic regression problem with 100 workers.

# B Compression Operators: Extra Commentary

Communication efficient distributed `SGD` methods based on the idea of communication compression exists in two distinct varieties: i) methods based on unbiased compression operators, and ii) methods based on biased compression operators. The first class of methods is much mire developed than the latter since it is easier to theoretically analyze unbiased operators. The subject of this paper is the study of the latter and dramatically less developed and understood class.

## B.1 Unbiased compressors

By unbiased compression operators we mean randomized mappings $\mathcal{Q} : \mathbb{R}^d \to \mathbb{R}$ satisfying the relations

$$\mathbf{E}\mathcal{Q}(x) = x \qquad \text{and} \qquad \mathbf{E}\|\mathcal{Q}(x) - x\|^2 \leq \omega\|x\|^2, \qquad \forall x \in \mathbb{R}^d$$

for some $\omega \geq 0$. While operators satisfying the above relations are often in the literature called *quantization operators*, this class includes compressors which perform sparsification as well.

Among the first methods using unbiased compressors developed in this field are `QSGD` [2], `TernGrad` [49] and `DQGD` [26]. The first analysis of `QSGD` and `TernGrad` without bounded gradients assumptions was proposed in [37], which contains the best known results for `QSGD` and `TernGrad`. However, existing guarantees in the strongly convex case for `QGSD`, `TernGrad`, and `DQGD` establish linear convergence to some neighborhood of the solution only, even if the workers quantize the full gradients of their functions. This problem was resolved by Mishchenko et al. [37], who proposed the first method, called `DIANA`, which uses quantization for communication and enjoys the linear rate of convergence to the exact optimum asymptotically in the strongly convex case when workers compute the full gradients of their functions in each iteration. Unlike all previous approaches, `DIANA` is based on the quantization of gradient differences rather than iterates or gradients. In essence, DIANA is a technique for reducing the variance introduced by quantization. Horváth et al. [19] generalized the `DIANA` method to the case of more general quantization operators. Moreover, the same authors developed a new method called `VR-DIANA` specially designed to solve problems (1) with the individual functions having the finite sum structure (3).

## B.2 Biased compressors

By biased compressors we mean (possibly) randomized mappings $\mathcal{C} : \mathbb{R}^d \to \mathbb{R}$ satisfying the average contraction relation

$$\mathbf{E}\left[\|\mathcal{C}(x) - x\|^2\right] \leq (1 - \delta)\|x\|^2, \qquad \forall x \in \mathbb{R}^d$$

for some $\delta > 0$.

Perhaps the most popular biased compression operator is TopK, which takes vector $x$ as input and substitutes all coordinates of $x$ by zero except the $k$ components with the largest absolute values. However, such a greedy approach applied to simple distributed `SGD` and even distributed `GD` can break the convergence of the method even when applied to simple functions in small dimensions, and may even lead to exponential divergence [7]. The *error-feedback* framework described in [23, 46, 47] and studies in this paper can fix this problem, and it remains the only known mechanism that does so for all compressors described in (8). This is one of the main motivations for the study of the error-feedback mechanism. For instance, error feedback can fix convergence issues with methods like `sign-SGD` [5]. The analysis of error feedback by Karimireddy et al. [23], Stich and Karimireddy [46], Stich et al. [47] works either under the assumption that the second moment of the stochastic gradient is uniformly bounded or only for the single-worker case. Recently Beznosikov et al. [7] proposed the first analysis of `SGD` with error feedback for the general case of multiple workers without bounded second moment assumption. There is another line of works [27, 28] where authors apply arbitrary compressions in the decentralized setup. This approach has better potential than a centralized one in terms of reducing the communication cost. However, in this paper, we study only centralized architecture.

## C  Further Notation and Definitions

In what follows it will be useful to denote

$$v^k \overset{\text{def}}{=} \tfrac{1}{n}\sum_i v_i^k, \quad g^k \overset{\text{def}}{=} \tfrac{1}{n}\sum_i g_i^k, \quad e^k \overset{\text{def}}{=} \tfrac{1}{n}\sum_i e_i^k.$$

By aggregating identities (5) across all $i$, we get $e^{k+1} = e^k + \gamma g^k - v^k$. In our proofs we also use the perturbed iterates technique [32, 36] based on the analysis of the following sequence

$$\tilde{x}^k = x^k - e^k. \tag{24}$$

This sequence satisfies very useful for the analysis relation:

$$\tilde{x}^{k+1} \overset{(24)}{=} x^{k+1} - e^{k+1} \overset{(4),(5)}{=} x^k - v^k - (e^k + \gamma g^k - v^k) = x^k - e^k - \gamma g^k \overset{(24)}{=} \tilde{x}^k - \gamma g^k. \tag{25}$$

### C.1  Quantization operators

**Definition C.1.** *We say that stochastic mapping $Q(x) : \mathbb{R}^d \to \mathbb{R}^d$ is a quantization operator if there exists such $\omega > 0$ that for any $x \in \mathbb{R}^d$*

$$\mathbf{E}\left[\mathcal{Q}(x)\right] = x, \quad \mathbf{E}\left[\|\mathcal{Q}(x) - x\|^2\right] \le \omega\|x\|^2. \tag{26}$$

Below we enumerate some classical compression and quantization operators (see more in [7]).

1. **TopK sparsification.** This compression operator is defined as follows:

$$\mathcal{C}(x) = \sum_{i=1}^K x_{(i)} e_{(i)}$$

   where $|x_{(1)}| \ge |x_{(2)}| \ge \ldots \ge |x_{(d)}|$ are components of $x$ sorted in the decreasing order of their absolute values, $e_1, \ldots, e_d$ is the standard basis in $\mathbb{R}^d$ and $K$ is some number from $[d]$. Clearly, TopK is a biased compression operator. One can show that TopK satisfies (8) with $\delta = \frac{K}{d}$ [7].

2. **RandK sparsification** operator is defined as

$$\mathcal{Q}(x) = \frac{d}{K}\sum_{i \in S} x_i e_i$$

   where $S$ is a random subset of $[d]$ sampled from the uniform distribution on the all subset of $[d]$ with cardinality $K$. RandK is an unbiased compression operator satisfying (26) with $\omega = \frac{d}{K}$.

3. **$\ell_p$-quantization.** By $\ell_2$-quantization we mean the following random operator:

$$\mathcal{Q}(x) = \|x\|_p \text{sign}(x) \circ \xi$$

   where $\|x\|_p = \left(\sum_{i=1}^d |x_i|^p\right)^{1/p}$ is an $\ell_p$-norm of vector $x$, $\text{sign}(x)$ is a component-wise sign of vector $x$, $a \circ b$ defines a component-wise product of vectors $a$ and $b$ and $\xi = (\xi_1, \ldots, \xi_d)^\top$ is a random vector such that

$$\xi_i = \begin{cases} 1, & \text{with probability } \frac{|x_i|}{\|x\|_p}, \\ 0, & \text{with probability } 1 - \frac{|x_i|}{\|x\|_p}. \end{cases}$$

   One can show that this operator satisfies (26). In particular, if $p = 2$ it satisfies (26) with $\omega = \sqrt{d} - 1$ and if $p = \infty$, then $\omega = \frac{1+\sqrt{d}}{2} - 1$ (see [37]).

We assume that $\mathcal{C}$ is any operator which enjoys the following contractive property: there exists a constant $0 < \delta \le 1$ such that

$$\mathbf{E}\left[\|x - \mathcal{C}(x)\|^2\right] \le (1-\delta)\|x\|^2, \quad \forall x \in \mathbb{R}^d.$$

# D  Basic Inequalities, Identities and Technical Lemmas

## D.1  Basic inequalities

For all $a, b, x_1, \ldots, x_n \in \mathbb{R}^d$, $\beta > 0$ and $p \in (0, 1]$ the following inequalities hold

$$\langle a, b \rangle \leq \frac{\|a\|^2}{2\beta} + \frac{\beta \|b\|^2}{2}, \tag{27}$$

$$\langle a - b, a + b \rangle = \|a\|^2 - \|b\|^2, \tag{28}$$

$$\frac{1}{2}\|a\|^2 - \|b\|^2 \leq \|a + b\|^2, \tag{29}$$

$$\|a + b\|^2 \leq (1 + \beta)\|a\|^2 + (1 + 1/\beta)\|b\|^2, \tag{30}$$

$$\left\| \sum_{i=1}^{n} x_n \right\|^2 \leq n \sum_{i=1}^{n} \|x_i\|^2, \tag{31}$$

$$\left(1 - \frac{p}{2}\right)^{-1} \leq 1 + p, \tag{32}$$

$$\left(1 + \frac{p}{2}\right)(1 - p) \leq 1 - \frac{p}{2}. \tag{33}$$

## D.2  Identities and inequalities involving random variables

**Variance decomposition.** For a random vector $\xi \in \mathbb{R}^d$ and any deterministic vector $x \in \mathbb{R}^d$ the variance can be decomposed as

$$\mathbf{E}\left[\|\xi - \mathbf{E}\xi\|^2\right] = \mathbf{E}\left[\|\xi - x\|^2\right] - \|\mathbf{E}\xi - x\|^2 \tag{34}$$

**Tower property of mathematical expectation.** For random variables $\xi, \eta \in \mathbb{R}^d$ we have

$$\mathbf{E}[\xi] = \mathbf{E}[\mathbf{E}[\xi \mid \eta]] \tag{35}$$

under assumption that all expectations in the expression above are well-defined.

**Lemma D.1** (Lemma 14 from [46]). *For any $\tau$ vectors $a_1, \ldots, a_\tau \in \mathbb{R}^d$ and $\xi_1, \ldots, \xi_\tau$ zero-mean random vectors in $\mathbb{R}^d$, each $\xi_t$ conditionally independent of $\{\xi_i\}_{i=1}^{t-1}$ for all $1 \leq t \leq \tau$ the following inequality holds*

$$\mathbf{E}\left[\left\|\sum_{t=1}^{\tau}(a_t + \xi_t)\right\|^2\right] \leq \tau \sum_{t=1}^{\tau} \|a_t\|^2 + \sum_{t=1}^{\tau} \mathbf{E}\|\xi_t\|^2. \tag{36}$$

## D.3  Technical lemmas

**Lemma D.2** (see also Lemma 2 from [44]). *Let $\{r_k\}_{k \geq 0}$ satisfy*

$$r_K \leq \frac{a}{\gamma W_K} + c_1 \gamma + c_2 \gamma^2 \tag{37}$$

*for all $K \geq 0$ with some constants $a, c_2 > 0$, $c_1 \geq 0$ where $\{w_k\}_{k \geq 0}$ and $\{W_K\}_{K \geq 0}$ are defined in (20), $\gamma \leq \frac{1}{d}$. Then for all $K$ such that $\frac{\ln\left(\max\{2, \min\{a\mu^2 K^2/c_1, a\mu^3 K^3/c_2\}\}\right)}{K} \leq \min\{\rho_1, \rho_2\}$ and*

$$\gamma = \min\left\{\frac{1}{d}, \frac{\ln\left(\max\{2, \min\{a\mu^2 K^2/c_1, a\mu^3 K^3/c_2\}\}\right)}{\mu K}\right\} \tag{38}$$

*we have that*

$$r_K = \widetilde{\mathcal{O}}\left(da \exp\left(-\min\left\{\frac{\mu}{d}, \rho_1, \rho_2\right\}K\right) + \frac{c_1}{\mu K} + \frac{c_2}{\mu^2 K^2}\right). \tag{39}$$

*Proof.* Since $W_K \geq w_K = (1-\eta)^{-(K+1)}$ we have

$$r_K \quad \leq \quad (1-\eta)^{K+1}\frac{a}{\gamma} + c_1\gamma + c_2\gamma^2 \leq \frac{a}{\gamma}\exp\left(-\eta(K+1)\right) + c_1\gamma + c_2\gamma^2. \tag{40}$$

Next we consider two possible situations.

1. If $\frac{1}{d} \geq \frac{\ln\left(\max\{2,\min\{a\mu^2 K^2/c_1, a\mu^3 K^3/c_2\}\}\right)}{\mu K}$ then we choose $\gamma = \frac{\ln\left(\max\{2,\min\{a\mu^2 K^2/c_1, a\mu^3 K^3/c_2\}\}\right)}{\mu K}$ and get that

$$
\begin{aligned}
r_K \quad &\overset{(40)}{\leq} \quad \frac{a}{\gamma}\exp\left(-\eta(K+1)\right) + c_1\gamma + c_2\gamma^2 \\
&= \quad \widetilde{\mathcal{O}}\left(a\mu K\exp\left(-\min\left\{\rho_1,\rho_2, \frac{\ln\left(\max\{2,\min\{a\mu^2 K^2/c_1, a\mu^3 K^3/c_2\}\}\right)}{K}\right\}K\right)\right) \\
&\quad + \widetilde{\mathcal{O}}\left(\frac{c_1}{\mu K} + \frac{c_2}{\mu^2 K^2}\right).
\end{aligned}
$$

Since $\frac{\ln\left(\max\{2,\min\{a\mu^2 K^2/c_1, a\mu^3 K^3/c_2\}\}\right)}{K} \leq \min\{\rho_1,\rho_2\}$ we have

$$
\begin{aligned}
r_K \quad &= \quad \widetilde{\mathcal{O}}\left(a\mu K\exp\left(-\ln\left(\max\left\{2,\min\left\{\frac{a\mu^2 K^2}{c_1}, \frac{a\mu^3 K^3}{c_2}\right\}\right\}\right)\right)\right) \\
&\quad + \widetilde{\mathcal{O}}\left(\frac{c_1}{\mu K} + \frac{c_2}{\mu^2 K^2}\right) \\
&= \quad \widetilde{\mathcal{O}}\left(\frac{c_1}{\mu K} + \frac{c_2}{\mu^2 K^2}\right).
\end{aligned}
$$

2. If $\frac{1}{d} \leq \frac{\ln\left(\max\{2,\min\{a\mu^2 K^2/c_1, a\mu^3 K^3/c_2\}\}\right)}{\mu K}$ then we choose $\gamma = \frac{1}{d}$ which implies that

$$
\begin{aligned}
r_K \quad &\overset{(40)}{\leq} \quad da\exp\left(-\min\left\{\frac{\mu}{d}, \frac{\rho_1}{4}, \frac{\rho_2}{4}\right\}(K+1)\right) + \frac{c_1}{d} + \frac{c_2}{d^2} \\
&= \quad \widetilde{\mathcal{O}}\left(da\exp\left(-\min\left\{\frac{\mu}{d}, \rho_1, \rho_2\right\}K\right) + \frac{c_1}{\mu K} + \frac{c_2}{\mu^2 K^2}\right).
\end{aligned}
$$

Combining the obtained bounds we get the result. $\qquad\square$

**Lemma D.3.** *Let $\{r_k\}_{k\geq 0}$ satisfy*

$$r_K \leq \frac{a}{\gamma K} + \frac{b_1\gamma}{K} + \frac{b_2\gamma^2}{K} + c_1\gamma + c_2\gamma^2 \tag{41}$$

*for all $K \geq 0$ with some constants $a > 0$, $b_1, b_2, c_1, c_2 \geq 0$ where $\gamma \leq \gamma_0$. Then for all $K$ and*

$$\gamma = \min\left\{\gamma_0, \sqrt{\frac{a}{b_1}}, \sqrt[3]{\frac{a}{b_2}}, \sqrt{\frac{a}{c_1 K}}, \sqrt[3]{\frac{a}{c_2 K}}\right\}$$

*we have that*

$$r_K = \mathcal{O}\left(\frac{a}{\gamma_0 K} + \frac{\sqrt{ab_1}}{K} + \frac{\sqrt[3]{a^2 b_2}}{K} + \sqrt{\frac{ac_1}{K}} + \frac{\sqrt[3]{a^2 c_2}}{K^{2/3}}\right). \tag{42}$$

*Proof.* We have

$$
\begin{aligned}
r_K \quad &\leq \quad \frac{a}{\gamma K} + \frac{b_1\gamma}{K} + \frac{b_2\gamma^2}{K} + c_1\gamma + c_2\gamma^2 \\
&\leq \quad \frac{a}{\min\left\{\gamma_0, \sqrt{\frac{a}{b_1}}, \sqrt[3]{\frac{a}{b_2}}, \sqrt{\frac{a}{c_1 K}}, \sqrt[3]{\frac{a}{c_2 K}}\right\}K} + \frac{b_1}{K}\cdot\sqrt{\frac{a}{b_1}} + \frac{b_2}{K}\cdot\sqrt[3]{\frac{a}{b_2}} \\
&\quad + c_1\cdot\sqrt{\frac{a}{c_1 K}} + c_2\left(\sqrt[3]{\frac{a}{c_2 K}}\right)^2 \\
&= \quad \mathcal{O}\left(\frac{a}{\gamma_0 K} + \frac{\sqrt{ab_1}}{K} + \frac{\sqrt[3]{a^2 b_2}}{K} + \sqrt{\frac{ac_1}{K}} + \frac{\sqrt[3]{a^2 c_2}}{K^{2/3}}\right).
\end{aligned}
$$

□

# E   Proofs for Section 3

## E.1   A lemma

**Lemma E.1** (See also Lemma 8 from [46])**.** *Let Assumptions 3.1, 3.4 and 3.2 be satisfied and $\gamma \leq 1/4(A' + C_1 M_1 + C_2 M_2)$. Then for all $k \geq 0$ we have*

$$\frac{\gamma}{2}\mathbf{E}\left[f(x^k) - f(x^*)\right] \leq (1-\eta)\mathbf{E}T^k - \mathbf{E}T^{k+1} + \gamma^2(D_1' + M_1 D_2) + 3L\gamma\mathbf{E}\|e^k\|^2, \qquad (43)$$

*where $T^k \stackrel{def}{=} \|\tilde{x}^k - x^*\|^2 + M_1\gamma^2\sigma_{1,k}^2 + M_2\gamma^2\sigma_{2,k}^2$ and $M_1 = \frac{4B_1'}{3\rho_1}$, $M_2 = \frac{4\left(B_2' + \frac{4}{3}G\right)}{3\rho_2}$.*

*Proof.* We start with the upper bound for $\mathbf{E}\|\tilde{x}^{k+1} - x^*\|^2$. First of all, by definition of $\tilde{x}^k$ we have

$$
\begin{aligned}
\|\tilde{x}^{k+1} - x^*\|^2 &\stackrel{(25)}{=} \|\tilde{x}^k - x^* - \gamma g^k\|^2 \\
&= \|\tilde{x}^k - x^*\|^2 - 2\gamma\langle\tilde{x}^k - x^*, g^k\rangle + \gamma^2\|g^k\|^2 \\
&= \|\tilde{x}^k - x^*\|^2 - 2\gamma\langle x^k - x^*, g^k\rangle + \gamma^2\|g^k\|^2 + 2\gamma\langle x^k - \tilde{x}^k, g^k\rangle.
\end{aligned}
$$

Taking conditional expectation $\mathbf{E}\left[\cdot \mid x^k\right]$ from the both sides of the previous inequality we get

$$
\begin{aligned}
\mathbf{E}\left[\|\tilde{x}^{k+1} - x^*\|^2 \mid x^k\right] &\stackrel{(18),(15)}{\leq} \|\tilde{x}^k - x^*\|^2 - 2\gamma\langle x^k - x^*, \nabla f(x^k)\rangle \\
&\quad + \gamma^2\left(2A'(f(x^k) - f(x^*)) + B_1'\sigma_{1,k}^2 + B_2'\sigma_{2,k}^2 + D_1'\right) \\
&\quad + 2\gamma\langle x^k - \tilde{x}^k, \nabla f(x^k)\rangle \\
&\stackrel{(9)}{\leq} \|\tilde{x}^k - x^*\|^2 - \gamma\mu\|x^k - x^*\|^2 - \gamma(2 - 2A'\gamma)(f(x^k) - f(x^*)) \\
&\quad + \gamma^2 B_1'\sigma_{1,k}^2 + \gamma^2 B_2'\sigma_{2,k}^2 + \gamma^2 D_1' \\
&\quad + 2\gamma\langle x^k - \tilde{x}^k, \nabla f(x^k)\rangle. \qquad (44)
\end{aligned}
$$

Next,

$$-\|x^k - x^*\|^2 = -\|\tilde{x}^k - x^* + x^k - \tilde{x}^k\|^2 \stackrel{(29)}{\leq} -\frac{1}{2}\|\tilde{x}^k - x^*\|^2 + \|x^k - \tilde{x}^k\|^2. \qquad (45)$$

Using Fenchel-Young inequality we derive an upper bound for the inner product from (44):

$$\langle x^k - \tilde{x}^k, \nabla f(x^k)\rangle \stackrel{(27)}{\leq} L\|x^k - \tilde{x}^k\|^2 + \frac{1}{4L}\|\nabla f(x^k)\|^2 \stackrel{(11)}{\leq} L\|x^k - \tilde{x}^k\|^2 + \frac{1}{2}(f(x^k) - f(x^*)). \qquad (46)$$

Combining previous three inequalities we get

$$
\begin{aligned}
\mathbf{E}\left[\|\tilde{x}^{k+1} - x^*\|^2 \mid x^k\right] &\stackrel{(44)-(46)}{\leq} \left(1 - \frac{\gamma\mu}{2}\right)\|\tilde{x}^k - x^*\|^2 - \gamma\left(1 - 2A'\gamma\right)(f(x^k) - f(x^*)) \\
&\quad + \gamma^2 B_1'\sigma_{1,k}^2 + \gamma^2 B_2'\sigma_{2,k}^2 + \gamma^2 D_1' \\
&\quad + \gamma(2L + \mu)\|x^k - \tilde{x}^k\|^2. \qquad (47)
\end{aligned}
$$

Taking into account that $T^k = \|\tilde{x}^k - x^*\|^2 + M_1\gamma^2\sigma_{1,k}^2 + M_2\gamma^2\sigma_{2,k}^2$ with $M_1 = \frac{4B_1'}{3\rho_1}$ and $M_2 = \frac{4\left(B_2' + \frac{4}{3}G\right)}{3\rho_2}$, using the tower property (35) of mathematical expectation together with

$\gamma \leq \frac{1}{4(A'+C_1M_1+C_2M_2)}$, we conclude

$$
\begin{aligned}
\mathbf{E}\left[T^{k+1}\right] \quad &\overset{(47)}{\leq} \quad \left(1-\frac{\gamma\mu}{2}\right)\mathbf{E}\|\tilde{x}^k-x^*\|^2 - \gamma\left(1-2A'\gamma\right)\mathbf{E}\left[f(x^k)-f(x^*)\right] + M_1\gamma^2\mathbf{E}\left[\sigma_{1,k+1}^2\right] \\
&\qquad M_2\gamma^2\mathbf{E}\left[\sigma_{2,k+1}^2\right] + \gamma^2 B_1'\sigma_{1,k}^2 + \gamma^2 B_2'\sigma_{2,k}^2 + \gamma^2 D_1' + \gamma(2L+\mu)\mathbf{E}\|x^k-\tilde{x}^k\|^2 \\
&\overset{(16),(17)}{\leq} \quad \left(1-\frac{\gamma\mu}{2}\right)\mathbf{E}\|\tilde{x}^k-x^*\|^2 + \left(1+\frac{B_1'}{M_1}-\rho_1\right)M_1\gamma^2\mathbf{E}\left[\sigma_{1,k}^2\right] \\
&\qquad + \left(1+\frac{B_2'+M_1G\rho_1}{M_2}-\rho_2\right)M_2\gamma^2\mathbf{E}\left[\sigma_{2,k}^2\right] + \gamma^2(D_1'+M_1D_2) \\
&\qquad - \gamma\left(1-2(A'+C_1M_1+C_2M_2)\gamma\right)\mathbf{E}\left[f(x^k)-f(x^*)\right] + \gamma(2L+\mu)\mathbf{E}\|x^k-\tilde{x}^k\|^2 \\
&\leq \quad \left(1-\frac{\gamma\mu}{2}\right)\mathbf{E}\|\tilde{x}^k-x^*\|^2 + \left(1-\frac{\rho_1}{4}\right)M_1\gamma^2\mathbf{E}\left[\sigma_{1,k}^2\right] + \left(1-\frac{\rho_2}{4}\right)M_2\gamma^2\mathbf{E}\left[\sigma_{2,k}^2\right] \\
&\qquad - \frac{\gamma}{2}\mathbf{E}\left[f(x^k)-f(x^*)\right] + \gamma(2L+\mu)\mathbf{E}\|x^k-\tilde{x}^k\|^2 + \gamma^2(D_1'+M_1D_2).
\end{aligned}
$$

Since $L \geq \mu$, $\tilde{x}^k = x^k - e^k$ and $\eta \overset{\text{def}}{=} \min\{\frac{\gamma\mu}{2}, \frac{\rho_1}{4}, \frac{\rho_2}{4}\}$ the last inequality implies

$$
\frac{\gamma}{2}\mathbf{E}\left[f(x^k)-f(x^*)\right] \leq (1-\eta)\mathbf{E}T^k - \mathbf{E}T^{k+1} + \gamma^2(D_1'+M_1D_2) + 3L\gamma\mathbf{E}\|e^k\|^2,
$$

which concludes the proof. $\qquad\square$

### E.2 Proof of Theorem 3.1

*Proof.* Form Lemma E.1 we have

$$
\frac{\gamma}{2}\mathbf{E}\left[f(x^k)-f(x^*)\right] \leq (1-\eta)\mathbf{E}T^k - \mathbf{E}T^{k+1} + \gamma^2(D_1'+M_1D_2) + 3L\gamma\mathbf{E}\|e^k\|^2.
$$

Summing up these inequalities for $k=0,\ldots,K$ with weights $w_k = (1-\eta)^{-(k+1)}$ we get

$$
\begin{aligned}
\frac{1}{2}\sum_{k=0}^K w_k\mathbf{E}\left[f(x^k)-f(x^*)\right] \quad &\leq \quad \sum_{k=0}^K\left(\frac{w_k(1-\eta)}{\gamma}\mathbf{E}T^k - \frac{w_k}{\gamma}\mathbf{E}T^{k+1}\right) + \gamma(D_1'+M_1D_2)\sum_{k=0}^K w_k \\
&\qquad + 3L\sum_{k=0}^K w_k\mathbf{E}\|e^k\|^2 \\
&\overset{(19),(20)}{\leq} \quad \sum_{k=0}^K\left(\frac{w_{k-1}}{\gamma}\mathbf{E}T^k - \frac{w_k}{\gamma}\mathbf{E}T^{k+1}\right) + F_1\sigma_{1,0}^2 + F_2\sigma_{2,0}^2 \\
&\qquad + \gamma^2(D_1'+M_1D_2+D_3)W_K + \frac{1}{4}\sum_{k=0}^K w_k\mathbf{E}\left[f(x^k)-f(x^*)\right].
\end{aligned}
$$

Rearranging the terms and using $\bar{x}^K = \frac{1}{W_K}\sum_{k=0}^K w_k x^k$ together with Jensen's inequality we obtain

$$
\mathbf{E}\left[f(\bar{x}^K)-f(x^*)\right] \quad \leq \quad \frac{4(T^0+\gamma F_1\sigma_{1,0}^2+\gamma F_2\sigma_{2,0}^2)}{\gamma W_K} + 4\gamma\left(D_1'+M_1D_2+D_3\right).
$$

Finally, using the definition of the sequences $\{W_K\}_{K\geq 0}$ and $\{w_k\}_{k\geq 0}$ we derive that if $\mu >$, then $W_K \geq w_K \geq (1-\eta)^{-K}$ and we get (21). In the case when $\mu = 0$ we have $w_k = 1$ and $W_K = K$ which implies (22). $\qquad\square$

# F    SGD as a Special Case

In this section we want to show that our approach is general enough to cover many existing methods of SGD type. Consider the following situation:

$$v^k = \gamma g^k, \quad e^0 = 0. \tag{48}$$

It implies that $e^k = 0$ for all $k \geq 0$ and the updates rules (4)-(5) gives us a simple SGD:

$$x^{k+1} = x^k - \gamma g^k. \tag{49}$$

The following lemma formally shows that SGD under general enough assumptions satisfies Assumption 3.4.

**Lemma F.1.** *Let Assumptions 3.1 and 3.2 be satisfies and inequalities* (18), (15), (16) *and* (17) *hold. Then for the method* (49) *inequality* (19) *holds with $F_1 = F_2 = 0$ and $D_3 = 0$ for all $k \geq 0$.*

*Proof.* Since $e^k = 0$ and $f(x^k) \geq f(x^*)$ for all $k \geq 0$ we get

$$3L \sum_{k=0}^{K} w_k \mathbf{E}\|e^k\|^2 = 0 \leq \frac{1}{4} \sum_{k=0}^{K} w_k \mathbf{E}\left[f(x^k) - f(x^*)\right]$$

which concludes the proof. $\qquad\qquad\qquad\qquad\qquad\qquad\qquad\qquad\qquad\qquad\square$

It implies that all methods considered in [11] fit our framework. Moreover, using Theorem 3.1 we derive the following result.

**Theorem F.1.** *Let Assumptions 3.1 and 3.2 be satisfied, inequalities* (18), (15), (16), (17) *hold and $\gamma \leq {}^{1}/_{4(A' + C_1 M_1 + C_2 M_2)}$. Then for the method* (49) *for all $K \geq 0$ we have*

$$\mathbf{E}\left[f(\bar{x}^K) - f(x^*)\right] \leq \left(1 - \min\left\{\frac{\gamma\mu}{2}, \frac{\rho_1}{4}, \frac{\rho_2}{4}\right\}\right)^K \frac{4T^0}{\gamma} + 4\gamma\left(D'_1 + M_1 D_2\right),$$

*when $\mu > 0$ and*

$$\mathbf{E}\left[f(\bar{x}^K) - f(x^*)\right] \leq \frac{4T^0}{\gamma K} + 4\gamma\left(D'_1 + M_1 D_2\right)$$

*when $\mu = 0$, where $T^k \overset{def}{=} \|x^k - x^*\|^2 + M_1 \gamma^2 \sigma_{1,k}^2 + M_2 \gamma^2 \sigma_{2,k}^2$ and $M_1 = \frac{4B'_1}{3\rho_1}$, $M_2 = \frac{4\left(B'_2 + \frac{4}{3}G\right)}{3\rho_2}$.*

In particular, if $\sigma_{2,k}^2 \equiv 0$, then our assumption coincides with the key assumption from [11] and our theorem recovers the same rates as in [11] when $\mu > 0$. The case when $\mu = 0$ was not considered in [11], while in our analysis we get it for free.

# G  Distributed `SGD` with Compression and Error Compensation

In this section we consider the scenario when compression and error-feedback is applied in order to reduce the communication cost of the method, i.e., we consider `SGD` with error compensation and compression (`EC-SGD`) which has updates of the form (4)-(5) with

$$g^k = \frac{1}{n}\sum_{i=1}^{n} g_i^k$$

$$v^k = \frac{1}{n}\sum_{i=1}^{n} v_i^k, \quad v_i^k = C(e_i^k + \gamma g_i^k) \tag{50}$$

$$e^k = \frac{1}{n}\sum_{i=1}^{n} e_i^k, \quad e_i^{k+1} = e_i^k + \gamma g_i^k - v_i^k = e_i^k + \gamma g_i^k - C(e_i^k + \gamma g_i^k). \tag{51}$$

Moreover, we assume that $e_i^0 = 0$ for $i = 1, \ldots, n$.

**Lemma G.1.** *Let Assumptions 3.1 and 3.2 be satisfied, Assumption 3.3 holds and*[5]

$$\gamma \le \min\left\{ \frac{\delta}{4\mu}, \sqrt{\frac{\delta}{96L\left(\frac{2A}{\delta} + \widetilde{A} + \frac{2}{1-\rho_1}\left(\frac{C_1}{\rho_1} + \frac{2GC_2}{\rho_2(1-\rho_2)}\right)\left(\frac{2B_1}{\delta} + \widetilde{B}_1\right) + \frac{2C_2\left(\frac{2B_2}{\delta} + \widetilde{B}_2\right)}{\rho_2(1-\rho_2)}\right)}} \right\}, \tag{52}$$

*where $M_1 = \frac{4B_1'}{3\rho_1}$ and $M_2 = \frac{4\left(B_2' + \frac{4}{3}G\right)}{3\rho_2}$. Then `EC-SGD` satisfies Assumption 3.3, i.e., inequality* (19) *holds with the following parameters:*

$$F_1 = \frac{24L\gamma^2}{\delta\rho_1(1-\eta)}\left(\frac{2B_1}{\delta} + \widetilde{B}_1\right), \quad F_2 = \frac{24L\gamma^2}{\delta\rho_2(1-\eta)}\left(\frac{2G}{1-\rho_1}\left(\frac{2B_1}{\delta} + \tilde{B}_1\right) + \frac{2B_2}{\delta} + \widetilde{B}_2\right), \tag{53}$$

$$D_3 = \frac{6L\gamma}{\delta}\left(\frac{D_2}{\rho_1}\left(\frac{2B_1}{\delta} + \widetilde{B}_1\right) + \frac{2D_1}{\delta} + \widetilde{D}_1\right). \tag{54}$$

*Proof.* First of all, we derive an upper bound for the second moment of $e_i^{k+1}$:

$$\mathbf{E}\|e_i^{k+1}\|^2 \overset{(51),(35)}{=} \mathbf{E}\left[\mathbf{E}\left[\|e_i^k + \gamma g_i^k - C(e_i^k + \gamma g_i^k)\|^2 \mid e_i^k, g_i^k\right]\right]$$

$$\overset{(8)}{\le} (1-\delta)\mathbf{E}\|e_i^k + \gamma g_i^k\|^2$$

$$\overset{(35),(34)}{=} (1-\delta)\mathbf{E}\|e_i^k + \gamma\bar{g}_i^k\|^2 + (1-\delta)\gamma^2\mathbf{E}\|g_i^k - \bar{g}_i^k\|^2$$

$$\overset{(30)}{\le} (1-\delta)(1+\beta)\mathbf{E}\|e_i^k\|^2 + (1-\delta)\left(1+\frac{1}{\beta}\right)\gamma^2\mathbf{E}\|\bar{g}_i^k\|^2$$

$$\qquad + (1-\delta)\gamma^2\mathbf{E}\|g_i^k - \bar{g}_i^k\|^2.$$

Summing up these inequalities for $i = 1, \ldots, n$ we get

$$\frac{1}{n}\sum_{i=1}^{n}\mathbf{E}\|e_i^{k+1}\|^2 \le (1-\delta)(1+\beta)\frac{1}{n}\sum_{i=1}^{n}\mathbf{E}\|e_i^k\|^2$$

$$+ (1-\delta)\left(1+\frac{1}{\beta}\right)\gamma^2\frac{1}{n}\sum_{i=1}^{n}\mathbf{E}\|\bar{g}_i^k\|^2 + (1-\delta)\gamma^2\frac{1}{n}\sum_{i=1}^{n}\mathbf{E}\|g_i^k - \bar{g}_i^k\|^2 \tag{55}$$

Consider $\beta = \frac{\delta}{2(1-\delta)}$. For this choice of $\beta$ we have

$$(1-\delta)(1+\beta) = (1-\delta)\left(1+\frac{\delta}{2(1-\delta)}\right) = 1 - \frac{\delta}{2}$$

$$(1-\delta)\left(1+\frac{1}{\beta}\right) = (1-\delta)\left(1+\frac{2(1-\delta)}{\delta}\right) = \frac{(1-\delta)(2-\delta)}{\delta} \leq \frac{2(1-\delta)}{\delta}.$$

Using this we continue our derivations:

$$
\frac{1}{n}\sum_{i=1}^{n}\mathbf{E}\|e_i^{k+1}\|^2 \quad \leq \quad \left(1-\frac{\delta}{2}\right)\frac{1}{n}\sum_{i=1}^{n}\mathbf{E}\|e_i^k\|^2 + \frac{2\gamma^2(1-\delta)}{\delta}\frac{1}{n}\sum_{i=1}^{n}\mathbf{E}\|\bar{g}_i^k\|^2
$$

$$
+(1-\delta)\gamma^2\frac{1}{n}\sum_{i=1}^{n}\mathbf{E}\|g_i^k - \bar{g}_i^k\|^2
$$

$$
\overset{(13),(14)}{\leq} \quad \left(1-\frac{\delta}{2}\right)\frac{1}{n}\sum_{i=1}^{n}\mathbf{E}\|e_i^k\|^2 + 2\gamma^2(1-\delta)\left(\frac{2A}{\delta}+\widetilde{A}\right)\mathbf{E}\left[f(x^k)-f(x^*)\right]
$$

$$
+\gamma^2(1-\delta)\left(\frac{2B_1}{\delta}+\widetilde{B}_1\right)\mathbf{E}\sigma_{1,k}^2 + \gamma^2(1-\delta)\left(\frac{2B_2}{\delta}+\widetilde{B}_2\right)\mathbf{E}\sigma_{2,k}^2
$$

$$
+\gamma^2(1-\delta)\left(\frac{2D_1}{\delta}+\widetilde{D}_1\right). \tag{56}
$$

Unrolling the recurrence above we get

$$
\frac{1}{n}\sum_{i=1}^{n}\mathbf{E}\|e_i^{k+1}\|^2 \overset{(56)}{\leq} 2\gamma^2(1-\delta)\left(\frac{2A}{\delta}+\widetilde{A}\right)\sum_{l=0}^{k}\left(1-\frac{\delta}{2}\right)^{k-l}\mathbf{E}\left[f(x^l)-f(x^*)\right]
$$

$$
+\gamma^2(1-\delta)\left(\frac{2B_1}{\delta}+\widetilde{B}_1\right)\sum_{l=0}^{k}\left(1-\frac{\delta}{2}\right)^{k-l}\mathbf{E}\sigma_{1,l}^2
$$

$$
+\gamma^2(1-\delta)\left(\frac{2B_2}{\delta}+\widetilde{B}_2\right)\sum_{l=0}^{k}\left(1-\frac{\delta}{2}\right)^{k-l}\mathbf{E}\sigma_{2,l}^2
$$

$$
+\gamma^2(1-\delta)\left(\frac{2D_1}{\delta}+\widetilde{D}_1\right)\sum_{l=0}^{k}\left(1-\frac{\delta}{2}\right)^{k-l} \tag{57}
$$

which implies

$$
3L\sum_{k=0}^{K}w_k\mathbf{E}\|e^k\|^2 \overset{(51)}{=} 3L\sum_{k=0}^{K}w_k\mathbf{E}\left\|\frac{1}{n}\sum_{i=1}^{n}e_i^k\right\|^2 \overset{(31)}{\leq} 3L\sum_{k=0}^{K}w_k\frac{1}{n}\sum_{i=1}^{n}\mathbf{E}\left\|e_i^k\right\|^2
$$

$$
\overset{(57)}{\leq} \frac{6L\gamma^2(1-\delta)}{1-\frac{\delta}{2}}\left(\frac{2A}{\delta}+\widetilde{A}\right)\sum_{k=0}^{K}\sum_{l=0}^{k}w_k\left(1-\frac{\delta}{2}\right)^{k-l}\mathbf{E}\left[f(x^l)-f(x^*)\right]
$$

$$
+\frac{3L\gamma^2(1-\delta)}{1-\frac{\delta}{2}}\left(\frac{2B_1}{\delta}+\widetilde{B}_1\right)\sum_{k=0}^{K}\sum_{l=0}^{k}w_k\left(1-\frac{\delta}{2}\right)^{k-l}\mathbf{E}\sigma_{1,l}^2
$$

$$
+\frac{3L\gamma^2(1-\delta)}{1-\frac{\delta}{2}}\left(\frac{2B_2}{\delta}+\widetilde{B}_2\right)\sum_{k=0}^{K}\sum_{l=0}^{k}w_k\left(1-\frac{\delta}{2}\right)^{k-l}\mathbf{E}\sigma_{2,l}^2
$$

$$
+\frac{3L\gamma^2(1-\delta)}{1-\frac{\delta}{2}}\left(\frac{2D_1}{\delta}+\widetilde{D}_1\right)\sum_{k=0}^{K}\sum_{l=0}^{k}w_k\left(1-\frac{\delta}{2}\right)^{k-l}. \tag{58}
$$

In the remaining part of the proof we derive upper bounds for three terms in the right-hand side of the previous inequality. First of all, recall that $w_k = (1-\eta)^{-(k+1)}$ and

$\eta = \min\left\{\frac{\gamma\mu}{2}, \frac{\rho_1}{4}, \frac{\rho_2}{4}\right\}$. It implies that for all $0 \le i < k$ we have

$$
\begin{aligned}
w_k &= (1-\eta)^{-(k-j+1)}(1-\eta)^{-j} \overset{(32)}{\le} w_{k-j}(1+2\eta)^j \\
&\le w_{k-j}(1+\gamma\mu)^j \overset{(52)}{\le} w_{k-j}\left(1+\frac{\delta}{4}\right)^j,
\end{aligned}
\tag{59}
$$

$$
\begin{aligned}
w_k &= (1-\eta)^{-(k-j+1)}(1-\eta)^{-j} \overset{(32)}{\le} w_{k-j}(1+2\eta)^j \\
&\le w_{k-j}\left(1+\frac{\min\{\rho_1,\rho_2\}}{2}\right)^j.
\end{aligned}
\tag{60}
$$

For simplicity, we introduce new notation: $r_k \overset{\text{def}}{=} \mathbf{E}\left[f(x^k) - f(x^*)\right]$. Using this we get

$$
\begin{aligned}
\sum_{k=0}^{K}\sum_{l=0}^{k} w_k\left(1-\frac{\delta}{2}\right)^{k-l} r_l &\overset{(59)}{\le} \sum_{k=0}^{K}\sum_{l=0}^{k} w_l r_l\left(1+\frac{\delta}{4}\right)^{k-l}\left(1-\frac{\delta}{2}\right)^{k-l} \\
&\overset{(33)}{\le} \sum_{k=0}^{K}\sum_{l=0}^{k} w_l r_l\left(1-\frac{\delta}{4}\right)^{k-l} \\
&\le \left(\sum_{k=0}^{K} w_k r_k\right)\left(\sum_{k=0}^{\infty}\left(1-\frac{\delta}{4}\right)^k\right) = \frac{4}{\delta}\sum_{k=0}^{K} w_k r_k.
\end{aligned}
\tag{61}
$$

Next, we apply our assumption on $\sigma_{2,k}^2$ and derive that

$$
\begin{aligned}
\mathbf{E}\sigma_{2,k+1}^2 &\overset{(17)}{\le} (1-\rho_2)\mathbf{E}\sigma_{2,k}^2 + 2C_2 \underbrace{\mathbf{E}\left[f(x^k)-f(x^*)\right]}_{r_k} \\
&\le (1-\rho_2)^{k+1}\sigma_{2,0}^2 + 2C_2\sum_{l=0}^{k}(1-\rho_2)^{k-l} r_l,
\end{aligned}
\tag{62}
$$

hence

$$
\begin{aligned}
\sum_{k=0}^{K}\sum_{l=0}^{k} w_k\left(1-\frac{\delta}{2}\right)^{k-l}\mathbf{E}\sigma_{2,l}^2 &\le \sum_{k=0}^{K}\sum_{l=0}^{k} w_k\left(1-\frac{\delta}{2}\right)^{k-l}(1-\rho_2)^l\sigma_{2,0}^2 \\
&\quad + \frac{2C_2}{1-\rho_2}\sum_{k=0}^{K}\sum_{l=0}^{k}\sum_{t=0}^{l} w_k\left(1-\frac{\delta}{2}\right)^{k-l}(1-\rho_2)^{l-t} r_t.
\end{aligned}
$$

Using this and

$$
\begin{aligned}
w_k\left(1-\frac{\delta}{2}\right)^{k-l}(1-\rho_2)^{l-t} &\overset{(59)}{\le} w_l\left(1+\frac{\delta}{4}\right)^{k-l}\left(1-\frac{\delta}{2}\right)^{k-l}(1-\rho_2)^{l-t} \\
&\overset{(33),(60)}{\le} \left(1-\frac{\delta}{4}\right)^{k-l}\left(1+\frac{\rho_2}{2}\right)^{l-t}(1-\rho_2)^{l-t} w_t \\
&\overset{(33)}{\le} \left(1-\frac{\delta}{4}\right)^{k-l}\left(1-\frac{\rho_2}{2}\right)^{l-t} w_t
\end{aligned}
$$

we derive

$$\sum_{k=0}^{K}\sum_{l=0}^{k} w_k \left(1-\frac{\delta}{2}\right)^{k-l} \mathbf{E}\sigma_{2,l}^2 \;\leq\; \sum_{k=0}^{K}\sum_{l=0}^{k} w_k \left(1-\frac{\delta}{4}\right)^{k-l}\left(1-\frac{\rho_2}{2}\right)^{l} w_0\sigma_{2,0}^2$$

$$+\frac{2C_2}{1-\rho_2}\sum_{k=0}^{K}\sum_{l=0}^{k}\sum_{t=0}^{l}\left(1-\frac{\delta}{4}\right)^{k-l}\left(1-\frac{\rho_2}{2}\right)^{l-t}w_t r_t$$

$$\leq\; w_0\sigma_{2,0}^2\left(\sum_{k=0}^{\infty}\left(1-\frac{\delta}{4}\right)^{k}\right)\left(\sum_{k=0}^{\infty}\left(1-\frac{\rho_2}{2}\right)^{k}\right)$$

$$\frac{2C_2}{1-\rho_2}\left(\sum_{k=0}^{K} w_k r_k\right)\left(\sum_{k=0}^{\infty}\left(1-\frac{\delta}{4}\right)^{k}\right)\left(\sum_{k=0}^{\infty}\left(1-\frac{\rho_2}{2}\right)^{k}\right)$$

$$=\; \frac{8\sigma_{2,0}^2}{\delta\rho_2(1-\eta)}+\frac{16C_2}{\delta\rho_2(1-\rho_2)}\sum_{k=0}^{K} w_k r_k. \tag{63}$$

Similarly, we estimate $\sigma_{1,k}^2$:

$$\mathbf{E}\sigma_{1,k+1}^2 \;\overset{(16)}{\leq}\; (1-\rho_1)\mathbf{E}\sigma_{1,k}^2 + 2C_1\underbrace{\mathbf{E}\left[f(x^k)-f(x^*)\right]}_{r_k}+G\rho_1\mathbf{E}\sigma_{2,k}^2 + D_2$$

$$\leq\; (1-\rho_1)^{k+1}\sigma_{1,0}^2 + 2C_1\sum_{l=0}^{k}(1-\rho_1)^{k-l}r_l + G\rho_1\sum_{l=0}^{k}(1-\rho_1)^{k-l}\mathbf{E}\sigma_{2,k}^2$$

$$+D_2\sum_{l=0}^{k}(1-\rho_1)^{l}$$

$$\leq\; (1-\rho_1)^{k+1}\sigma_{1,0}^2 + 2C_1\sum_{l=0}^{k}(1-\rho_1)^{k-l}r_l + G\rho_1\sum_{l=0}^{k}(1-\rho_1)^{k-l}\mathbf{E}\sigma_{2,k}^2$$

$$+D_2\sum_{l=0}^{\infty}(1-\rho_1)^{l}$$

$$=\; (1-\rho_1)^{k+1}\sigma_{1,0}^2 + 2C_1\sum_{l=0}^{k}(1-\rho_1)^{k-l}r_l + G\rho_1\sum_{l=0}^{k}(1-\rho_1)^{k-l}\mathbf{E}\sigma_{2,k}^2$$

$$+\frac{D_2}{\rho_1}. \tag{64}$$

Using this we get

$$\sum_{k=0}^{K}\sum_{l=0}^{k} w_k \left(1-\frac{\delta}{2}\right)^{k-l}\mathbf{E}\sigma_{1,l}^2 \;\leq\; \sigma_{1,0}^2\sum_{k=0}^{K}\sum_{l=0}^{k} w_k \left(1-\frac{\delta}{2}\right)^{k-l}(1-\rho_1)^{l}$$

$$+\frac{2C_1}{1-\rho_1}\sum_{k=0}^{K}\sum_{l=0}^{k}\sum_{t=0}^{l} w_k \left(1-\frac{\delta}{2}\right)^{k-l}(1-\rho_1)^{l-t}r_t$$

$$+\frac{G\rho_1}{1-\rho_1}\sum_{k=0}^{K}\sum_{l=0}^{k}\sum_{t=0}^{l} w_k \left(1-\frac{\delta}{2}\right)^{k-l}(1-\rho_1)^{l-t}\mathbf{E}\sigma_{2,t}^2$$

$$+\frac{D_2}{\rho_1}\sum_{k=0}^{K}\sum_{l=0}^{k}\sum_{t=0}^{l} w_k \left(1-\frac{\delta}{2}\right)^{k-l}(1-\rho_1)^{l-t}. \tag{65}$$

Moreover,

$$
\begin{aligned}
w_k \left(1 - \frac{\delta}{2}\right)^{k-l} (1 - \rho_1)^{l-t} \quad &\overset{(59)}{\leq} \quad w_l \left(1 + \frac{\delta}{4}\right)^{k-l} \left(1 - \frac{\delta}{2}\right)^{k-l} (1 - \rho_1)^{l-t} \\
&\overset{(33),(60)}{\leq} \quad \left(1 - \frac{\delta}{4}\right)^{k-l} \left(1 + \frac{\rho_1}{2}\right)^{l-t} (1 - \rho_1)^{l-t} w_t \\
&\overset{(33)}{\leq} \quad \left(1 - \frac{\delta}{4}\right)^{k-l} \left(1 - \frac{\rho_1}{2}\right)^{l-t} w_t,
\end{aligned}
$$

hence

$$
\begin{aligned}
\sum_{k=0}^{K} \sum_{l=0}^{k} w_k \left(1 - \frac{\delta}{2}\right)^{k-l} \mathbf{E}\sigma_{1,l}^2 \quad &\overset{(65)}{\leq} \quad w_0 \sigma_{1,0}^2 \sum_{k=0}^{K} \sum_{l=0}^{k} \left(1 - \frac{\delta}{4}\right)^{k-l} \left(1 - \frac{\rho_1}{2}\right)^{l} \\
&\quad + \frac{2C_1}{1 - \rho_1} \sum_{k=0}^{K} \sum_{l=0}^{k} \sum_{t=0}^{l} \left(1 - \frac{\delta}{4}\right)^{k-l} \left(1 - \frac{\rho_1}{2}\right)^{l-t} w_t r_t \\
&\quad + \frac{G\rho_1}{1 - \rho_1} \sum_{k=0}^{K} \sum_{l=0}^{k} \sum_{t=0}^{l} \left(1 - \frac{\delta}{4}\right)^{k-l} \left(1 - \frac{\rho_1}{2}\right)^{l-t} w_t \mathbf{E}\sigma_{2,t}^2 \\
&\quad + \frac{D_2}{\rho_1} \left(\sum_{k=0}^{K} w_k\right) \left(\sum_{k=0}^{\infty} \left(1 - \frac{\delta}{2}\right)^{k}\right) \left(\sum_{k=0}^{\infty} (1 - \rho_1)^{k}\right) \\
&\leq \quad w_0 \sigma_{1,0}^2 \left(\sum_{k=0}^{\infty} \left(1 - \frac{\delta}{4}\right)^{k}\right) \left(\sum_{k=0}^{\infty} \left(1 - \frac{\rho_1}{2}\right)^{k}\right) \\
&\quad + \frac{2C_1}{1 - \rho_1} \left(\sum_{k=0}^{K} w_k r_k\right) \left(\sum_{k=0}^{\infty} \left(1 - \frac{\delta}{4}\right)^{k}\right) \left(\sum_{k=0}^{\infty} \left(1 - \frac{\rho_1}{2}\right)^{k}\right) \\
&\quad + \frac{G\rho_1}{1 - \rho_1} \left(\sum_{k=0}^{K} w_k \mathbf{E}\sigma_{2,k}^2\right) \left(\sum_{k=0}^{\infty} \left(1 - \frac{\delta}{4}\right)^{k}\right) \left(\sum_{k=0}^{\infty} \left(1 - \frac{\rho_1}{2}\right)^{k}\right) \\
&\quad + \frac{2D_2}{\delta\rho_1} W_K \\
&= \quad \frac{8\sigma_{1,0}^2}{\delta\rho_1(1 - \eta)} + \frac{16C_1}{\delta\rho_1(1 - \rho_1)} \sum_{k=0}^{K} w_k r_k + \frac{8G}{\delta(1 - \rho_1)} \sum_{k=0}^{K} w_k \mathbf{E}\sigma_{2,k}^2 \\
&\quad + \frac{2D_2}{\delta\rho_1} W_K. \tag{66}
\end{aligned}
$$

For the third term in the right-hand side of previous inequality we have

$$\frac{8G}{\delta(1-\rho_1)}\sum_{k=0}^{K}w_k\mathbf{E}\sigma_{2,k}^2 \overset{(62)}{\leq} \frac{8G\sigma_{2,0}^2}{\delta(1-\rho_1)}\sum_{k=0}^{K}w_k(1-\rho_2)^k$$

$$+\frac{16GC_2}{\delta(1-\rho_1)(1-\rho_2)}\sum_{k=0}^{K}\sum_{l=0}^{k}w_k(1-\rho_2)^{k-l}r_l$$

$$\overset{(60)}{\leq}\frac{8G\sigma_{2,0}^2 w_0}{\delta(1-\rho_1)}\sum_{k=0}^{K}\left(1+\frac{\rho_2}{2}\right)^k(1-\rho_2)^k$$

$$+\frac{16GC_2}{\delta(1-\rho_1)(1-\rho_2)}\sum_{k=0}^{K}\sum_{l=0}^{k}\left(1+\frac{\rho_2}{2}\right)^{k-l}(1-\rho_2)^{k-l}w_l r_l$$

$$\overset{(33)}{\leq}\frac{8G\sigma_{2,0}^2 w_0}{\delta(1-\rho_1)}\sum_{k=0}^{\infty}\left(1-\frac{\rho_2}{2}\right)^k$$

$$+\frac{16GC_2}{\delta(1-\rho_1)(1-\rho_2)}\sum_{k=0}^{K}\sum_{l=0}^{k}\left(1-\frac{\rho_2}{2}\right)^{k-l}w_l r_l$$

$$\leq\frac{16G\sigma_{2,0}^2 w_0}{\delta\rho_2(1-\rho_1)}+\frac{16GC_2}{\delta(1-\rho_1)(1-\rho_2)}\left(\sum_{k=0}^{K}w_k r_k\right)\left(\sum_{k=0}^{\infty}\left(1-\frac{\rho_2}{2}\right)^k\right)$$

$$=\frac{16G\sigma_{2,0}^2}{\delta\rho_2(1-\rho_1)(1-\eta)}+\frac{32GC_2}{\delta\rho_2(1-\rho_1)(1-\rho_2)}\sum_{k=0}^{K}w_k r_k \qquad (67)$$

Combining inequalities (66) and (67) we get

$$\sum_{k=0}^{K}\sum_{l=0}^{k}w_k\left(1-\frac{\delta}{2}\right)^{k-l}\mathbf{E}\sigma_{1,l}^2 \leq \frac{8\sigma_{1,0}^2}{\delta\rho_1(1-\eta)}+\frac{16}{\delta(1-\rho_1)}\left(\frac{C_1}{\rho_1}+\frac{2GC_2}{\rho_2(1-\rho_2)}\right)\sum_{k=0}^{K}w_k r_k$$

$$+\frac{16G\sigma_{2,0}^2}{\delta\rho_2(1-\rho_1)(1-\eta)}+\frac{2D_2}{\delta\rho_1}W_K \qquad (68)$$

Finally, we estimate the last term in the right-hand side of (58):

$$\sum_{k=0}^{K}\sum_{l=0}^{k}w_k\left(1-\frac{\delta}{2}\right)^{k-l} \leq \left(\sum_{k=0}^{K}w_k\right)\left(\sum_{k=0}^{\infty}\left(1-\frac{\delta}{2}\right)^k\right)=\frac{2}{\delta}W_K. \qquad (69)$$

Plugging inequalities (61), (63), (68), (69) and $\frac{1-\delta}{1-\frac{\delta}{2}}\leq 1$ in (58) we obtain

$$3L\sum_{k=0}^{K}w_k\mathbf{E}\|e^k\|^2 \leq \frac{24L\left(\frac{2A}{\delta}+\widetilde{A}+\frac{2}{1-\rho_1}\left(\frac{C_1}{\rho_1}+\frac{2GC_2}{\rho_2(1-\rho_2)}\right)\left(\frac{2B_1}{\delta}+\widetilde{B}_1\right)+\frac{2C_2\left(\frac{2B_2}{\delta}+\widetilde{B}_2\right)}{\rho_2(1-\rho_2)}\right)\gamma^2}{\delta}\sum_{k=0}^{K}w_k r_k$$

$$+\frac{24L\gamma^2}{\delta\rho_1(1-\eta)}\left(\frac{2B_1}{\delta}+\widetilde{B}_1\right)\sigma_{1,0}^2$$

$$+\frac{24L\gamma^2}{\delta\rho_2(1-\eta)}\left(\frac{2G}{1-\rho_1}\left(\frac{2B_1}{\delta}+\tilde{B}_1\right)+\frac{2B_2}{\delta}+\widetilde{B}_2\right)\sigma_{2,0}^2$$

$$+\frac{6L\gamma^2}{\delta}\left(\frac{D_2}{\rho_1}\left(\frac{2B_1}{\delta}+\widetilde{B}_1\right)+\frac{2D_1}{\delta}+\widetilde{D}_1\right)W_K.$$

Taking into account that $\gamma \leq \sqrt{\dfrac{\delta}{96L\left(\frac{2A}{\delta}+\widetilde{A}+\frac{2}{1-\rho_1}\left(\frac{C_1}{\rho_1}+\frac{2GC_2}{\rho_2(1-\rho_2)}\right)\left(\frac{2B_1}{\delta}+\widetilde{B}_1\right)+\frac{2C_2\left(\frac{2B_2}{\delta}+\widetilde{B}_2\right)}{\rho_2(1-\rho_2)}\right)}}$,

$F_1 = \frac{24L\gamma^2}{\delta\rho_1(1-\eta)}\left(\frac{2B_1}{\delta}+\widetilde{B}_1\right)$, $F_2 = \frac{24L\gamma^2}{\delta\rho_2(1-\eta)}\left(\frac{2G}{1-\rho_1}\left(\frac{2B_1}{\delta}+\tilde{B}_1\right)+\frac{2B_2}{\delta}+\widetilde{B}_2\right)$ and $D_3 =$

$\frac{6L\gamma}{\delta}\left(\frac{D_2}{\rho_1}\left(\frac{2B_1}{\delta}+\widetilde{B}_1\right)+\frac{2D_1}{\delta}+\widetilde{D}_1\right)$ we get

$$3L\sum_{k=0}^{K}w_k\mathbf{E}\|e^k\|^2 \leq \frac{1}{4}\sum_{k=0}^{K}w_k r_k + F_1\sigma_{1,0}^2 + F_2\sigma_{2,0}^2 + \gamma D_3.$$

$\square$

As a direct application of Lemma G.1 and Theorem 3.1 we get the following result.

**Theorem G.1.** *Let Assumptions 3.1 and 3.2 be satisfied, Assumption 3.3 holds and*

$$\gamma \leq \frac{1}{4(A'+C_1 M_1 + C_2 M_2)},$$

$$\gamma \leq \min\left\{\frac{\delta}{4\mu},\sqrt{\frac{\delta}{96L\left(\frac{2A}{\delta}+\widetilde{A}+\frac{2}{1-\rho_1}\left(\frac{C_1}{\rho_1}+\frac{2GC_2}{\rho_2(1-\rho_2)}\right)\left(\frac{2B_1}{\delta}+\widetilde{B}_1\right)+\frac{2C_2\left(\frac{2B_2}{\delta}+\widetilde{B}_2\right)}{\rho_2(1-\rho_2)}\right)}}\right\},$$

*where $M_1 = \frac{4B_1'}{3\rho_1}$ and $M_2 = \frac{4\left(B_2'+\frac{4}{3}G\right)}{3\rho_2}$. Then for all $K\geq 0$ we have*

$$\mathbf{E}\left[f(\bar{x}^K)-f(x^*)\right] \leq (1-\eta)^K \frac{4(T^0+\gamma F_1\sigma_{1,0}^2+\gamma F_2\sigma_{2,0}^2)}{\gamma}+4\gamma\left(D_1'+M_1 D_2 + D_3\right),$$

*when $\mu>0$ and*

$$\mathbf{E}\left[f(\bar{x}^K)-f(x^*)\right] \leq \frac{4(T^0+\gamma F_1\sigma_{1,0}^2+\gamma F_2\sigma_{2,0}^2)}{\gamma K}+4\gamma\left(D_1'+M_1 D_2 + D_3\right)$$

*when $\mu=0$, where $\eta = \min\{\gamma\mu/2, \rho_1/4, \rho_2/4\}$, $T^k \overset{def}{=} \|\tilde{x}^k - x^*\|^2 + M_1\gamma^2\sigma_{1,k}^2 + M_2\gamma^2\sigma_{2,k}^2$ and*

$$F_1 = \frac{24L\gamma^2}{\delta\rho_1(1-\eta)}\left(\frac{2B_1}{\delta}+\widetilde{B}_1\right),\quad F_2 = \frac{24L\gamma^2}{\delta\rho_2(1-\eta)}\left(\frac{2G}{1-\rho_1}\left(\frac{2B_1}{\delta}+\tilde{B}_1\right)+\frac{2B_2}{\delta}+\widetilde{B}_2\right),$$

$$D_3 = \frac{6L\gamma}{\delta}\left(\frac{D_2}{\rho_1}\left(\frac{2B_1}{\delta}+\widetilde{B}_1\right)+\frac{2D_1}{\delta}+\widetilde{D}_1\right).$$

## H   SGD with Delayed Updates

In this section we consider the SGD with delayed updates (D-SGD) [1, 33, 10, 3, 46]. This method has updates of the form (4)-(5) with

$$
\begin{aligned}
g^k &= \frac{1}{n}\sum_{i=1}^{n} g_i^k \\
v^k &= \frac{1}{n}\sum_{i=1}^{n} v_i^k, \quad v_i^k = \begin{cases} \gamma g_i^{k-\tau}, & \text{if } t \geq \tau, \\ 0, & \text{if } t < \tau \end{cases}
\end{aligned}
\tag{70}
$$

$$
e^k = \frac{1}{n}\sum_{i=1}^{n} e_i^k, \quad e_i^{k+1} = e_i^k + \gamma g_i^k - v_i^k = \gamma \sum_{t=1}^{\tau} g_i^{k+1-t},
\tag{71}
$$

where the summation is performed only for non-negative indices. Moreover, we assume that $e_i^0 = 0$ for $i = 1, \ldots, n$.

For convenience we also introduce new constant:

$$
\hat{A} = A' + L\tau.
\tag{72}
$$

**Lemma H.1.** *Let Assumptions 3.1 and 3.2 be satisfied, inequalities* (15), (16) *and* (17) *hold and*[6]

$$
\gamma \leq \min\left\{ \frac{1}{2\tau\mu}, \frac{1}{8\sqrt{L\tau\left(\hat{A} + \frac{2B_1'C_1}{\rho_1(1-\rho_1)} + \frac{2B_2'C_2}{\rho_2(1-\rho_2)} + \frac{4B_1'GC_2}{\rho_2(1-\rho_1)(1-\rho_2)}\right)}} \right\},
\tag{73}
$$

*where $M_1 = \frac{4B_1'}{3\rho_1}$ and $M_2 = \frac{4\left(B_2' + \frac{4}{3}G\right)}{3\rho_2}$. Then* D-SGD *satisfies Assumption 3.4, i.e., inequality* (19) *holds with the following parameters:*

$$
F_1 = \frac{6\gamma^2 LB_1'\tau(2+\rho_1)}{\rho_1}, \quad F_2 = \frac{6\gamma^2\tau L(2+\rho_2)}{\rho_2}\left(\frac{2B_1'G}{1-\rho_1} + B_2'\right),
\tag{74}
$$

$$
D_3 = 3\gamma\tau L\left(D_1' + \frac{2B_1'D_2}{\rho_1}\right).
\tag{75}
$$

*Proof.* First of all, we derive an upper bound for the second moment of $e_i^k$:

$$
\begin{aligned}
\mathbf{E}\|e^k\|^2 &\overset{(71)}{=} \gamma^2 \mathbf{E}\left[\left\|\sum_{t=1}^{\tau} g^{k-t}\right\|^2\right] \\
&\overset{(36)}{\leq} \gamma^2\tau\sum_{t=1}^{\tau}\mathbf{E}\left[\left\|\nabla f(x^{k-t})\right\|^2\right] + \gamma^2\sum_{t=1}^{\tau}\mathbf{E}\left[\left\|g^{k-t} - \nabla f(x^{k-t})\right\|^2\right] \\
&\overset{(34)}{\leq} \gamma^2\tau\sum_{t=1}^{\tau}\mathbf{E}\left[\left\|\nabla f(x^{k-t})\right\|^2\right] + \gamma^2\sum_{t=1}^{\tau}\mathbf{E}\left[\left\|g^{k-t}\right\|^2\right] \\
&\overset{(15),(11)}{\leq} 2\gamma^2\underbrace{(A'+L\tau)}_{\hat{A}}\sum_{t=1}^{\tau}\mathbf{E}\left[f(x^{k-t}) - f(x^*)\right] + \gamma^2 B_1'\sum_{t=1}^{\tau}\mathbf{E}\sigma_{1,k-t}^2 \\
&\quad + \gamma^2 B_2'\sum_{t=1}^{\tau}\mathbf{E}\sigma_{2,k-t}^2 + \gamma^2\tau D_1'
\end{aligned}
\tag{76}
$$

which implies

$$3L \sum_{k=0}^{K} w_k \mathbf{E} \|e^k\|^2 \overset{(76)}{\leq} 6\gamma^2 L \hat{A} \sum_{k=0}^{K} \sum_{t=1}^{\tau} w_k \mathbf{E} \left[ f(x^{k-t}) - f(x^*) \right]$$

$$+ 3\gamma^2 L B_1' \sum_{k=0}^{K} \sum_{t=1}^{\tau} w_k \mathbf{E} \sigma_{1,k-t}^2$$

$$+ 3\gamma^2 L B_2' \sum_{k=0}^{K} \sum_{t=1}^{\tau} w_k \mathbf{E} \sigma_{2,k-t}^2 + 3\gamma^2 \tau L D_1' W_K \qquad (77)$$

In the remaining part of the proof we derive upper bounds for four terms in the right-hand side of the previous inequality. First of all, recall that $w_k = (1 - \eta)^{-(k+1)}$ and $\eta = \min \left\{ \frac{\gamma \mu}{2}, \frac{\rho_1}{4}, \frac{\rho_2}{4} \right\}$. It implies that for all $0 \leq i < k$ and $0 \leq t \leq \tau$ we have

$$w_k = (1 - \eta)^{-(k-t+1)} (1 - \eta)^{-t} \overset{(32)}{\leq} w_{k-t} (1 + 2\eta)^t$$

$$\leq w_{k-t} (1 + \gamma\mu)^t \overset{(73)}{\leq} w_{k-t} \left( 1 + \frac{1}{2\tau} \right)^t \leq w_{k-t} \exp \left( \frac{t}{2\tau} \right) \leq 2 w_{k-t}, \qquad (78)$$

$$w_k = (1 - \eta)^{-(k-j+1)} (1 - \eta)^{-j} \overset{(32)}{\leq} w_{k-j} (1 + 2\eta)^j \leq w_{k-j} \left( 1 + \frac{\min\{\rho_1, \rho_2\}}{2} \right)^j . (79)$$

For simplicity, we introduce new notation: $r_k \overset{\text{def}}{=} \mathbf{E} \left[ f(x^k) - f(x^*) \right]$. Using this we get

$$\sum_{k=0}^{K} \sum_{t=1}^{\tau} w_k r_{k-t} \overset{(78)}{\leq} \sum_{k=0}^{K} \sum_{t=1}^{\tau} 2 w_{k-t} r_{k-t} \leq 2\tau \sum_{k=0}^{K} w_k r_k \qquad (80)$$

Similarly, we estimate the second term in the right-hand side of (79):

$$\sum_{k=0}^{K} \sum_{t=1}^{\tau} w_k \mathbf{E} \sigma_{1,k-t}^2 \leq \sum_{k=0}^{K} \sum_{t=1}^{\tau} 2 w_{k-t} \mathbf{E} \sigma_{1,k-t}^2 \leq 2\tau \sum_{k=0}^{K} w_k \mathbf{E} \sigma_{1,k}^2$$

$$\overset{(64)}{\leq} 2\tau \sigma_{1,0}^2 \sum_{k=0}^{K} w_k (1 - \rho_1)^k + \frac{4 C_1 \tau}{1 - \rho_1} \sum_{k=0}^{K} \sum_{l=0}^{k} w_k (1 - \rho_1)^{k-l} r_l$$

$$+ \frac{2 G \rho_1 \tau}{1 - \rho_1} \sum_{k=0}^{K} \sum_{l=0}^{k} w_k (1 - \rho_1)^{k-l} \mathbf{E} \sigma_{2,l}^2 + \frac{2\tau D_2}{\rho} W_K. \qquad (81)$$

For the first term in the right-hand side of previous inequality we have

$$2\tau \sigma_{1,0}^2 \sum_{k=0}^{K} w_k (1 - \rho_1)^k \overset{(79)}{\leq} 2\tau \sigma_{1,0}^2 \sum_{k=0}^{K} \left( 1 + \frac{\rho_1}{2} \right)^{k+1} (1 - \rho_1)^k$$

$$\overset{(33)}{\leq} 2\tau \left( 1 + \frac{\rho_1}{2} \right) \sigma_{1,0}^2 \sum_{k=0}^{K} \left( 1 - \frac{\rho_1}{2} \right)^k$$

$$\leq \tau (2 + \rho_1) \sigma_{1,0}^2 \sum_{k=0}^{\infty} \left( 1 - \frac{\rho_1}{2} \right)^k \leq \frac{2\tau (2 + \rho_1) \sigma_{1,0}^2}{\rho_1}. \qquad (82)$$

The second term in the right-hand side of (81) can be upper bounded in the following way:

$$
\frac{4C_1\tau}{1-\rho_1}\sum_{k=0}^{K}\sum_{l=0}^{k}w_k(1-\rho_1)^{k-l}r_l \overset{(79)}{\leq} \frac{4C_1\tau}{1-\rho_1}\sum_{k=0}^{K}\sum_{l=0}^{k}w_lr_l\left(1+\frac{\rho_1}{2}\right)^{k-l}(1-\rho_1)^{k-l}
$$

$$
\overset{(33)}{\leq} \frac{4C_1\tau}{1-\rho_1}\sum_{k=0}^{K}\sum_{l=0}^{k}w_lr_l\left(1-\frac{\rho_1}{2}\right)^{k-l}
$$

$$
\leq \frac{4C_1\tau}{1-\rho_1}\left(\sum_{k=0}^{K}w_kr_k\right)\left(\sum_{k=0}^{\infty}\left(1-\frac{\rho_1}{2}\right)^{k}\right)
$$

$$
\leq \frac{8C_1\tau}{\rho_1(1-\rho_1)}\sum_{k=0}^{K}w_kr_k. \tag{83}
$$

Repeating similar steps we estimate the third term in the right-hand side of (81):

$$
\frac{2G\rho_1\tau}{1-\rho_1}\sum_{k=0}^{K}\sum_{l=0}^{k}w_k(1-\rho_1)^{k-l}\mathbf{E}\sigma_{2,l}^2 \leq \frac{4G\tau}{1-\rho_1}\sum_{k=0}^{K}w_k\mathbf{E}\sigma_{2,k}^2
$$

$$
\overset{(62)}{\leq} \frac{4G\tau\sigma_{2,0}^2}{1-\rho_1}\sum_{k=0}^{K}w_k(1-\rho_2)^k
$$

$$
+\frac{8GC_2}{(1-\rho_1)(1-\rho_2)}\sum_{k=0}^{K}\sum_{l=0}^{k}w_k(1-\rho_2)^{k-l}r_l
$$

$$
\overset{(79)}{\leq} \frac{4G\tau\sigma_{2,0}^2}{1-\rho_1}\sum_{k=0}^{K}\left(1+\frac{\rho_2}{2}\right)^{k+1}(1-\rho_2)^k
$$

$$
+\frac{8GC_2\tau}{(1-\rho_1)(1-\rho_2)}\sum_{k=0}^{K}\sum_{l=0}^{k}\left(1+\frac{\rho_2}{2}\right)^{k-l}(1-\rho_2)^{k-l}w_lr_l
$$

$$
\overset{(33)}{\leq} \frac{2G\tau(2+\rho_2)\sigma_{2,0}^2}{1-\rho_1}\sum_{k=0}^{\infty}\left(1-\frac{\rho_2}{2}\right)^k
$$

$$
+\frac{8GC_2\tau}{(1-\rho_1)(1-\rho_2)}\sum_{k=0}^{K}\sum_{l=0}^{k}\left(1-\frac{\rho_2}{2}\right)^{k-l}w_lr_l
$$

$$
\leq \frac{4G\tau(2+\rho_2)\sigma_{2,0}^2}{\rho_2(1-\rho_1)}
$$

$$
+\frac{8GC_2\tau}{(1-\rho_1)(1-\rho_2)}\left(\sum_{k=0}^{K}w_kr_k\right)\left(\sum_{k=0}^{\infty}\left(1-\frac{\rho_2}{2}\right)^k\right)
$$

$$
= \frac{4G\tau(2+\rho_2)\sigma_{2,0}^2}{\rho_2(1-\rho_1)}
$$

$$
+\frac{16GC_2\tau}{\rho_2(1-\rho_1)(1-\rho_2)}\sum_{k=0}^{K}w_kr_k \tag{84}
$$

Combining inequalities (81), (82), (83) and (84) we get

$$
\sum_{k=0}^{K}\sum_{t=1}^{\tau}w_k\mathbf{E}\sigma_{1,k-t}^2 \leq \frac{2\tau(2+\rho_1)\sigma_{1,0}^2}{\rho_1}+\frac{8\tau}{1-\rho_1}\left(\frac{C_1}{\rho_1}+\frac{2GC_2}{\rho_2(1-\rho_2)}\right)\sum_{k=0}^{K}w_kr_k
$$

$$
+\frac{4G\tau(2+\rho_2)\sigma_{2,0}^2}{\rho_2(1-\rho_1)}+\frac{2\tau D_2}{\rho}W_K. \tag{85}
$$

Next, we derive

$$\sum_{k=0}^{K}\sum_{t=1}^{\tau} w_k \mathbf{E}\sigma_{2,k-t}^2 \;\leq\; \sum_{k=0}^{K}\sum_{t=1}^{\tau} 2w_{k-t}\mathbf{E}\sigma_{2,k-t}^2 \leq 2\tau \sum_{k=0}^{K} w_k \mathbf{E}\sigma_{2,k}^2$$

$$\overset{(62)}{\leq}\; 2\tau\sigma_{2,0}^2 \sum_{k=0}^{K} w_k(1-\rho_1)^k$$

$$+\frac{4C_2\tau}{1-\rho_2}\sum_{k=0}^{K}\sum_{l=0}^{k} w_k(1-\rho_2)^{k-l}r_l. \qquad (86)$$

For the first term in the right-hand side of previous inequality we have

$$2\tau\sigma_{2,0}^2 \sum_{k=0}^{K} w_k(1-\rho_2)^k \;\overset{(79)}{\leq}\; 2\tau\sigma_{2,0}^2 \sum_{k=0}^{K}\left(1+\frac{\rho_2}{2}\right)^{k+1}(1-\rho_2)^k$$

$$\overset{(33)}{\leq}\; 2\tau\left(1+\frac{\rho_2}{2}\right)\sigma_{2,0}^2 \sum_{k=0}^{K}\left(1-\frac{\rho_2}{2}\right)^k$$

$$\leq\; \tau(2+\rho_2)\sigma_{2,0}^2 \sum_{k=0}^{\infty}\left(1-\frac{\rho_2}{2}\right)^k \leq \frac{2\tau(2+\rho_2)\sigma_{2,0}^2}{\rho_2}.$$

The second term in the right-hand side of (86) can be upper bounded in the following way:

$$\frac{4C_2\tau}{1-\rho_2}\sum_{k=0}^{K}\sum_{l=0}^{k} w_k(1-\rho_2)^{k-l}r_l \;\overset{(79)}{\leq}\; \frac{4C_2\tau}{1-\rho_2}\sum_{k=0}^{K}\sum_{l=0}^{k} w_l r_l\left(1+\frac{\rho_2}{2}\right)^{k-l}(1-\rho_2)^{k-l}$$

$$\overset{(33)}{\leq}\; \frac{4C_2\tau}{1-\rho_2}\sum_{k=0}^{K}\sum_{l=0}^{k} w_l r_l\left(1-\frac{\rho_2}{2}\right)^{k-l}$$

$$\leq\; \frac{4C_2\tau}{1-\rho_2}\left(\sum_{k=0}^{K} w_k r_k\right)\left(\sum_{k=0}^{\infty}\left(1-\frac{\rho_2}{2}\right)^k\right)$$

$$\leq\; \frac{8C_2\tau}{\rho_2(1-\rho_2)}\sum_{k=0}^{K} w_k r_k,$$

hence

$$\sum_{k=0}^{K}\sum_{t=1}^{\tau} w_k\mathbf{E}\sigma_{2,k-t}^2 \;\overset{(86)}{\leq}\; \frac{2\tau(2+\rho_2)\sigma_{2,0}^2}{\rho_2} + \frac{8C_2\tau}{\rho_2(1-\rho_2)}\sum_{k=0}^{K} w_k r_k. \qquad (87)$$

Plugging inequalities (80), (85) and (87) in (77) we obtain

$$3L\sum_{k=0}^{K} w_k\mathbf{E}\|e^k\|^2 \;\leq\; 12\gamma^2 L\tau\left(\hat{A} + \frac{2B_1'C_1}{\rho_1(1-\rho_1)} + \frac{2B_2'C_2}{\rho_2(1-\rho_2)} + \frac{4B_1'GC_2}{\rho_2(1-\rho_1)(1-\rho_2)}\right)\sum_{k=0}^{K} w_k r_k$$

$$+\frac{6\gamma^2 LB_1'\tau(2+\rho_1)}{\rho_1}\sigma_0^2 + \frac{6\gamma^2\tau L(2+\rho_2)}{\rho_2}\left(\frac{2B_1'G}{1-\rho_1}+B_2'\right)\sigma_{2,0}^2$$

$$+3\gamma^2\tau L\left(D_1' + \frac{2B_1'D_2}{\rho}\right)W_K.$$

Taking into account that $\gamma \leq \dfrac{1}{4\sqrt{4L\tau\left(\hat{A}+\frac{2B_1'C_1}{\rho_1(1-\rho_1)}+\frac{2B_2'C_2}{\rho_2(1-\rho_2)}+\frac{4B_1'GC_2}{\rho_2(1-\rho_1)(1-\rho_2)}\right)}}$, $F_1 = \dfrac{6\gamma^2 LB_1'\tau(2+\rho_1)}{\rho_1}$, $F_2 = \dfrac{6\gamma^2\tau L}{\rho_2}\left(\frac{2B_1'G(2+\rho_2)}{1-\rho_1}+B_2'\right)$ and $D_3 = 3\gamma\tau L\left(D_1' + \frac{2B_1'D_2}{\rho}\right)$ we get

$$3L\sum_{k=0}^{K} w_k\mathbf{E}\|e^k\|^2 \;\leq\; \frac{1}{4}\sum_{k=0}^{K} w_k r_k + F_1\sigma_{1,0}^2 + F_2\sigma_{2,0}^2 + \gamma D_3.$$

$\square$

As a direct application of Lemma H.1 and Theorem 3.1 we get the following result.

**Theorem H.1.** *Let Assumptions 3.1 and 3.2 be satisfied, inequalities* (15), (16) *and* (17) *hold and*

$$\gamma \leq \min\left\{\frac{1}{4(A' + C_1 M_1 + C_2 M_2)}, \frac{1}{2\tau\mu}, \frac{1}{8\sqrt{L\tau\left(\hat{A} + \frac{2B_1'C_1}{\rho_1(1-\rho_1)} + \frac{2B_2'C_2}{\rho_2(1-\rho_2)} + \frac{4B_1'GC_2}{\rho_2(1-\rho_1)(1-\rho_2)}\right)}}\right\},$$

*where $M_1 = \frac{4B_1'}{3\rho_1}$ and $M_2 = \frac{4\left(B_2' + \frac{4}{3}G\right)}{3\rho_2}$. Then for all $K \geq 0$ we have*

$$\mathbf{E}\left[f(\bar{x}^K) - f(x^*)\right] \leq (1-\eta)^K \frac{4(T^0 + \gamma F_1 \sigma_{1,0}^2 + \gamma F_2 \sigma_{2,0}^2)}{\gamma} + 4\gamma\left(D_1' + MD_2 + D_3\right)$$

*when $\mu > 0$ and*

$$\mathbf{E}\left[f(\bar{x}^K) - f(x^*)\right] \leq \frac{4(T^0 + \gamma F_1 \sigma_{1,0}^2 + \gamma F_2 \sigma_{2,0}^2)}{\gamma K} + 4\gamma\left(D_1' + MD_2 + D_3\right)$$

*when $\mu = 0$, where $\eta = \min\left\{\gamma\mu/2, \rho_1/4, \rho_2/4\right\}$, $T^k \stackrel{\text{def}}{=} \|\tilde{x}^k - x^*\|^2 + M_1\gamma^2\sigma_{1,k}^2 + M_2\gamma^2\sigma_{2,k}^2$ and*

$$F_1 = \frac{6\gamma^2 LB_1'\tau(2+\rho_1)}{\rho_1}, \quad F_2 = \frac{6\gamma^2\tau L(2+\rho_2)}{\rho_2}\left(\frac{2B_1'G}{1-\rho_1} + B_2'\right),$$

$$D_3 = 3\gamma\tau L\left(D_1' + \frac{2B_1'D_2}{\rho_1}\right).$$

---

**Algorithm 1** `DIANAsr` with Double Compression (`DIANAsr-DQ`)

---

**Input:** learning rates $\gamma > 0$, $\alpha \in (0,1]$, initial vectors $x^0, h_1^0, \ldots, h_n^0 \in \mathbb{R}^d$

1: Set $h^0 = \frac{1}{n}\sum_{i=1}^n h_i^0$
2: **for** $k = 0, 1, \ldots$ **do**
3:      Broadcast $g^{k-1}$ to all workers                        $\triangleright$ If $k = 0$, then broadcast $x^0$
4:      **for** $i = 1, \ldots, n$ in parallel **do**
5:          $x^k = x^{k-1} - \gamma g^{k-1}$                          $\triangleright$ Ignore this line if $k = 0$
6:          Sample $g_i^{k,1} = \nabla f_{\xi_i^k}(x^k)$ satisfying Assumption I.1 independtently from other workers
7:          $\hat{\Delta}_i^k = g_i^{k,1} - h_i^k$
8:          Sample $\Delta_i^k \sim Q_1(\hat{\Delta}_i^k)$ indepently from other workers
9:          $g_i^{k,2} = h_i^k + \Delta_i^k$
10:         $h_i^{k+1} = h_i^k + \alpha \Delta_i^k$
11:      **end for**
12:      $g^{k,2} = \frac{1}{n}\sum_{i=1}^n g_i^{k,2} = h^k + \frac{1}{n}\sum_{i=1}^n \Delta_i^k$
13:      $h^{k+1} = \frac{1}{n}\sum_{i=1}^n h_i^{k+1} = h^k + \alpha\frac{1}{n}\sum_{i=1}^n \Delta_i^k$
14:      Sample $g^k \sim Q_2(g^{k,2})$
15:      $x^{k+1} = x^k - \gamma g^{k-1}$
16: **end for**

---

# I   Special Cases: `SGD`

To illustrate the generality of our approach, we develop and analyse a new special case of `SGD` without error-feedback and show that in some cases, our framework recovers tighter rates than the framework from [11].

## I.1   `DIANA` with Arbitrary Sampling and Double Quantization

In this section we consider problem (1) with $f(x)$ being $\mu$-quasi strongly convex and $f_i(x)$ satisfying (3) where functions $f_{ij}(x)$ are differentiable, but not necessary convex. Following [13] we construct a stochastic reformulation of this problem:

$$f(x) = \mathbf{E}_{\mathcal{D}}\left[f_\xi(x)\right], \quad f_\xi(x) = \frac{1}{n}\sum_{i=1}^n f_{\xi_i}(x), \quad f_{\xi_i}(x) = \frac{1}{m}\sum_{j=1}^m \xi_{ij} f_{ij}(x), \qquad (88)$$

where $\xi = (\xi_1^\top, \ldots, \xi_n^\top), \xi_i = (\xi_{i1}, \ldots, \xi_{im})^\top$ is a random vector with distribution $\mathcal{D}_i$ such that $\mathbf{E}_{\mathcal{D}_i}[\xi_{ij}] = 1$ for all $i \in [n], j \in [m]$ and the following assumption holds.

**Assumption I.1** (Expected smoothness). *We assume that functions $f_1, \ldots, f_n$ are $\mathcal{L}$-smooth in expectation w.r.t. distributions $\mathcal{D}_1, \ldots, \mathcal{D}_n$, i.e., there exists constant $\mathcal{L} = \mathcal{L}(f, \mathcal{D}_1, \ldots, \mathcal{D}_n)$ such that*

$$\mathbf{E}_{\mathcal{D}_i}\left[\|\nabla f_{\xi_i}(x) - \nabla f_{\xi_i}(x^*)\|^2\right] \leq 2\mathcal{L}D_{f_i}(x, x^*) \qquad (89)$$

*for all $i \in [n]$ and $x \in \mathbb{R}^d$.*

To solve this problem, we consider `DIANA` [37, 19] — a distributed stochastic method using unbiased compressions or *quantizations* for communication between workers and master. We start with the formal definition of quantization. In [37, 19] `DIANA` was analyzed under the assumption that stochastic gradients $g_i^k$ have uniformly bounded variances which is not very practical.

Therefore, we consider a slightly different method called `DIANAsr-DQ` which works with the stochastic reformulation (88) of problem (1)+(3), see Algorithm 1. Moreover, to illustrate the flexibility of our approach, we consider compression not only on the workers' side but also on the master side. To perform an update of `DIANAsr-DQ` master needs to gather quantized gradient differences $\Delta_i^k$ and the to broadcast quantized stochastic gradient $g^k$ to all workers. Clearly, in this case, only compressed vectors participate in communication.

In the concurrent work [40] the same method was independently proposed under the name of `Artemis`. However, our analysis is slightly more general: it is based on Assumption I.1 while in [40] authors assume $L$-cocoercivity of stochastic gradients almost surely. Next, a very similar approach was considered in [48], where authors present a method with error compensation on master and worker sides. Moreover, recently another method called `DORE` was developed in [35], which uses `DIANA`-trick on the worker side and error compensation on the master side. However, in these methods, compression operators are the same on both sides, despite the fact that gathering the information often costs much more than broadcasting. Therefore, the natural idea is in using different quantization for gathering and broadcasting, and it is what `DIANAsr-DQ` does. Moreover, we do not assume uniform boundedness of the second moment of the stochastic gradient like in [48], and we also do not assume uniform boundedness of the variance of the stochastic gradient like in [35]. Assumption I.1 is more natural and always holds for the problems (1)+(3) when $f_{ij}$ are convex and $L$-smooth for each $i \in [n]$, $j \in [m]$. In contrast, in the same setup, there exist such problems that the variance of the stochastic gradients is not uniformly upper bounded by any finite constant.

We assume that $Q_1$ and $Q_2$ satisfy (26) with parameters $\omega_1$ and $\omega_2$ respectively.

**Lemma I.1.** *Let Assumption I.1 be satisfied. Then, for all $k \geq 0$ we have*

$$\mathbf{E}\left[g^k \mid x^k\right] = \nabla f(x^k), \tag{90}$$

$$\mathbf{E}\left[\|g^k\|^2 \mid x^k\right] \leq 2\mathcal{L}(1+\omega_2)\left(2 + \frac{3\omega_1}{n}\right)\left(f(x^k) - f(x^*)\right) + \frac{3\omega_1(1+\omega_2)}{n}\sigma_k^2 + D_1', \tag{91}$$

*where* $\sigma_k^2 = \frac{1}{n}\sum_{i=1}^n \|h_i^k - \nabla f(x^*)\|^2$ *and* $D_1' = \frac{(2+3\omega_1)(1+\omega_2)}{n^2}\sum_{i=1}^n \mathbf{E}_{\mathcal{D}_i}\left[\|\nabla f_{\xi_i}(x^*) - \nabla f_i(x^*)\|^2\right].$

*Proof.* First of all, we show inbiasedness of $g^k$:

$$\mathbf{E}\left[g^k \mid x^k\right] \overset{(35),(26)}{=} \mathbf{E}\left[g^{k,2} \mid x^k\right] = h^k + \frac{1}{n}\sum_{i=1}^n \mathbf{E}\left[\Delta_i^k \mid x^k\right]$$

$$\overset{(35),(26)}{=} h^k + \frac{1}{n}\sum_{i=1}^n \mathbf{E}\left[\hat{\Delta}_i^k \mid x^k\right]$$

$$= h^k + \frac{1}{n}\sum_{i=1}^n \left(\nabla f_i(x^k) - h_i^k\right) = \nabla f(x^k).$$

Next, to denote mathematical expectation w.r.t. the randomness coming from quantizations $Q_1$ and $Q_2$ at iteration $k$ we use $\mathbf{E}_{Q_1^k}[\cdot]$ and $\mathbf{E}_{Q_2^k}[\cdot]$ respectively. Using these notations and the definition of quantization we derive

$$\mathbf{E}_{Q_2^k}[\|g^k\|^2] \overset{(34),(26)}{=} \|g^{k,1}\|^2 + \mathbf{E}_{Q_2^k}\left[\|g^{k,2} - g^{k,1}\|^2\right]$$

$$\overset{(26)}{\leq} (1+\omega_2)\|g^{k,1}\|^2.$$

Taking the conditopnal mathematical expectation $\mathbf{E}_{Q_1^k}[\cdot]$ from the both sides of previous inequality and using the independence of $\Delta_i^1, \ldots, \Delta_i^n$ we get

$$\mathbf{E}_{Q_1^k,Q_2^k}\left[\|g^k\|^2\right] \overset{(35)}{=} (1+\omega_2)\mathbf{E}_{Q_1^k}\left[\|g^{k,1}\|^2\right] = (1+\omega_2)\mathbf{E}_{Q_1^k}\left[\left\|\frac{1}{n}\sum_{i=1}^n(h_i^k+\Delta_i^k)\right\|^2\right]$$

$$\overset{(34)}{=} (1+\omega_2)\left\|\frac{1}{n}\sum_{i=1}^n\left(h_i^k+\hat{\Delta}_i^k\right)\right\|^2 + (1+\omega_2)\mathbf{E}_{Q_1^k}\left[\left\|\frac{1}{n}\sum_{i=1}^n(\Delta_i^k-\hat{\Delta}_i^k)\right\|^2\right]$$

$$= (1+\omega_2)\left\|\frac{1}{n}\sum_{i=1}^n\left(\nabla f_{\xi_i^k}(x^k)-\nabla f_{\xi_i^k}(x^*)+\nabla f_{\xi_i^k}(x^*)-\nabla f_i(x^*)\right)\right\|^2$$

$$+\frac{(1+\omega_2)}{n^2}\sum_{i=1}^n\mathbf{E}_{Q_1^k}\left[\|\Delta_i^k-\hat{\Delta}_i^k\|^2\right]$$

$$\overset{(31),(26)}{\leq} \frac{2(1+\omega_2)}{n}\sum_{i=1}^n\|\nabla f_{\xi_i^k}(x^k)-\nabla f_{\xi_i^k}(x^*)\|^2$$

$$+2(1+\omega_2)\left\|\frac{1}{n}\sum_{i=1}^n\left(\nabla f_{\xi_i^k}(x^*)-\nabla f_i(x^*)\right)\right\|^2$$

$$+\frac{\omega_1(1+\omega_2)}{n^2}\sum_{i=1}^n\|\nabla f_{\xi_i^k}(x^k)-h_i^k\|^2$$

$$\overset{(31)}{\leq} \frac{2(1+\omega_2)}{n}\sum_{i=1}^n\|\nabla f_{\xi_i^k}(x^k)-\nabla f_{\xi_i^k}(x^*)\|^2$$

$$+2(1+\omega_2)\left\|\frac{1}{n}\sum_{i=1}^n\left(\nabla f_{\xi_i^k}(x^*)-\nabla f_i(x^*)\right)\right\|^2$$

$$+\frac{3\omega_1(1+\omega_2)}{n^2}\sum_{i=1}^n\|\nabla f_{\xi_i^k}(x^k)-\nabla f_{\xi_i^k}(x^*)\|^2$$

$$+\frac{3\omega_1(1+\omega_2)}{n^2}\sum_{i=1}^n\|\nabla f_{\xi_i^k}(x^*)-\nabla f_i(x^*)\|^2$$

$$+\frac{3\omega_1(1+\omega_2)}{n^2}\sum_{i=1}^n\|h_i^k-\nabla f_i(x^*)\|^2.$$

Finally, we take conditional mathematical expectation $\mathbf{E}[\cdot \mid x^k]$ from the both sides of the inequality above and use the independece of $\xi_1^k, \ldots, \xi_n^k$:

$$\mathbf{E}\left[\|g^k\|^2 \mid x^k\right] \overset{(89)}{\leq} 2\mathcal{L}(1+\omega_2)\left(2+\frac{3\omega_1}{n}\right)(f(x^k)-f(x^*)) + \frac{3\omega_1(1+\omega_2)}{n}\sigma_k^2$$

$$+2(1+\omega_2)\mathbf{E}\left[\left\|\frac{1}{n}\sum_{i=1}^n\left(\nabla f_{\xi_i^k}(x^*)-\nabla f_i(x^*)\right)\right\|^2 \mid x^k\right]$$

$$+\frac{3\omega_1(1+\omega_2)}{n^2}\sum_{i=1}^n\mathbf{E}_{\mathcal{D}_i}\left[\|\nabla f_{\xi_i}(x^*)-\nabla f_i(x^*)\|^2\right]$$

$$= 2\mathcal{L}(1+\omega_2)\left(2+\frac{3\omega_1}{n}\right)(f(x^k)-f(x^*)) + \frac{3\omega_1(1+\omega_2)}{n}\sigma_k^2$$

$$+\frac{(1+\omega_2)(2+3\omega_1)}{n^2}\sum_{i=1}^n\mathbf{E}_{\mathcal{D}_i}\left[\|\nabla f_{\xi_i}(x^*)-\nabla f_i(x^*)\|^2\right].$$

$\square$

**Lemma I.2.** *Let $f_i$ be convex and L-smooth, Assumption I.1 holds and $\alpha \leq \frac{1}{(\omega_1+1)}$. Then, for all $k \geq 0$ we have*

$$\mathbf{E}\left[\sigma_{k+1}^2 \mid x^k\right] \leq (1-\alpha)\sigma_k^2 + 2\alpha(3\mathcal{L} + 4L)(f(x^k) - f(x^*)) + D_2, \qquad (92)$$

*where $\sigma_k^2 = \frac{1}{n}\sum_{i=1}^{n} \|h_i^k - \nabla f_i(x^*)\|^2$ and $D_2 = \frac{3\alpha}{n}\sum_{i=1}^{n} \mathbf{E}_{\mathcal{D}_i}\left[\|\nabla f_{\xi_i}(x^*) - \nabla f_i(x^*)\|^2\right]$.*

*Proof.* For simplicity, we introduce new notation: $h_i^* \stackrel{\text{def}}{=} \nabla f_i(x^*)$. Using this we derive an upper bound for the second moment of $h_i^{k+1} - h_i^*$:

$$
\begin{aligned}
\mathbf{E}\left[\|h_i^{k+1} - h_i^*\|^2 \mid x^k\right] &= \mathbf{E}\left[\left\|h_i^k - h_i^* + \alpha\Delta_i^k\right\|^2 \mid x^k\right] \\
&\overset{(26)}{=} \|h_i^k - h_i^*\|^2 + 2\alpha\langle h_i^k - h_i^*, \nabla f_i(x^k) - h_i^k\rangle + \alpha^2\mathbf{E}\left[\|\Delta_i^k\|^2 \mid x^k\right] \\
&\overset{(26),(35)}{\leq} \|h_i^k - h_i^*\|^2 + 2\alpha\langle h_i^k - h_i^*, \nabla f_i(x^k) - h_i^k\rangle \\
&\qquad + \alpha^2(\omega_1+1)\mathbf{E}\left[\|\nabla f_{\xi_i^k}(x^k) - h_i^k\|^2 \mid x^k\right].
\end{aligned}
$$

Using variance decomposition (34) and $\alpha \leq \frac{1}{(\omega_1+1)}$ we get

$$
\begin{aligned}
\alpha^2(\omega_1+1)\mathbf{E}_{\mathcal{D}_i}\left[\|\nabla f_{\xi_i^k}(x^k) - h_i^k\|^2\right] &\overset{(34)}{=} \alpha^2(\omega_1+1)\mathbf{E}_{\mathcal{D}_i}\left[\|\nabla f_{\xi_i^k}(x^k) - \nabla f_i(x^k)\|^2\right] \\
&\qquad + \alpha^2(\omega_1+1)\|\nabla f_i(x^k) - h_i^k\|^2 \\
&\overset{(31)}{\leq} 3\alpha\mathbf{E}_{\mathcal{D}_i}\left[\|\nabla f_{\xi_i^k}(x^k) - \nabla f_{\xi_i^k}(x^*)\|^2\right] \\
&\qquad + 3\alpha\mathbf{E}_{\mathcal{D}_i}\left[\|\nabla f_{\xi_i^k}(x^*) - \nabla f_i(x^*)\|^2\right] \\
&\qquad + 3\alpha\|\nabla f_i(x^k) - \nabla f_i(x^*)\|^2 \\
&\qquad + \alpha\|\nabla f_i(x^k) - h_i^k\|^2 \\
&\overset{(11),(89)}{\leq} 6\alpha(\mathcal{L} + L)D_{f_i}(x^k, x^*) + \alpha\|\nabla f_i(x^k) - h_i^k\|^2 \\
&\qquad + 3\alpha\mathbf{E}_{\mathcal{D}_i}\left[\|\nabla f_{\xi_i^k}(x^*) - \nabla f_i(x^*)\|^2\right]
\end{aligned}
$$

Putting all together we obtain

$$
\begin{aligned}
\mathbf{E}\left[\|h_i^{k+1} - h_i^*\|^2 \mid x^k\right] &\leq \|h_i^k - h_i^*\|^2 + \alpha\left\langle \nabla f_i(x^k) - h_i^k, f_i(x^k) + h_i^k - 2h_i^*\right\rangle \\
&\qquad + 6\alpha(\mathcal{L} + L)D_{f_i}(x^k, x^*) + 3\alpha\mathbf{E}_{\mathcal{D}_i}\left[\|\nabla f_{\xi_i^k}(x^*) - \nabla f_i(x^*)\|^2\right] \\
&\overset{(28)}{=} \|h_i^k - h_i^*\|^2 + \alpha\|\nabla f_i(x^k) - h_i^*\|^2 - \alpha\|h_i^k - h_i^*\|^2 \\
&\qquad + 6\alpha(\mathcal{L} + L)D_{f_i}(x^k, x^*) + 3\alpha\mathbf{E}_{\mathcal{D}_i}\left[\|\nabla f_{\xi_i^k}(x^*) - \nabla f_i(x^*)\|^2\right] \\
&\overset{(11)}{\leq} (1-\alpha)\|h_i^k - h_i^*\|^2 + \alpha(6\mathcal{L} + 8L)D_{f_i}(x^k, x^*) \\
&\qquad + 3\alpha\mathbf{E}_{\mathcal{D}_i}\left[\|\nabla f_{\xi_i^k}(x^*) - \nabla f_i(x^*)\|^2\right].
\end{aligned}
$$

Summing up the above inequality for $i = 1, \ldots, n$ we derive

$$
\begin{aligned}
\frac{1}{n}\sum_{i=1}^{n}\mathbf{E}\left[\|h_i^{k+1} - h_i^*\|^2 \mid x^k\right] &\leq \frac{1-\alpha}{n}\sum_{i=1}^{n}\|h_i^k - h_i^*\|^2 + \alpha(6\mathcal{L} + 8L)(f(x^k) - f(x^*)) \\
&\qquad + \frac{3\alpha}{n}\sum_{i=1}^{n}\mathbf{E}_{\mathcal{D}_i}\left[\|\nabla f_{\xi_i^k}(x^*) - \nabla f_i(x^*)\|^2\right].
\end{aligned}
$$

$\square$

**Theorem I.1.** *Assume that $f_i(x)$ is convex and $L$-smooth for all $i = 1, \ldots, n$, $f(x)$ is $\mu$-quasi strongly convex and Assumption I.1 holds. Then* `DIANAsr-DQ` *satisfies Assumption 3.4 with*

$$A' = \mathcal{L}(1 + \omega_2)\left(2 + \frac{3\omega_1}{n}\right), \quad B'_1 = \frac{3\omega_1(1 + \omega_2)}{n},$$

$$D'_1 = \frac{(2 + 3\omega_1)(1 + \omega_2)}{n^2} \sum_{i=1}^{n} \mathbf{E}_{\mathcal{D}_i}\left[\|\nabla f_{\xi_i}(x^*) - \nabla f_i(x^*)\|^2\right],$$

$$\sigma_{1,k}^2 = \sigma_k^2 = \frac{1}{n}\sum_{i=1}^{n}\|h_i^k - \nabla f_i(x^*)\|^2, \quad B'_2 = 0, \quad \sigma_{2,k}^2 \equiv 0, \quad \rho_1 = \alpha, \quad \rho_2 = 1,$$

$$C_1 = \alpha(3\mathcal{L} + 4L), \quad C_2 = 0, \quad D_2 = \frac{3\alpha}{n}\sum_{i=1}^{n}\mathbf{E}_{\mathcal{D}_i}\left[\|\nabla f_{\xi_i}(x^*) - \nabla f_i(x^*)\|^2\right],$$

$$G = 0, \quad F_1 = F_2 = 0, \quad D_3 = 0,$$

*with $\gamma$ and $\alpha$ satisfying*

$$\gamma \le \frac{1}{4(1 + \omega_2)\left(\mathcal{L}\left(2 + \frac{15\omega_1}{n}\right) + \frac{16L\omega_1}{n}\right)}, \quad \alpha \le \frac{1}{\omega + 1}, \quad M_1 = \frac{4\omega_1(1 + \omega_2)}{n\alpha}, \quad M_2 = 0$$

*and for all $K \ge 0$*

$$\mathbf{E}\left[f(\bar{x}^K) - f(x^*)\right] \le \left(1 - \min\left\{\frac{\gamma\mu}{2}, \frac{\alpha}{4}\right\}\right)^K \frac{4T^0}{\gamma} + 4\gamma\left(D'_1 + M_1 D_2\right),$$

*when $\mu > 0$ and*

$$\mathbf{E}\left[f(\bar{x}^K) - f(x^*)\right] \le \frac{4T^0}{\gamma K} + 4\gamma\left(D'_1 + M_1 D_2\right)$$

*when $\mu = 0$, where $T^k \overset{def}{=} \|x^k - x^*\|^2 + M_1\gamma^2\sigma_{1,k}^2$.*

In other words, if

$$\gamma = \frac{1}{4(1 + \omega_2)\left(\mathcal{L}\left(2 + \frac{15\omega_1}{n}\right) + \frac{16L\omega_1}{n}\right)}, \quad \alpha = \frac{1}{\omega + 1}$$

and $D_1 = 0$, i.e., $\nabla f_{\xi_i^k}(x^k) = \nabla f_i(x^k)$ almost surely, `DIANAsr-DQ` converges with the linear rate

$$\mathcal{O}\left(\left(\omega_1 + \frac{\mathcal{L}}{\mu}(1 + \omega_2)\left(1 + \frac{\omega_1}{n}\right)\right)\ln\frac{1}{\varepsilon}\right)$$

to the exact solution. Applying Lemma D.2 we establish the rate of convergence to $\varepsilon$-solution.

**Corollary I.1.** *Let the assumptions of Theorem I.1 hold and $\mu > 0$. Then after $K$ iterations of* `DIANAsq-DQ` *with the stepsize*

$$\gamma_0 = \frac{1}{4(1 + \omega_2)\left(\mathcal{L}\left(2 + \frac{15\omega_1}{n}\right) + \frac{16L\omega_1}{n}\right)}$$

$$\gamma = \min\left\{\gamma_0, \frac{\ln\left(\max\left\{2, \frac{\mu^2 K^2(\|x^0 - x^*\|^2 + M_1\gamma_0^2\sigma_{1,0}^2)}{D'_1 + M_1 D_2}\right\}\right)}{\mu K}\right\}, \quad M_1 = \frac{4\omega_1(1 + \omega_2)}{n\alpha}$$

*and $\alpha = \frac{1}{\omega + 1}$ we have*

$$\mathbf{E}\left[f(\bar{x}^K) - f(x^*)\right] = \widetilde{\mathcal{O}}\left(A'\|x^0 - x^*\|^2 \exp\left(-\min\left\{\frac{\mu}{A'}, \frac{1}{\omega_1}\right\}K\right) + \frac{D'_1 + M_1 D_2}{\mu K}\right).$$

*That is, to achive* $\mathbf{E}\left[f(\bar{x}^K) - f(x^*)\right] \le \varepsilon$ `DIANAsq-DQ` *requires*

$$\widetilde{\mathcal{O}}\left(\omega_1 + \frac{\mathcal{L}\left(1 + \frac{\omega_1}{n}\right)(1 + \omega_2)}{\mu} + \frac{(1 + \omega_1)(1 + \omega_2)}{n^2\mu\varepsilon}\sum_{i=1}^{n}\mathbf{E}_{\mathcal{D}_i}\|\nabla f_{\xi_i}(x^*) - \nabla f_i(x^*)\|^2\right) \text{ iterations.}$$

Applying Lemma D.3 we get the complexity result in the case when $\mu = 0$.

**Corollary I.2.** *Let the assumptions of Theorem I.1 hold and $\mu = 0$. Then after $K$ iterations of* `DIANAsq-DQ` *with the stepsize*

$$\gamma_0 = \frac{1}{4(1 + \omega_2)\left(\mathcal{L}\left(2 + \frac{15\omega_1}{n}\right) + \frac{16L\omega_1}{n}\right)}$$

$$\gamma = \min\left\{\gamma_0, \sqrt{\frac{\|x^0 - x^*\|^2}{M_1\sigma_{1,0}^2}}, \sqrt{\frac{\|x^0 - x^*\|^2}{(D_1' + M_1 D_2)K}}\right\}, \quad M_1 = \frac{4\omega_1(1 + \omega_2)}{n\alpha}$$

*and $\alpha = \frac{1}{\omega+1}$ we have $\mathbf{E}\left[f(\bar{x}^K) - f(x^*)\right]$ of order*

$$\mathcal{O}\left(\frac{\mathcal{L}R_0^2(1 + \omega_2)\left(1 + \frac{\omega_1}{n}\right)}{K} + \frac{R_0\sigma_{1,0}(1 + \omega_1)\sqrt{1 + \omega_2}}{\sqrt{nK}} + \frac{R_0\sqrt{(1 + \omega_1)(1 + \omega_2)D_{opt}}}{\sqrt{nK}}\right)$$

*where $R_0 = \|x^0 - x^*\|^2, D_{opt} = \frac{1}{n}\sum\limits_{i=1}^{n}\mathbf{E}_{\mathcal{D}_i}\|\nabla f_{\xi_i}(x^*) - \nabla f_i(x^*)\|^2$. That is, to achieve $\mathbf{E}\left[f(\bar{x}^K) - f(x^*)\right] \leq \varepsilon$* `DIANAsq-DQ` *requires*

$$\mathcal{O}\left(\frac{\mathcal{L}R_0^2(1 + \omega_2)\left(1 + \frac{\omega_1}{n}\right)}{\varepsilon} + \frac{R_0\sigma_{1,0}(1 + \omega_1)\sqrt{1 + \omega_2}}{\sqrt{n}\varepsilon} + \frac{R_0^2(1 + \omega_1)(1 + \omega_2)D_{opt}}{n\varepsilon^2}\right)$$

*iterations.*

## I.2 Recovering Tight Complexity Bounds for `VR-DIANA`

In this section we consider the same problem (1)+(3) and variance reduced version of `DIANA` called `VR-DIANA` [19], see Algorithm 2. For simplicity we assume that each $f_{ij}$ is convex and $L$-smooth and $f_i$ is additionally $\mu$-strongly convex.

**Lemma I.3** (Lemmas 3, 5, 6 and 7 from [19]). *Let $\alpha \leq \frac{1}{\omega+1}$. Then for all iterates $k \geq 0$ of Algorithm 2 the following inequalities hold:*

$$\mathbf{E}\left[g^k \mid x^k\right] = \nabla f(x^k), \tag{93}$$

$$\mathbf{E}\left[H^{k+1} \mid x^k\right] \leq (1 - \alpha)H^k + \frac{2\alpha}{m}D^k + 8\alpha Ln\left(f(x^k) - f(x^*)\right), \tag{94}$$

$$\mathbf{E}\left[D^{k+1} \mid x^k\right] \leq \left(1 - \frac{1}{m}\right)D^k + 2Ln\left(f(x^k) - f(x^*)\right), \tag{95}$$

$$\mathbf{E}\left[\|g^k\|^2 \mid x^k\right] \leq 2L\left(1 + \frac{4\omega + 2}{n}\right)\left(f(x^k) - f(x^*)\right) + \frac{2\omega}{n^2}\frac{D^k}{m} + \frac{2(\omega + 1)}{n^2}H^k, \tag{96}$$

*where $H^k = \sum\limits_{i=1}^{n}\|h_i^k - \nabla f_i(x^*)\|^2$ and $D^k = \sum\limits_{i=1}^{n}\sum\limits_{j=1}^{m}\|\nabla f_{ij}(w_{ij}^k) - \nabla f_{ij}(x^*)\|^2$.*

This lemma shows that `VR-DIANA` satisfies (15), (16) and (17). Applying Theorem F.1 we get the following result.

**Theorem I.2.** *Assume that $f_{ij}(x)$ is convex and $L$-smooth for all $i = 1, \ldots, n$ and $f_i(x)$ is $\mu$-strongly convex for all $i = 1, \ldots, n$. Then* `VR-DIANA` *satisfies Assumption 3.4 with*

$$A' = L\left(1 + \frac{4\omega + 2}{n}\right), \quad B_1' = \frac{2(\omega + 1)}{n}, \quad D_1' = 0,$$

$$\sigma_{1,k}^2 = H^k = \frac{1}{n}\sum_{i=1}^{n}\|h_i^k - \nabla f_i(x^*)\|^2, \quad B_2' = \frac{2\omega}{n},$$

$$\sigma_{2,k}^2 = D^k = \frac{1}{nm}\sum_{i=1}^{n}\sum_{j=1}^{m}\|\nabla f_{ij}(w_{ij}^k) - \nabla f_{ij}(x^*)\|^2, \quad \rho_1 = \alpha, \quad \rho_2 = \frac{1}{m},$$

$$C_1 = 4\alpha L, \quad C_2 = \frac{L}{m}, \quad D_2 = 0, \quad G = 2, \quad F_1 = F_2 = 0, \quad D_3 = 0,$$

**Algorithm 2** VR-DIANA based on LSVRG (Variant 1), SAGA (Variant 2), [19]

**Input:** learning rates $\alpha > 0$ and $\gamma > 0$, initial vectors $x^0, h_1^0, \ldots, h_n^0$, $h^0 = \frac{1}{n} \sum_{i=1}^n h_i^0$

1: **for** $k = 0, 1, \ldots$ **do**

2:   Sample random $u^k = \begin{cases} 1, & \text{with probability } \frac{1}{m} \\ 0, & \text{with probability } 1 - \frac{1}{m} \end{cases}$   $\triangleright$ only for Variant 1

3:   Broadcast $x^k$, $u^k$ to all workers

4:   **for** $i = 1, \ldots, n$ in parallel **do**   $\triangleright$ Worker side

5:     Pick $j_i^k$ uniformly at random from $[m]$

6:     $\mu_i^k = \frac{1}{m} \sum_{j=1}^m \nabla f_{ij}(w_{ij}^k)$

7:     $g_i^k = \nabla f_{ij_i^k}(x^k) - \nabla f_{ij_i^k}(w_{ij_i^k}^k) + \mu_i^k$

8:     $\hat{\Delta}_i^k = Q(g_i^k - h_i^k)$

9:     $h_i^{k+1} = h_i^k + \alpha \hat{\Delta}_i^k$

10:     **for** $j = 1, \ldots, m$ **do**

11:       $w_{ij}^{k+1} = \begin{cases} x^k, & \text{if } u^k = 1 \\ w_{ij}^k, & \text{if } u^k = 0 \end{cases}$   $\triangleright$ Variant 1 (L-SVRG): update epoch gradient if $u^k = 1$

12:       $w_{ij}^{k+1} = \begin{cases} x^k, & j = j_i^k \\ w_{ij}^k, & j \neq j_i^k \end{cases}$   $\triangleright$ Variant 2 (SAGA): update gradient table

13:     **end for**

14:   **end for**

15:   $h^{k+1} = h^k + \frac{\alpha}{n} \sum_{i=1}^n \hat{\Delta}_i^k$   $\triangleright$ Gather quantized updates

16:   $g^k = \frac{1}{n} \sum_{i=1}^n (\hat{\Delta}_i^k + h_i^k)$

17:   $x^{k+1} = x^k - \gamma g^k$

18: **end for**

---

*with $\gamma$ and $\alpha$ satisfying*

$$\gamma \leq \frac{3}{L\left(\frac{41}{3} + \frac{52\omega + 35}{n}\right)}, \quad \alpha \leq \frac{1}{\omega + 1}, \quad M_1 = \frac{8(\omega + 1)}{3n\alpha}, \quad M_2 = \frac{8\omega m}{3n} + \frac{32m}{9}$$

*and for all $K \geq 0$*

$$\mathbf{E}\left[f(\bar{x}^K) - f(x^*)\right] \leq \left(1 - \min\left\{\frac{\gamma\mu}{2}, \frac{\alpha}{4}, \frac{1}{4m}\right\}\right)^K \frac{4T^0}{\gamma},$$

*when $\mu > 0$ and*

$$\mathbf{E}\left[f(\bar{x}^K) - f(x^*)\right] \leq \frac{4T^0}{\gamma K}$$

*when $\mu = 0$, where $T^k \overset{def}{=} \|x^k - x^*\|^2 + M_1 \gamma^2 \sigma_{1,k}^2 + M_2 \gamma^2 \sigma_{2,k}^2$.*

In other words, if $\mu > 0$ and

$$\gamma = \frac{3}{L\left(\frac{41}{3} + \frac{52\omega + 35}{n}\right)}, \quad \alpha = \frac{1}{\omega + 1},$$

then VR-DIANA converges with the linear rate

$$\mathcal{O}\left(\left(\omega + m + \kappa\left(1 + \frac{\omega}{n}\right)\right) \ln \frac{1}{\varepsilon}\right)$$

to the exact solution which coincides with the rate obtained in [19]. We notice that the framework from [11] establishes slightly worse guarantee:

$$\mathcal{O}\left(\left(\omega + m + \kappa\left(1 + \frac{\omega}{n}\right)\frac{\max\{m, \omega + 1\}}{m}\right) \ln \frac{1}{\varepsilon}\right)$$

This guarantee is strictly worse than our bound when $m \leq 1 + \omega$. The key tool that helps us to improve the rate is two sequences of $\{\sigma_{1,k}^2\}_{k \geq 0}$, $\{\sigma_{2,k}^2\}_{k \geq 0}$ instead of one sequence $\{\sigma_k^2\}_{k \geq 0}$ as in [11].

Applying Lemma D.3 we get the complexity result in the case when $\mu = 0$.

**Corollary I.3.** *Let the assumptions of Theorem I.2 hold and $\mu = 0$. Then after $K$ iterations of* VR-DIANA *with the stepsize*

$$
\gamma_0 = \frac{3}{L\left(\frac{41}{3} + \frac{52\omega + 35}{n}\right)}
$$

$$
\gamma = \min\left\{\gamma_0, \sqrt{\frac{\|x^0 - x^*\|^2}{M_1 \sigma_{1,0}^2 + M_2 \sigma_{2,0}^2}}\right\}, \quad M_1 = \frac{8(\omega + 1)}{3n\alpha}, \quad M_2 = \frac{8\omega m}{3n} + \frac{32m}{9}
$$

*and $\alpha = \frac{1}{\omega + 1}$ we have $\mathbf{E}\left[f(\bar{x}^K) - f(x^*)\right]$ of order*

$$
\mathcal{O}\left(\frac{LR_0^2\left(1 + \frac{\omega}{n}\right)}{K} + \frac{R_0\sqrt{\frac{(1+\omega)^2}{n}\sigma_{1,0}^2 + \left(1 + \frac{\omega}{n}\right)m\sigma_{2,0}^2}}{K}\right)
$$

*where $R_0 = \|x^0 - x^*\|^2$. That is, to achive $\mathbf{E}\left[f(\bar{x}^K) - f(x^*)\right] \leq \varepsilon$* VR-DIANA *requires*

$$
\mathcal{O}\left(\frac{LR_0^2\left(1 + \frac{\omega}{n}\right)}{\varepsilon} + \frac{R_0\sqrt{\frac{(1+\omega)^2}{n}\sigma_{1,0}^2 + \left(1 + \frac{\omega}{n}\right)m\sigma_{2,0}^2}}{\varepsilon}\right)
$$

*iterations.*

---
**Algorithm 3** `EC-SGDsr`
---
**Input:** learning rate $\gamma > 0$, initial vector $x^0 \in \mathbb{R}^d$
1: Set $e_i^0 = 0$ for all $i = 1, \ldots, n$
2: **for** $k = 0, 1, \ldots$ **do**
3:      Broadcast $x^k$ to all workers
4:      **for** $i = 1, \ldots, n$ in parallel **do**
5:          Sample $g_i^k = \nabla f_{\xi_i}(x^k)$
6:          $v_i^k = C(e_i^k + \gamma g_i^k)$
7:          $e_i^{k+1} = e_i^k + \gamma g_i^k - v_i^k$
8:      **end for**
9:      $e^k = \frac{1}{n} \sum_{i=1}^n e_i^k$, $g^k = \frac{1}{n} \sum_{i=1}^n g_i^k$, $v^k = \frac{1}{n} \sum_{i=1}^n v_i^k$
10:    $x^{k+1} = x^k - v^k$
11: **end for**
---

## J    Special Cases: Error Compensated Methods

### J.1    `EC-SGDsr`

In this section we consider the same setup as in Section I.1 and assume additionally that $f_1, \ldots, f_n$ are $L$-smooth.

**Lemma J.1.** *For all $k \geq 0$ we have*

$$
\frac{1}{n} \sum_{i=1}^n \mathbf{E}\left[\|g_i^k\|^2 \mid x^k\right] \leq 4L\left(f(x^k) - f(x^*)\right) + \frac{2}{n} \sum_{i=1}^n \|\nabla f_i(x^*)\|^2,
$$

$$
\frac{1}{n} \sum_{i=1}^n \mathbf{E}\left[\|g_i^k - \bar{g}_i^k\|^2 \mid x^k\right] \leq 6(\mathcal{L} + L)\left(f(x^k) - f(x^*)\right) + \frac{3}{n} \sum_{i=1}^n \mathbf{E}_{\mathcal{D}}\left[\|\nabla f_{\xi_i}(x^*) - \nabla f_i(x^*)\|^2\right],
$$

$$
\mathbf{E}\left[\|g^k\|^2 \mid x^k\right] \leq 4\mathcal{L}\left(f(x^k) - f(x^*)\right) + \frac{2}{n^2} \sum_{i=1}^n \mathbf{E}_{\mathcal{D}}\left[\|\nabla f_{\xi_i}(x^*) - \nabla f_i(x^*)\|^2\right].
$$

*Proof.* Applying straightforward inequality $\|a + b\|^2 \leq 2\|a\|^2 + 2\|b\|^2$ for $a, b \in \mathbb{R}^d$ we get

$$
\begin{aligned}
\frac{1}{n} \sum_{i=1}^n \|\bar{g}_i^k\|^2 &= \frac{1}{n} \sum_{i=1}^n \|\nabla f_i(x^k) - \nabla f_i(x^*) + \nabla f_i(x^*)\|^2 \\
&\overset{(31)}{\leq} \frac{1}{n} \sum_{i=1}^n \|\nabla f_i(x^k) - \nabla f_i(x^*)\|^2 + \frac{2}{n} \sum_{i=1}^n \|\nabla f_i(x^*)\|^2 \\
&\overset{(11)}{\leq} 4L\left(f(x^k) - f(x^*)\right) + \frac{2}{n} \sum_{i=1}^n \|\nabla f_i(x^*)\|^2.
\end{aligned} \tag{97}
$$

Similarly we obtain

$$
\begin{aligned}
\frac{1}{n}\sum_{i=1}^{n}\mathbf{E}\left[\|g_i^k - \bar{g}_i^k\|^2 \mid x^k\right] &= \frac{1}{n}\sum_{i=1}^{n}\mathbf{E}_{\mathcal{D}}\left[\|\nabla f_{\xi_i}(x^k) - \nabla f_i(x^k)\|^2\right] \\
&\stackrel{(31)}{\leq} \frac{3}{n}\sum_{i=1}^{n}\mathbf{E}_{\mathcal{D}}\left[\|\nabla f_{\xi_i}(x^k) - \nabla f_{\xi_i}(x^*)\|^2\right] \\
&\quad + \frac{3}{n}\sum_{i=1}^{n}\mathbf{E}_{\mathcal{D}}\left[\|\nabla f_{\xi_i}(x^*) - \nabla f_i(x^*)\|^2\right] \\
&\quad + \frac{3}{n}\sum_{i=1}^{n}\|\nabla f_i(x^*) - \nabla f_i(x^k)\|^2 \\
&\stackrel{(11),(89)}{\leq} 6(\mathcal{L} + L)\left(f(x^k) - f(x^*)\right) \\
&\quad + \frac{3}{n}\sum_{i=1}^{n}\mathbf{E}_{\mathcal{D}}\left[\|\nabla f_{\xi_i}(x^*) - \nabla f_i(x^*)\|^2\right].
\end{aligned}
$$

Next, using the independence of $\xi_1^k, \ldots, \xi_n^k$ we derive

$$
\begin{aligned}
\mathbf{E}\left[\|g^k\|^2 \mid x^k\right] &= \mathbf{E}\left[\left\|\frac{1}{n}\sum_{i=1}^{n}\left(\nabla f_{\xi_i^k}(x^k) - \nabla f_{\xi_i^k}(x^*) + \nabla f_{\xi_i^k}(x^*) - \nabla f_i(x^*)\right)\right\|^2 \mid x^k\right] \\
&\stackrel{(31)}{\leq} \frac{2}{n}\sum_{i=1}^{n}\mathbf{E}\left[\left\|\nabla f_{\xi_i^k}(x^k) - \nabla f_{\xi_i^k}(x^*)\right\|^2 \mid x^k\right] \\
&\quad + 2\mathbf{E}\left[\left\|\frac{1}{n}\sum_{i=1}^{n}\left(\nabla f_{\xi_i^k}(x^*) - \nabla f_i(x^*)\right)\right\|^2 \mid x^k\right] \\
&\stackrel{(89)}{\leq} 4\mathcal{L}\left(f(x^k) - f(x^*)\right) + \frac{2}{n^2}\sum_{i=1}^{n}\mathbf{E}_{\mathcal{D}_i}\left[\|\nabla f_{\xi_i}(x^*) - \nabla f_i(x^*)\|^2\right].
\end{aligned}
$$

$\square$

Applying Theorem G.1 we get the following result.

**Theorem J.1.** *Assume that $f(x)$ is $\mu$-quasi strongly convex, $f_1, \ldots, f_n$ are $L$-smooth and Assumption I.1 holds. Then* EC-SGDsr *satisfies Assumption 3.3 with*

$$
A = 2L, \quad \widetilde{A} = 3(\mathcal{L} + L), \quad A' = 2\mathcal{L}, \quad B_1 = \widetilde{B}_1 = B_1' = B_2 = \widetilde{B}_2 = B_2' = 0,
$$

$$
D_1 = \frac{2}{n}\sum_{i=1}^{n}\|\nabla f_i(x^*)\|^2, \quad \widetilde{D}_1 = \frac{3}{n}\sum_{i=1}^{n}\mathbf{E}_{\mathcal{D}}\left[\|\nabla f_{\xi_i}(x^*) - \nabla f_i(x^*)\|^2\right], \quad \sigma_{1,k}^2 \equiv \sigma_{2,k}^2 \equiv 0,
$$

$$
D_1' = \frac{2}{n^2}\sum_{i=1}^{n}\mathbf{E}_{\mathcal{D}}\left[\|\nabla f_{\xi_i}(x^*) - \nabla f_i(x^*)\|^2\right], \quad \rho_1 = \rho_2 = 1, \quad C_1 = C_2 = 0, \quad G = 0, \quad D_2 = 0,
$$

$$
F_1 = F_2 = 0, \quad D_3 = \frac{6L\gamma}{\delta}\left(\frac{2D_1}{\delta} + \widetilde{D}_1\right),
$$

*with $\gamma$ satisfying*

$$
\gamma \leq \min\left\{\frac{1}{8\mathcal{L}}, \frac{\delta}{4\sqrt{6L\left(4L + 3\delta(\mathcal{L} + L)\right)}}\right\}
$$

*and for all $K \geq 0$*

$$
\mathbf{E}\left[f(\bar{x}^K) - f(x^*)\right] \leq \left(1 - \frac{\gamma\mu}{2}\right)^K \frac{4\|x^0 - x^*\|^2}{\gamma} + 4\gamma\left(D_1' + \frac{12L\gamma}{\delta^2}D_1 + \frac{6L\gamma}{\delta}\widetilde{D}_1\right)
$$

*when $\mu > 0$ and*

$$\mathbf{E}\left[f(\bar{x}^K) - f(x^*)\right] \leq \frac{4\|x^0 - x^*\|^2}{K\gamma} + 4\gamma\left(D_1' + \frac{12L\gamma}{\delta^2}D_1 + \frac{6L\gamma}{\delta}\widetilde{D}_1\right)$$

*when $\mu = 0$.*

In other words, `EC-SGDsr` converges with linear rate $\mathcal{O}\left(\left(\frac{\mathcal{L}}{\mu} + \frac{L+\sqrt{\delta L \mathcal{L}}}{\mu\delta}\right)\ln\frac{1}{\varepsilon}\right)$ to the neighbourhood of the solution when $\mu > 0$. Applying Lemma D.2 we establish the rate of convergence to $\varepsilon$-solution.

**Corollary J.1.** *Let the assumptions of Theorem J.1 hold and $\mu > 0$. Then after $K$ iterations of* `EC-SGDsr` *with the stepsize*

$$\gamma = \min\left\{\frac{1}{8\mathcal{L}}, \frac{\delta}{4\sqrt{6L\left(4L + 3\delta(\mathcal{L}+L)\right)}}, \frac{\ln\left(\max\left\{2, \min\left\{\frac{\|x^0-x^*\|^2\mu^2 K^2}{D_1'}, \frac{\delta\|x^0-x^*\|^2\mu^3 K^3}{6L(2D_1/\delta+\widetilde{D}_1)}\right\}\right\}\right)}{\mu K}\right\}$$

*we have $\mathbf{E}\left[f(\bar{x}^K) - f(x^*)\right]$ of order*

$$\widetilde{\mathcal{O}}\left(\left(\mathcal{L} + \frac{L+\sqrt{\delta L\mathcal{L}}}{\delta}\right)\|x^0 - x^*\|^2 \exp\left(-\frac{\mu}{\mathcal{L} + \frac{L+\sqrt{\delta L\mathcal{L}}}{\delta}}K\right) + \frac{D_1'}{\mu K} + \frac{L(\widetilde{D}_1 + {}^{D_1}/\delta)}{\delta\mu^2 K^2}\right).$$

*That is, to achive $\mathbf{E}\left[f(\bar{x}^K) - f(x^*)\right] \leq \varepsilon$* `EC-SGDsr` *requires*

$$\widetilde{\mathcal{O}}\left(\frac{\mathcal{L}}{\mu} + \frac{L + \sqrt{\delta L\mathcal{L}}}{\delta\mu} + \frac{D_1'}{\mu\varepsilon} + \frac{\sqrt{L(\widetilde{D}_1 + {}^{D_1}/\delta)}}{\mu\sqrt{\delta\varepsilon}}\right) \quad iterations.$$

Applying Lemma D.3 we get the complexity result in the case when $\mu = 0$.

**Corollary J.2.** *Let the assumptions of Theorem J.1 hold and $\mu = 0$. Then after $K$ iterations of* `EC-SGDsr` *with the stepsize*

$$\gamma_0 = \min\left\{\frac{1}{8\mathcal{L}}, \frac{\delta}{4\sqrt{6L\left(4L + 3\delta(\mathcal{L}+L)\right)}}\right\}$$

$$\gamma = \min\left\{\gamma_0, \sqrt{\frac{\|x^0-x^*\|^2}{D_1' K}}, \sqrt[3]{\frac{\|x^0-x^*\|^2\delta}{6L(2D_1/\delta + \widetilde{D}_1)K}}\right\}$$

*we have $\mathbf{E}\left[f(\bar{x}^K) - f(x^*)\right]$ of order*

$$\mathcal{O}\left(\frac{R_0^2\left(\mathcal{L} + \frac{L+\sqrt{\delta L\mathcal{L}}}{\delta}\right)}{K} + \sqrt{\frac{R_0^2 D_1'}{K}} + \frac{\sqrt[3]{LR_0^4(2D_1/\delta + \widetilde{D}_1)}}{(\delta K^2)^{1/3}}\right)$$

*where $R_0 = \|x^0 - x^*\|^2$. That is, to achive $\mathbf{E}\left[f(\bar{x}^K) - f(x^*)\right] \leq \varepsilon$* `EC-SGDsr` *requires*

$$\mathcal{O}\left(\frac{R_0^2\left(\mathcal{L} + \frac{L+\sqrt{\delta L\mathcal{L}}}{\delta}\right)}{\varepsilon} + \frac{R_0^2 D_1'}{\varepsilon^2} + \frac{R_0^2\sqrt{L(2D_1/\delta + \widetilde{D}_1)}}{\sqrt{\delta\varepsilon^3}}\right)$$

*iterations.*

## J.2 `EC-SGD`

In this section we consider problem (1) with $f_i(x)$ satisfying (2) where functions $f_{\xi_i}(x)$ are differentiable and $L$-smooth almost surely in $\xi_i$, $i = 1, \ldots, n$.

---
**Algorithm 4** EC-SGD
---
**Input:** learning rate $\gamma > 0$, initial vector $x^0 \in \mathbb{R}^d$

1: Set $e_i^0 = 0$ for all $i = 1, \ldots, n$
2: **for** $k = 0, 1, \ldots$ **do**
3:     Broadcast $x^k$ to all workers
4:     **for** $i = 1, \ldots, n$ in parallel **do**
5:         Sample $g_i^k = \nabla f_{\xi_i}(x^k)$ independently from other workers
6:         $v_i^k = C(e_i^k + \gamma g_i^k)$
7:         $e_i^{k+1} = e_i^k + \gamma g_i^k - v_i^k$
8:     **end for**
9:     $e^k = \frac{1}{n} \sum_{i=1}^n e_i^k$, $g^k = \frac{1}{n} \sum_{i=1}^n g_i^k$, $v^k = \frac{1}{n} \sum_{i=1}^n v_i^k$
10:    $x^{k+1} = x^k - v^k$
11: **end for**
---

**Lemma J.2** (See also Lemmas 1,2 from [39]). *Assume that $f_{\xi_i}(x)$ are convex in $x$ for every $\xi_i$, $i = 1, \ldots, n$. Then for every $x \in \mathbb{R}^d$ and $i = 1, \ldots, n$*

$$\frac{1}{n} \sum_{i=1}^n \|\nabla f_i(x)\|^2 \leq 4L \left( f(x) - f(x^*) \right) + \frac{2}{n} \sum_{i=1}^n \|\nabla f_i(x^*)\|^2,$$

$$\frac{1}{n} \sum_{i=1}^n \mathbf{E}_{\xi_i \sim \mathcal{D}_i} \|\nabla f_{\xi_i}(x) - \nabla f_i(x)\|^2 \leq 12L \left( f(x) - f(x^*) \right) + \frac{3}{n} \sum_{i=1}^n \mathbf{E} \left[ \|\nabla f_{\xi_i}(x^*) - \nabla f_i(x^*)\|^2 \right],$$

$$\mathbf{E}_{\xi_1, \ldots, \xi_n} \left\| \frac{1}{n} \sum_{i=1}^n \nabla f_{\xi_i}(x) \right\|^2 \leq 4L \left( f(x) - f(x^*) \right) + \frac{2}{n^2} \sum \mathbf{E} \left[ \|\nabla f_{\xi_i}(x^*) - \nabla f_i(x^*)\|^2 \right].$$

*If further $f(x)$ is $\mu$-strongly convex with $\mu > 0$ and possibly non-convex $f_i, f_{\xi_i}$, then for every $x \in \mathbb{R}^d$ and $i = 1, \ldots, n$*

$$\frac{1}{n} \sum_{i=1}^n \|\nabla f_i(x)\|^2 \leq 4L\kappa \left( f(x) - f(x^*) \right) + \frac{2}{n} \sum_{i=1}^n \|\nabla f_i(x^*)\|^2,$$

$$\frac{1}{n} \sum_{i=1}^n \mathbf{E}_{\xi_i \sim \mathcal{D}_i} \|\nabla f_{\xi_i}(x) - \nabla f_i(x)\|^2 \leq 12L\kappa \left( f(x) - f(x^*) \right) + \frac{3}{n} \sum_{i=1}^n \mathbf{E} \left[ \|\nabla f_{\xi_i}(x^*) - \nabla f_i(x^*)\|^2 \right],$$

$$\mathbf{E}_{\xi_1, \ldots, \xi_n} \left\| \frac{1}{n} \sum_{i=1}^n \nabla f_{\xi_i}(x) \right\|^2 \leq 4L\kappa \left( f(x) - f(x^*) \right) + \frac{2}{n^2} \sum_{i=1}^n \mathbf{E} \left[ \|\nabla f_{\xi_i}(x^*) - \nabla f_i(x^*)\|^2 \right].$$

*where $\kappa = \frac{L}{\mu}$.*

*Proof.* We start with the case when functions $f_{\xi_i}(x)$ are convex in $x$ for every $\xi_i$. The first inequality follows from (97). Next, we derive

$$\frac{1}{n} \sum_{i=1}^n \mathbf{E}_{\xi_i \sim \mathcal{D}_i} \|\nabla f_{\xi_i}(x) - \nabla f_i(x)\|^2 \overset{(31)}{\leq} \frac{3}{n} \sum_{i=1}^n \mathbf{E}_{\xi_i \sim \mathcal{D}_i} \|\nabla f_{\xi_i}(x) - \nabla f_{\xi_i}(x^*)\|^2$$

$$+ \frac{3}{n} \sum_{i=1}^n \mathbf{E}_{\xi_i \sim \mathcal{D}_i} \|\nabla f_{\xi_i}(x^*) - \nabla f_i(x^*)\|^2$$

$$+ \frac{3}{n} \sum_{i=1}^n \|\nabla f_i(x^*) - \nabla f_i(x)\|^2$$

$$\overset{(11)}{\leq} 12L \left( f(x) - f(x^*) \right) + \frac{3}{n} \sum_{i=1}^n \mathbf{E} \|\nabla f_{\xi_i}(x^*)\|^2.$$

Due to independence of $\xi_1^k, \ldots, \xi_n^k$ we get

$$
\begin{aligned}
\mathbf{E}_{\xi_1,\ldots,\xi_n} \left\| \frac{1}{n} \sum_{i=1}^n \nabla f_{\xi_i}(x) \right\|^2 &= \mathbf{E}_{\xi_1,\ldots,\xi_n} \left\| \frac{1}{n} \sum_{i=1}^n \left( \nabla f_{\xi_i}(x) - \nabla f_{\xi_i}(x^*) + \nabla f_{\xi_i}(x^*) - \nabla f_i(x^*) \right) \right\|^2 \\
&\overset{(31)}{\leq} \frac{2}{n} \sum_{i=1}^n \mathbf{E}_{\xi_i \sim \mathcal{D}_i} \left[ \| \nabla f_{\xi_i}(x) - \nabla f_{\xi_i}(x^*) \|^2 \right] \\
&\quad + 2 \mathbf{E}_{\xi_1,\ldots,\xi_n} \left\| \frac{1}{n} \sum_{i=1}^n \left( \nabla f_{\xi_i}(x^*) - \nabla f_i(x^*) \right) \right\|^2 \\
&\overset{(11)}{\leq} 4L \left( f(x) - f(x^*) \right) + \frac{2}{n^2} \sum \mathbf{E} \left[ \| \nabla f_{\xi_i}(x^*) - \nabla f_i(x^*) \|^2 \right].
\end{aligned}
$$

Next, we consider the second case: $f(x)$ is $\mu$-strongly convex with possibly non-convex $f_i, f_{\xi_i}$. In this case

$$
\begin{aligned}
\frac{1}{n} \sum_{i=1}^n \| \nabla f_i(x) \|^2 &\overset{(31)}{\leq} \frac{2}{n} \sum_{i=1}^n \| \nabla f_i(x) - \nabla f_i(x^*) \|^2 + \frac{2}{n} \sum_{i=1}^n \| \nabla f_i(x^*) \|^2 \\
&\overset{(10)}{\leq} 2L^2 \| x - x^* \|^2 + \frac{2}{n} \sum_{i=1}^n \| \nabla f_i(x^*) \|^2 \\
&\leq \frac{4L^2}{\mu} \left( f(x) - f(x^*) \right) + \frac{2}{n} \sum_{i=1}^n \| \nabla f_i(x^*) \|^2
\end{aligned}
$$

where the last inequality follows from $\mu$-strong convexity of $f$. Similarly, we get

$$
\begin{aligned}
\frac{1}{n} \sum_{i=1}^n \mathbf{E}_{\xi_i \sim \mathcal{D}_i} \left[ \| \nabla f_{\xi_i}(x) - \nabla f_i(x) \|^2 \right] &\overset{(31)}{\leq} \frac{3}{n} \sum_{i=1}^n \mathbf{E}_{\xi_i \sim \mathcal{D}_i} \left[ \| \nabla f_{\xi_i}(x) - \nabla f_{\xi_i}(x^*) \|^2 \right] \\
&\quad + \frac{3}{n} \sum_{i=1}^n \mathbf{E}_{\xi_i \sim \mathcal{D}_i} \left[ \| \nabla f_{\xi_i}(x^*) - \nabla f_i(x^*) \|^2 \right] \\
&\quad + \frac{3}{n} \sum_{i=1}^n \| \nabla f_i(x^*) - \nabla f_i(x) \|^2 \\
&\overset{(10)}{\leq} 6L^2 \| x - x^* \|^2 \\
&\quad + \frac{3}{n} \sum_{i=1}^n \mathbf{E}_{\xi_i \sim \mathcal{D}_i} \left[ \| \nabla f_{\xi_i}(x^*) - \nabla f_i(x^*) \|^2 \right] \\
&\leq \frac{12L^2}{\mu} \left( f(x) - f(x^*) \right) \\
&\quad + \frac{3}{n} \sum_{i=1}^n \mathbf{E}_{\xi_i \sim \mathcal{D}_i} \left[ \| \nabla f_{\xi_i}(x^*) - \nabla f_i(x^*) \|^2 \right].
\end{aligned}
$$

Finally, using independence of $\xi_1^k, \dots, \xi_n^k$ we derive

$$
\mathbf{E}_{\xi_1,\dots,\xi_n} \left\| \frac{1}{n} \sum_{i=1}^n \nabla f_{\xi_i}(x) \right\|^2 = \mathbf{E}_{\xi_1,\dots,\xi_n} \left\| \frac{1}{n} \sum_{i=1}^n \left( \nabla f_{\xi_i}(x) - \nabla f_{\xi_i}(x^*) + \nabla f_{\xi_i}(x^*) - \nabla f_i(x^*) \right) \right\|^2
$$

$$
\overset{(31)}{\leq} \frac{2}{n} \sum_{i=1}^n \mathbf{E}_{\xi_i \sim \mathcal{D}_i} \left[ \| \nabla f_{\xi_i}(x) - \nabla f_{\xi_i}(x^*) \|^2 \right]
$$

$$
+ 2 \mathbf{E}_{\xi_1,\dots,\xi_n} \left\| \frac{1}{n} \sum_{i=1}^n \left( \nabla f_{\xi_i}(x^*) - \nabla f_i(x^*) \right) \right\|^2
$$

$$
\overset{(10)}{\leq} 2L^2 \|x - x^*\|^2 + \frac{2}{n^2} \sum \mathbf{E} \left[ \| \nabla f_{\xi_i}(x^*) - \nabla f_i(x^*) \|^2 \right]
$$

$$
\leq \frac{4L^2}{\mu} \left( f(x) - f(x^*) \right) + \frac{2}{n^2} \sum \mathbf{E} \left[ \| \nabla f_{\xi_i}(x^*) - \nabla f_i(x^*) \|^2 \right].
$$

$\square$

Applying Theorem G.1 we get the following result.

**Theorem J.2.** *Assume that $f_\xi(x)$ is convex and L-smooth in x for every $\xi$ and $f(x)$ is $\mu$-quasi strongly convex. Then* `EC-SGD` *satisfies Assumption 3.3 with*

$$
A = A' = 2L, \quad \widetilde{A} = 6L, \quad B_1 = \widetilde{B}_1 = B_1' = B_2 = \widetilde{B}_2 = B_2' = 0,
$$

$$
D_1 = \frac{2}{n} \sum_{i=1}^n \| \nabla f_i(x^*) \|^2, \quad \widetilde{D}_1 = \frac{2}{n} \sum_{i=1}^n \mathbf{E} \left[ \| \nabla f_{\xi_i}(x^*) - \nabla f_i(x^*) \|^2 \right], \quad \sigma_{1,k}^2 \equiv \sigma_{2,k}^2 \equiv 0,
$$

$$
D_1' = \frac{2}{n^2} \sum_{i=1}^n \mathbf{E} \left[ \| \nabla f_{\xi_i}(x^*) - \nabla f_i(x^*) \|^2 \right], \quad \rho_1 = \rho_2 = 1, \quad C_1 = C_2 = 0, \quad G = 0, \quad D_2 = 0,
$$

$$
F_1 = F_2 = 0, \quad D_3 = \frac{6L\gamma}{\delta} \left( \frac{2D_1}{\delta} + \widetilde{D}_1 \right),
$$

*with $\gamma$ satisfying*

$$
\gamma \leq \frac{\delta}{8L\sqrt{6 + 9\delta}}
$$

*and for all $K \geq 0$*

$$
\mathbf{E} \left[ f(\bar{x}^K) - f(x^*) \right] \leq \left( 1 - \frac{\gamma \mu}{2} \right)^K \frac{4\|x^0 - x^*\|^2}{\gamma} + 4\gamma \left( D_1' + \frac{12L\gamma}{\delta^2} D_1 + \frac{6L\gamma}{\delta} \widetilde{D}_1 \right)
$$

*when $\mu > 0$ and*

$$
\mathbf{E} \left[ f(\bar{x}^K) - f(x^*) \right] \leq \frac{4\|x^0 - x^*\|^2}{K\gamma} + 4\gamma \left( D_1' + \frac{12L\gamma}{\delta^2} D_1 + \frac{6L\gamma}{\delta} \widetilde{D}_1 \right)
$$

*when $\mu = 0$. If further $f(x)$ is $\mu$-strongly convex with $\mu > 0$ and possibly non-convex $f_i, f_{\xi_i}$, then* `EC-SGD` *satisfies Assumption 3.3 with*

$$
A = A' = 2L\kappa, \quad \widetilde{A} = 6L\kappa, \quad B_1 = \widetilde{B}_1 = B_1' = B_2 = \widetilde{B}_2 = B_2' = 0,
$$

$$
D_1 = \frac{2}{n} \sum_{i=1}^n \| \nabla f_i(x^*) \|^2, \quad \widetilde{D}_1 = \frac{2}{n} \sum_{i=1}^n \mathbf{E} \left[ \| \nabla f_{\xi_i}(x^*) - \nabla f_i(x^*) \|^2 \right], \quad \sigma_{1,k}^2 \equiv \sigma_{2,k}^2 \equiv 0,
$$

$$
D_1' = \frac{2}{n^2} \sum_{i=1}^n \mathbf{E} \left[ \| \nabla f_{\xi_i}(x^*) - \nabla f_i(x^*) \|^2 \right], \quad \rho_1 = \rho_2 = 1, \quad C_1 = C_2 = 0, \quad G = 0, \quad D_2 = 0,
$$

$$
F_1 = F_2 = 0, \quad D_3 = \frac{6L\gamma}{\delta} \left( \frac{2D_1}{\delta} + \widetilde{D}_1 \right),
$$

*with $\gamma$ satisfying*

$$\gamma \le \min\left\{\frac{1}{8\kappa L}, \frac{\delta}{8L\sqrt{3\kappa(2+3\delta)}}\right\}$$

*and for all $K \ge 0$*

$$\mathbf{E}\left[f(\bar{x}^K) - f(x^*)\right] \le \left(1 - \frac{\gamma\mu}{2}\right)^K \frac{4\|x^0 - x^*\|^2}{\gamma} + 4\gamma\left(D_1' + \frac{12L\gamma}{\delta^2}D_1 + \frac{6L\gamma}{\delta}\widetilde{D}_1\right).$$

In other words, `EC-SGD` converges with linear rate $\mathcal{O}\left(\frac{\kappa}{\delta}\ln\frac{1}{\varepsilon}\right)$ to the neighbourhood of the solution when $f_\xi(x)$ are convex for each $\xi$ and $\mu > 0$. Applying Lemma D.2 we establish the rate of convergence to $\varepsilon$-solution.

**Corollary J.3.** *Let the assumptions of Theorem J.2 hold, $f_\xi(x)$ are convex for each $\xi$ and $\mu > 0$. Then after $K$ iterations of* `EC-SGD` *with the stepsize*

$$\gamma = \min\left\{\frac{\delta}{8L\sqrt{6+9\delta}}, \frac{\ln\left(\max\left\{2, \min\left\{\frac{\|x^0-x^*\|^2\mu^2 K^2}{D_1'}, \frac{\delta\|x^0-x^*\|^2\mu^3 K^3}{6L(2D_1/\delta+\widetilde{D}_1)}\right\}\right\}\right)}{\mu K}\right\}$$

*we have*

$$\mathbf{E}\left[f(\bar{x}^K) - f(x^*)\right] = \widetilde{\mathcal{O}}\left(\frac{L}{\delta}\|x^0 - x^*\|^2 \exp\left(-\frac{\delta\mu}{L}K\right) + \frac{D_1'}{\mu K} + \frac{L(\widetilde{D}_1 + D_1/\delta)}{\delta\mu^2 K^2}\right).$$

*That is, to achive $\mathbf{E}\left[f(\bar{x}^K) - f(x^*)\right] \le \varepsilon$* `EC-SGD` *requires*

$$\widetilde{\mathcal{O}}\left(\frac{L}{\delta\mu} + \frac{D_1'}{\mu\varepsilon} + \frac{\sqrt{L(\widetilde{D}_1 + D_1/\delta)}}{\mu\sqrt{\delta\varepsilon}}\right) \quad \text{iterations.}$$

**Corollary J.4.** *Let the assumptions of Theorem J.2 hold and $f(x)$ is $\mu$-strongly convex with $\mu > 0$ and possibly non-convex $f_i, f_{\xi_i}$. Then after $K$ iterations of* `EC-SGD` *with the stepsize*

$$\gamma = \min\left\{\frac{1}{8\kappa L}, \frac{\delta}{8L\sqrt{3\kappa(2+3\delta)}}, \frac{\ln\left(\max\left\{2, \min\left\{\frac{\|x^0-x^*\|^2\mu^2 K^2}{D_1'}, \frac{\delta\|x^0-x^*\|^2\mu^3 K^3}{6L(2D_1/\delta+\widetilde{D}_1)}\right\}\right\}\right)}{\mu K}\right\}$$

*we have $\mathbf{E}\left[f(\bar{x}^K) - f(x^*)\right]$ of order*

$$\widetilde{\mathcal{O}}\left(\left(L\kappa + \frac{L\sqrt{\kappa}}{\delta}\right)\|x^0 - x^*\|^2 \exp\left(-\min\left\{\frac{\delta\mu}{L\sqrt{\kappa}}, \frac{1}{\kappa^2}\right\}K\right) + \frac{D_1'}{\mu K} + \frac{L(\widetilde{D}_1 + D_1/\delta)}{\delta\mu^2 K^2}\right).$$

*That is, to achive $\mathbf{E}\left[f(\bar{x}^K) - f(x^*)\right] \le \varepsilon$* `EC-SGD` *requires*

$$\widetilde{\mathcal{O}}\left(\kappa^2 + \frac{\kappa^{3/2}}{\delta} + \frac{D_1'}{\mu\varepsilon} + \frac{\sqrt{L(\widetilde{D}_1 + D_1/\delta)}}{\mu\sqrt{\delta\varepsilon}}\right) \quad \text{iterations.}$$

Applying Lemma D.3 we get the complexity result in the case when $\mu = 0$.

**Corollary J.5.** *Let the assumptions of Theorem J.2 hold, $f_\xi(x)$ are convex for each $\xi$ and $\mu = 0$. Then after $K$ iterations of* `EC-SGD` *with the stepsize*

$$\gamma = \min\left\{\frac{\delta}{8L\sqrt{6+9\delta}}, \sqrt{\frac{\|x^0 - x^*\|^2}{D_1' K}}, \sqrt[3]{\frac{\|x^0 - x^*\|^2\delta}{6L(2D_1/\delta + \widetilde{D}_1)K}}\right\}$$

*we have $\mathbf{E}\left[f(\bar{x}^K) - f(x^*)\right]$ of order*

$$\mathcal{O}\left(\frac{LR_0^2}{\delta K} + \sqrt{\frac{R_0^2 D_1'}{K}} + \frac{\sqrt[3]{LR_0^4(2D_1/\delta + \widetilde{D}_1)}}{(\delta K^2)^{1/3}}\right)$$

**Algorithm 5** `EC-GDstar` (see also [11])

---

**Input:** learning rate $\gamma > 0$, initial vector $x^0 \in \mathbb{R}^d$
1: Set $e_i^0 = 0$ for all $i = 1, \dots, n$
2: **for** $k = 0, 1, \dots$ **do**
3:      Broadcast $x^k$ to all workers
4:      **for** $i = 1, \dots, n$ in parallel **do**
5:          $g_i^k = \nabla f_i(x^k) - \nabla f_i(x^*)$
6:          $v_i^k = C(e_i^k + \gamma g_i^k)$
7:          $e_i^{k+1} = e_i^k + \gamma g_i^k - v_i^k$
8:      **end for**
9:      $e^k = \frac{1}{n} \sum_{i=1}^n e_i^k$, $g^k = \frac{1}{n} \sum_{i=1}^n g_i^k$, $v^k = \frac{1}{n} \sum_{i=1}^n v_i^k$
10:    $x^{k+1} = x^k - v^k$
11: **end for**

---

where $R_0 = \|x^0 - x^*\|^2$. That is, to achive $\mathbf{E}\left[f(\bar{x}^K) - f(x^*)\right] \leq \varepsilon$ `EC-SGD` requires

$$\mathcal{O}\left(\frac{LR_0^2}{\delta\varepsilon} + \frac{R_0^2 D_1'}{\varepsilon^2} + \frac{R_0^2 \sqrt{L(2D_1/\delta + \widetilde{D}_1)}}{\sqrt{\delta\varepsilon^3}}\right)$$

*iterations.*

### J.3   `EC-GDstar`

We assume that $i$-th node has access to the gradient of $f_i$ at the optimality, i.e., to the $\nabla f_i(x^*)$. It is unrealistic scenario but it gives some insights that we will use next in order to design the method that converges asymptotically to *the exact solution.*

Assume that $f(x)$ is $\mu$-quasi strongly convex and each $f_i$ is convex and $L$-smooth. By definition of $g_i^k$ it trivially follows that

$$g^k = \frac{1}{n} \sum_{i=1}^n g_i^k = \frac{1}{n} \sum_{i=1}^n \left(\nabla f_i(x^k) - \nabla f_i(x^*)\right) = \nabla f(x^k) - \nabla f(x^*) = \nabla f(x^k),$$

$g_i^k = \bar{g}_i^k$, and

$$
\begin{aligned}
\frac{1}{n} \sum_{i=1}^n \|g_i^k\|^2 \;&=\; \frac{1}{n} \sum_{i=1}^n \|\nabla f_i(x^k) - \nabla f_i(x^*)\|^2 \\
&\overset{(11)}{\leq}\; \frac{2L}{n} \sum_{i=1}^n \left(f_i(x^k) - f_i(x^*) - \langle \nabla f_i(x^*), x^k - x^* \rangle\right) = 2L\left(f(x^k) - f(x^*)\right), \\
\|g^k\|^2 \;&=\; \|\nabla f(x^k)\|^2 \overset{(11)}{\leq} 2L\left(f(x^k) - f(x^*)\right).
\end{aligned}
$$

Applying Theorem G.1 we get the following result.

**Theorem J.3.** *Assume that $f_i(x)$ is convex and $L$-smooth for all $i = 1, \dots, n$ and $f(x)$ is $\mu$-quasi strongly convex. Then* `EC-GDstar` *satisfies Assumption 3.3 with*

$$A = A' = L, \quad \widetilde{A} = 0, \quad B_1 = B_2 = \widetilde{B}_1 = \widetilde{B}_2 = B_1' = B_2' = 0,$$

$$D_1 = \widetilde{D}_1 = D_1' = 0, \quad \sigma_{1,k}^2 \equiv \sigma_{2,k}^2 \equiv 0,$$

$$\rho_1 = \rho_2 = 1, \quad C_1 = C_2 = 0, \quad G = 0, \quad D_2 = 0, \quad F_1 = F_2 = 0, \quad D_3 = 0,$$

*with $\gamma$ satisfying*

$$\gamma \leq \frac{\delta}{8L\sqrt{3}}$$

*and for all $K \geq 0$*

$$\mathbf{E}\left[f(\bar{x}^K) - f(x^*)\right] \leq \left(1 - \frac{\gamma\mu}{2}\right)^K \frac{4\|x^0 - x^*\|^2}{\gamma},$$

**Algorithm 6** `EC-SGD-DIANA`

---

**Input:** learning rates $\gamma > 0$, $\alpha \in (0, 1]$, initial vectors $x^0, h_1^0, \ldots, h_n^0 \in \mathbb{R}^d$
1: Set $e_i^0 = 0$ for all $i = 1, \ldots, n$
2: Set $h^0 = \frac{1}{n} \sum_{i=1}^n h_i^0$
3: **for** $k = 0, 1, \ldots$ **do**
4:     Broadcast $x^k, h^k$ to all workers
5:     **for** $i = 1, \ldots, n$ in parallel **do**
6:         Sample $\hat{g}_i^k$ such that $\mathbf{E}[\hat{g}_i^k \mid x^k] = \nabla f_i(x^k)$ and $\mathbf{E}\left[\|\hat{g}_i^k - \nabla f_i(x^k)\|^2 \mid x^k\right] \leq \widetilde{D}_{1,i}$
    independently from other workers
7:         $g_i^k = \hat{g}_i^k - h_i^k + h^k$
8:         $v_i^k = C(e_i^k + \gamma g_i^k)$
9:         $e_i^{k+1} = e_i^k + \gamma g_i^k - v_i^k$
10:        $h_i^{k+1} = h_i^k + \alpha Q(\hat{g}_i^k - h_i^k)$
11:     **end for**
12:    $e^k = \frac{1}{n} \sum_{i=1}^n e_i^k, \; g^k = \frac{1}{n} \sum_{i=1}^n g_i^k, \; v^k = \frac{1}{n} \sum_{i=1}^n v_i^k, \; h^{k+1} = \frac{1}{n} \sum_{i=1}^n h_i^{k+1} = h^k + \alpha \frac{1}{n} \sum_{i=1}^n Q(\hat{g}_i^k - h_i^k)$
13:    $x^{k+1} = x^k - v^k$
14: **end for**

---

*when $\mu > 0$ and*

$$\mathbf{E}\left[f(\bar{x}^K) - f(x^*)\right] \leq \frac{4\|x^0 - x^*\|^2}{K\gamma}$$

*when $\mu = 0$.*

In other words, `EC-GDstar` converges with linear rate $\mathcal{O}\left(\frac{\kappa}{\delta} \ln \frac{1}{\varepsilon}\right)$ to the exact solution when $\mu > 0$ removing the drawback of `EC-SGD` and `EC-GD`. If $\mu = 0$ then the rate of convergence is $\mathcal{O}\left(\frac{L\|x^0 - x^*\|^2}{\delta\varepsilon}\right)$. However, `EC-GDstar` relies on the fact that $i$-th node knows $\nabla f_i(x^*)$ which is not realistic.

## J.4 `EC-SGD-DIANA`

In this section we present a new method that converges to the exact optimum asymptotically but does not need to know $\nabla f_i(x^*)$ and instead of this it learns the gradients at the optimum. This method is inspired by another method called `DIANA` (see [37, 19]).

We notice that master needs to gather only $C(e_i^k + \gamma g_i^k)$ and $Q(\hat{g}_i^k - h_i^k)$ from all nodes in order to perform an update.

**Lemma J.3.** *Assume that $f_i(x)$ is convex and $L$-smooth for all $i = 1, \ldots, n$. Then, for all $k \geq 0$ we have*

$$\mathbf{E}\left[g^k \mid x^k\right] = \nabla f(x^k), \tag{98}$$

$$\frac{1}{n} \sum_{i=1}^n \|\bar{g}_i^k\|^2 \leq 4L\left(f(x^k) - f(x^*)\right) + 2\sigma_k^2, \tag{99}$$

$$\frac{1}{n} \sum_{i=1}^n \mathbf{E}\left[\|g_i^k - \bar{g}_i^k\|^2 \mid x^k\right] \leq \widetilde{D}_1, \tag{100}$$

$$\mathbf{E}\left[\|g^k\|^2 \mid x^k\right] \leq 2L\left(f(x^k) - f(x^*)\right) + \frac{\widetilde{D}_1}{n} \tag{101}$$

*where $\widetilde{D}_1 = \frac{1}{n} \sum_{i=1}^n \widetilde{D}_{1,i}$ and $\sigma_k^2 = \frac{1}{n} \sum_{i=1}^n \|h_i^k - \nabla f(x^*)\|^2$.*

*Proof.* First of all, we show unbiasedness of $g^k$:

$$\mathbf{E}\left[g^k \mid x^k\right] = \frac{1}{n}\sum_{i=1}^n \mathbf{E}\left[g_i^k \mid x^k\right] = \frac{1}{n}\sum_{i=1}^n \left(\nabla f_i(x^k) - h_i^k + h^k\right) = \nabla f(x^k).$$

Next, we derive the upper bound for $\|\bar{g}_i^k\|^2$:

$$
\begin{aligned}
\|\bar{g}_i^k\|^2 &= \|\nabla f_i(x^k) - h_i^k - h^k\|^2 \\
&\overset{(31)}{\leq} 2\|\nabla f_i(x^k) - \nabla f_i(x^*)\|^2 + 2\left\|h_i^k - \nabla f_i(x^*) - \left(h^k + \nabla f(x^*)\right)\right\|^2 \\
&\overset{(11)}{\leq} 4L\left(f_i(x^k) - \nabla f_i(x^*) - \langle \nabla f_i(x^*), x^k - x^* \rangle\right) \\
&\quad + 2\left\|h_i^k - \nabla f_i(x^*) - \left(h^k + \nabla f(x^*)\right)\right\|^2.
\end{aligned}
$$

Summing up previous inequality for $i = 1, \ldots, n$ we get

$$
\begin{aligned}
\frac{1}{n}\sum_{i=1}^n \|\bar{g}_i^k\|^2 &\leq 4L(f(x^k) - f(x^*)) + \frac{2}{n}\sum_{i=1}^n \left\|h_i^k - \nabla f_i(x^*) - \left(\frac{1}{n}\sum_{i=1}^n (h_i^k - \nabla f_i(x^*))\right)\right\|^2 \\
&\overset{(34)}{\leq} 4L\left(f(x^k) - f(x^*)\right) + \frac{2}{n}\sum_{i=1}^n \|h_i^k - \nabla f(x^*)\|^2. \quad\quad (102)
\end{aligned}
$$

Using the unbiasedness of $\hat{g}_i^k$ we derive

$$\frac{1}{n}\sum_{i=1}^n \mathbf{E}\left[\|g_i^k - \bar{g}_i^k\|^2 \mid x^k\right] = \frac{1}{n}\sum_{i=1}^n \mathbf{E}\left[\|\hat{g}_i^k - \nabla f_i(x^k)\|^2 \mid x^k\right] \leq \frac{1}{n}\sum_{i=1}^n \widetilde{D}_{1,i} = \widetilde{D}_1.$$

Finally, we obtain the upper bound for the second moment of $g^k$ using the independence of $\hat{g}_1^k, \ldots, \hat{g}_n^k$:

$$
\begin{aligned}
\mathbf{E}\left[\|g^k\|^2 \mid x^k\right] &\overset{(34)}{=} \|\nabla f(x^k)\|^2 + \mathbf{E}\left[\|g^k - \nabla f(x^k)\|^2\right] \\
&\overset{(11)}{\leq} 2L(f(x^k) - f(x^*)) + \mathbf{E}\left[\left\|\frac{1}{n}\sum_{i=1}^n (\hat{g}_i^k - \nabla f_i(x^k))\right\|^2 \mid x^k\right] \\
&= 2L(f(x^k) - f(x^*)) + \frac{1}{n^2}\sum_{i=1}^n \mathbf{E}\left[\|\hat{g}_i^k - \nabla f_i(x^k)\|^2 \mid x^k\right] \\
&\leq 2L(f(x^k) - f(x^*)) + \frac{1}{n^2}\sum_{i=1}^n \widetilde{D}_{1,i}.
\end{aligned}
$$

$\square$

**Lemma J.4.** *Let assumptions of Lemma J.3 hold and $\alpha \leq 1/(\omega+1)$. Then, for all $k \geq 0$ we have*

$$\mathbf{E}\left[\sigma_{k+1}^2 \mid x^k\right] \leq (1-\alpha)\sigma_k^2 + 2L\alpha(f(x^k) - f(x^*)) + \alpha^2(\omega+1)\widetilde{D}_1, \quad\quad (103)$$

*where $\sigma_k^2 = \frac{1}{n}\sum_{i=1}^n \|h_i^k - \nabla f_i(x^*)\|^2$ and $\widetilde{D}_1 = \frac{1}{n}\sum_{i=1}^n \widetilde{D}_{1,i}$.*

*Proof.* For simplicity, we introduce new notation: $h_i^* \overset{\text{def}}{=} \nabla f_i(x^*)$. Using this we derive an upper bound for the second moment of $h_i^{k+1} - h_i^*$:

$$
\begin{aligned}
\mathbf{E}\left[\|h_i^{k+1} - h_i^*\|^2 \mid x^k\right] &= \mathbf{E}\left[\|h_i^k - h_i^* + \alpha Q(\hat{g}_i^k - h_i^k)\|^2 \mid x^k\right] \\
&\overset{(26)}{=} \|h_i^k - h_i^*\|^2 + 2\alpha\langle h_i^k - h_i^*, \nabla f_i(x^k) - h_i^k \rangle \\
&\quad + \alpha^2 \mathbf{E}\left[\|Q(\hat{g}_i^k - h_i^k)\|^2 \mid x^k\right] \\
&\overset{(26),(35)}{\leq} \|h_i^k - h_i^*\|^2 + 2\alpha\langle h_i^k - h_i^*, \nabla f_i(x^k) - h_i^k \rangle \\
&\quad + \alpha^2(\omega+1)\mathbf{E}\left[\|\hat{g}_i^k - h_i^k\|^2 \mid x^k\right].
\end{aligned}
$$

Using variance decomposition (34) and $\alpha \le {1}/{(\omega+1)}$ we get

$$\alpha^2(\omega+1)\mathbf{E}\left[\|\hat{g}_i^k - h_i^k\|^2 \mid x^k\right] \overset{(34)}{=} \alpha^2(\omega+1)\mathbf{E}\left[\|\hat{g}_i^k - \nabla f_i(x^k)\|^2 \mid x^k\right]$$
$$+\alpha^2(\omega+1)\|\nabla f_i(x^k) - h_i^k\|^2$$
$$\le \alpha^2(\omega+1)\widetilde{D}_{1,i} + \alpha\|\nabla f_i(x^k) - h_i^k\|^2.$$

Putting all together we obtain

$$\mathbf{E}\left[\|h_i^{k+1} - h_i^*\|^2 \mid x^k\right] \le \|h_i^k - h_i^*\|^2 + \alpha\left\langle\nabla f_i(x^k) - h_i^k, f_i(x^k) + h_i^k - 2h_i^*\right\rangle + \alpha^2(\omega+1)\widetilde{D}_{1,i}$$
$$\overset{(28)}{=} \|h_i^k - h_i^*\|^2 + \alpha\|\nabla f_i(x^k) - h_i^*\|^2 - \alpha\|h_i^k - h_i^*\|^2 + \alpha^2(\omega+1)\widetilde{D}_{1,i}$$
$$\overset{(11)}{\le} (1-\alpha)\|h_i^k - h_i^*\|^2 + 2L\alpha\left(f_i(x^k) - f_i(x^*) - \langle\nabla f_i(x^*), x^k - x^*\rangle\right)$$
$$+\alpha^2(\omega+1)\widetilde{D}_{1,i}.$$

Summing up the above inequality for $i = 1, \ldots, n$ we derive

$$\frac{1}{n}\sum_{i=1}^{n}\mathbf{E}\left[\|h_i^{k+1} - h_i^*\|^2 \mid x^k\right] \le \frac{1-\alpha}{n}\sum_{i=1}^{n}\|h_i^k - h_i^*\|^2 + 2L\alpha(f(x^k) - f(x^*)) + \frac{\alpha^2(\omega+1)}{n}\sum_{i=1}^{n}\widetilde{D}_{1,i}.$$

$\square$

Applying Theorem G.1 we get the following result.

**Theorem J.4.** *Assume that $f_i(x)$ is convex and $L$-smooth for all $i = 1, \ldots, n$ and $f(x)$ is $\mu$-quasi strongly convex. Then* EC-SGD-DIANA *satisfies Assumption 3.3 with*

$$A = 2L, \quad \widetilde{A} = 0, \quad A' = L, \quad B_1 = 2, \quad \widetilde{D}_1 = \frac{1}{n}\sum_{i=1}^{n}\widetilde{D}_{1,i}, \quad \sigma_{1,k}^2 = \sigma_k^2 = \frac{1}{n}\sum_{i=1}^{n}\|h_i^k - \nabla f_i(x^*)\|^2,$$

$$B_1' = B_2' = B_2 = \widetilde{B}_1 = \widetilde{B}_2 = 0, \quad \sigma_{2,k}^2 \equiv 0, \quad \rho_1 = \alpha, \quad \rho_2 = 1, \quad C_1 = L\alpha, \quad C_2 = 0, \quad D_1 = 0,$$

$$D_2 = \alpha^2(\omega+1)\widetilde{D}_1, \quad D_1' = \frac{D_1}{n}, \quad G = 0,$$

$$F_1 = \frac{96L\gamma^2}{\delta^2\alpha\left(1 - \min\left\{\frac{\gamma\mu}{2}, \frac{\alpha}{4}\right\}\right)}, \quad F_2 = 0, \quad D_3 = \frac{6L\gamma}{\delta}\left(\frac{4\alpha(\omega+1)}{\delta} + 1\right)\widetilde{D}_1,$$

*with $\gamma$ and $\alpha$ satisfying*

$$\gamma \le \min\left\{\frac{1}{4L}, \frac{\delta\sqrt{1-\alpha}}{8L\sqrt{6(3-\alpha)}}\right\}, \quad \alpha \le \frac{1}{\omega+1}, \quad M_1 = M_2 = 0$$

*and for all $K \ge 0$*

$$\mathbf{E}\left[f(\bar{x}^K) - f(x^*)\right] \le \left(1 - \min\left\{\frac{\gamma\mu}{2}, \frac{\alpha}{4}\right\}\right)^K \frac{4(\|x^0 - x^*\|^2 + \gamma F_1\sigma_0^2)}{\gamma} + 4\gamma\left(D_1' + D_3\right),$$

*when $\mu > 0$ and*

$$\mathbf{E}\left[f(\bar{x}^K) - f(x^*)\right] \le \frac{4(\|x^0 - x^*\|^2 + \gamma F_1\sigma_0^2)}{\gamma K} + 4\gamma\left(D_1' + D_3\right)$$

*when $\mu = 0$.*

In other words, if

$$\gamma = \min\left\{\frac{1}{4L}, \frac{\delta\sqrt{1-\alpha}}{8L\sqrt{6(3-\alpha)}}\right\}, \quad \alpha = \min\left\{\frac{1}{\omega+1}, \frac{1}{2}\right\}$$

and $\widetilde{D}_1 = 0$, i.e., $\hat{g}_i^k = \nabla f_i(x^k)$ almost surely (this is the setup of EC-GD-DIANA), EC-SGD-DIANA converges with the linear rate

$$\mathcal{O}\left(\left(\omega + \frac{\kappa}{\delta}\right)\ln\frac{1}{\varepsilon}\right)$$

to the exact solution. Applying Lemma D.2 we establish the rate of convergence to $\varepsilon$-solution in the case when $\mu > 0$.

**Corollary J.6.** *Let the assumptions of Theorem J.4 hold and $\mu > 0$. Then after $K$ iterations of* `EC-SGD-DIANA` *with the stepsize*

$$\gamma_0 = \min\left\{\frac{1}{4L}, \frac{\delta\sqrt{1-\alpha}}{8L\sqrt{6(3-\alpha)}}\right\}, \quad R_0 = \|x^0 - x^*\|, \quad \tilde{F}_1 = \frac{784L\gamma^2}{7\delta^2\alpha},$$

$$\gamma = \min\left\{\gamma_0, \frac{\ln\left(\max\left\{2, \min\left\{\frac{n\left(R_0^2 + \tilde{F}_1\gamma_0\sigma_{1,0}^2\right)\mu^2 K^2}{\widetilde{D}_1}, \frac{\delta\left(R_0^2 + \tilde{F}_1\gamma_0\sigma_{1,0}^2\right)\mu^3 K^3}{6L\widetilde{D}_1(4\alpha(\omega+1)/\delta+1)}\right\}\right\}\right)}{\mu K}\right\},$$

*and $\alpha \leq \frac{1}{\omega+1}$ we have*

$$\mathbf{E}\left[f(\bar{x}^K) - f(x^*)\right] = \widetilde{\mathcal{O}}\left(\frac{L}{\delta}R_0^2 \exp\left(-\min\left\{\frac{\delta\mu}{L}, \alpha\right\}K\right) + \frac{\widetilde{D}_1}{n\mu K} + \frac{L\widetilde{D}_1\left(\alpha(\omega+1)/\delta + 1\right)}{\delta\mu^2 K^2}\right).$$

*That is, to achive $\mathbf{E}\left[f(\bar{x}^K) - f(x^*)\right] \leq \varepsilon$* `EC-SGD-DIANA` *requires*

$$\widetilde{\mathcal{O}}\left(\frac{1}{\alpha} + \frac{L}{\delta\mu} + \frac{D_1}{n\mu\varepsilon} + \frac{\sqrt{L\widetilde{D}_1\left(\alpha(\omega+1)/\delta + 1\right)}}{\mu\sqrt{\delta\varepsilon}}\right) \quad \text{iterations.}$$

*In particular, if $\alpha = \frac{1}{\omega+1}$, then to achive $\mathbf{E}\left[f(\bar{x}^K) - f(x^*)\right] \leq \varepsilon$* `EC-SGD-DIANA` *requires*

$$\widetilde{\mathcal{O}}\left(\omega + \frac{L}{\delta\mu} + \frac{\widetilde{D}_1}{n\mu\varepsilon} + \frac{\sqrt{L\widetilde{D}_1}}{\delta\mu\sqrt{\varepsilon}}\right) \quad \text{iterations,}$$

*and if $\alpha = \frac{\delta}{\omega+1}$, then to achive $\mathbf{E}\left[f(\bar{x}^K) - f(x^*)\right] \leq \varepsilon$* `EC-SGD-DIANA` *requires*

$$\widetilde{\mathcal{O}}\left(\frac{\omega+1}{\delta} + \frac{L}{\delta\mu} + \frac{\widetilde{D}_1}{n\mu\varepsilon} + \frac{\sqrt{L\widetilde{D}_1}}{\mu\sqrt{\delta\varepsilon}}\right) \quad \text{iterations.}$$

Applying Lemma D.3 we get the complexity result in the case when $\mu = 0$.

**Corollary J.7.** *Let the assumptions of Theorem J.4 hold and $\mu = 0$. Then after $K$ iterations of* `EC-SGD-DIANA` *with the stepsize*

$$\gamma_0 = \min\left\{\frac{1}{4L}, \frac{\delta\sqrt{1-\alpha}}{8L\sqrt{6(3-\alpha)}}\right\}, \quad R_0 = \|x^0 - x^*\|,$$

$$\gamma = \min\left\{\gamma_0, \sqrt[3]{\frac{R_0^2\delta^2\alpha\left(1 - \min\left\{\frac{\gamma_0\mu}{2}, \frac{\alpha}{4}\right\}\right)}{96L\sigma_0^2}}, \sqrt{\frac{nR_0^2}{\widetilde{D}_1 K}}, \sqrt[3]{\frac{\delta R_0^2}{6L\widetilde{D}_1\left(\frac{4\alpha(\omega+1)}{\delta} + 1\right)K}}\right\},$$

*and $\alpha \leq \frac{1}{\omega+1}$ we have $\mathbf{E}\left[f(\bar{x}^K) - f(x^*)\right]$ of order*

$$\mathcal{O}\left(\frac{LR_0^2}{\delta K} + \frac{\sqrt[3]{LR_0^4\sigma_0^2}}{K\sqrt[3]{\delta^2\alpha}} + \sqrt{\frac{R_0^2\widetilde{D}_1}{nK}} + \sqrt[3]{\frac{LR_0^4\widetilde{D}_1\left(\frac{\alpha(\omega+1)}{\delta} + 1\right)}{\delta K^2}}\right).$$

*That is, to achive $\mathbf{E}\left[f(\bar{x}^K) - f(x^*)\right] \leq \varepsilon$* `EC-SGD-DIANA` *requires*

$$\mathcal{O}\left(\frac{LR_0^2}{\delta\varepsilon} + \frac{\sqrt[3]{LR_0^4\sigma_0^2}}{\varepsilon\sqrt[3]{\delta^2\alpha}} + \frac{R_0^2\widetilde{D}_1}{n\varepsilon^2} + \frac{R_0^2\sqrt{L\widetilde{D}_1\left(\frac{\alpha(\omega+1)}{\delta} + 1\right)}}{\sqrt{\delta\varepsilon^3}}\right)$$

**Algorithm 7** EC-SGDsr-DIANA

**Input:** learning rates $\gamma > 0$, $\alpha \in (0, 1]$, initial vectors $x^0, h_1^0, \dots, h_n^0 \in \mathbb{R}^d$
1: Set $e_i^0 = 0$ for all $i = 1, \dots, n$
2: Set $h^0 = \frac{1}{n} \sum_{i=1}^n h_i^0$
3: **for** $k = 0, 1, \dots$ **do**
4:    Broadcast $x^k, h^k$ to all workers
5:    **for** $i = 1, \dots, n$ in parallel **do**
6:        Sample $\hat{g}_i^k = \nabla f_{\xi_i^k}(x^k)$ satisfying Assumption I.1 independtently from other workers
7:        $g_i^k = \hat{g}_i^k - h_i^k + h^k$
8:        $v_i^k = C(e_i^k + \gamma g_i^k)$
9:        $e_i^{k+1} = e_i^k + \gamma g_i^k - v_i^k$
10:        $h_i^{k+1} = h_i^k + \alpha Q(\hat{g}_i^k - h_i^k)$    $\triangleright$ $Q(\cdot)$ is calculated independtly from other workers
11:    **end for**
12:    $e^k = \frac{1}{n} \sum_{i=1}^n e_i^k$, $g^k = \frac{1}{n} \sum_{i=1}^n g_i^k$, $v^k = \frac{1}{n} \sum_{i=1}^n v_i^k$, $h^{k+1} = \frac{1}{n} \sum_{i=1}^n h_i^{k+1} = h^k + \alpha \frac{1}{n} \sum_{i=1}^n Q(\hat{g}_i^k - h_i^k)$
13:    $x^{k+1} = x^k - v^k$
14: **end for**

---

*iterations. In particular, if $\alpha = \frac{1}{\omega+1}$, then to achive $\mathbf{E}\left[ f(\bar{x}^K) - f(x^*) \right] \leq \varepsilon$* EC-SGD-DIANA *requires*

$$\mathcal{O}\left( \frac{LR_0^2}{\delta \varepsilon} + \frac{\sqrt[3]{LR_0^4(\omega+1)\sigma_0^2}}{\varepsilon \sqrt[3]{\delta^2}} + \frac{R_0^2 \widetilde{D}_1}{n\varepsilon^2} + \frac{R_0^2\sqrt{L\widetilde{D}_1}}{\delta\sqrt{\varepsilon^3}} \right) \text{ iterations,}$$

*and if $\alpha = \frac{\delta}{\omega+1}$, then to achive $\mathbf{E}\left[ f(\bar{x}^K) - f(x^*) \right] \leq \varepsilon$* EC-SGD-DIANA *requires*

$$\mathcal{O}\left( \frac{LR_0^2}{\delta \varepsilon} + \frac{\sqrt[3]{LR_0^4(\omega+1)\sigma_0^2}}{\delta \varepsilon} + \frac{R_0^2 \widetilde{D}_1}{n\varepsilon^2} + \frac{R_0^2\sqrt{L\widetilde{D}_1}}{\sqrt{\delta\varepsilon^3}} \right) \text{ iterations.}$$

## J.5  EC-SGDsr-DIANA

In this section we consider the same setup as in Section I.1 and consider EC-SGD-DIANA adjusted to this setup. The resulting algorithm is called EC-SGDsr-DIANA, see

**Lemma J.5.** *Let Assumption I.1 be satisfied and $f_i$ be convex and $L$-smooth for all $i \in [n]$. Then, for all $k \geq 0$ we have*

$$\mathbf{E}\left[ g^k \mid x^k \right] = \nabla f(x^k), \tag{104}$$

$$\frac{1}{n}\sum_{i=1}^n \|\bar{g}_i^k\|^2 \leq 4L\left( f(x^k) - f(x^*) \right) + 2\sigma_k^2, \tag{105}$$

$$\frac{1}{n}\sum_{i=1}^n \mathbf{E}\left[ \|g_i^k - \bar{g}_i^k\|^2 \mid x^k \right] \leq 6(\mathcal{L} + L)\left( f(x^k) - f(x^*) \right) + \widetilde{D}_1, \tag{106}$$

$$\mathbf{E}\left[ \|g^k\|^2 \mid x^k \right] \leq 4\mathcal{L}\left( f(x^k) - f(x^*) \right) + D_1' \tag{107}$$

*where $\sigma_k^2 = \frac{1}{n}\sum_{i=1}^n \|h_i^k - \nabla f(x^*)\|^2$, $\widetilde{D}_1 = \frac{3}{n}\sum_{i=1}^n \mathbf{E}_{\mathcal{D}_i}\left[ \|\nabla f_{\xi_i}(x^*) - \nabla f_i(x^*)\|^2 \right]$ and $D_1' = \frac{2}{n^2}\sum_{i=1}^n \mathbf{E}_{\mathcal{D}_i}\left[ \|\nabla f_{\xi_i}(x^*) - \nabla f_i(x^*)\|^2 \right]$.*

*Proof.* First of all, we show unbiasedness of $g^k$:

$$\mathbf{E}\left[ g^k \mid x^k \right] = \frac{1}{n}\sum_{i=1}^n \mathbf{E}\left[ g_i^k \mid x^k \right] = \frac{1}{n}\sum_{i=1}^n \left( \nabla f_i(x^k) - h_i^k + h^k \right) = \nabla f(x^k).$$

Following the same steps as in the proof of (102) we derive (105). Next, we establish (106):

$$
\frac{1}{n}\sum_{i=1}^{n}\mathbf{E}\left[\|g_i^k - \bar{g}_i^k\|^2 \mid x^k\right] \quad = \quad \frac{1}{n}\sum_{i=1}^{n}\mathbf{E}_{\mathcal{D}_i}\left[\|\nabla f_{\xi_i^k}(x^k) - \nabla f_i(x^k)\|^2\right]
$$

$$
\overset{(31)}{\leq} \quad \frac{3}{n}\sum_{i=1}^{n}\mathbf{E}_{\mathcal{D}_i}\left[\|\nabla f_{\xi_i^k}(x^k) - \nabla f_{\xi_i^k}(x^*)\|^2\right]
$$

$$
+\frac{3}{n}\sum_{i=1}^{n}\mathbf{E}_{\mathcal{D}_i}\left[\|\nabla f_{\xi_i^k}(x^*) - \nabla f_i(x^*)\|^2\right]
$$

$$
+\frac{3}{n}\sum_{i=1}^{n}\|\nabla f_i(x^*) - \nabla f_i(x^k)\|^2
$$

$$
\overset{(11),(89)}{\leq} \quad 6(\mathcal{L}+L)\left(f(x^k) - f(x^*)\right)
$$

$$
+\frac{3}{n}\sum_{i=1}^{n}\mathbf{E}_{\mathcal{D}_i}\left[\|\nabla f_{\xi_i}(x^*) - \nabla f_i(x^*)\|^2\right].
$$

Finally, we obtain the upper bound for the second moment of $g^k$ using the independence of $\xi_1^k, \ldots, \xi_n^k$:

$$
\mathbf{E}\left[\|g^k\|^2 \mid x^k\right] \quad = \quad \mathbf{E}\left[\left\|\frac{1}{n}\sum_{i=1}^{n}(\nabla f_{\xi_i^k}(x^k) - \nabla f_{\xi_i^k}(x^*) + \nabla f_{\xi_i^k}(x^*) - \nabla f_i(x^*))\right\|^2 \mid x^k\right]
$$

$$
\overset{(31)}{\leq} \quad \frac{2}{n}\sum_{i=1}^{n}\mathbf{E}\left[\|\nabla f_{\xi_i^k}(x^k) - \nabla f_{\xi_i^k}(x^*)\|^2 \mid x^k\right]
$$

$$
+2\mathbf{E}\left[\left\|\frac{1}{n}\sum_{i=1}^{n}(\nabla f_{\xi_i^k}(x^*) - \nabla f_i(x^*))\right\|^2 \mid x^k\right]
$$

$$
\overset{(89)}{\leq} \quad 4\mathcal{L}\left(f(x^k) - f(x^*)\right) + \frac{2}{n^2}\sum_{i=1}^{n}\mathbf{E}_{\mathcal{D}_i}\left[\|\nabla f_{\xi_i}(x^*) - \nabla f_i(x^*)\|^2\right].
$$

$\square$

**Lemma J.6.** *Let $f_i$ be convex and L-smooth, Assumption I.1 holds and $\alpha \leq 1/(\omega+1)$. Then, for all $k \geq 0$ we have*

$$
\mathbf{E}\left[\sigma_{k+1}^2 \mid x^k\right] \leq (1-\alpha)\sigma_k^2 + 2\alpha(3\mathcal{L}+4L)(f(x^k) - f(x^*)) + D_2, \tag{108}
$$

*where $\sigma_k^2 = \frac{1}{n}\sum_{i=1}^{n}\|h_i^k - \nabla f_i(x^*)\|^2$ and $D_2 = \alpha^2(\omega+1)\widetilde{D}_1$.*

*Proof.* The proof is identical to the proof of Lemma I.2 up to the following changes in the notation: $\omega_1 = \omega$, $\Delta_i^k = Q(\hat{g}_i^k - h_i^k)$ and $\hat{\Delta}_i^k = \hat{g}_i^k - h_i^k$. $\square$

Applying Theorem G.1 we get the following result.

**Theorem J.5.** *Assume that $f_i(x)$ is convex and L-smooth for all $i = 1,\ldots,n$, $f(x)$ is $\mu$-quasi strongly convex and Assumption I.1 holds. Then* `EC-SGDsr-DIANA` *satisfies Assumption 3.3 with*

$$
A = 2L, \quad \widetilde{A} = 3(\mathcal{L}+L), \quad A' = 2\mathcal{L}, \quad B_1 = 2, \quad \widetilde{D}_1 = \frac{3}{n}\sum_{i=1}^{n}\mathbf{E}_{\mathcal{D}_i}\left[\|\nabla f_{\xi_i}(x^*) - \nabla f_i(x^*)\|^2\right],
$$

$$
\sigma_{1,k}^2 = \sigma_k^2 = \frac{1}{n}\sum_{i=1}^{n}\|h_i^k - \nabla f_i(x^*)\|^2, \quad D_1 = 0, \quad D_1' = \frac{2}{3n}\widetilde{D}_1, \quad D_2 = \alpha^2(\omega+1)\widetilde{D}_1
$$

$$
\widetilde{B}_1 = B_1' = B_2' = B_2 = \widetilde{B}_2 = 0, \quad \sigma_{2,k}^2 \equiv 0, \quad \rho_1 = \alpha, \quad \rho_2 = 1, \quad C_1 = 2\alpha(3\mathcal{L}+4L), \quad C_2 = 0,
$$

$$
G = 0, \quad F_1 = \frac{96L\gamma^2}{\delta^2\alpha\left(1 - \min\left\{\frac{\gamma\mu}{2}, \frac{\alpha}{4}\right\}\right)}, \quad F_2 = 0, \quad D_3 = \frac{6L\gamma}{\delta}\left(\frac{4\alpha(\omega+1)}{\delta} + 1\right)\widetilde{D}_1,
$$

*with $\gamma$ and $\alpha$ satisfying*

$$\gamma \leq \min\left\{ \frac{1}{4\mathcal{L}}, \frac{\delta}{4\sqrt{6L\left(4L + 3\delta(\mathcal{L}+L) + \frac{16(3\mathcal{L}+4L)}{1-\alpha}\right)}} \right\}, \quad \alpha \leq \frac{1}{\omega+1}, \quad M_1 = M_2 = 0.$$

*and for all $K \geq 0$*

$$\mathbf{E}\left[f(\bar{x}^K) - f(x^*)\right] \leq \left(1 - \min\left\{\frac{\gamma\mu}{2}, \frac{\alpha}{4}\right\}\right)^K \frac{4(\|x^0 - x^*\|^2 + \gamma F_1\sigma_0^2)}{\gamma} + 4\gamma\left(D_1' + D_3\right),$$

*when $\mu > 0$ and*

$$\mathbf{E}\left[f(\bar{x}^K) - f(x^*)\right] \leq \frac{4(\|x^0 - x^*\|^2 + \gamma F_1\sigma_0^2)}{\gamma K} + 4\gamma\left(D_1' + D_3\right)$$

*when $\mu = 0$.*

Applying Lemma D.2 we establish the rate of convergence to $\varepsilon$-solution in the case when $\mu > 0$.

**Corollary J.8.** *Let the assumptions of Theorem J.5 hold and $\mu > 0$. Then after $K$ iterations of* `EC-SGDsr-DIANA` *with the stepsize*

$$\gamma_0 = \min\left\{ \frac{1}{4\mathcal{L}}, \frac{\delta}{4\sqrt{6L\left(4L + 3\delta(\mathcal{L}+L) + \frac{16(3\mathcal{L}+4L)}{1-\alpha}\right)}} \right\},$$

$$R_0 = \|x^0 - x^*\|, \quad \tilde{F}_1 = \frac{96L\gamma_0^2}{\delta^2\alpha\left(1 - \min\left\{\frac{\gamma_0\mu}{2}, \frac{\alpha}{4}\right\}\right)},$$

$$\gamma = \min\left\{ \gamma_0, \frac{\ln\left(\max\left\{2, \min\left\{\frac{3n\left(R_0^2 + \tilde{F}_1\gamma_0\sigma_{1,0}^2\right)\mu^2 K^2}{2\widetilde{D}_1}, \frac{\delta\left(R_0^2 + \tilde{F}_1\gamma_0\sigma_{1,0}^2\right)\mu^3 K^3}{6L\widetilde{D}_1\left(\frac{4\alpha(\omega+1)}{\delta}+1\right)}\right\}\right\}\right)}{\mu K} \right\},$$

*and $\alpha \leq \frac{1}{\omega+1}$ we have $\mathbf{E}\left[f(\bar{x}^K) - f(x^*)\right]$ of order*

$$\widetilde{\mathcal{O}}\left(\left(\mathcal{L} + \frac{\sqrt{L\mathcal{L}}}{\delta}\right) R_0^2 \exp\left(-\min\left\{\frac{\mu}{\mathcal{L}+\frac{\sqrt{L\mathcal{L}}}{\delta}}, \alpha\right\} K\right) + \frac{\widetilde{D}_1}{n\mu K} + \frac{L\widetilde{D}_1\left(\frac{\alpha(\omega+1)}{\delta}+1\right)}{\delta\mu^2 K^2}\right)$$

*That is, to achive $\mathbf{E}\left[f(\bar{x}^K) - f(x^*)\right] \leq \varepsilon$* `EC-SGDsr-DIANA` *requires*

$$\widetilde{\mathcal{O}}\left(\frac{1}{\alpha} + \frac{\mathcal{L}}{\mu} + \frac{\sqrt{L\mathcal{L}}}{\delta\mu} + \frac{\widetilde{D}_1}{n\mu\varepsilon} + \frac{\sqrt{L\widetilde{D}_1\left(\frac{\alpha(\omega+1)}{\delta}+1\right)}}{\mu\sqrt{\delta\varepsilon}}\right) \quad \text{iterations.}$$

*In particular, if $\alpha = \frac{1}{\omega+1}$, then to achive $\mathbf{E}\left[f(\bar{x}^K) - f(x^*)\right] \leq \varepsilon$* `EC-SGDsr-DIANA` *requires*

$$\widetilde{\mathcal{O}}\left(\omega + \frac{\mathcal{L}}{\mu} + \frac{\sqrt{L\mathcal{L}}}{\delta\mu} + \frac{\widetilde{D}_1}{n\mu\varepsilon} + \frac{\sqrt{L\widetilde{D}_1}}{\delta\mu\sqrt{\varepsilon}}\right) \quad \text{iterations,}$$

*and if $\alpha = \frac{\delta}{\omega+1}$, then to achive $\mathbf{E}\left[f(\bar{x}^K) - f(x^*)\right] \leq \varepsilon$* `EC-SGDsr-DIANA` *requires*

$$\widetilde{\mathcal{O}}\left(\frac{\omega+1}{\delta} + \frac{\mathcal{L}}{\mu} + \frac{\sqrt{L\mathcal{L}}}{\delta\mu} + \frac{\widetilde{D}_1}{n\mu\varepsilon} + \frac{\sqrt{L\widetilde{D}_1}}{\mu\sqrt{\delta\varepsilon}}\right) \quad \text{iterations.}$$

Applying Lemma D.3 we get the complexity result in the case when $\mu = 0$.

**Corollary J.9.** *Let the assumptions of Theorem J.5 hold and $\mu = 0$. Then after $K$ iterations of* `EC-SGDsr-DIANA` *with the stepsize*

$$
\gamma_0 \;=\; \min\left\{\frac{1}{4\mathcal{L}}, \frac{\delta}{4\sqrt{6L\left(4L + 3\delta(\mathcal{L}+L) + \frac{16(3\mathcal{L}+4L)}{1-\alpha}\right)}}\right\}, \quad R_0 = \|x^0 - x^*\|,
$$

$$
\gamma \;=\; \min\left\{\gamma_0, \sqrt[3]{\frac{R_0^2\delta^2\alpha\left(1 - \min\left\{\frac{\gamma_0\mu}{2}, \frac{\alpha}{4}\right\}\right)}{96L\sigma_0^2}}, \sqrt{\frac{3nR_0^2}{2\widetilde{D}_1 K}}, \sqrt[3]{\frac{\delta R_0^2}{6L\widetilde{D}_1\left(\frac{4\alpha(\omega+1)}{\delta}+1\right)K}}\right\},
$$

*and $\alpha \le \frac{1}{\omega+1}$ we have $\mathbf{E}\left[f(\bar{x}^K) - f(x^*)\right]$ of order*

$$
\mathcal{O}\left(\frac{\mathcal{L}R_0^2}{K} + \frac{\sqrt{\mathcal{L}L}R_0^2}{\delta K} + \frac{\sqrt[3]{LR_0^4\sigma_0^2}}{K\sqrt[3]{\delta^2\alpha}} + \sqrt{\frac{R_0^2\widetilde{D}_1}{nK}} + \sqrt[3]{\frac{LR_0^4\widetilde{D}_1\left(\frac{\alpha(\omega+1)}{\delta}+1\right)}{\delta K^2}}\right).
$$

*That is, to achive $\mathbf{E}\left[f(\bar{x}^K) - f(x^*)\right] \le \varepsilon$* `EC-SGDsr-DIANA` *requires*

$$
\mathcal{O}\left(\frac{\mathcal{L}R_0^2}{\varepsilon} + \frac{\sqrt{\mathcal{L}L}R_0^2}{\delta\varepsilon} + \frac{\sqrt[3]{LR_0^4\sigma_0^2}}{\varepsilon\sqrt[3]{\delta^2\alpha}} + \frac{R_0^2\widetilde{D}_1}{n\varepsilon^2} + \frac{R_0^2\sqrt{L\widetilde{D}_1\left(\frac{\alpha(\omega+1)}{\delta}+1\right)}}{\sqrt{\delta\varepsilon^3}}\right)
$$

*iterations. In particular, if $\alpha = \frac{1}{\omega+1}$, then to achive $\mathbf{E}\left[f(\bar{x}^K) - f(x^*)\right] \le \varepsilon$* `EC-SGDsr-DIANA` *requires*

$$
\mathcal{O}\left(\frac{\mathcal{L}R_0^2}{\varepsilon} + \frac{\sqrt{\mathcal{L}L}R_0^2}{\delta\varepsilon} + \frac{\sqrt[3]{LR_0^4(\omega+1)\sigma_0^2}}{\varepsilon\sqrt[3]{\delta^2}} + \frac{R_0^2\widetilde{D}_1}{n\varepsilon^2} + \frac{R_0^2\sqrt{L\widetilde{D}_1}}{\delta\sqrt{\varepsilon^3}}\right) \quad \text{iterations,}
$$

*and if $\alpha = \frac{\delta}{\omega+1}$, then to achive $\mathbf{E}\left[f(\bar{x}^K) - f(x^*)\right] \le \varepsilon$* `EC-SGDsr-DIANA` *requires*

$$
\mathcal{O}\left(\frac{\mathcal{L}R_0^2}{\varepsilon} + \frac{\sqrt{\mathcal{L}L}R_0^2}{\delta\varepsilon} + \frac{\sqrt[3]{LR_0^4(\omega+1)\sigma_0^2}}{\delta\varepsilon} + \frac{R_0^2\widetilde{D}_1}{n\varepsilon^2} + \frac{R_0^2\sqrt{L\widetilde{D}_1}}{\sqrt{\delta\varepsilon^3}}\right) \quad \text{iterations.}
$$

### J.6 `EC-LSVRG`

In this section we consider problem (1) with $f(x)$ being $\mu$-quasi strongly convex and $f_i(x)$ satisfying (3) where functions $f_{ij}(x)$ are convex and $L$-smooth. For this problem we propose a new method called `EC-LSVRG` which takes for the origin another method called `LSVRG` (see [17, 31]).

**Lemma J.7.** *For all $k \ge 0$, $i \in [n]$ we have*

$$
\bar{g}_i^k = \mathbf{E}\left[g_i^k \mid x^k\right] = \nabla f_i(x^k) \tag{109}
$$

*and*

$$
\frac{1}{n}\sum_{i=1}^{n}\|\bar{g}_i^k\|^2 \;\le\; 4L\left(f(x^k) - f(x^*)\right) + D_1, \tag{110}
$$

$$
\frac{1}{n}\sum_{i=1}^{n}\mathbf{E}\left[\|g_i^k - \bar{g}_i^k\|^2 \mid x^k\right] \;\le\; 12L\left(f(x^k) - f(x^*)\right) + 3\sigma_k^2, \tag{111}
$$

$$
\mathbf{E}\left[\|g^k\|^2 \mid x^k\right] \;\le\; 4L\left(f(x^k) - f(x^*)\right) + 2\sigma_k^2 \tag{112}
$$

*where $\sigma_k^2 = \frac{1}{nm}\sum_{i=1}^{n}\sum_{j=1}^{n}\|\nabla f_{ij}(w_i^k) - \nabla f_{ij}(x^*)\|^2$ and $D_1 = \frac{2}{n}\sum_{i=1}^{n}\|\nabla f_i(x^*)\|^2$.*

**Algorithm 8** EC-LSVRG

**Input:** learning rate $\gamma > 0$, initial vector $x^0 \in \mathbb{R}^d$
1: Set $e_i^0 = 0$ for all $i = 1, \dots, n$
2: **for** $k = 0, 1, \dots$ **do**
3:      Broadcast $x^k$ to all workers
4:      **for** $i = 1, \dots, n$ in parallel **do**
5:          Pick $l$ uniformly at random from $[m]$
6:          Set $g_i^k = \nabla f_{il}(x^k) - \nabla f_{il}(w_i^k) + \nabla f_i(w_i^k)$
7:          $v_i^k = C(e_i^k + \gamma g_i^k)$
8:          $e_i^{k+1} = e_i^k + \gamma g_i^k - v_i^k$
9:          $w_i^{k+1} = \begin{cases} x^k, & \text{with probability } p, \\ w_i^k, & \text{with probability } 1 - p \end{cases}$
10:      **end for**
11:      $e^k = \frac{1}{n} \sum_{i=1}^n e_i^k$, $g^k = \frac{1}{n} \sum_{i=1}^n g_i^k$, $v^k = \frac{1}{n} \sum_{i=1}^n v_i^k$
12:      $x^{k+1} = x^k - v^k$
13: **end for**

*Proof.* First of all, we derive unbiasedness of $g_i^k$:

$$\mathbf{E}\left[g_i^k \mid x^k\right] = \frac{1}{m} \sum_{j=1}^m \left(\nabla f_{ij}(x^k) - \nabla f_{ij}(w_i^k) + \nabla f_i(w_i^k)\right) = \nabla f_i(x^k).$$

Next, we get an upper bound for $\frac{1}{n} \sum_{i=1}^n \|\bar{g}_i^k\|^2$:

$$
\begin{aligned}
\frac{1}{n} \sum_{i=1}^n \|\bar{g}_i^k\|^2 &= \frac{1}{n} \sum_{i=1}^n \|\nabla f_i(x^k)\|^2 \\
&\overset{(31)}{\leq} \frac{2}{n} \sum_{i=1}^n \|\nabla f_i(x^k) - \nabla f_i(x^*)\|^2 + \frac{2}{n} \sum_{i=1}^n \|\nabla f_i(x^*)\|^2 \\
&\overset{(11)}{\leq} 4L\left(f(x^k) - f(x^*)\right) + \frac{2}{n} \sum_{i=1}^n \|\nabla f_i(x^*)\|^2.
\end{aligned}
$$

Using (109) we establish the following inequality:

$$
\begin{aligned}
\frac{1}{n} \sum_{i=1}^n \mathbf{E}\left[\|g_i^k - \bar{g}_i^k\|^2 \mid x^k\right] &\overset{(31)}{\leq} \frac{3}{n} \sum_{i=1}^n \mathbf{E}\left[\|\nabla f_{il}(x^k) - \nabla f_{il}(x^*)\|^2 \mid x^k\right] \\
&\quad + \frac{3}{n} \sum_{i=1}^n \mathbf{E}\left[\left\|\nabla f_{il}(w_i^k) - \nabla f_{il}(x^*) - \left(\nabla f_i(w_i^k) - \nabla f_i(x^*)\right)\right\|^2 \mid x^k\right] \\
&\quad + \frac{3}{n} \sum_{i=1}^n \|\nabla f_i(x^*) - \nabla f_i(x^k)\|^2 \\
&\overset{(11),(34)}{\leq} 12L\left(f(x^k) - f(x^*)\right) + \frac{3}{nm} \sum_{i=1}^n \sum_{j=1}^m \|\nabla f_{ij}(w_i^k) - \nabla f_{ij}(x^*)\|^2.
\end{aligned}
$$

Finally, we derive (112):

$$\mathbf{E}\left[\|g^k\|^2 \mid x^k\right] = \mathbf{E}\left[\left\|\frac{1}{n}\sum_{i=1}^{n}\left(\nabla f_{il}(x^k) - \nabla f_{il}(w_i^k) + \nabla f_i(w_i^k) - \nabla f_i(x^*)\right)\right\|^2 \mid x^k\right]$$

$$\overset{(31)}{\leq} \frac{2}{n}\sum_{i=1}^{n}\mathbf{E}\left[\|\nabla f_{il}(x^k) - \nabla f_{il}(x^*)\|^2 \mid x^k\right]$$

$$+ \frac{2}{n}\sum_{i=1}^{n}\mathbf{E}\left[\left\|\nabla f_{il}(w_i^k) - \nabla f_{il}(x^*) - \left(\nabla f_i(w_i^k) - \nabla f_i(x^*)\right)\right\|^2 \mid x^k\right]$$

$$= \frac{2}{nm}\sum_{i=1}^{n}\sum_{j=1}^{m}\|\nabla f_{ij}(x^k) - \nabla f_{ij}(x^*)\|^2$$

$$+ \frac{2}{nm}\sum_{i=1}^{n}\sum_{j=1}^{m}\left\|\nabla f_{ij}(w_i^k) - \nabla f_{ij}(x^*) - \frac{1}{m}\sum_{j=1}^{m}\left(\nabla f_{ij}(w_i^k) - \nabla f_{ij}(x^*)\right)\right\|^2$$

$$\overset{(11),(34)}{\leq} 4L\left(f(x^k) - f(x^*)\right) + \frac{2}{nm}\sum_{i=1}^{n}\sum_{j=1}^{m}\|\nabla f_{ij}(w_i^k) - \nabla f_{ij}(x^*)\|^2.$$

$\square$

**Lemma J.8.** *For all $k \geq 0$, $i \in [n]$ we have*

$$\mathbf{E}\left[\sigma_{k+1}^2 \mid x^k\right] \leq (1-p)\sigma_k^2 + 2Lp\left(f(x^k) - f(x^*)\right), \tag{113}$$

*where $\sigma_k^2 = \frac{1}{nm}\sum_{i=1}^{n}\sum_{j=1}^{n}\|\nabla f_{ij}(w_i^k) - \nabla f_{ij}(x^*)\|^2$.*

*Proof.* By definition of $w_i^{k+1}$ we get

$$\mathbf{E}\left[\sigma_{k+1}^2 \mid x^k\right] = \frac{1}{nm}\sum_{i=1}^{n}\sum_{j=1}^{m}\mathbf{E}\left[\|\nabla f_{ij}(w_i^{k+1}) - \nabla f_{ij}(x^*)\|^2 \mid x^k\right]$$

$$= \frac{1-p}{nm}\sum_{i=1}^{n}\sum_{j=1}^{m}\|\nabla f_{ij}(w_i^k) - \nabla f_{ij}(x^*)\|^2 + \frac{p}{nm}\sum_{i=1}^{n}\sum_{j=1}^{m}\|\nabla f_{ij}(x^k) - \nabla f_{ij}(x^*)\|^2$$

$$\overset{(11)}{\leq} (1-p)\sigma_k^2 + \frac{2Lp}{nm}\sum_{i=1}^{n}\sum_{j=1}^{m}D_{f_{ij}}(x^k, x^*)$$

$$= (1-p)\sigma_k^2 + 2Lp\left(f(x^k) - f(x^*)\right).$$

$\square$

Applying Theorem G.1 we get the following result.

**Theorem J.6.** *Assume that $f(x)$ is $\mu$-quasi strongly convex and functions $f_{ij}$ are convex and $L$-smooth for all $i \in [n], j \in [m]$. Then* `EC-LSVRG` *satisfies Assumption 3.3 with*

$$A = 2L, \quad \widetilde{A} = 12L, \quad A' = 2L, \quad B_1 = \widetilde{B}_1 = B_1' = B_2 = 0, \quad D_1 = \frac{2}{n}\sum_{i=1}^{n}\|\nabla f_i(x^*)\|^2,$$

$$D_1' = \widetilde{D}_1 = 0, \quad \widetilde{B}_2 = 3, \quad B_2' = 2, \quad \sigma_{1,k}^2 \equiv 0, \quad C_1 = 0,$$

$$\sigma_{2,k}^2 = \sigma_k^2 = \frac{1}{nm}\sum_{i=1}^{n}\sum_{j=1}^{m}\|\nabla f_{ij}(w_i^k) - \nabla f_{ij}(x^*)\|^2, \quad \rho_1 = 1, \quad \rho_2 = p, \quad C_2 = Lp, \quad D_2 = 0,$$

$$G = 0, \quad F_1 = 0, \quad F_2 = \frac{72L\gamma^2}{\delta p\left(1 - \min\left\{\frac{\gamma\mu}{2}, \frac{p}{4}\right\}\right)}, \quad D_3 = \frac{12L\gamma}{\delta^2}D_1,$$

*with $\gamma$ satisfying*

$$\gamma \leq \min\left\{\frac{1}{24L}, \frac{\delta}{8L\sqrt{3\left(2 + 3\delta\left(2 + \frac{1}{1-p}\right)\right)}}\right\}, \quad M_2 = \frac{4}{p}.$$

*and for all $K \geq 0$*

$$\mathbf{E}\left[f(\bar{x}^K) - f(x^*)\right] \leq \left(1 - \min\left\{\frac{\gamma\mu}{2}, \frac{p}{4}\right\}\right)^K \frac{4(T^0 + \gamma F_2 \sigma_0^2)}{\gamma} + \frac{48L\gamma^2}{\delta^2}D_1$$

*when $\mu > 0$ and*

$$\mathbf{E}\left[f(\bar{x}^K) - f(x^*)\right] \leq \frac{4(T^0 + \gamma F_2 \sigma_0^2)}{\gamma K} + \frac{48L\gamma^2}{\delta^2}D_1$$

*when $\mu = 0$, where $T^k \stackrel{def}{=} \|x^k - x^*\|^2 + M_2\gamma^2\sigma_k^2$.*

In other words, `EC-LSVRG` converges with linear rate $\mathcal{O}\left(\left(\frac{1}{p} + \frac{\kappa}{\delta\sqrt{1-p}}\right)\ln\frac{1}{\varepsilon}\right)$ to the neighbourhood of the solution. If $m \geq 2$ then taking $p = \frac{1}{m}$ we get that in expectation the sample complexity of one iteration of `EC-LSVRG` is $\mathcal{O}(1)$ gradients calculations per node as for `EC-SGDsr` with standard sampling and the rate of convergence to the neighbourhood becomes $\mathcal{O}\left(\left(m + \frac{\kappa}{\delta}\right)\ln\frac{1}{\varepsilon}\right)$. We notice that the size of this neighbourhood is typically smaller than for `EC-SGDsr`, but still the method fails to converge to the exact solution with linear rate. Applying Lemma D.2 we establish the rate of convergence to $\varepsilon$-solution in the case when $\mu > 0$.

**Corollary J.10.** *Let the assumptions of Theorem J.6 hold and $\mu > 0$. Then after $K$ iterations of `EC-LSVRG` with the stepsize*

$$\gamma_0 = \min\left\{\frac{1}{24L}, \frac{\delta}{8L\sqrt{3\left(2 + 3\delta\left(2 + \frac{1}{1-p}\right)\right)}}\right\},$$

$$\tilde{T}^0 = \|x^0 - x^*\|^2 + M_2\gamma_0^2\sigma_0^2, \quad \tilde{F}_2 = \frac{72L\gamma_0^2}{\delta p\left(1 - \min\left\{\frac{\gamma_0\mu}{2}, \frac{p}{4}\right\}\right)},$$

$$\gamma = \min\left\{\gamma_0, \frac{\ln\left(\max\left\{2, \frac{\delta^2\left(\tilde{T}^0 + \tilde{F}_2\gamma_0\sigma_0^2\right)\mu^3 K^3}{48LD_1}\right\}\right)}{\mu K}\right\},$$

*and $p = \frac{1}{m}$, $m \geq 2$ we have*

$$\mathbf{E}\left[f(\bar{x}^K) - f(x^*)\right] = \tilde{\mathcal{O}}\left(\frac{L}{\delta}\left(\tilde{T}^0 + \tilde{F}_2\gamma_0\sigma_0^2\right)\exp\left(-\min\left\{\frac{\delta\mu}{L}, \frac{1}{m}\right\}K\right) + \frac{LD_1}{\delta^2\mu^2 K^2}\right).$$

*That is, to achive $\mathbf{E}\left[f(\bar{x}^K) - f(x^*)\right] \leq \varepsilon$ `EC-LSVRG` requires*

$$\tilde{\mathcal{O}}\left(m + \frac{L}{\delta\mu} + \frac{\sqrt{LD_1}}{\delta\mu\sqrt{\varepsilon}}\right) \quad \text{iterations.}$$

Applying Lemma D.3 we get the complexity result in the case when $\mu = 0$.

**Corollary J.11.** *Let the assumptions of Theorem J.6 hold and $\mu = 0$. Then after $K$ iterations of `EC-LSVRG` with the stepsize*

$$\gamma_0 = \min\left\{\frac{1}{24L}, \frac{\delta}{8L\sqrt{3\left(2 + 3\delta\left(2 + \frac{1}{1-p}\right)\right)}}\right\}, \quad R_0 = \|x^0 - x^*\|,$$

$$\gamma = \min\left\{\gamma_0, \sqrt{\frac{R_0^2 p}{4\sigma_0^2}}, \sqrt[3]{\frac{R_0^2 \delta p\left(1 - \min\left\{\frac{\gamma_0\mu}{2}, \frac{p}{4}\right\}\right)}{72L\sigma_0^2}}, \sqrt[3]{\frac{\delta^2 R_0^2}{12LD_1 K}}\right\},$$

---

**Algorithm 9** `EC-LSVRGstar`

---

**Input:** learning rate $\gamma > 0$, initial vector $x^0 \in \mathbb{R}^d$
 1: Set $e_i^0 = 0$ for all $i = 1, \ldots, n$
 2: **for** $k = 0, 1, \ldots$ **do**
 3:     Broadcast $x^k$ to all workers
 4:     **for** $i = 1, \ldots, n$ in parallel **do**
 5:         Pick $l$ uniformly at random from $[m]$
 6:         Set $g_i^k = \nabla f_{il}(x^k) - \nabla f_{il}(w_i^k) + \nabla f_i(w_i^k) - \nabla f_i(x^*)$
 7:         $v_i^k = C(e_i^k + \gamma g_i^k)$
 8:         $e_i^{k+1} = e_i^k + \gamma g_i^k - v_i^k$
 9:         $w_i^{k+1} = \begin{cases} x^k, & \text{with probability } p, \\ w_i^k, & \text{with probability } 1 - p \end{cases}$
10:     **end for**
11:     $e^k = \frac{1}{n}\sum_{i=1}^n e_i^k$, $g^k = \frac{1}{n}\sum_{i=1}^n g_i^k$, $v^k = \frac{1}{n}\sum_{i=1}^n v_i^k$
12:     $x^{k+1} = x^k - v^k$
13: **end for**

---

and $p = \frac{1}{m}$, $m \geq 2$ we have $\mathbf{E}\left[f(\bar{x}^K) - f(x^*)\right]$ of order

$$\mathcal{O}\left(\frac{LR_0^2}{\delta K} + \frac{\sqrt{mR_0^2\sigma_0^2}}{K} + \frac{\sqrt[3]{LR_0^4 m\sigma_0^2}}{\sqrt[3]{\delta}K} + \frac{\sqrt[3]{LR_0^4}}{(\delta K)^{2/3}}\sqrt[3]{\frac{1}{n}\sum_{i=1}^n \|\nabla f_i(x^*)\|^2}\right).$$

That is, to achive $\mathbf{E}\left[f(\bar{x}^K) - f(x^*)\right] \leq \varepsilon$ `EC-LSVRG` requires

$$\mathcal{O}\left(\frac{LR_0^2}{\delta\varepsilon} + \frac{\sqrt{mR_0^2\sigma_0^2}}{\varepsilon} + \frac{\sqrt[3]{LR_0^4 m\sigma_0^2}}{\sqrt[3]{\delta}\varepsilon} + \frac{R_0^2}{\delta\varepsilon^{3/2}}\sqrt{\frac{L}{n}\sum_{i=1}^n \|\nabla f_i(x^*)\|^2}\right)$$

iterations.

## J.7 `EC-LSVRGstar`

In the setup of Section J.6 we now assume that $i$-th node has an access to the $\nabla f_i(x^*)$. Under this unrealistic assumption we construct the method called `EC-LSVRGstar` that asymptotically converges to the exact solution.

**Lemma J.9.** *For all $k \geq 0$, $i \in [n]$ we have*

$$\mathbf{E}\left[g^k \mid x^k\right] = \nabla f(x^k) \tag{114}$$

*and*

$$\frac{1}{n}\sum_{i=1}^n \|\bar{g}_i^k\|^2 \leq 2L\left(f(x^k) - f(x^*)\right), \tag{115}$$

$$\frac{1}{n}\sum_{i=1}^n \mathbf{E}\left[\|g_i^k - \bar{g}_i^k\|^2 \mid x^k\right] \leq 4L\left(f(x^k) - f(x^*)\right) + 2\sigma_k^2, \tag{116}$$

$$\mathbf{E}\left[\|g^k\|^2 \mid x^k\right] \leq 4L\left(f(x^k) - f(x^*)\right) + 2\sigma_k^2, \tag{117}$$

*where* $\sigma_k^2 = \frac{1}{nm}\sum_{i=1}^n\sum_{j=1}^n \|\nabla f_{ij}(w_i^k) - \nabla f_{ij}(x^*)\|^2$.

*Proof.* First of all, we derive unbiasedness of $g^k$:

$$\begin{aligned}
\mathbf{E}\left[g^k \mid x^k\right] &= \frac{1}{n}\sum_{i=1}^n \mathbf{E}\left[\nabla f_{il}(x^k) - \nabla f_{il}(w_i^k) + \nabla f_i(w_i^k) - \nabla f_i(x^*) \mid x^k\right] \\
&= \frac{1}{nm}\sum_{i=1}^n\sum_{j=1}^m \left(\nabla f_{ij}(x^k) - \nabla f_{ij}(w_i^k) + \nabla f_i(w_i^k) - \nabla f_i(x^*)\right) \\
&= \nabla f(x^k) + \frac{1}{n}\sum_{i=1}^n \left(-\nabla f_i(w_i^k) + \nabla f_i(w_i^k)\right) - \nabla f(x^*) = \nabla f(x^k).
\end{aligned}$$

Next, we get an upper bound for $\frac{1}{n}\sum_{i=1}^{n}\|\bar{g}_i^k\|^2$:

$$\frac{1}{n}\sum_{i=1}^{n}\|\bar{g}_i^k\|^2 = \frac{1}{n}\sum_{i=1}^{n}\|\nabla f_i(x^k) - \nabla f_i(x^*)\|^2 \stackrel{(11)}{\leq} 2L\left(f(x^k) - f(x^*)\right).$$

Since the variance of random vector is not greater than its second moment we obtain:

$$
\begin{aligned}
\frac{1}{n}\sum_{i=1}^{n}\mathbf{E}\left[\|g_i^k - \bar{g}_i^k\|^2 \mid x^k\right] &\stackrel{(34)}{\leq} \frac{1}{n}\sum_{i=1}^{n}\mathbf{E}\left[\|g_i^k\|^2 \mid x^k\right] \\
&\stackrel{(31)}{\leq} \frac{2}{n}\sum_{i=1}^{n}\mathbf{E}\left[\|\nabla f_{il}(x^k) - \nabla f_{il}(x^*)\|^2 \mid x^k\right] \\
&\quad + \frac{2}{n}\sum_{i=1}^{n}\mathbf{E}\left[\left\|\nabla f_{il}(w_i^k) - \nabla f_{il}(x^*) - \left(\nabla f_i(w_i^k) - \nabla f_i(x^*)\right)\right\|^2 \mid x^k\right] \\
&\stackrel{(11),(34)}{\leq} 4L\left(f(x^k) - f(x^*)\right) + \frac{2}{nm}\sum_{i=1}^{n}\sum_{j=1}^{m}\|\nabla f_{ij}(w_i^k) - \nabla f_{ij}(x^*)\|^2.
\end{aligned}
$$

Inequality (117) trivially follows from the inequality above by Jensen's inequality and convexity of $\|\cdot\|^2$. $\qquad\square$

**Lemma J.10.** *For all $k \geq 0$, $i \in [n]$ we have*

$$\mathbf{E}\left[\sigma_{k+1}^2 \mid x^k\right] \leq (1-p)\sigma_k^2 + 2Lp\left(f(x^k) - f(x^*)\right), \qquad (118)$$

*where $\sigma_k^2 = \frac{1}{nm}\sum_{i=1}^{n}\sum_{j=1}^{n}\|\nabla f_{ij}(w_i^k) - \nabla f_{ij}(x^*)\|^2$.*

*Proof.* The proof of this lemma is identical to the proof of Lemma J.8. $\qquad\square$

Applying Theorem G.1 we get the following result.

**Theorem J.7.** *Assume that $f(x)$ is $\mu$-quasi strongly convex and functions $f_{ij}$ are convex and $L$-smooth for all $i \in [n], j \in [m]$. Then* `EC-LSVRGstar` *satisfies Assumption 3.3 with*

$$A = L, \quad \widetilde{A} = A' = 2L, \quad B_1 = \widetilde{B}_1 = B_1' = B_2 = 0, \quad \widetilde{B}_2 = B_2' = 2, \quad D_1 = D_1' = 0,$$

$$\sigma_{1,k}^2 \equiv 0,, \quad C_1 = 0, \quad \sigma_{2,k}^2 = \sigma_k^2 = \frac{1}{nm}\sum_{i=1}^{n}\sum_{j=1}^{m}\|\nabla f_{ij}(w_i^k) - \nabla f_{ij}(x^*)\|^2, \quad \rho_1 = 1,$$

$$\rho_2 = p, \quad C_2 = Lp, \quad D_2 = 0, \quad G = 0, \quad F_1 = 0, \quad F_2 = \frac{48L\gamma^2(2+p)}{\delta p}, \quad D_3 = 0,$$

*with $\gamma$ satisfying*

$$\gamma \leq \min\left\{\frac{3}{56L}, \frac{\delta}{8L\sqrt{3\left(1 + \delta\left(1 + \frac{2}{1-p}\right)\right)}}\right\}, \quad M_2 = \frac{8}{3p}.$$

*and for all $K \geq 0$*

$$\mathbf{E}\left[f(\bar{x}^K) - f(x^*)\right] \leq \left(1 - \min\left\{\frac{\gamma\mu}{2}, \frac{p}{4}\right\}\right)^K \frac{4(T^0 + \gamma F_2\sigma_0^2)}{\gamma}$$

*when $\mu > 0$ and*

$$\mathbf{E}\left[f(\bar{x}^K) - f(x^*)\right] \leq \frac{4(T^0 + \gamma F_2\sigma_0^2)}{\gamma K}$$

*when $\mu = 0$, where $T^k \stackrel{def}{=} \|x^k - x^*\|^2 + M_2\gamma^2\sigma_k^2$.*

In other words, `EC-LSVRGstar` converges with linear rate $\mathcal{O}\left(\left(\frac{1}{p} + \frac{\kappa}{\delta\sqrt{1-p}}\right) \ln \frac{1}{\varepsilon}\right)$ exactly to the solution when $\mu > 0$. If $m \geq 2$ then taking $p = \frac{1}{m}$ we get that in expectation the sample complexity of one iteration of `EC-LSVRGstar` is $\mathcal{O}(1)$ gradients calculations per node as for `EC-SGDsr` with standard sampling and the rate of convergence becomes $\mathcal{O}\left(\left(m + \frac{\kappa}{\delta}\right) \ln \frac{1}{\varepsilon}\right)$.

Applying Lemma D.3 we get the complexity result in the case when $\mu = 0$.

**Corollary J.12.** *Let the assumptions of Theorem J.7 hold and $\mu = 0$. Then after $K$ iterations of* `EC-LSVRGstar` *with the stepsize*

$$\gamma_0 = \min\left\{\frac{3}{56L}, \frac{\delta}{8L\sqrt{3\left(1 + \delta\left(1 + \frac{2}{1-p}\right)\right)}}\right\}, \quad R_0 = \|x^0 - x^*\|,$$

$$\gamma = \min\left\{\gamma_0, \sqrt{\frac{3pR_0^2}{8\sigma_0^2}}, \sqrt[3]{\frac{R_0^2\delta p\left(1 - \min\left\{\frac{\gamma_0\mu}{2}, \frac{p}{4}\right\}\right)}{72L\sigma_0^2}}\right\},$$

*and $p = \frac{1}{m}$, $m \geq 2$ we have $\mathbf{E}\left[f(\bar{x}^K) - f(x^*)\right]$ of order*

$$\mathcal{O}\left(\frac{LR_0^2}{\delta K} + \frac{\sqrt{R_0^2 m\sigma_0^2}}{K} + \frac{\sqrt[3]{LR_0^4 m\sigma_0^2}}{\sqrt[3]{\delta}K}\right).$$

*That is, to achive $\mathbf{E}\left[f(\bar{x}^K) - f(x^*)\right] \leq \varepsilon$* `EC-LSVRGstar` *requires*

$$\mathcal{O}\left(\frac{LR_0^2}{\delta\varepsilon} + \frac{\sqrt{R_0^2 m\sigma_0^2}}{\varepsilon} + \frac{\sqrt[3]{LR_0^4 m\sigma_0^2}}{\sqrt[3]{\delta}\varepsilon}\right)$$

*iterations.*

However, such convergence guarantees are obtained under very restrictive assumption: the method requires to know vectors $\nabla f_i(x^*)$.

### J.8 `EC-LSVRG-DIANA`

In the setup of Section J.6 we construct a new method called `EC-LSVRG-DIANA` which does not require to know $\nabla f_i(x^*)$ and has linear convergence to the exact solution. As in `EC-SGD-DIANA` the master needs to gather only $C(e_i^k + \gamma g_i^k)$ and $Q(\hat{g}_i^k - h_i^k)$ from all nodes in order to perform an update.

**Lemma J.11.** *Assume that $f_{ij}(x)$ is convex and $L$-smooth for all $i = 1, \ldots, n$, $j = 1, \ldots, m$. Then, for all $k \geq 0$ we have*

$$\mathbf{E}\left[g^k \mid x^k\right] = \nabla f(x^k), \tag{119}$$

$$\frac{1}{n}\sum_{i=1}^{n}\|\bar{g}_i^k\|^2 \leq 4L\left(f(x^k) - f(x^*)\right) + 2\sigma_{1,k}^2, \tag{120}$$

$$\frac{1}{n}\sum_{i=1}^{n}\mathbf{E}\left[\|g_i^k - \bar{g}_i^k\|^2 \mid x^k\right] \leq 6L\left(f(x^k) - f(x^*)\right) + 3\sigma_{1,k}^2 + 3\sigma_{2,k}^2, \tag{121}$$

$$\mathbf{E}\left[\|g^k\|^2 \mid x^k\right] \leq 4L\left(f(x^k) - f(x^*)\right) + 2\sigma_{2,k}^2 \tag{122}$$

*where*

$$\sigma_{1,k}^2 = \frac{1}{n}\sum_{i=1}^{n}\|h_i^k - \nabla f(x^*)\|^2, \quad \sigma_{2,k}^2 = \frac{1}{nm}\sum_{i=1}^{n}\sum_{j=1}^{m}\|\nabla f_{ij}(w_i^k) - \nabla f_{ij}(x^*)\|^2.$$

*Proof.* First of all, we show unbiasedness of $g^k$:

$$\mathbf{E}\left[g^k \mid x^k\right] = \frac{1}{n}\sum_{i=1}^{n}\mathbf{E}\left[\hat{g}_i^k - h_i^k + h^k \mid x^k\right]$$

$$= \frac{1}{nm}\sum_{i=1}^{n}\sum_{j=1}^{m}\left(\nabla f_{ij}(x^k) - \nabla f_{ij}(w_i^k) + \nabla f_i(w_i^k) - h_i^k + h^k\right) = \nabla f(x^k).$$

---

**Algorithm 10** `EC-LSVRG-DIANA`

---

**Input:** learning rates $\gamma > 0$, $\alpha \in (0, 1]$, initial vectors $x^0, h_1^0, \ldots, h_n^0 \in \mathbb{R}^d$

1: Set $e_i^0 = 0$ for all $i = 1, \ldots, n$
2: Set $h^0 = \frac{1}{n} \sum_{i=1}^n h_i^0$
3: **for** $k = 0, 1, \ldots$ **do**
4:     Broadcast $x^k, h^k$ to all workers
5:     **for** $i = 1, \ldots, n$ in parallel **do**
6:         Pick $l$ uniformly at random from $[m]$
7:         Set $\hat{g}_i^k = \nabla f_{il}(x^k) - \nabla f_{il}(w_i^k) + \nabla f_i(w_i^k)$
8:         $g_i^k = \hat{g}_i^k - h_i^k + h^k$
9:         $v_i^k = C(e_i^k + \gamma g_i^k)$
10:         $e_i^{k+1} = e_i^k + \gamma g_i^k - v_i^k$
11:         $h_i^{k+1} = h_i^k + \alpha Q(\hat{g}_i^k - h_i^k)$
12:         $w_i^{k+1} = \begin{cases} x^k, & \text{with probability } p, \\ w_i^k, & \text{with probability } 1 - p \end{cases}$
13:     **end for**
14:     $e^k = \frac{1}{n} \sum_{i=1}^n e_i^k$, $g^k = \frac{1}{n} \sum_{i=1}^n g_i^k$, $v^k = \frac{1}{n} \sum_{i=1}^n v_i^k$, $h^{k+1} = \frac{1}{n} \sum_{i=1}^n h_i^{k+1} = h^k + \alpha \frac{1}{n} \sum_{i=1}^n Q(\hat{g}_i^k - h_i^k)$
15:     $x^{k+1} = x^k - v^k$
16: **end for**

---

Next, we derive the upper bound for $\frac{1}{n} \sum_{i=1}^n \|\bar{g}_i^k\|^2$:

$$\frac{1}{n} \sum_{i=1}^n \|\bar{g}_i^k\|^2 \quad = \quad \frac{1}{n} \sum_{i=1}^n \|\nabla f_i(x^k) - h_i^k + h^k\|^2$$

$$\overset{(31)}{\le} \quad \frac{2}{n} \sum_{i=1}^n \|\nabla f_i(x^k) - \nabla f_i(x^*)\|^2 + \frac{2}{n} \sum_{i=1}^n \left\| h_i^k - \nabla f_i(x^*) - \left( h^k - \nabla f(x^*) \right) \right\|^2$$

$$\overset{(11),(34)}{\le} \quad 4L \left( f(x^k) - f(x^*) \right) + \frac{2}{n} \sum_{i=1}^n \|h_i^k - \nabla f_i(x^*)\|^2.$$

Since the variance of random vector is not greater than its second moment we obtain:

$$\frac{1}{n} \sum_{i=1}^n \mathbf{E} \left[ \|g_i^k - \bar{g}_i^k\|^2 \mid x^k \right] \quad \overset{(34)}{\le} \quad \frac{1}{n} \sum_{i=1}^n \mathbf{E} \left[ \|g_i^k\|^2 \mid x^k \right]$$

$$= \quad \frac{1}{n} \sum_{i=1}^n \mathbf{E} \left[ \|\nabla f_{il}(x^k) - \nabla f_{il}(w_i^k) + \nabla f_i(w_i^k) - h_i^k + h^k\|^2 \mid x^k \right]$$

$$\overset{(31)}{\le} \quad \frac{3}{n} \sum_{i=1}^n \mathbf{E} \left[ \left\| \nabla f_{il}(x^k) - \nabla f_{il}(x^*) \right\|^2 \mid x^k \right]$$

$$+ \frac{3}{n} \sum_{i=1}^n \mathbf{E} \left[ \left\| \nabla f_{il}(w_i^k) - \nabla f_{il}(x^*) - \left( \nabla f_i(w_i^k) - \nabla f_i(x^*) \right) \right\|^2 \mid x^k \right]$$

$$+ \frac{3}{n} \sum_{i=1}^n \left\| h_i^k - \nabla f_i(x^*) - \left( h^k - \nabla f(x^*) \right) \right\|^2$$

$$\overset{(11),(34)}{\le} \quad 6L \left( f(x^k) - f(x^*) \right) + \frac{3}{nm} \sum_{i=1}^n \sum_{j=1}^m \|\nabla f_{ij}(w_i^k) - \nabla f_{ij}(x^*)\|^2$$

$$+ \frac{3}{n} \sum_{i=1}^n \left\| h_i^k - \nabla f_i(x^*) \right\|^2.$$

Finally, we obtain an upper boud for the second moment of $g^k$:

$$\mathbf{E}\left[\|g^k\|^2 \mid x^k\right] \quad = \quad \mathbf{E}\left[\left\|\frac{1}{n}\sum_{i=1}^{n}\left(\nabla f_{il}(x^k) - \nabla f_{il}(w_i^k) + \nabla f_i(w_i^k) - \nabla f_i(x^*)\right)\right\|^2 \mid x^k\right]$$

$$\overset{(31)}{\leq} \quad \frac{2}{n}\sum_{i=1}^{n}\mathbf{E}\left[\|\nabla f_{il}(x^k) - \nabla f_{il}(x^*)\|^2 \mid x^k\right]$$

$$+\frac{2}{n}\sum_{i=1}^{n}\mathbf{E}\left[\left\|\nabla f_{il}(w_i^k) - \nabla f_{il}(x^*) - \left(\nabla f_i(w_i^k) - \nabla f_i(x^*)\right)\right\|^2 \mid x^k\right]$$

$$= \quad \frac{2}{nm}\sum_{i=1}^{n}\sum_{j=1}^{m}\|\nabla f_{ij}(x^k) - \nabla f_{ij}(x^*)\|^2$$

$$+\frac{2}{nm}\sum_{i=1}^{n}\sum_{j=1}^{m}\left\|\nabla f_{ij}(w_i^k) - \nabla f_{ij}(x^*) - \frac{1}{m}\sum_{j=1}^{m}\left(\nabla f_{ij}(w_i^k) - \nabla f_{ij}(x^*)\right)\right\|^2$$

$$\overset{(11),(34)}{\leq} \quad 4L\left(f(x^k) - f(x^*)\right) + \frac{2}{nm}\sum_{i=1}^{n}\sum_{j=1}^{m}\left\|\nabla f_{ij}(w_i^k) - \nabla f_{ij}(x^*)\right\|^2.$$

$\square$

**Lemma J.12.** *Assume that $\alpha \leq 1/(\omega+1)$. Then, for all $k \geq 0$ we have*

$$\mathbf{E}\left[\sigma_{1,k+1}^2 \mid x^k\right] \leq (1-\alpha)\sigma_{1,k}^2 + 6L\alpha(f(x^k) - f(x^*)) + 2\alpha\sigma_{2,k}^2, \tag{123}$$

$$\mathbf{E}\left[\sigma_{2,k+1}^2 \mid x^k\right] \leq (1-p)\sigma_{k,2}^2 + 2Lp\left(f(x^k) - f(x^*)\right) \tag{124}$$

*where $\sigma_{1,k}^2 = \frac{1}{n}\sum_{i=1}^{n}\|h_i^k - \nabla f_i(x^*)\|^2$ and $\sigma_{2,k}^2 = \frac{1}{nm}\sum_{i=1}^{n}\sum_{j=1}^{m}\|\nabla f_{ij}(w_i^k) - \nabla f_{ij}(x^*)\|^2$.*

*Proof.* First of all, we derive an upper bound for the second moment of $h_i^{k+1} - h_i^*$:

$$\mathbf{E}\left[\|h_i^{k+1} - h_i^*\|^2 \mid x^k\right] \quad = \quad \mathbf{E}\left[\left\|h_i^k - h_i^* + \alpha Q(\hat{g}_i^k - h_i^k)\right\|^2 \mid x^k\right]$$

$$\overset{(26)}{=} \quad \|h_i^k - h_i^*\|^2 + 2\alpha\langle h_i^k - h_i^*, \nabla f_i(x^k) - h_i^k\rangle$$
$$+\alpha^2\mathbf{E}\left[\|Q(\hat{g}_i^k - h_i^k)\|^2 \mid x^k\right]$$

$$\overset{(26),(35)}{\leq} \quad \|h_i^k - h_i^*\|^2 + 2\alpha\langle h_i^k - h_i^*, \nabla f_i(x^k) - h_i^k\rangle$$
$$+\alpha^2(\omega + 1)\mathbf{E}\left[\|\hat{g}_i^k - h_i^k\|^2 \mid x^k\right].$$

Using variance decomposition (34) and $\alpha \leq 1/(\omega+1)$ we get

$$\alpha^2(\omega+1)\mathbf{E}\left[\|\hat{g}_i^k - h_i^k\|^2 \mid x^k\right] \quad \overset{(34)}{=} \quad \alpha^2(\omega+1)\mathbf{E}\left[\|\hat{g}_i^k - \nabla f_i(x^k)\|^2 \mid x^k\right] + \alpha^2(\omega+1)\|\nabla f_i(x^k) - h_i^k\|^2$$

$$\leq \quad \alpha\mathbf{E}\left[\|\hat{g}_i^k - \nabla f_i(x^k)\|^2 \mid x^k\right] + \alpha\|\nabla f_i(x^k) - h_i^k\|^2$$

$$\overset{(31)}{\leq} \quad 2\alpha\mathbf{E}\left[\left\|\nabla f_{il}(x^k) - \nabla f_{il}(x^*) - \left(\nabla f_i(x^k) - \nabla f_i(x^*)\right)\right\|^2 \mid x^k\right]$$

$$+2\alpha\mathbf{E}\left[\left\|\nabla f_{il}(w_i^k) - \nabla f_{il}(x^*) - \left(\nabla f_i(w_i^k) - \nabla f_i(x^*)\right)\right\|^2 \mid x^k\right]$$

$$+\alpha\|\nabla f_i(x^k) - h_i^k\|^2$$

$$\overset{(34)}{\leq} \quad 2\alpha\mathbf{E}\left[\left\|\nabla f_{il}(x^k) - \nabla f_{il}(x^*)\right\|^2 \mid x^k\right]$$

$$+2\alpha\mathbf{E}\left[\left\|\nabla f_{il}(w_i^k) - \nabla f_{il}(x^*)\right\|^2 \mid x^k\right] + \alpha\|\nabla f_i(x^k) - h_i^k\|^2$$

$$\overset{(11)}{\leq} \quad 4L\alpha D_{f_i}(x^k, x^*) + \frac{2\alpha}{m}\sum_{j=1}^{m}\|\nabla f_{ij}(w_i^k) - \nabla f_{ij}(x^*)\|^2$$

$$+\alpha\|\nabla f_i(x^k) - h_i^k\|^2$$

Putting all together we obtain

$$
\begin{aligned}
\mathbf{E}\left[\|h_i^{k+1} - h_i^*\|^2 \mid x^k\right] \quad &\leq \quad \|h_i^k - h_i^*\|^2 + \alpha\left\langle \nabla f_i(x^k) - h_i^k, f_i(x^k) + h_i^k - 2h_i^*\right\rangle \\
&\qquad + 4L\alpha D_{f_i}(x^k, x^*) + \frac{2\alpha}{m}\sum_{j=1}^{m}\|\nabla f_{ij}(w_i^k) - \nabla f_{ij}(x^*)\|^2 \\[2mm]
&\overset{(28)}{=} \quad \|h_i^k - h_i^*\|^2 + \alpha\|\nabla f_i(x^k) - h_i^*\|^2 - \alpha\|h_i^k - h_i^*\|^2 \\
&\qquad + 4L\alpha D_{f_i}(x^k, x^*) + \frac{2\alpha}{m}\sum_{j=1}^{m}\|\nabla f_{ij}(w_i^k) - \nabla f_{ij}(x^*)\|^2 \\[2mm]
&\overset{(11)}{\leq} \quad (1-\alpha)\|h_i^k - h_i^*\|^2 + 6L\alpha D_{f_i}(x^k, x^*) \\
&\qquad + \frac{2\alpha}{m}\sum_{j=1}^{m}\|\nabla f_{ij}(w_i^k) - \nabla f_{ij}(x^*)\|^2.
\end{aligned}
$$

Summing up the above inequality for $i = 1, \ldots, n$ we derive

$$
\mathbf{E}\left[\sigma_{1,k+1}^2 \mid x^k\right] \quad \leq \quad (1-\alpha)\sigma_{1,k}^2 + 6L\alpha(f(x^k) - f(x^*)) + 2\alpha\sigma_{2,k}^2.
$$

Similarly to the proof of Lemma J.8 we get

$$
\begin{aligned}
\mathbf{E}\left[\sigma_{2,k+1}^2 \mid x^k\right] \quad &= \quad \frac{1}{nm}\sum_{i=1}^{n}\sum_{j=1}^{m}\mathbf{E}\left[\|\nabla f_{ij}(w_i^{k+1}) - \nabla f_{ij}(x^*)\|^2 \mid x^k\right] \\[2mm]
&= \quad \frac{1-p}{nm}\sum_{i=1}^{n}\sum_{j=1}^{m}\|\nabla f_{ij}(w_i^k) - \nabla f_{ij}(x^*)\|^2 \\[2mm]
&\qquad + \frac{p}{nm}\sum_{i=1}^{n}\sum_{j=1}^{m}\|\nabla f_{ij}(x^k) - \nabla f_{ij}(x^*)\|^2 \\[2mm]
&\overset{(11)}{\leq} \quad (1-p)\sigma_{2,k}^2 + \frac{2Lp}{nm}\sum_{i=1}^{n}\sum_{j=1}^{m}D_{f_{ij}}(x^k, x^*) \\[2mm]
&= \quad (1-p)\sigma_{2,k}^2 + 2Lp\left(f(x^k) - f(x^*)\right).
\end{aligned}
$$

$\qquad\qquad\qquad\qquad\qquad\qquad\qquad\qquad\qquad\qquad\qquad\qquad\qquad\qquad\qquad\qquad\qquad\qquad\qquad\qquad$ $\square$

Applying Theorem G.1 we get the following result.

**Theorem J.8.** *Assume that $f_{ij}(x)$ is convex and $L$-smooth for all $i = 1, \ldots, n$, $j = 1, \ldots, m$ and $f(x)$ is $\mu$-quasi strongly convex. Then* EC-LSVRG-DIANA *satisfies Assumption 3.3 with*

$$
A = A' = 2L, \quad B_1' = B_2 = 0, \quad B_1 = B_2' = 2, \quad D_1 = \widetilde{D}_1 = D_1' = D_2 = D_3 = 0,
$$

$$
\widetilde{A} = 3L, \quad \widetilde{B}_1 = \widetilde{B}_2 = 3, \quad \sigma_{1,k}^2 = \frac{1}{n}\sum_{i=1}^{n}\|h_i^k - \nabla f_i(x^*)\|^2, \quad \rho_1 = \alpha,
$$

$$
\sigma_{2,k}^2 = \frac{1}{nm}\sum_{i=1}^{n}\sum_{j=1}^{m}\|\nabla f_{ij}(w_i^k) - \nabla f_{ij}(x^*)\|^2, \quad \rho_2 = p, \quad C_1 = 3L\alpha, \quad C_2 = Lp,
$$

$$
G = 2, \quad F_1 = \frac{24L\gamma^2\left(\frac{4}{\delta} + 3\right)}{\delta\alpha\left(1 - \min\left\{\frac{\gamma\mu}{2}, \frac{\alpha}{4}, \frac{p}{4}\right\}\right)}, \quad F_2 = \frac{24L\gamma^2\left(\frac{4}{1-\alpha}\left(\frac{4}{\delta} + 3\right) + 3\right)}{\delta p\left(1 - \min\left\{\frac{\gamma\mu}{2}, \frac{\alpha}{4}, \frac{p}{4}\right\}\right)},
$$

*with $\gamma$ and $\alpha$ satisfying*

$$
\gamma \leq \min\left\{\frac{9}{296L}, \frac{\delta}{4L\sqrt{6\left(4 + 3\delta + \frac{2}{1-\alpha}\left(3 + \frac{4}{1-p}\right)(4 + 3\delta) + \frac{6\delta}{1-p}\right)}}\right\}, \quad \alpha \leq \frac{1}{\omega + 1}
$$

*with $M_1 = 0$ and $M_2 = \frac{8}{3p} + \frac{32}{9p}$ and for all $K \geq 0$*

$$\mathbf{E}\left[f(\bar{x}^K) - f(x^*)\right] \leq \left(1 - \min\left\{\frac{\gamma\mu}{2}, \frac{\alpha}{4}, \frac{p}{4}\right\}\right)^K \frac{4(T^0 + \gamma F_1 \sigma_{1,0}^2 + \gamma F_2 \sigma_{2,0}^2)}{\gamma},$$

*when $\mu > 0$ and*

$$\mathbf{E}\left[f(\bar{x}^K) - f(x^*)\right] \leq \frac{4(T^0 + \gamma F_1 \sigma_{1,0}^2 + \gamma F_2 \sigma_{2,0}^2)}{K\gamma}$$

*when $\mu = 0$, where $T^k \stackrel{def}{=} \|x^k - x^*\|^2 + M_2 \gamma^2 \sigma_{2,k}^2$.*

In other words, if $p = 1/m$, $m \geq 2$ and

$$\gamma = \min\left\{\frac{9}{296L}, \frac{\delta}{4L\sqrt{6\left(4 + 3\delta + \frac{2}{1-\alpha}\left(3 + \frac{4}{1-p}\right)(4 + 3\delta) + \frac{6\delta}{1-p}\right)}}\right\}, \quad \alpha = \min\left\{\frac{1}{\omega + 1}, \frac{1}{2}\right\},$$

then `EC-LSVRG-DIANA` converges with the linear rate

$$\mathcal{O}\left(\left(\omega + m + \frac{\kappa}{\delta}\right)\ln\frac{1}{\varepsilon}\right)$$

to the exact solution when $\mu > 0$.

Applying Lemma D.3 we get the complexity result in the case when $\mu = 0$.

**Corollary J.13.** *Let the assumptions of Theorem J.8 hold and $\mu = 0$. Then after $K$ iterations of* `EC-LSVRG-DIANA` *with the stepsize*

$$\gamma_0 = \min\left\{\frac{9}{296L}, \frac{\delta}{4L\sqrt{6\left(4 + 3\delta + \frac{2}{1-\alpha}\left(3 + \frac{4}{1-p}\right)(4 + 3\delta) + \frac{6\delta}{1-p}\right)}}\right\}, \quad R_0 = \|x^0 - x^*\|,$$

$$\gamma = \min\left\{\gamma_0, \sqrt{\frac{9pR_0^2}{56\sigma_{2,0}^2}}, \sqrt[3]{\frac{R_0^2}{\frac{24L\left(\frac{4}{\delta}+3\right)}{\delta\alpha\left(1-\min\left\{\frac{\gamma_0\mu}{2}, \frac{\alpha}{4}, \frac{p}{4}\right\}\right)}\sigma_{1,0}^2 + \frac{24L\left(\frac{4}{1-\alpha}\left(\frac{4}{\delta}+3\right)+3\right)}{\delta p\left(1-\min\left\{\frac{\gamma_0\mu}{2}, \frac{\alpha}{4}, \frac{p}{4}\right\}\right)}\sigma_{2,0}^2}}\right\},$$

*and $p = \frac{1}{m}$, $m \geq 2$, $\alpha = \min\left\{\frac{1}{\omega+1}, \frac{1}{2}\right\}$ we have $\mathbf{E}\left[f(\bar{x}^K) - f(x^*)\right]$ of order*

$$\mathcal{O}\left(\frac{LR_0^2}{\delta K} + \frac{\sqrt{R_0^2 m \sigma_{2,0}^2}}{K} + \frac{\sqrt[3]{LR_0^4((\omega+1)\sigma_{1,0}^2 + m\sigma_{2,0}^2)}}{\delta^{2/3}K}\right).$$

*That is, to achive $\mathbf{E}\left[f(\bar{x}^K) - f(x^*)\right] \leq \varepsilon$* `EC-LSVRG-DIANA` *requires*

$$\mathcal{O}\left(\frac{LR_0^2}{\delta\varepsilon} + \frac{\sqrt{R_0^2 m \sigma_{2,0}^2}}{\varepsilon} + \frac{\sqrt[3]{LR_0^4((\omega+1)\sigma_{1,0}^2 + m\sigma_{2,0}^2)}}{\delta^{2/3}\varepsilon}\right)$$

*iterations.*

Table 4: Complexity of SGD methods with delayed updates established in this paper. Symbols: $\varepsilon$ = error tolerance; $\delta$ = contraction factor of compressor $\mathcal{C}$; $\omega$ = variance parameter of compressor $\mathcal{Q}$; $\kappa = L/\mu$; $\mathcal{L}$ = expected smoothness constant; $\sigma_*^2$ = variance of the stochastic gradients in the solution; $\zeta_*^2$ = average of $\|\nabla f_i(x^*)\|^2$; $\sigma^2$ = average of the uniform bounds for the variances of stochastic gradients of workers; $\mathcal{M}_{2,q} = (\omega+1)\sigma^2 + \omega\zeta_*^2$; $\sigma_q^2 = (1+\omega)\left(1+\frac{\omega}{n}\right)\sigma^2$. †D-QGDstar is a special case of D-QSGDstar where each worker $i$ computes the full gradient $\nabla f_i(x^k)$; ‡D-GD-DIANA is a special case of D-SGD-DIANA where each worker $i$ computes the full gradient $\nabla f_i(x^k)$.

| Problem | Method | Alg # | Citation | Sec # | Rate (constants ignored) |
|---|---|---|---|---|---|
| (1)+(3) | D-SGDsr | Alg 15 | **new** | K.5 | $\widetilde{\mathcal{O}}\left(\frac{\mathcal{L}+\sqrt{L^2\tau^2+L\mathcal{L}\tau}}{\mu} + \frac{\sigma_*^2}{n\mu\varepsilon} + \frac{\sqrt{L\tau\sigma_*^2}}{\mu\sqrt{n\varepsilon}}\right)$ |
| (1)+(2) | D-SGD | Alg 11 | [46] | K.1 | $\widetilde{\mathcal{O}}\left(\tau\kappa + \frac{\sigma_*^2}{n\mu\varepsilon} + \frac{\sqrt{L\tau\sigma_*^2}}{\mu\sqrt{n\varepsilon}}\right)$ |
| (1)+(2) | D-QSGD | Alg 12 | **new** | K.2 | $\widetilde{\mathcal{O}}\left(\kappa\left(\tau+\frac{\omega}{n}\right) + \frac{\mathcal{M}_{2,q}}{n\mu\varepsilon} + \frac{\sqrt{L\tau\mathcal{M}_{2,q}}}{\mu\sqrt{n\varepsilon}}\right)$ |
| (1)+(2) | D-QSGDstar | Alg 13 | **new** | K.3 | $\widetilde{\mathcal{O}}\left(\kappa\left(\tau+\frac{\omega}{n}\right) + \frac{\sigma^2}{n\mu\varepsilon} + \frac{\sqrt{L\tau\sigma^2}}{\mu\sqrt{n\varepsilon}}\right)$ |
| (1)+(2) | D-QGDstar† | Alg 13 | **new** | K.3 | $\mathcal{O}\left(\kappa\left(\tau+\frac{\omega}{n}\right)\log\frac{1}{\varepsilon}\right)$ |
| (1)+(2) | D-SGD-DIANA | Alg 14 | **new** | K.4 | $\widetilde{\mathcal{O}}\left(\omega + \kappa\left(\tau+\frac{\omega}{n}\right) + \frac{\sigma^2}{n\mu\varepsilon} + \frac{\sqrt{L\tau\sigma_q^2}}{\mu\sqrt{n\varepsilon}}\right)$ |
| (1)+(2) | D-GD-DIANA‡ | Alg 14 | **new** | K.4 | $\mathcal{O}\left(\left(\omega + \kappa\left(\tau+\frac{\omega}{n}\right)\right)\log\frac{1}{\varepsilon}\right)$ |
| (1)+(3) | D-LSVRG | Alg 16 | **new** | K.6 | $\mathcal{O}\left((m+\kappa\tau)\log\frac{1}{\varepsilon}\right)$ |
| (1)+(3) | D-QLSVRG | Alg 17 | **new** | K.7 | $\widetilde{\mathcal{O}}\left(m + \kappa\left(\tau+\frac{\omega}{n}\right) + \frac{\zeta_{**}^2}{n\mu\varepsilon} + \frac{\sqrt{L\tau\zeta_*^2}}{\mu\sqrt{n\varepsilon}}\right)$ |
| (1)+(3) | D-QLSVRGstar | Alg 18 | **new** | K.8 | $\mathcal{O}\left(\left(m+\kappa\left(\tau+\frac{\omega}{n}\right)\right)\log\frac{1}{\varepsilon}\right)$ |
| (1)+(3) | D-LSVRG-DIANA | Alg 19 | **new** | K.9 | $\mathcal{O}\left(\left(\omega+m+\kappa\left(\tau+\frac{\omega}{n}\right)\right)\log\frac{1}{\varepsilon}\right)$ |

---

**Algorithm 11** D-SGD

---

**Input:** learning rate $\gamma > 0$, initial vector $x^0 \in \mathbb{R}^d$
1: Set $e_i^0 = 0$ for all $i = 1, \ldots, n$
2: **for** $k = 0, 1, \ldots$ **do**
3:     Broadcast $x^k$ to all workers
4:     **for** $i = 1, \ldots, n$ in parallel **do**
5:         Sample $g_i^k = \nabla f_{\xi_i}(x^k) - \nabla f_i(x^*)$
6:         $v_i^k = \begin{cases} \gamma g_i^{k-\tau}, & \text{if } k \geq \tau, \\ 0, & \text{if } k < \tau \end{cases}$
7:         $e_i^{k+1} = e_i^k + \gamma g_i^k - v_i^k$
8:     **end for**
9:     $e^k = \frac{1}{n}\sum_{i=1}^n e_i^k$, $g^k = \frac{1}{n}\sum_{i=1}^n g_i^k$, $v^k = \frac{1}{n}\sum_{i=1}^n v_i^k = \frac{1}{n}\sum_{i=1}^n \nabla f_{\xi_i}(x^{k-\tau})$
10:     $x^{k+1} = x^k - v^k$
11: **end for**

---

# K  Special Cases: Delayed Updates Methods

## K.1  D-SGD

In this section we consider the same setup as in Section J.2. We notice that vectors $e_i^k$ appear only in the analysis and there is no need to compute them. Moreover, we use $\nabla f_i(x^*)$ in the definition of $g_i^k$ which is problematic at the firt glance. Indeed, workers do not know $\nabla f_i(x^*)$. However, since $0 = \nabla f(x^*) = \frac{1}{n}\nabla f_i(x^*)$ and master node uses averages of $g_i^k$ for the updates one can ignore $\nabla f_i(x^*)$ in $g_i^k$ in the implementation of D-SGD and get exactly the same method. We define $g_i^k$ in such a way only for the theoretical analysis.

**Lemma K.1** (see also Lemmas 1,2 from [39]). *Assume that $f_{\xi_i}(x)$ are convex in $x$ for every $\xi_i$, $i = 1, \ldots, n$. Then for every $x \in \mathbb{R}^d$ and $i = 1, \ldots, n$*

$$\mathbf{E}\left[\|g^k\|^2 \mid x^k\right] \le 4L(f(x^k) - f(x^*)) + \frac{2}{n^2} \sum_{i=1}^n \mathrm{Var}\left[\nabla f_{\xi_i}(x^*)\right]. \tag{125}$$

*If further $f(x)$ is $\mu$-quasi strongly convex with possibly non-convex $f_i, f_{\xi_i}$ and $\mu > 0$, then for every $x \in \mathbb{R}^d$ and $i = 1, \ldots, n$*

$$\mathbf{E}\left[\|g^k\|^2 \mid x^k\right] \le 4L\kappa(f(x^k) - f(x^*)) + \frac{2}{n^2} \sum_{i=1}^n \mathrm{Var}\left[\nabla f_{\xi_i}(x^*)\right], \tag{126}$$

*where $\kappa = \frac{L}{\mu}$.*

*Proof.* By definition of $g^k$ we have

$$
\begin{aligned}
\mathbf{E}\left[\|g^k\|^2 \mid x^k\right] \quad &= \quad \mathbf{E}\left[\left\|\frac{1}{n}\sum_{i=1}^n \left(\nabla f_{\xi_i}(x^k) - \nabla f_{\xi_i}(x^*) + \nabla f_{\xi_i}(x^*) - \nabla f_i(x^*)\right)\right\|^2 \mid x^k\right] \\
&\overset{(31)}{\le} \quad 2\mathbf{E}\left[\left\|\frac{1}{n}\sum_{i=1}^n \left(\nabla f_{\xi_i}(x^k) - \nabla f_{\xi_i}(x^*)\right)\right\|^2 \mid x^k\right] \\
&\qquad + 2\,\mathbf{E}\left[\underbrace{\left\|\frac{1}{n}\sum_{i=1}^n \left(\nabla f_{\xi_i}(x^*) - \nabla f_i(x^*)\right)\right\|^2}_{\mathrm{Var}\left[\frac{1}{n}\sum_{i=1}^n \nabla f_{\xi_i}(x^*)\right]}\right] \\
&\overset{(31)}{\le} \quad \frac{2}{n}\sum_{i=1}^n \mathbf{E}\left[\|\nabla f_{\xi_i}(x^k) - \nabla f_{\xi_i}(x^*)\|^2 \mid x^k\right] \\
&\qquad + \frac{2}{n^2}\sum_{i=1}^n \underbrace{\mathbf{E}\left[\|\nabla f_{\xi_i}(x^*) - \nabla f_i(x^*)\|^2\right]}_{\mathrm{Var}\left[\nabla f_{\xi_i}(x^*)\right]}, \tag{127}
\end{aligned}
$$

where in the last inequality we use independence of $\nabla f_{\xi_i}(x^*)$, $i = 1, \ldots, n$. Using this we derive inequality (125):

$$
\begin{aligned}
\mathbf{E}\left[\|g^k\|^2 \mid x^k\right] \quad &\overset{(127),(11)}{\le} \quad \frac{4L}{n}\sum_{i=1}^n \mathbf{E}\left[D_{f_{\xi_i}}(x^k, x^*) \mid x^k\right] + \frac{2}{n^2}\sum_{i=1}^n \mathrm{Var}\left[\nabla f_{\xi_i}(x^*)\right] \\
&\quad = \quad \frac{4L}{n}\sum_{i=1}^n D_{f_i}(x^k, x^*) + \frac{2}{n^2}\sum_{i=1}^n \mathrm{Var}\left[\nabla f_{\xi_i}(x^*)\right] \\
&\quad = \quad 4L\left(f(x^k) - f(x^*)\right) + \frac{2}{n^2}\sum_{i=1}^n \mathrm{Var}\left[\nabla f_{\xi_i}(x^*)\right].
\end{aligned}
$$

Next, if $f(x)$ is $\mu$-quasi strongly convex, but $f_i, f_{\xi_i}$ are not necessary convex, we obtain

$$
\begin{aligned}
\mathbf{E}\left[\|g^k\|^2 \mid x^k\right] \quad &\overset{(127),(10)}{\le} \quad \frac{2L^2}{n}\sum_{i=1}^n \|x^k - x^*\|^2 + \frac{2}{n^2}\sum_{i=1}^n \mathrm{Var}\left[\nabla f_{\xi_i}(x^*)\right] \\
&\quad \overset{(9)}{\le} \quad \frac{4L^2}{\mu}\left(f(x^k) - f(x^*)\right) + \frac{2}{n^2}\sum_{i=1}^n \mathrm{Var}\left[\nabla f_{\xi_i}(x^*)\right].
\end{aligned}
$$

$\square$

**Theorem K.1.** *Assume that $f_\xi(x)$ is convex in $x$ for every $\xi$. Then* `D-SGD` *satisfies Assumption 3.4 with*

$$A' = 2L, \quad B_1' = B_2' = 0, \quad D_1' = \frac{2}{n^2} \sum_{i=1}^{n} \mathrm{Var}\left[\nabla f_{\xi_i}(x^*)\right], \quad \sigma_{1,k}^2 \equiv \sigma_{2,k}^2 \equiv 0$$

$$\rho_1 = \rho_2 = 1, \quad C_1 = C_2 = 0, \quad D_2 = 0$$

$$F_1 = F_2 = 0, \quad D_3 = \frac{6\gamma\tau L}{n^2} \sum_{i=1}^{n} \mathrm{Var}\left[\nabla f_{\xi_i}(x^*)\right]$$

*with $\gamma$ satisfying*

$$\gamma \leq \frac{1}{8L\sqrt{2\tau(\tau+2)}}$$

*and for all $K \geq 0$*

$$\mathbf{E}\left[f(\bar{x}^K) - f(x^*)\right] \leq \left(1 - \frac{\gamma\mu}{2}\right)^K \frac{4\|x^0 - x^*\|^2}{\gamma} + \frac{8\gamma}{n^2}(1 + 3L\gamma\tau) \sum_{i=1}^{n} \mathrm{Var}\left[\nabla f_{\xi_i}(x^*)\right]$$

*when $\mu > 0$ and*

$$\mathbf{E}\left[f(\bar{x}^K) - f(x^*)\right] \leq \frac{4\|x^0 - x^*\|^2}{\gamma K} + \frac{8\gamma}{n^2}(1 + 3L\gamma\tau) \sum_{i=1}^{n} \mathrm{Var}\left[\nabla f_{\xi_i}(x^*)\right]$$

*when $\mu = 0$. If further $f_i(x)$ are $\mu$-strongly convex with possibly non-convex $f_{\xi_i}$ and $\mu > 0$, then* `D-SGD` *satisfies Assumption 3.4 with*

$$A' = 2\kappa L, \quad B_1' = B_2' = 0, \quad D_1' = \frac{2}{n^2} \sum_{i=1}^{n} \mathrm{Var}\left[\nabla f_{\xi_i}(x^*)\right], \quad \sigma_{1,k}^2 \equiv \sigma_{2,k}^2 \equiv 0,$$

$$\rho_1 = \rho_2 = 1, \quad C_1 = C_2 = 0, \quad D_2 = 0, \quad G = 0,$$

$$F_1 = F_2 = 0, \quad D_3 = \frac{6\gamma\tau L}{n^2} \sum_{i=1}^{n} \mathrm{Var}\left[\nabla f_{\xi_i}(x^*)\right]$$

*with $\gamma$ satisfying*

$$\gamma \leq \min\left\{\frac{1}{8\kappa L}, \frac{1}{8L\sqrt{2\tau(\tau+2\kappa)}}\right\}$$

*and for all $K \geq 0$*

$$\mathbf{E}\left[f(\bar{x}^K) - f(x^*)\right] \leq \left(1 - \frac{\gamma\mu}{2}\right)^K \frac{4\|x^0 - x^*\|^2}{\gamma} + \frac{8\gamma}{n^2}(1 + 3L\gamma\tau) \sum_{i=1}^{n} \mathrm{Var}\left[\nabla f_{\xi_i}(x^*)\right].$$

In other words, `D-SGD` converges with linear rate $\mathcal{O}\left(\tau\kappa\ln\frac{1}{\varepsilon}\right)$ to the neighbourhood of the solution when $\mu > 0$. Applying Lemma D.2 we establish the rate of convergence to $\varepsilon$-solution.

**Corollary K.1.** *Let the assumptions of Theorem K.1 hold, $f_\xi(x)$ are convex for each $\xi$ and $\mu > 0$. Then after $K$ iterations of* `D-SGD` *with the stepsize*

$$\gamma = \min\left\{\frac{1}{8L\sqrt{2\tau(\tau+2)}}, \frac{\ln\left(\max\left\{2, \min\left\{\frac{\|x^0-x^*\|^2\mu^2 K^2}{D_1'}, \frac{\|x^0-x^*\|^2\mu^3 K^3}{3\tau L D_1}\right\}\right\}\right)}{\mu K}\right\}$$

*we have*

$$\mathbf{E}\left[f(\bar{x}^K) - f(x^*)\right] = \widetilde{\mathcal{O}}\left(L\tau\|x^0 - x^*\|^2 \exp\left(-\frac{\mu}{\tau L}K\right) + \frac{D_1'}{\mu K} + \frac{L\tau D_1'}{\mu^2 K^2}\right).$$

*That is, to achive $\mathbf{E}\left[f(\bar{x}^K) - f(x^*)\right] \leq \varepsilon$* `D-SGD` *requires*

$$\widetilde{\mathcal{O}}\left(\frac{\tau L}{\mu} + \frac{D_1'}{\mu\varepsilon} + \frac{\sqrt{L\tau D_1'}}{\mu\sqrt{\varepsilon}}\right) \quad \text{iterations.}$$

**Corollary K.2.** *Let the assumptions of Theorem K.1 hold and $f(x)$ is $\mu$-strongly convex with $\mu > 0$ and possibly non-convex $f_i, f_{\xi_i}$. Then after $K$ iterations of* `D-SGD` *with the stepsize*

$$\gamma = \min\left\{\frac{1}{8\kappa L}, \frac{1}{8L\sqrt{2\tau(\tau + 2\kappa)}}, \frac{\ln\left(\max\left\{2, \min\left\{\frac{\|x^0 - x^*\|^2 \mu^2 K^2}{D_1'}, \frac{\|x^0 - x^*\|^2 \mu^3 K^3}{L\tau D_1'}\right\}\right\}\right)}{\mu K}\right\}$$

*we have* $\mathbf{E}\left[f(\bar{x}^K) - f(x^*)\right]$ *of order*

$$\widetilde{\mathcal{O}}\left(L\left(\kappa + \tau\sqrt{\kappa}\right)\|x^0 - x^*\|^2 \exp\left(-\min\left\{\frac{\mu}{\tau L\sqrt{\kappa}}, \frac{1}{\kappa^2}\right\}K\right) + \frac{D_1'}{\mu K} + \frac{L\tau D_1'}{\mu^2 K^2}\right).$$

*That is, to achive* $\mathbf{E}\left[f(\bar{x}^K) - f(x^*)\right] \leq \varepsilon$ `D-SGD` *requires*

$$\widetilde{\mathcal{O}}\left(\kappa^2 + \tau\kappa^{3/2} + \frac{D_1'}{\mu\varepsilon} + \frac{\sqrt{L\tau D_1'}}{\mu\sqrt{\varepsilon}}\right) \quad \text{iterations.}$$

Applying Lemma D.3 we get the complexity result in the case when $\mu = 0$.

**Corollary K.3.** *Let the assumptions of Theorem K.1 hold, $f_\xi(x)$ are convex for each $\xi$ and $\mu = 0$. Then after $K$ iterations of* `D-SGD` *with the stepsize*

$$\gamma = \min\left\{\frac{1}{8L\sqrt{2\tau(\tau + 2)}}, \sqrt{\frac{\|x^0 - x^*\|^2}{D_1'K}}, \sqrt[3]{\frac{\|x^0 - x^*\|^2}{3L\tau D_1'K}}\right\}$$

*we have* $\mathbf{E}\left[f(\bar{x}^K) - f(x^*)\right]$ *of order*

$$\mathcal{O}\left(\frac{\tau L R_0^2}{K} + \sqrt{\frac{R_0^2 \tau D_1'}{K}} + \frac{\sqrt[3]{L R_0^4 \tau D_1'}}{K^{2/3}}\right)$$

*where $R_0 = \|x^0 - x^*\|$. That is, to achive* $\mathbf{E}\left[f(\bar{x}^K) - f(x^*)\right] \leq \varepsilon$ `D-SGD` *requires*

$$\mathcal{O}\left(\frac{\tau L R_0^2}{\varepsilon} + \frac{R_0^2 D_1'}{\varepsilon^2} + \frac{R_0^2 \sqrt{L\tau D_1'}}{\varepsilon^{3/2}}\right)$$

*iterations.*

## K.2 `D-QSGD`

In this section we show how one can combine delayed updates with quantization using our scheme.

**Lemma K.2.** *Assume that $f_i(x)$ is convex and $L$-smooth for all $i = 1, \ldots, n$. Then, for all $k \geq 0$ we have*

$$\mathbf{E}\left[g^k \mid x^k\right] = \nabla f(x^k),$$

$$\mathbf{E}\left[\|g^k\|^2 \mid x^k\right] \leq 2L\left(1 + \frac{2\omega}{n}\right)\left(f(x^k) - f(x^*)\right) + \frac{(\omega + 1)D}{n} + \frac{2\omega}{n^2}\sum_{i=1}^{n}\|\nabla f_i(x^*)\|^2$$

*where $D = \frac{1}{n}\sum_{i=1}^{n} D_i$.*

*Proof.* First of all, we show unbiasedness of $g^k$:

$$\mathbf{E}\left[g^k \mid x^k\right] = \frac{1}{n}\sum_{i=1}^{n}\mathbf{E}\left[g_i^k \mid x^k\right] = \frac{1}{n}\sum_{i=1}^{n}\mathbf{E}\left[\mathbf{E}_Q\left[Q(\hat{g}_i^k) - \nabla f_i(x^*)\right] \mid x^k\right]$$

$$\overset{(26)}{=} \frac{1}{n}\sum_{i=1}^{n}\left(\nabla f_i(x^k) - \nabla f_i(x^*)\right) = \nabla f(x^k),$$

---

**Algorithm 12** D-QSGD

---

**Input:** learning rate $\gamma > 0$, initial vector $x^0 \in \mathbb{R}^d$

1: Set $e_i^0 = 0$ for all $i = 1, \ldots, n$
2: **for** $k = 0, 1, \ldots$ **do**
3:     Broadcast $x^{k-\tau}$ to all workers
4:     **for** $i = 1, \ldots, n$ **do**
5:         Sample $\hat{g}_i^{k-\tau}$ independently from other nodes such that $\mathbf{E}[\hat{g}_i^{k-\tau} \mid x^{k-\tau}] = \nabla f_i(x^{k-\tau})$ and $\mathbf{E}\left[\|\hat{g}_i^{k-\tau} - \nabla f_i(x^{k-\tau})\|^2 \mid x^{k-\tau}\right] \leq D_i$
6:         $g_i^{k-\tau} = Q(\hat{g}_i^{k-\tau}) - \nabla f_i(x^*)$ (quantization is performed independently from other nodes)
7:         $v_i^k = \gamma g_i^{k-\tau}$
8:         $e_i^{k+1} = e_i^k + \gamma g_i^k - v_i^k$
9:     **end for**
10:    $e^k = \frac{1}{n}\sum_{i=1}^{n} e_i^k, \quad g^k = \frac{1}{n}\sum_{i=1}^{n} g_i^k, \quad v^k = \frac{1}{n}\sum_{i=1}^{n} v_i^k = \frac{\gamma}{n}\sum_{i=1}^{n} g_i^{k-\tau} = \frac{\gamma}{n}\sum_{i=1}^{n} Q(\hat{g}_i^{k-\tau})$
11:    $x^{k+1} = x^k - v^k$
12: **end for**

---

where $\mathbf{E}_Q\left[\cdot\right]$ denotes mathematical expectation w.r.t. the randomness coming only from the quantization. Next, we derive the upper bound for the second moment of $g^k$:

$$
\begin{aligned}
\mathbf{E}_Q\left[\|g^k\|^2\right] &= \mathbf{E}_Q\left[\left\|\frac{1}{n}\sum_{i=1}^{n}\left(Q(\hat{g}_i^k) - \nabla f_i(x^*)\right)\right\|^2\right] \\
&\overset{(34)}{=} \mathbf{E}_Q\left[\left\|\frac{1}{n}\sum_{i=1}^{n}\left(Q(\hat{g}_i^k) - \hat{g}_i^k\right)\right\|^2\right] + \left\|\frac{1}{n}\sum_{i=1}^{n}\left(\hat{g}_i^k - \nabla f_i(x^*)\right)\right\|^2. \quad (128)
\end{aligned}
$$

Since $Q(\hat{g}_1^k), \ldots, Q(\hat{g}_n^k)$ are independent quantizations, we get

$$
\begin{aligned}
\mathbf{E}_Q\left[\|g^k\|^2\right] &\overset{(128)}{\leq} \frac{1}{n^2}\sum_{i=1}^{n}\mathbf{E}_Q\left[\|Q(\hat{g}_i^k) - \hat{g}_i^k\|^2\right] + \left\|\frac{1}{n}\sum_{i=1}^{n}\left(\hat{g}_i^k - \nabla f_i(x^*)\right)\right\|^2 \\
&\overset{(26)}{\leq} \frac{\omega}{n^2}\sum_{i=1}^{n}\|\hat{g}_i^k\|^2 + \left\|\frac{1}{n}\sum_{i=1}^{n}\left(\hat{g}_i^k - \nabla f_i(x^*)\right)\right\|^2.
\end{aligned}
$$

Taking conditional expectation $\mathbf{E}\left[\cdot \mid x^k\right]$ from the both sides of the previous inequality we obtain

$$
\begin{aligned}
\mathbf{E}\left[\|g^k\|^2 \mid x^k\right] &\leq \frac{\omega}{n^2}\sum_{i=1}^{n}\mathbf{E}\left[\|\hat{g}_i^k\|^2 \mid x^k\right] + \mathbf{E}\left[\left\|\frac{1}{n}\sum_{i=1}^{n}\left(\hat{g}_i^k - \nabla f_i(x^*)\right)\right\|^2 \mid x^k\right] \\
&\overset{(34)}{\leq} \frac{\omega}{n^2}\sum_{i=1}^{n}\|\nabla f_i(x^k)\|^2 + \frac{\omega}{n^2}\sum_{i=1}^{n}\mathbf{E}\left[\|\hat{g}_i^k - \nabla f_i(x^k)\|^2 \mid x^k\right] \\
&\quad + \underbrace{\left\|\frac{1}{n}\sum_{i=1}^{n}\left(\nabla f_i(x^k) - \nabla f_i(x^*)\right)\right\|^2}_{\|\nabla f(x^k) - \nabla f(x^*)\|^2} + \mathbf{E}\left[\left\|\frac{1}{n}\sum_{i=1}^{n}\left(\hat{g}_i^k - \nabla f_i(x^k)\right)\right\|^2 \mid x^k\right].
\end{aligned}
$$

It remains to estimate terms in the second and the third lines of the previous inequality:

$$\frac{\omega}{n^2} \sum_{i=1}^{n} \|\nabla f_i(x^k)\|^2 \overset{(31)}{\leq} \frac{2\omega}{n^2} \sum_{i=1}^{n} \|\nabla f_i(x^k) - \nabla f_i(x^*)\|^2 + \frac{2\omega}{n^2} \sum_{i=1}^{n} \|\nabla f_i(x^*)\|^2$$

$$\overset{(11)}{\leq} \frac{4\omega L}{n} \left( f(x^k) - f(x^*) \right) + \frac{2\omega}{n^2} \sum_{i=1}^{n} \|\nabla f_i(x^*)\|^2,$$

$$\frac{\omega}{n} \sum_{i=1}^{n} \mathbf{E}\left[ \|\hat{g}_i^k - \nabla f_i(x^k)\|^2 \mid x^k \right] \leq \frac{\omega}{n^2} \sum_{i=1}^{n} D_i = \frac{\omega D}{n},$$

$$\|\nabla f(x^k) - \nabla f(x^*)\|^2 \overset{(11)}{\leq} 2L \left( f(x^k) - f(x^*) \right),$$

$$\mathbf{E}\left[ \left\| \frac{1}{n} \sum_{i=1}^{n} \left( \hat{g}_i^k - \nabla f_i(x^k) \right) \right\|^2 \mid x^k \right] = \frac{1}{n^2} \sum_{i=1}^{n} \mathbf{E}\left[ \|\hat{g}_i^k - \nabla f_i(x^k)\|^2 \mid x^k \right]$$

$$\leq \frac{1}{n^2} \sum_{i=1}^{n} D_i = \frac{D}{n}.$$

Putting all together we get

$$\mathbf{E}\left[ \|g^k\|^2 \mid x^k \right] \leq 2L \left( 1 + \frac{2\omega}{n} \right) \left( f(x^k) - f(x^*) \right) + \frac{(\omega+1)D}{n} + \frac{2\omega}{n^2} \sum_{i=1}^{n} \|\nabla f_i(x^*)\|^2.$$

$\square$

**Theorem K.2.** *Assume that $f_i(x)$ is convex and $L$-smooth for all $i = 1, \ldots, n$ and $f(x)$ is $\mu$-quasi strongly convex. Then* D-QSGD *satisfies Assumption 3.4 with*

$$A' = L\left( 1 + \frac{2\omega}{n} \right), \quad B_1' = B_2' = 0, \quad D_1' = \frac{(\omega+1)D}{n} + \frac{2\omega}{n^2} \sum_{i=1}^{n} \|\nabla f_i(x^*)\|^2,$$

$$\sigma_{1,k}^2 \equiv \sigma_{2,k}^2 \equiv 0, \quad \rho_1 = \rho_2 = 1, \quad C_1 = C_2 = 0, \quad D_2 = 0$$

$$F_1 = F_2 = 0, \quad G = 0, \quad D_3 = \frac{3\gamma\tau L}{n} \left( (\omega+1)D + \frac{2\omega}{n} \sum_{i=1}^{n} \|\nabla f_i(x^*)\|^2 \right)$$

*with $\gamma$ satisfying*

$$\gamma \leq \min\left\{ \frac{1}{4L(1 + 2\omega/n)}, \frac{1}{8L\sqrt{2\tau\left( \tau + 1 + 2\omega/n \right)}} \right\}$$

*and for all $K \geq 0$*

$$\mathbf{E}\left[ f(\bar{x}^K) - f(x^*) \right] \leq \left( 1 - \frac{\gamma\mu}{2} \right)^K \frac{4\|x^0 - x^*\|^2}{\gamma} + \gamma\left( D_1' + D_3 \right)$$

*when $\mu > 0$ and*

$$\mathbf{E}\left[ f(\bar{x}^K) - f(x^*) \right] \leq \frac{4\|x^0 - x^*\|^2}{\gamma K} + \gamma\left( D_1' + D_3 \right)$$

*when $\mu = 0$.*

In other words, D-QSGD converges with the linear rate

$$\mathcal{O}\left( \left( \kappa\left( 1 + \frac{\omega}{n} \right) + \kappa\sqrt{\tau\left( \tau + \frac{\omega}{n} \right)} \right) \ln\frac{1}{\varepsilon} \right)$$

to the neighbourhood of the solution when $\mu > 0$. Applying Lemma D.2 we establish the rate of convergence to $\varepsilon$-solution.

**Corollary K.4.** *Let the assumptions of Theorem K.2 hold, $f_\xi(x)$ are convex for each $\xi$ and $\mu > 0$. Then after $K$ iterations of* D-QSGD *with the stepsize*

$$\gamma_0 = \min\left\{\frac{1}{4L(1 + 2\omega/n)}, \frac{1}{8L\sqrt{2\tau\left(\tau + 1 + 2\omega/n\right)}}\right\}, \quad R_0 = \|x^0 - x^*\|,$$

$$\gamma = \min\left\{\gamma_0, \frac{\ln\left(\max\left\{2, \min\left\{\frac{R_0^2\mu^2 K^2}{D_1'}, \frac{R_0^2\mu^3 K^3}{3\tau L D_1'}\right\}\right\}\right)}{\mu K}\right\}$$

*we have* $\mathbf{E}\left[f(\bar{x}^K) - f(x^*)\right]$ *of order*

$$\widetilde{\mathcal{O}}\left(LR_0^2\left(1 + \frac{\omega}{n} + \sqrt{\tau\left(\tau + \frac{\omega}{n}\right)}\right)\exp\left(-\frac{\mu}{L\left(1 + \frac{\omega}{n} + \sqrt{\tau\left(\tau + \frac{\omega}{n}\right)}\right)}K\right) + \frac{D_1'}{\mu K} + \frac{L\tau D_1'}{\mu^2 K^2}\right).$$

*That is, to achive* $\mathbf{E}\left[f(\bar{x}^K) - f(x^*)\right] \leq \varepsilon$ D-QSGD *requires*

$$\widetilde{\mathcal{O}}\left(\frac{L}{\mu}\left(1 + \frac{\omega}{n}\right) + \frac{L}{\mu}\sqrt{\tau\left(\tau + \frac{\omega}{n}\right)} + \frac{D_1'}{\mu\varepsilon} + \frac{\sqrt{L\tau D_1'}}{\mu\sqrt{\varepsilon}}\right) \quad \text{iterations.}$$

Applying Lemma D.3 we get the complexity result in the case when $\mu = 0$.

**Corollary K.5.** *Let the assumptions of Theorem K.2 hold and $\mu = 0$. Then after $K$ iterations of* D-QSGD *with the stepsize*

$$\gamma_0 = \min\left\{\frac{1}{4L(1 + 2\omega/n)}, \frac{1}{8L\sqrt{2\tau\left(\tau + 1 + 2\omega/n\right)}}\right\},$$

$$\gamma = \min\left\{\gamma_0, \sqrt{\frac{\|x^0 - x^*\|^2}{D_1' K}}, \sqrt[3]{\frac{\|x^0 - x^*\|^2}{3L\tau D_1' K}}\right\}$$

*we have* $\mathbf{E}\left[f(\bar{x}^K) - f(x^*)\right]$ *of order*

$$\mathcal{O}\left(\frac{LR_0^2\left(1 + \frac{\omega}{n}\right)}{K} + \frac{LR_0^2\sqrt{\tau\left(\tau + \frac{\omega}{n}\right)}}{K} + \sqrt{\frac{R_0^2 D_1'}{K}} + \frac{\sqrt[3]{LR_0^4\tau D_1'}}{K^{2/3}}\right)$$

*where $R_0 = \|x^0 - x^*\|$. That is, to achive* $\mathbf{E}\left[f(\bar{x}^K) - f(x^*)\right] \leq \varepsilon$ D-QSGD *requires*

$$\mathcal{O}\left(\frac{LR_0^2\left(1 + \frac{\omega}{n}\right)}{\varepsilon} + \frac{LR_0^2\sqrt{\tau\left(\tau + \frac{\omega}{n}\right)}}{\varepsilon} + \frac{R_0^2 D_1'}{\varepsilon^2} + \frac{R_0^2\sqrt{L\tau D_1'}}{\varepsilon^{3/2}}\right)$$

*iterations.*

### K.3 D-QSGDstar

As we saw in Section K.2 D-QSGD fails to converge to the exact optimum asymptotically even if $\hat{g}_i^k = \nabla f_i(x^k)$ for all $i = 1, \ldots, n$ almost surely, i.e., all $D_i = 0$ for all $i = 1, \ldots, n$. As for EC-GDstar we assume now that $i$-th worker has an access to $\nabla f_i(x^*)$. Using this one can construct the method with delayed updates that converges asymptotically to the exact solution when the full gradients are available.

**Lemma K.3.** *Assume that $f_i(x)$ is convex and $L$-smooth for all $i = 1, \ldots, n$. Then, for all $k \geq 0$ we have*

$$\mathbf{E}\left[g^k \mid x^k\right] = \nabla f(x^k), \tag{129}$$

$$\mathbf{E}\left[\|g^k\|^2 \mid x^k\right] \leq 2L\left(1 + \frac{\omega}{n}\right)\left(f(x^k) - f(x^*)\right) + \frac{(\omega + 1)D}{n} \tag{130}$$

*where $D = \frac{1}{n}\sum_{i=1}^n D_i$.*

**Algorithm 13** D-QSGDstar

**Input:** learning rate $\gamma > 0$, initial vector $x^0 \in \mathbb{R}^d$
1: Set $e_i^0 = 0$ for all $i = 1, \ldots, n$
2: **for** $k = 0, 1, \ldots$ **do**
3:     Broadcast $x^{k-\tau}$ to all workers
4:     **for** $i = 1, \ldots, n$ **do**
5:         Sample $\hat{g}_i^{k-\tau}$ independently from other nodes such that $\mathbf{E}[\hat{g}_i^{k-\tau} \mid x^{k-\tau}] = \nabla f_i(x^{k-\tau})$ and $\mathbf{E}\left[\|\hat{g}_i^{k-\tau} - \nabla f_i(x^{k-\tau})\|^2 \mid x^{k-\tau}\right] \leq D_i$
6:         $g_i^{k-\tau} = Q(\hat{g}_i^{k-\tau} - \nabla f_i(x^*))$ (quantization is performed independently from other nodes)
7:         $v_i^k = \gamma g_i^{k-\tau}$
8:         $e_i^{k+1} = e_i^k + \gamma g_i^k - v_i^k$
9:     **end for**
10:     $e^k = \frac{1}{n}\sum_{i=1}^n e_i^k, \quad g^k = \frac{1}{n}\sum_{i=1}^n g_i^k, \quad v^k = \frac{1}{n}\sum_{i=1}^n v_i^k = \frac{\gamma}{n}\sum_{i=1}^n g_i^{k-\tau} = \frac{\gamma}{n}\sum_{i=1}^n Q(\hat{g}_i^{k-\tau} - \nabla f_i(x^*))$
11:     $x^{k+1} = x^k - v^k$
12: **end for**

*Proof.* First of all, we show unbiasedness of $g^k$:

$$
\begin{aligned}
\mathbf{E}\left[g^k \mid x^k\right] &= \frac{1}{n}\sum_{i=1}^n \mathbf{E}\left[g_i^k \mid x^k\right] = \frac{1}{n}\sum_{i=1}^n \mathbf{E}\left[\mathbf{E}_Q\left[Q(\hat{g}_i^k - \nabla f_i(x^*))\right] \mid x^k\right] \\
&\overset{(26)}{=} \frac{1}{n}\sum_{i=1}^n \left(\nabla f_i(x^k) - \nabla f_i(x^*)\right) = \nabla f(x^k),
\end{aligned}
$$

where $\mathbf{E}_Q\left[\cdot\right]$ denotes mathematical expectation w.r.t. the randomness coming only from the quantization. Next, we derive the upper bound for the second moment of $g^k$:

$$
\begin{aligned}
\mathbf{E}_Q\left[\|g^k\|^2\right] &= \mathbf{E}_Q\left[\left\|\frac{1}{n}\sum_{i=1}^n \left(Q\left(\hat{g}_i^k - \nabla f_i(x^*)\right)\right)\right\|^2\right] \\
&\overset{(34)}{=} \mathbf{E}_Q\left[\left\|\frac{1}{n}\sum_{i=1}^n \left(Q\left(\hat{g}_i^k - \nabla f_i(x^*)\right) - \left(\hat{g}_i^k - \nabla f_i(x^*)\right)\right)\right\|^2\right] \\
&\quad + \left\|\frac{1}{n}\sum_{i=1}^n \left(\hat{g}_i^k - \nabla f_i(x^*)\right)\right\|^2.
\end{aligned}
\tag{131}
$$

Since $Q\left(\hat{g}_1^k - \nabla f_1(x^*)\right), \ldots, Q\left(\hat{g}_n^k - \nabla f_n(x^*)\right)$ are independent quantizations, we get

$$
\begin{aligned}
\mathbf{E}_Q\left[\|g^k\|^2\right] &\overset{(131)}{\leq} \frac{1}{n^2}\sum_{i=1}^n \mathbf{E}_Q\left[\left\|Q\left(\hat{g}_i^k - \nabla f_i(x^*)\right) - \left(\hat{g}_i^k - \nabla f_i(x^*)\right)\right\|^2\right] \\
&\quad + \left\|\frac{1}{n}\sum_{i=1}^n \left(\hat{g}_i^k - \nabla f_i(x^*)\right)\right\|^2 \\
&\overset{(26)}{\leq} \frac{\omega}{n^2}\sum_{i=1}^n \|\hat{g}_i^k - \nabla f_i(x^*)\|^2 + \left\|\frac{1}{n}\sum_{i=1}^n \left(\hat{g}_i^k - \nabla f_i(x^*)\right)\right\|^2.
\end{aligned}
$$

Taking conditional expectation $\mathbf{E}\left[\cdot \mid x^k\right]$ from the both sides of the previous inequality we obtain

$$
\begin{aligned}
\mathbf{E}\left[\|g^k\|^2 \mid x^k\right] \;\leq\; & \frac{\omega}{n^2}\sum_{i=1}^{n}\mathbf{E}\left[\|\hat{g}_i^k - \nabla f_i(x^*)\|^2 \mid x^k\right] + \mathbf{E}\left[\left\|\frac{1}{n}\sum_{i=1}^{n}\left(\hat{g}_i^k - \nabla f_i(x^*)\right)\right\|^2 \mid x^k\right] \\
\overset{(34)}{\leq} \; & \frac{\omega}{n^2}\sum_{i=1}^{n}\|\nabla f_i(x^k) - \nabla f_i(x^*)\|^2 + \frac{\omega}{n^2}\sum_{i=1}^{n}\mathbf{E}\left[\|\hat{g}_i^k - \nabla f_i(x^k)\|^2 \mid x^k\right] \\
& + \underbrace{\left\|\frac{1}{n}\sum_{i=1}^{n}\left(\nabla f_i(x^k) - \nabla f_i(x^*)\right)\right\|^2}_{\|\nabla f(x^k) - \nabla f(x^*)\|^2} + \mathbf{E}\left[\left\|\frac{1}{n}\sum_{i=1}^{n}\left(\hat{g}_i^k - \nabla f_i(x^k)\right)\right\|^2 \mid x^k\right].
\end{aligned}
$$

It remains to estimate terms in the second and the third lines of the previous inequality:

$$
\frac{\omega}{n^2}\sum_{i=1}^{n}\|\nabla f_i(x^k) - \nabla f_i(x^*)\|^2 \;\overset{(11)}{\leq}\; \frac{2\omega L}{n}\left(f(x^k) - f(x^*)\right),
$$

$$
\frac{\omega}{n}\sum_{i=1}^{n}\mathbf{E}\left[\|\hat{g}_i^k - \nabla f_i(x^k)\|^2 \mid x^k\right] \;\leq\; \frac{\omega}{n^2}\sum_{i=1}^{n}D_i = \frac{\omega D}{n},
$$

$$
\|\nabla f(x^k) - \nabla f(x^*)\|^2 \;\overset{(11)}{\leq}\; 2L\left(f(x^k) - f(x^*)\right),
$$

$$
\mathbf{E}\left[\left\|\frac{1}{n}\sum_{i=1}^{n}\left(\hat{g}_i^k - \nabla f_i(x^k)\right)\right\|^2 \mid x^k\right] \;=\; \frac{1}{n^2}\sum_{i=1}^{n}\mathbf{E}\left[\|\hat{g}_i^k - \nabla f_i(x^k)\|^2 \mid x^k\right]
$$

$$
\;\leq\; \frac{1}{n^2}\sum_{i=1}^{n}D_i = \frac{D}{n}.
$$

Putting all together we get

$$
\mathbf{E}\left[\|g^k\|^2 \mid x^k\right] \;\leq\; 2L\left(1 + \frac{\omega}{n}\right)\left(f(x^k) - f(x^*)\right) + \frac{(\omega+1)D}{n}.
$$

$\square$

**Theorem K.3.** *Assume that $f_i(x)$ is convex and $L$-smooth for all $i = 1, \ldots, n$ and $f(x)$ is $\mu$-quasi strongly convex. Then* `D-QSGDstar` *satisfies Assumption 3.4 with*

$$
A' = L\left(1 + \frac{\omega}{n}\right), \quad B_1' = B_2' = 0, \quad D_1' = \frac{(\omega+1)D}{n}, \quad \sigma_{1,k}^2 \equiv \sigma_{2,k}^2 \equiv 0,
$$

$$
\rho_1 = \rho_2 = 1, \quad C_1 = C_2 = 0, \quad D_2 = 0, \quad G = 0,
$$

$$
F_1 = F_2 = 0, \quad D_3 = \frac{3\gamma\tau L(\omega+1)D}{n}
$$

*with $\gamma$ satisfying*

$$
\gamma \leq \min\left\{\frac{1}{4L(1 + \omega/n)}, \frac{1}{8L\sqrt{\tau\left(\tau + 1 + \omega/n\right)}}\right\}.
$$

*and for all $K \geq 0$*

$$
\mathbf{E}\left[f(\bar{x}^K) - f(x^*)\right] \leq \left(1 - \frac{\gamma\mu}{2}\right)^K \frac{4\|x^0 - x^*\|^2}{\gamma} + 4\gamma\left(D_1' + D_3\right)
$$

*when $\mu > 0$ and*

$$
\mathbf{E}\left[f(\bar{x}^K) - f(x^*)\right] \leq \frac{4\|x^0 - x^*\|^2}{\gamma K} + 4\gamma\left(D_1' + D_3\right)
$$

*when $\mu = 0$.*

In other words, `D-QSGDstar` converges with the linear rate

$$\mathcal{O}\left(\left(\tau + \kappa\left(1 + \frac{\omega}{n}\right) + \kappa\sqrt{\tau\left(\tau + \frac{\omega}{n}\right)}\right)\ln\frac{1}{\varepsilon}\right)$$

to the exact solution when $\mu > 0$ and $D = 0$, i.e., $\hat{g}_i^k = \nabla f_i(x^k)$ for all $i = 1, \ldots, n$ almost surely. Applying Lemma D.2 we establish the rate of convergence to $\varepsilon$-solution.

**Corollary K.6.** *Let the assumptions of Theorem K.3 hold and $\mu > 0$. Then after $K$ iterations of* `D-QSGDstar` *with the stepsize*

$$\gamma_0 = \min\left\{\frac{1}{4L(1 + \omega/n)}, \frac{1}{8L\sqrt{\tau(\tau + 1 + \omega/n)}}\right\}, \quad R_0 = \|x^0 - x^*\|,$$

$$\gamma = \min\left\{\gamma_0, \frac{\ln\left(\max\left\{2, \min\left\{\frac{nR_0^2\mu^2K^2}{D}, \frac{nR_0^2\mu^3K^3}{3\tau LD}\right\}\right\}\right)}{\mu K}\right\}$$

*we have* $\mathbf{E}\left[f(\bar{x}^K) - f(x^*)\right]$ *of order*

$$\widetilde{\mathcal{O}}\left(LR_0^2\left(1 + \frac{\omega}{n} + \sqrt{\tau\left(\tau + \frac{\omega}{n}\right)}\right)\exp\left(-\frac{\mu}{L\left(1 + \frac{\omega}{n} + \sqrt{\tau\left(\tau + \frac{\omega}{n}\right)}\right)}K\right) + \frac{D}{n\mu K} + \frac{L\tau D}{n\mu^2 K^2}\right).$$

*That is, to achive* $\mathbf{E}\left[f(\bar{x}^K) - f(x^*)\right] \leq \varepsilon$ `D-QSGDstar` *requires*

$$\widetilde{\mathcal{O}}\left(\frac{L}{\mu}\left(1 + \frac{\omega}{n}\right) + \frac{L}{\mu}\sqrt{\tau\left(\tau + \frac{\omega}{n}\right)} + \frac{D}{n\mu\varepsilon} + \frac{\sqrt{L\tau D}}{\mu\sqrt{n\varepsilon}}\right) \quad \text{iterations.}$$

Applying Lemma D.3 we get the complexity result in the case when $\mu = 0$.

**Corollary K.7.** *Let the assumptions of Theorem K.3 hold and $\mu = 0$. Then after $K$ iterations of* `D-QSGDstar` *with the stepsize*

$$\gamma_0 = \min\left\{\frac{1}{4L(1 + 2\omega/n)}, \frac{1}{8L\sqrt{\tau(\tau + 1 + \omega/n)}}\right\},$$

$$\gamma = \min\left\{\gamma_0, \sqrt{\frac{n\|x^0 - x^*\|^2}{DK}}, \sqrt[3]{\frac{n\|x^0 - x^*\|^2}{3L\tau DK}}\right\}$$

*we have* $\mathbf{E}\left[f(\bar{x}^K) - f(x^*)\right]$ *of order*

$$\mathcal{O}\left(\frac{LR_0^2\left(1 + \frac{\omega}{n}\right)}{K} + \frac{LR_0^2\sqrt{\tau\left(\tau + \frac{\omega}{n}\right)}}{K} + \sqrt{\frac{R_0^2 D}{nK}} + \frac{\sqrt[3]{LR_0^4\tau D}}{n^{1/3}K^{2/3}}\right)$$

*where* $R_0 = \|x^0 - x^*\|$. *That is, to achive* $\mathbf{E}\left[f(\bar{x}^K) - f(x^*)\right] \leq \varepsilon$ `D-QSGDstar` *requires*

$$\mathcal{O}\left(\frac{LR_0^2\left(1 + \frac{\omega}{n}\right)}{\varepsilon} + \frac{LR_0^2\sqrt{\tau\left(\tau + \frac{\omega}{n}\right)}}{\varepsilon} + \frac{R_0^2 D}{n\varepsilon^2} + \frac{R_0^2\sqrt{L\tau D}}{\sqrt{n}\varepsilon^{3/2}}\right)$$

*iterations.*

### K.4  D-SGD-DIANA

In this section we present a practical version of `D-QSGDstar`: `D-SGD-DIANA`.

**Lemma K.4** (Lemmas 1 and 2 from [19]). *Assume that $f_i(x)$ is convex and $L$-smooth for all $i = 1, \ldots, n$ and $\alpha \leq 1/(\omega+1)$. Then, for all $k \geq 0$ we have*

$$\mathbf{E}\left[g^k \mid x^k\right] = \nabla f(x^k), \tag{132}$$

$$\mathbf{E}\left[\|g^k\|^2 \mid x^k\right] \leq 2L\left(1 + \frac{2\omega}{n}\right)\left(f(x^k) - f(x^*)\right) + \frac{2\omega\sigma_k^2}{n} + \frac{(\omega+1)D}{n} \tag{133}$$

$$\mathbf{E}\left[\sigma_{k+1}^2 \mid x^k\right] \leq (1 - \alpha)\sigma_k^2 + 2L\alpha\left(f(x^k) - f(x^*)\right) + \alpha D \tag{134}$$

*where* $\sigma_k^2 = \frac{1}{n}\sum_{i=1}^n \|h_i^k - \nabla f_i(x^*)\|^2$ *and* $D = \frac{1}{n}\sum_{i=1}^n D_i$.

**Algorithm 14** D-SGD-DIANA

**Input:** learning rates $\gamma > 0, \alpha \in (0, 1]$, initial vectors $x^0, h_1^0, \ldots, h_n^0 \in \mathbb{R}^d$
1: Set $e_i^0 = 0$ for all $i = 1, \ldots, n$
2: Set $h^0 = \frac{1}{n} \sum_{i=1}^n h_i^0$
3: **for** $k = 0, 1, \ldots$ **do**
4:      Broadcast $x^{k-\tau}$ to all workers
5:      **for** $i = 1, \ldots, n$ **do**
6:          Sample $\hat{g}_i^{k-\tau}$ independently from other nodes such that $\mathbf{E}[\hat{g}_i^{k-\tau} \mid x^{k-\tau}] = \nabla f_i(x^{k-\tau})$ and $\mathbf{E}\left[\|\hat{g}_i^{k-\tau} - \nabla f_i(x^{k-\tau})\|^2 \mid x^{k-\tau}\right] \leq D_i$
7:          $\hat{\Delta}_i^{k-\tau} = Q(\hat{g}_i^{k-\tau} - h_i^{k-\tau})$ (quantization is performed independently from other nodes)
8:          $g_i^{k-\tau} = h_i^{k-\tau} + \hat{\Delta}_i^{k-\tau}$
9:          $v_i^k = \gamma g_i^{k-\tau}$
10:         $e_i^{k+1} = e_i^k + \gamma g_i^k - v_i^k$
11:         $h_i^{k-\tau+1} = h_i^{k-\tau} + \alpha \hat{\Delta}_i^{k-\tau}$
12:      **end for**
13:      $h^{k-\tau} = \frac{1}{n} \sum_{i=1}^n h_i^{k-\tau}$, $e^k = \frac{1}{n} \sum_{i=1}^n e_i^k$, $g^k = \frac{1}{n} \sum_{i=1}^n g_i^k$, $v^k = \frac{1}{n} \sum_{i=1}^n v_i^k = \frac{\gamma}{n} \sum_{i=1}^n g_i^{k-\tau} = \gamma h^{k-\tau} + \frac{\gamma}{n} \sum_{i=1}^n \hat{\Delta}_i^{k-\tau}$
14:      $x^{k+1} = x^k - v^k$
15:      $h^{k-\tau+1} = h^{k-\tau} + \frac{\alpha}{n} \sum_{i=1}^n \hat{\Delta}_i^{k-\tau}$
16: **end for**

**Theorem K.4.** *Assume that $f_i(x)$ is convex and $L$-smooth for all $i = 1, \ldots, n$ and $f(x)$ is $\mu$-quasi strongly convex. Then* D-SGD-DIANA *satisfies Assumption 3.4 with*

$$A' = L\left(1 + \frac{2\omega}{n}\right), \quad B_1' = \frac{2\omega}{n}, \quad D_1' = \frac{(\omega + 1)D}{n}, \quad \sigma_{1,k}^2 = \sigma_k^2 = \frac{1}{n}\sum_{i=1}^n \|h_i^k - \nabla f_i(x^*)\|^2,$$

$$B_2' = 0, \quad \rho_1 = \alpha, \quad \rho_2 = 1, \quad C_1 = L\alpha, \quad C_2 = 0, \quad D_2 = \frac{\alpha(\omega + 1)D}{n}, \quad G = 0,$$

$$F_1 = \frac{12\gamma^2 L\omega\tau(2 + \alpha)}{n\alpha}, \quad F_2 = 0, \quad D_3 = 3\gamma\tau L\left(1 + \frac{4\omega}{n}\right)\frac{(\omega + 1)D}{n}$$

*with $\gamma$ and $\alpha$ satisfying*

$$\gamma \leq \min\left\{\frac{1}{4L(1 + {}^{14\omega}/_{3n})}, \frac{1}{8L\sqrt{2\tau\left(1 + \tau + {}^{2\omega}/_n + {}^{4\omega}/_{n(1-\alpha)}\right)}}\right\}, \quad \alpha \leq \frac{1}{\omega + 1}, \quad M_1 = \frac{8\omega}{3n\alpha}$$

*and for all $K \geq 0$*

$$\mathbf{E}\left[f(\bar{x}^K) - f(x^*)\right] \leq \left(1 - \min\left\{\frac{\gamma\mu}{2}, \frac{\alpha}{4}\right\}\right)^K \frac{4(T^0 + \gamma F_1 \sigma_0^2)}{\gamma} + 4\gamma\left(D_1' + M_1 D_2 + D_3\right)$$

*when $\mu > 0$ and*

$$\mathbf{E}\left[f(\bar{x}^K) - f(x^*)\right] \leq \frac{4(T^0 + \gamma F_1 \sigma_0^2)}{\gamma K} + 4\gamma\left(D_1' + M_1 D_2 + D_3\right)$$

*when $\mu = 0$, where $T^k \overset{def}{=} \|\tilde{x}^k - x^*\|^2 + M_1 \gamma^2 \sigma_k^2$.*

In other words, if

$$\gamma \leq \min\left\{\frac{1}{4L(1 + {}^{14\omega}/_{3n})}, \frac{1}{8L\sqrt{2\tau\left(1 + \tau + {}^{10\omega}/_n\right)}}\right\}, \quad \alpha \leq \min\left\{\frac{1}{\omega + 1}, \frac{1}{2}\right\}$$

then D-SGD-DIANA *converges with the linear rate*

$$\mathcal{O}\left(\left(\omega + \kappa\left(1 + \frac{\omega}{n}\right) + \kappa\sqrt{\tau\left(\tau + \frac{\omega}{n}\right)}\right)\ln\frac{1}{\varepsilon}\right)$$

*to the exact solution when $\mu > 0$. Applying Lemma D.2 we establish the rate of convergence to $\varepsilon$-solution.*

**Corollary K.8.** *Let the assumptions of Theorem K.4 hold and $\mu > 0$. Then after $K$ iterations of* `D-SGD-DIANA` *with the stepsize*

$$
\gamma_0 = \min\left\{\frac{1}{4L(1 + {}^{14\omega}\!/_{3n})}, \frac{1}{8L\sqrt{2\tau\left(1 + \tau + {}^{10\omega}\!/_{n}\right)}}\right\}, \quad R_0 = \|x^0 - x^*\|,
$$

$$
\tilde{F}_1 = \frac{12L\omega\tau(2 + \alpha)\gamma_0^2}{n\alpha}, \quad \tilde{T}^0 = R_0^2 + M_1\gamma_0^2\sigma_0^2,
$$

$$
\gamma = \min\left\{\gamma_0, \frac{\ln\left(\max\left\{2, \min\left\{\frac{(\tilde{T}^0 + \gamma_0\tilde{F}_1\sigma_0^2)\mu^2 K^2}{D_1' + M_1 D_2}, \frac{(\tilde{T}^0 + \gamma_0\tilde{F}_1\sigma_0^2)\mu^3 K^3}{3\tau L\left(D_1' + \frac{2B_1' D_2}{\alpha}\right)}\right\}\right\}\right)}{\mu K}\right\}
$$

*and $\alpha \leq \min\left\{\frac{1}{\omega + 1}, \frac{1}{2}\right\}$ we have $\mathbf{E}\left[f(\bar{x}^K) - f(x^*)\right]$ of order*

$$
\tilde{\mathcal{O}}\left(LR_0^2\left(1 + \frac{\omega}{n} + \sqrt{\tau\left(\tau + \frac{\omega}{n}\right)}\right)\exp\left(-\min\left\{\frac{\mu}{L\left(1 + \frac{\omega}{n} + \sqrt{\tau\left(\tau + \frac{\omega}{n}\right)}\right)}, \frac{1}{1 + \omega}\right\}K\right)\right)
$$

$$
+ \tilde{\mathcal{O}}\left(\frac{D_1' + M_1 D_2}{\mu K} + \frac{\tau L\left(D_1' + \frac{B_1' D_2}{\alpha}\right)}{\mu^2 K^2}\right).
$$

*That is, to achive $\mathbf{E}\left[f(\bar{x}^K) - f(x^*)\right] \leq \varepsilon$* `D-SGD-DIANA` *requires*

$$
\tilde{\mathcal{O}}\left(\omega + \frac{L}{\mu}\left(1 + \frac{\omega}{n}\right) + \frac{L}{\mu}\sqrt{\tau\left(\tau + \frac{\omega}{n}\right)} + \frac{(\omega + 1)\left(1 + \frac{\omega}{n}\right)D}{n\mu\varepsilon} + \frac{\sqrt{L\tau(\omega + 1)\left(1 + \frac{\omega}{n}\right)D}}{\mu\sqrt{n\varepsilon}}\right)
$$

*iterations.*

Applying Lemma D.3 we get the complexity result in the case when $\mu = 0$.

**Corollary K.9.** *Let the assumptions of Theorem K.4 hold and $\mu = 0$. Then after $K$ iterations of* `D-SGD-DIANA` *with the stepsize*

$$
\gamma_0 = \min\left\{\frac{1}{4L(1 + {}^{14\omega}\!/_{3n})}, \frac{1}{8L\sqrt{2\tau\left(1 + \tau + {}^{10\omega}\!/_{n}\right)}}\right\}, \quad R_0 = \|x^0 - x^*\|,
$$

$$
\gamma = \min\left\{\gamma_0, \sqrt{\frac{R_0^2}{M_1\sigma_0^2}}, \sqrt[3]{\frac{R_0^2 n\alpha}{12L\omega\tau(2 + \alpha)\sigma_0^2}}, \sqrt{\frac{R_0^2}{(D_1' + M_1 D_2)K}}, \sqrt[3]{\frac{R_0^2}{3\tau L\left(D_1' + \frac{2B_1' D_2}{\alpha}\right)K}}\right\}
$$

*we have $\mathbf{E}\left[f(\bar{x}^K) - f(x^*)\right]$ of order*

$$
\mathcal{O}\left(\frac{L\left(1 + \frac{\omega}{n}\right)R_0^2}{K} + \frac{L\sqrt{\tau\left(\tau + \frac{\omega}{n}\right)}R_0^2}{K} + \frac{\sqrt{R_0^2\omega(1 + \omega)\sigma_0^2}}{\sqrt{n}K} + \frac{\sqrt[3]{R_0^4 L\tau\omega(1 + \omega)\sigma_0^2}}{\sqrt[3]{n}K}\right)
$$

$$
+ \mathcal{O}\left(\sqrt{\frac{(1 + \omega)\left(1 + \frac{\omega}{n}\right)R_0^2 D}{nK}} + \frac{\sqrt[3]{R_0^4 \tau L(1 + \omega)\left(1 + \frac{\omega}{n}\right)D}}{n^{1/3}K^{2/3}}\right).
$$

---

**Algorithm 15** D-SGDsr

---

**Input:** learning rate $\gamma > 0$, initial vector $x^0 \in \mathbb{R}^d$
  1: Set $e_i^0 = 0$ for all $i = 1, \ldots, n$
  2: **for** $k = 0, 1, \ldots$ **do**
  3:     Broadcast $x^{k-\tau}$ to all workers
  4:     **for** $i = 1, \ldots, n$ in parallel **do**
  5:         Sample $g_i^{k-\tau} = \nabla f_{\xi_i}(x^{k-\tau}) - \nabla f_i(x^*)$
  6:         $v_i^k = \gamma g_i^{k-\tau}$
  7:         $e_i^{k+1} = e_i^k + \gamma g_i^k - v_i^k$
  8:     **end for**
  9:     $e^k = \frac{1}{n} \sum_{i=1}^n e_i^k, \ g^k = \frac{1}{n} \sum_{i=1}^n g_i^k, \ v^k = \frac{1}{n} \sum_{i=1}^n v_i^k = \frac{1}{n} \sum_{i=1}^n \nabla f_{\xi_i}(x^{k-\tau})$
 10:     $x^{k+1} = x^k - v^k$
 11: **end for**

---

*That is, to achive* $\mathbf{E}\left[f(\bar{x}^K) - f(x^*)\right] \leq \varepsilon$ D-SGD-DIANA *requires*

$$\mathcal{O}\left(\frac{L\left(1 + \frac{\omega}{n}\right) R_0^2}{\varepsilon} + \frac{L\sqrt{\tau\left(\tau + \frac{\omega}{n}\right)} R_0^2}{\varepsilon} + \frac{\sqrt{R_0^2 \omega (1 + \omega) \sigma_0^2}}{\sqrt{n}\varepsilon} + \frac{\sqrt[3]{R_0^4 L \tau \omega (1 + \omega) \sigma_0^2}}{\sqrt[3]{n}\varepsilon}\right)$$

$$+ \mathcal{O}\left(\frac{(1 + \omega)\left(1 + \frac{\omega}{n}\right) R_0^2 D}{n\varepsilon^2} + \frac{R_0^2 \sqrt{\tau L (1 + \omega)\left(1 + \frac{\omega}{n}\right) D}}{n^{1/2}\varepsilon^{3/2}}\right) \quad iterations.$$

### K.5  D-SGDsr

In this section we consider the same settings as in Section J.1, but this time we consider delayed updates. Moreover, in this section we need slightly weaker assumption.

**Assumption K.1** (Expected smoothness). *We assume that function $f$ is $\mathcal{L}$-smooth in expectation w.r.t. distribution $\mathcal{D}$, i.e., there exists constant $\mathcal{L} = \mathcal{L}(f, \mathcal{D})$ such that*

$$\mathbf{E}_{\mathcal{D}}\left[\|\nabla f_\xi(x) - \nabla f_\xi(x^*)\|^2\right] \leq 2\mathcal{L}\left(f(x) - f(x^*)\right) \tag{135}$$

*for all $i \in [n]$ and $x \in \mathbb{R}^d$.*

**Lemma K.5.** *For all $k \geq 0$ we have*

$$\mathbf{E}\left[\|g^k\|^2 \mid x^k\right] \leq 4\mathcal{L}\left(f(x^k) - f(x^*)\right) + 2\mathbf{E}_{\mathcal{D}}\left[\|\nabla f_\xi(x^*)\|^2\right]. \tag{136}$$

*Proof.* Applying straightforward inequality $\|a + b\|^2 \leq 2\|a\|^2 + 2\|b\|^2$ for $a, b \in \mathbb{R}^d$ we get

$$
\begin{aligned}
\mathbf{E}\left[\|g^k\|^2 \mid x^k\right] \ &= \ \mathbf{E}\left[\left\|\frac{1}{n}\sum_{i=1}^n \left(\nabla f_{\xi_i}(x^k) - \nabla f_i(x^*)\right)\right\|^2 \mid x^k\right] \\
&\overset{(31)}{\leq} \ 2\mathbf{E}_{\mathcal{D}}\left[\|\nabla f_\xi(x^k) - \nabla f_\xi(x^*)\|^2\right] + 2\mathbf{E}_{\mathcal{D}}\left[\|\nabla f_\xi(x^*) - \nabla f(x^*)\|^2\right] \\
&\overset{(135)}{\leq} \ 4\mathcal{L}\left(f(x^k) - f(x^*)\right) + 2\mathbf{E}_{\mathcal{D}}\left[\|\nabla f_\xi(x^*)\|^2\right].
\end{aligned}
$$

$\square$

**Theorem K.5.** *Assume that $f(x)$ is $\mu$-quasi strongly convex, $L$-smooth and Assumption K.1 holds. Then* D-SGDsr *satisfies Assumption 3.4 with*

$$A' = 2\mathcal{L}, \quad B_1' = B_2' = 0, \quad D_1' = 2\mathbf{E}_{\mathcal{D}}\|\nabla f_\xi(x^*)\|^2, \quad \sigma_{1,k}^2 \equiv \sigma_{2,k}^2 \equiv 0$$

$$\rho_1 = \rho_2 = 1, \quad C_1 = C_2 = 0, \quad D_2 = 0, \quad G = 0,$$

$$F_1 = F_2 = 0, \quad D_3 = 6\gamma\tau L \mathbf{E}_{\mathcal{D}}\|\nabla f_\xi(x^*)\|^2$$

*with $\gamma$ satisfying*

$$\gamma \le \min\left\{\frac{1}{8\mathcal{L}}, \frac{1}{8\sqrt{L\tau\left(L\tau + 2\mathcal{L}\right)}}\right\}$$

*and for all $K \ge 0$*

$$\mathbf{E}\left[f(\bar{x}^K) - f(x^*)\right] \le \left(1 - \frac{\gamma\mu}{2}\right)^K \frac{4\|x^0 - x^*\|^2}{\gamma} + 8\gamma(1 + 3\gamma\tau L)\mathbf{E}_{\mathcal{D}}\|\nabla f_\xi(x^*)\|^2$$

*when $\mu > 0$ and*

$$\mathbf{E}\left[f(\bar{x}^K) - f(x^*)\right] \le \frac{4\|x^0 - x^*\|^2}{\gamma K} + 8\gamma(1 + 3\gamma\tau L)\mathbf{E}_{\mathcal{D}}\|\nabla f_\xi(x^*)\|^2$$

*when $\mu = 0$.*

In other words, D-SGDsr converges with linear rate $\mathcal{O}\left(\left(\frac{\mathcal{L}}{\mu} + \frac{\sqrt{L\mathcal{L}\tau + L^2\tau^2}}{\mu}\right)\ln\frac{1}{\varepsilon}\right)$ to the neighbourhood of the solution when $\mu > 0$. Applying Lemma D.2 we establish the rate of convergence to $\varepsilon$-solution.

**Corollary K.10.** *Let the assumptions of Theorem K.5 hold and $\mu > 0$. Then after $K$ iterations of* D-SGDsr *with the stepsize*

$$\gamma_0 = \min\left\{\frac{1}{8\mathcal{L}}, \frac{1}{8\sqrt{L\tau\left(L\tau + 2\mathcal{L}\right)}}\right\}, \quad R_0 = \|x^0 - x^*\|,$$

$$\gamma = \min\left\{\gamma_0, \frac{\ln\left(\max\left\{2, \min\left\{\frac{R_0^2\mu^2 K^2}{D_1'}, \frac{R_0^2\mu^3 K^3}{3\tau L D_1'}\right\}\right\}\right)}{\mu K}\right\}$$

*we have $\mathbf{E}\left[f(\bar{x}^K) - f(x^*)\right]$ of order*

$$\widetilde{\mathcal{O}}\left(R_0^2\left(\mathcal{L} + \sqrt{L^2\tau^2 + L\mathcal{L}\tau}\right)\exp\left(-\frac{\mu}{\tau L}K\right) + \frac{\mathbf{E}_{\mathcal{D}}\|\nabla f_\xi(x^*)\|^2}{\mu K} + \frac{L\tau\mathbf{E}_{\mathcal{D}}\|\nabla f_\xi(x^*)\|^2}{\mu^2 K^2}\right).$$

*That is, to achive $\mathbf{E}\left[f(\bar{x}^K) - f(x^*)\right] \le \varepsilon$* D-SGDsr *requires*

$$\widetilde{\mathcal{O}}\left(\frac{\mathcal{L} + \sqrt{L^2\tau^2 + L\mathcal{L}\tau}}{\mu} + \frac{\mathbf{E}_{\mathcal{D}}\|\nabla f_\xi(x^*)\|^2}{\mu\varepsilon} + \frac{\sqrt{L\tau\mathbf{E}_{\mathcal{D}}\|\nabla f_\xi(x^*)\|^2}}{\mu\sqrt{\varepsilon}}\right) \quad \textit{iterations.}$$

Applying Lemma D.3 we get the complexity result in the case when $\mu = 0$.

**Corollary K.11.** *Let the assumptions of Theorem K.5 hold and $\mu = 0$. Then after $K$ iterations of* D-SGDsr *with the stepsize*

$$\gamma = \min\left\{\frac{1}{8\mathcal{L}}, \frac{1}{8\sqrt{L\tau\left(L\tau + 2\mathcal{L}\right)}}, \sqrt{\frac{\|x^0 - x^*\|^2}{D_1' K}}, \sqrt[3]{\frac{\|x^0 - x^*\|^2}{3L\tau D_1' K}}\right\}$$

*we have $\mathbf{E}\left[f(\bar{x}^K) - f(x^*)\right]$ of order*

$$\mathcal{O}\left(\frac{\mathcal{L}R_0^2}{K} + \frac{\sqrt{L^2\tau^2 + L\mathcal{L}\tau}R_0^2}{K} + \sqrt{\frac{R_0^2\tau\mathbf{E}_{\mathcal{D}}\|\nabla f_\xi(x^*)\|^2}{K}} + \frac{\sqrt[3]{LR_0^4\tau\mathbf{E}_{\mathcal{D}}\|\nabla f_\xi(x^*)\|^2}}{K^{2/3}}\right)$$

*where $R_0 = \|x^0 - x^*\|$. That is, to achive $\mathbf{E}\left[f(\bar{x}^K) - f(x^*)\right] \le \varepsilon$* D-SGDsr *requires*

$$\mathcal{O}\left(\frac{\mathcal{L}R_0^2}{\varepsilon} + \frac{\sqrt{L^2\tau^2 + L\mathcal{L}\tau}R_0^2}{\varepsilon} + \frac{R_0^2\mathbf{E}_{\mathcal{D}}\|\nabla f_\xi(x^*)\|^2}{\varepsilon^2} + \frac{R_0^2\sqrt{L\tau\mathbf{E}_{\mathcal{D}}\|\nabla f_\xi(x^*)\|^2}}{\varepsilon^{3/2}}\right)$$

*iterations.*

---
**Algorithm 16** D-LSVRG
---
**Input:** learning rate $\gamma > 0$, initial vector $x^0 \in \mathbb{R}^d$
 1: Set $e_i^0 = 0$ for all $i = 1, \ldots, n$
 2: **for** $k = 0, 1, \ldots$ **do**
 3:     Broadcast $x^{k-\tau}$ to all workers
 4:     **for** $i = 1, \ldots, n$ in parallel **do**
 5:         Pick $l$ uniformly at random from $[m]$
 6:         Set $g_i^{k-\tau} = \nabla f_{il}(x^{k-\tau}) - \nabla f_{il}(w_i^{k-\tau}) + \nabla f_i(w_i^{k-\tau})$
 7:         $v_i^k = \gamma g_i^{k-\tau}$
 8:         $e_i^{k+1} = e_i^k + \gamma g_i^k - v_i^k$
 9:         $w_i^{k-\tau+1} = \begin{cases} x^{k-\tau}, & \text{with probability } p, \\ w_i^{k-\tau}, & \text{with probability } 1-p \end{cases}$
10:     **end for**
11:     $e^k = \frac{1}{n} \sum_{i=1}^n e_i^k, \ g^k = \frac{1}{n} \sum_{i=1}^n g_i^k, \ v^k = \frac{1}{n} \sum_{i=1}^n v_i^k$
12:     $x^{k+1} = x^k - v^k$
13: **end for**
---

## K.6   D-LSVRG

In the same settings as in Section J.6 we now consider a new method called D-LSVRG which is another modification of LSVRG that works with delayed updates.

**Lemma K.6.** *For all $k \geq 0$, $i \in [n]$ we have*

$$\mathbf{E}\left[g_i^k \mid x^k\right] = \nabla f_i(x^k) \tag{137}$$

*and*

$$\mathbf{E}\left[\|g^k\|^2 \mid x^k\right] \leq 4L\left(f(x^k) - f(x^*)\right) + 2\sigma_k^2, \tag{138}$$

*where $\sigma_k^2 = \frac{1}{nm} \sum_{i=1}^n \sum_{j=1}^n \|\nabla f_{ij}(w_i^k) - \nabla f_{ij}(x^*)\|^2$.*

*Proof.* First of all, we derive unbiasedness of $g_i^k$:

$$\mathbf{E}\left[g_i^k \mid x^k\right] = \frac{1}{m} \sum_{j=1}^m \left(\nabla f_{ij}(x^k) - \nabla f_{ij}(w_i^k) + \nabla f_i(w_i^k)\right) = \nabla f_i(x^k).$$

Next, we estimate the second moment of $g^k$:

$$
\begin{aligned}
\mathbf{E}\left[\|g^k\|^2 \mid x^k\right] \ &= \ \mathbf{E}\left[\left\|\frac{1}{n} \sum_{i=1}^n \left(\nabla f_{il}(x^k) - \nabla f_{il}(w_i^k) + \nabla f_i(w_i^k)\right)\right\|^2\right] \\[2mm]
&= \ \mathbf{E}\left[\left\|\frac{1}{n} \sum_{i=1}^n \left(\nabla f_{il}(x^k) - \nabla f_{il}(x^*) + \nabla f_{il}(x^*) - \nabla f_{il}(w_i^k) + \nabla f_i(w_i^k) - \nabla f_i(x^*)\right)\right\|^2\right] \\[2mm]
&\overset{(31)}{\leq} \ \frac{2}{n} \sum_{i=1}^n \mathbf{E}\left[\|\nabla f_{il}(x^k) - \nabla f_{il}(x^*)\|^2 \mid x^k\right] \\[2mm]
&\qquad + \frac{2}{n} \sum_{i=1}^n \mathbf{E}\left[\left\|\nabla f_{il}(w_i^k) - \nabla f_{il}(x^*) - \left(\nabla f_i(w_i^k) - \nabla f_i(x^*)\right)\right\|^2 \mid x^k\right] \\[2mm]
&\overset{(34)}{\leq} \ \frac{2}{nm} \sum_{i=1}^n \sum_{j=1}^m \|\nabla f_{ij}(x^k) - \nabla f_{ij}(x^*)\|^2 + \frac{2}{n} \mathbf{E}\left[\|\nabla f_{il}(w_i^k) - \nabla f_{il}(x^*)\|^2 \mid x^k\right] \\[2mm]
&\overset{(11)}{\leq} \ \frac{4L}{nm} \sum_{i=1}^n \sum_{j=1}^m D_{f_{ij}}(x^k, x^*) + \frac{2}{nm} \sum_{i=1}^n \sum_{j=1}^m \|\nabla f_{ij}(w_i^k) - \nabla f_{ij}(x^*)\|^2 \\[2mm]
&= \ 4L\left(f(x^k) - f(x^*)\right) + 2\sigma_k^2.
\end{aligned}
$$

$\square$

**Lemma K.7.** *For all $k \geq 0$, $i \in [n]$ we have*

$$\mathbf{E}\left[\sigma_{k+1}^2 \mid x^k\right] \leq (1-p)\sigma_k^2 + 2Lp\left(f(x^k) - f(x^*)\right), \tag{139}$$

*where $\sigma_k^2 = \frac{1}{nm}\sum_{i=1}^n\sum_{j=1}^n \|\nabla f_{ij}(w_i^k) - \nabla f_{ij}(x^*)\|^2$.*

*Proof.* The proof is identical to the proof of Lemma J.8. $\qquad\square$

**Theorem K.6.** *Assume that $f(x)$ is $\mu$-quasi strongly convex and functions $f_{ij}$ are convex and $L$-smooth for all $i \in [n], j \in [m]$. Then* D-LSVRG *satisfies Assumption 3.4 with*

$$A' = 2L, \quad B_1' = 0, \quad B_2' = 2, \quad D_1' = 0, \quad \sigma_{2,k}^2 = \sigma_k^2 = \frac{1}{nm}\sum_{i=1}^n\sum_{j=1}^m \|\nabla f_{ij}(w_i^k) - \nabla f_{ij}(x^*)\|^2,$$

$$\sigma_{1,k}^2 \equiv 0, \quad \rho_1 = 1, \quad \rho_2 = p, \quad C_1 = 0, \quad C_2 = Lp, \quad D_2 = 0,$$

$$G = 0, \quad F_1 = 0, \quad F_2 = \frac{12\gamma^2 L\tau(2+p)}{p}, \quad D_3 = 0$$

*with $\gamma$ satisfying*

$$\gamma \leq \min\left\{\frac{3}{56L}, \frac{1}{8L\sqrt{\tau\left(2 + \tau + 4/(1-p)\right)}}\right\}, \quad M_2 = \frac{8}{3p}$$

*and for all $K \geq 0$*

$$\mathbf{E}\left[f(\bar{x}^K) - f(x^*)\right] \leq \left(1 - \min\left\{\frac{\gamma\mu}{2}, \frac{p}{4}\right\}\right)^K \frac{4(T^0 + \gamma F_2\sigma_0^2)}{\gamma}$$

*when $\mu > 0$ and*

$$\mathbf{E}\left[f(\bar{x}^K) - f(x^*)\right] \leq \frac{4(T^0 + \gamma F_2\sigma_0^2)}{\gamma K}$$

*when $\mu = 0$, where $T^k \stackrel{def}{=} \|\tilde{x}^k - x^*\|^2 + M_2\gamma^2\sigma_k^2$.*

In other words, D-LSVRG converges with linear rate $\mathcal{O}\left(\left(\frac{1}{p} + \kappa\sqrt{\tau\left(\tau + \frac{1}{(1-p)}\right)}\right)\ln\frac{1}{\varepsilon}\right)$ to

the exact solution when $\mu > 0$. If $m \geq 2$ then taking $p = \frac{1}{m}$ we get that in expectation the sample complexity of one iteration of D-LSVRG is $\mathcal{O}(1)$ gradients calculations per node as for D-SGDsr with standard sampling and the rate of convergence to the exact solution becomes $\mathcal{O}\left((m + \kappa\tau)\ln\frac{1}{\varepsilon}\right)$.

Applying Lemma D.3 we get the complexity result in the case when $\mu = 0$.

**Corollary K.12.** *Let the assumptions of Theorem K.6 hold and $\mu = 0$. Then after $K$ iterations of* D-LSVRG *with the stepsize*

$$\gamma = \min\left\{\frac{3}{56L}, \frac{1}{8L\sqrt{\tau\left(2 + \tau + 4/(1-p)\right)}}, \sqrt{\frac{\|x^0 - x^*\|^2}{M_2\sigma_0^2}}, \sqrt[3]{\frac{\|x^0 - x^*\|^2 p}{12L\tau(2+p)\sigma_0^2}}\right\}$$

*and $p = \frac{1}{m}$, $m \geq 2$ we have $\mathbf{E}\left[f(\bar{x}^K) - f(x^*)\right]$ of order*

$$\mathcal{O}\left(\frac{L\tau R_0^2}{K} + \frac{\sqrt{R_0^2 m\sigma_0^2}}{K} + \frac{\sqrt[3]{R_0^4 L\tau\sigma_0^2}}{K}\right)$$

*where $R_0 = \|x^0 - x^*\|$. That is, to achive $\mathbf{E}\left[f(\bar{x}^K) - f(x^*)\right] \leq \varepsilon$* D-LSVRG *requires*

$$\mathcal{O}\left(\frac{L\tau R_0^2}{\varepsilon} + \frac{\sqrt{R_0^2 m\sigma_0^2}}{\varepsilon} + \frac{\sqrt[3]{R_0^4 L\tau\sigma_0^2}}{\varepsilon}\right)$$

*iterations.*

---

**Algorithm 17** D-QLSVRG

---

**Input:** learning rate $\gamma > 0$, initial vector $x^0 \in \mathbb{R}^d$
1: Set $e_i^0 = 0$ for all $i = 1, \ldots, n$
2: **for** $k = 0, 1, \ldots$ **do**
3:     Broadcast $x^{k-\tau}$ to all workers
4:     **for** $i = 1, \ldots, n$ in parallel **do**
5:         Pick $l$ uniformly at random from $[m]$
6:         Set $\hat{g}_i^{k-\tau} = \nabla f_{il}(x^{k-\tau}) - \nabla f_{il}(w_i^{k-\tau}) + \nabla f_i(w_i^{k-\tau})$
7:         Set $g_i^{k-\tau} = Q(\hat{g}_i^{k-\tau})$ (quantization is performed independently from other nodes)
8:         $v_i^k = \gamma g_i^{k-\tau}$
9:         $e_i^{k+1} = e_i^k + \gamma g_i^k - v_i^k$
10:        $w_i^{k-\tau+1} = \begin{cases} x^{k-\tau}, & \text{with probability } p, \\ w_i^{k-\tau}, & \text{with probability } 1-p \end{cases}$
11:     **end for**
12:     $e^k = \frac{1}{n}\sum_{i=1}^n e_i^k$, $g^k = \frac{1}{n}\sum_{i=1}^n g_i^k$, $v^k = \frac{1}{n}\sum_{i=1}^n v_i^k$
13:     $x^{k+1} = x^k - v^k$
14: **end for**

---

## K.7   D-QLSVRG

In this section we add a quantization to D-LSVRG.

**Lemma K.8.** *For all $k \geq 0$, $i \in [n]$ we have*
$$\mathbf{E}\left[g_i^k \mid x^k\right] = \nabla f_i(x^k)$$
*and*
$$\mathbf{E}\left[\|g^k\|^2 \mid x^k\right] \leq 4L\left(1 + \frac{2\omega}{n}\right)\left(f(x^k) - f(x^*)\right) + 2\left(1 + \frac{2\omega}{n}\right)\sigma_k^2 + \frac{2\omega}{n^2}\sum_{i=1}^n \|\nabla f_i(x^*)\|^2,$$
*where $\sigma_k^2 = \frac{1}{nm}\sum_{i=1}^n \sum_{j=1}^n \|\nabla f_{ij}(w_i^k) - \nabla f_{ij}(x^*)\|^2$.*

*Proof.* First of all, we derive unbiasedness of $g_i^k$:
$$\mathbf{E}\left[g_i^k \mid x^k\right] \stackrel{(35)}{=} \mathbf{E}\left[\mathbf{E}_Q\left[Q(\hat{g}_i^k)\right] \mid x^k\right] \stackrel{(26)}{=} \mathbf{E}\left[\hat{g}_i^k \mid x^k\right]$$
$$= \frac{1}{m}\sum_{j=1}^m \left(\nabla f_{ij}(x^k) - \nabla f_{ij}(w_i^k) + \nabla f_i(w_i^k)\right) = \nabla f_i(x^k).$$

Next, we estimate the second moment of $g^k$:
$$\mathbf{E}_Q\left[\|g^k\|^2\right] = \mathbf{E}_Q\left[\left\|\frac{1}{n}\sum_{i=1}^n Q(\hat{g}_i^k)\right\|^2\right]$$
$$\stackrel{(34)}{=} \mathbf{E}_Q\left[\left\|\frac{1}{n}\sum_{i=1}^n \left(Q(\hat{g}_i^k) - \hat{g}_i^k\right)\right\|^2\right] + \left\|\frac{1}{n}\sum_{i=1}^n \hat{g}_i^k\right\|^2.$$

Since quantization on nodes is performed independently we can decompose the first term from the last row of the previous inequality into the sum of variances:
$$\mathbf{E}_Q\left[\|g^k\|^2\right] = \frac{1}{n^2}\sum_{i=1}^n \mathbf{E}_Q\left\|Q(\hat{g}_i^k) - \hat{g}_i^k\right\|^2 + \left\|\frac{1}{n}\sum_{i=1}^n \hat{g}_i^k\right\|^2$$
$$\stackrel{(26)}{\leq} \frac{\omega}{n^2}\sum_{i=1}^n \|\hat{g}_i^k\|^2 + \left\|\frac{1}{n}\sum_{i=1}^n \left(\hat{g}_i^k - \nabla f_i(x^*)\right)\right\|^2$$
$$\stackrel{(31)}{\leq} \left(1 + \frac{2\omega}{n}\right)\frac{1}{n}\sum_{i=1}^n \|\hat{g}_i^k - \nabla f_i(x^*)\|^2 + \frac{2\omega}{n^2}\sum_{i=1}^n \|\nabla f_i(x^*)\|^2.$$

Taking conditional mathematical expectation $\mathbf{E}\left[\cdot \mid x^k\right]$ from the both sides of previous inequality we get

$$
\begin{aligned}
\mathbf{E}\left[\|g^k\|^2 \mid x^k\right] \leq\ & \left(1+\frac{2\omega}{n}\right)\frac{2}{n}\sum_{i=1}^{n}\mathbf{E}\left[\|\nabla f_{il}(x^k)-\nabla f_{il}(x^*)\|^2 \mid x^k\right] \\
& +\left(1+\frac{2\omega}{n}\right)\frac{2}{n}\sum_{i=1}^{n}\mathbf{E}\left[\left\|\nabla f_{il}(w_i^k)-\nabla f_{il}(x^*)-\left(\nabla f_i(w_i^k)-\nabla f_i(x^*)\right)\right\|^2 \mid x^k\right] \\
& +\frac{2\omega}{n^2}\sum_{i=1}^{n}\|\nabla f_i(x^*)\|^2 \\
\leq\ & \left(1+\frac{2\omega}{n}\right)\frac{2}{nm}\sum_{i=1}^{n}\sum_{j=1}^{m}\|\nabla f_{ij}(x^k)-\nabla f_{ij}(x^*)\|^2 \\
& +\left(1+\frac{2\omega}{n}\right)\frac{2}{n}\sum_{i=1}^{n}\mathbf{E}\left[\|\nabla f_{il}(w_i^k)-\nabla f_{il}(x^*)\|^2 \mid x^k\right]+\frac{2\omega}{n^2}\sum_{i=1}^{n}\|\nabla f_i(x^*)\|^2 \\
\overset{(11)}{\leq}\ & \left(1+\frac{2\omega}{n}\right)\frac{4L}{nm}\sum_{i=1}^{n}\sum_{j=1}^{m}D_{f_{ij}}(x^k,x^*) \\
& +\left(1+\frac{2\omega}{n}\right)\frac{2}{nm}\sum_{i=1}^{n}\sum_{j=1}^{m}\|\nabla f_{ij}(w_i^k)-\nabla f_{ij}(x^*)\|^2+\frac{2\omega}{n^2}\sum_{i=1}^{n}\|\nabla f_i(x^*)\|^2 \\
=\ & 4L\left(1+\frac{2\omega}{n}\right)\left(f(x^k)-f(x^*)\right)+2\left(1+\frac{2\omega}{n}\right)\sigma_k^2+\frac{2\omega}{n^2}\sum_{i=1}^{n}\|\nabla f_i(x^*)\|^2.
\end{aligned}
$$

$\square$

**Lemma K.9.** *For all $k \geq 0$, $i \in [n]$ we have*
$$
\mathbf{E}\left[\sigma_{k+1}^2 \mid x^k\right] \leq (1-p)\sigma_k^2 + 2Lp\left(f(x^k)-f(x^*)\right), \tag{140}
$$
*where $\sigma_k^2 = \frac{1}{nm}\sum_{i=1}^{n}\sum_{j=1}^{n}\|\nabla f_{ij}(w_i^k)-\nabla f_{ij}(x^*)\|^2$.*

*Proof.* The proof is identical to the proof of Lemma J.8. $\square$

**Theorem K.7.** *Assume that $f(x)$ is $\mu$-quasi strongly convex and functions $f_{ij}$ are convex and $L$-smooth for all $i \in [n], j \in [m]$. Then* D-QLSVRG *satisfies Assumption 3.4 with*

$$
A' = 2L\left(1+\frac{2\omega}{n}\right), \quad B_1' = 0, \quad B_2' = 2\left(1+\frac{2\omega}{n}\right), \quad D_1' = \frac{2\omega}{n^2}\sum_{i=1}^{n}\|\nabla f_i(x^*)\|^2, \quad \sigma_{1,0}^2 \equiv 0,
$$

$$
\sigma_{2,k}^2 = \sigma_k^2 = \frac{1}{nm}\sum_{i=1}^{n}\sum_{j=1}^{m}\|\nabla f_{ij}(w_i^k)-\nabla f_{ij}(x^*)\|^2, \quad \rho_1 = 1, \quad \rho_2 = p, \quad C_2 = Lp, \quad D_2 = 0,
$$

$$
C_1 = 0, \quad G = 0, \quad F_1 = 0, \quad F_2 = \frac{12\gamma^2 L\tau\left(1+\frac{2\omega}{n}\right)(2+p)}{p}, \quad D_3 = \frac{6\gamma\tau L\omega}{n^2}\sum_{i=1}^{n}\|\nabla f_i(x^*)\|^2
$$

*with $\gamma$ satisfying*

$$
\gamma \leq \min\left\{\frac{3}{56L(1+2\omega/n)}, \frac{1}{8L\sqrt{\tau\left(\tau+2\left(1+2\omega/n\right)\left(1+2/(1-p)\right)\right)}}\right\}, \quad M_2 = \frac{8\left(1+\frac{2\omega}{n}\right)}{3p}
$$

*and for all $K \geq 0$*

$$
\mathbf{E}\left[f(\bar{x}^K)-f(x^*)\right] \leq \left(1-\min\left\{\frac{\gamma\mu}{2},\frac{p}{4}\right\}\right)^K \frac{4(T^0+\gamma F_2\sigma_0^2)}{\gamma}+4\gamma\left(D_1'+D_3\right)
$$

*when $\mu > 0$ and*

$$
\mathbf{E}\left[f(\bar{x}^K)-f(x^*)\right] \leq \frac{4(T^0+\gamma F_2\sigma_0^2)}{\gamma K}+4\gamma\left(D_1'+D_3\right)
$$

*when $\mu = 0$, where $T^k \overset{def}{=} \|\tilde{x}^k - x^*\|^2 + M_2\gamma^2\sigma_k^2$.*

In other words, `D-QLSVRG` converges with linear rate

$$\mathcal{O}\left(\left(\frac{1}{p} + \kappa\left(1 + \frac{\omega}{n}\right) + \kappa\sqrt{\tau\left(\tau + \left(1 + \frac{\omega}{n}\right)\left(1 + \frac{1}{(1-p)}\right)\right)}\right)\ln\frac{1}{\varepsilon}\right)$$

to neighbourhood the solution when $\mu > 0$. If $m \geq 2$ then taking $p = \frac{1}{m}$ we get that in expectation the sample complexity of one iteration of `D-QLSVRG` is $\mathcal{O}(1)$ gradients calculations per node as for `D-QSGDsr` with standard sampling and the rate of convergence to the neighbourhood of the solution becomes

$$\mathcal{O}\left(\left(m + \kappa\left(1 + \frac{\omega}{n}\right) + \kappa\sqrt{\tau\left(\tau + \frac{\omega}{n}\right)}\right)\ln\frac{1}{\varepsilon}\right).$$

Applying Lemma D.2 we establish the rate of convergence to $\varepsilon$-solution.

**Corollary K.13.** *Let the assumptions of Theorem K.7 hold, $f_\xi(x)$ are convex for each $\xi$ and $\mu > 0$. Then after $K$ iterations of `D-QLSVRG` with the stepsize*

$$\gamma_0 = \min\left\{\frac{3}{56L(1 + 2\omega/n)}, \frac{1}{8L\sqrt{\tau\left(\tau + 2\left(1 + 2\omega/n\right)\left(1 + 2/(1-p)\right)\right)}}\right\}, \quad R_0 = \|x^0 - x^*\|,$$

$$\gamma = \min\left\{\gamma_0, \frac{\ln\left(\max\left\{2, \min\left\{\frac{R_0^2\mu^2 K^2}{D_1'}, \frac{R_0^2\mu^3 K^3}{3\tau L D_1'}\right\}\right\}\right)}{\mu K}\right\}$$

*and $p = \frac{1}{m}$, $m \geq 2$ we have $\mathbf{E}\left[f(\bar{x}^K) - f(x^*)\right]$ of order*

$$\widetilde{\mathcal{O}}\left(LR_0^2\left(1 + \frac{\omega}{n} + \sqrt{\tau\left(\tau + \frac{\omega}{n}\right)}\right)\exp\left(-\frac{\mu}{L\left(1 + \frac{\omega}{n} + \sqrt{\tau\left(\tau + \frac{\omega}{n}\right)}\right)}K\right) + \frac{D_1'}{\mu K} + \frac{L\tau D_1'}{\mu^2 K^2}\right).$$

*That is, to achive $\mathbf{E}\left[f(\bar{x}^K) - f(x^*)\right] \leq \varepsilon$ `D-QLSVRG` requires*

$$\widetilde{\mathcal{O}}\left(\frac{L}{\mu}\left(1 + \frac{\omega}{n}\right) + \frac{L}{\mu}\sqrt{\tau\left(\tau + \frac{\omega}{n}\right)} + \frac{D_1'}{\mu\varepsilon} + \frac{\sqrt{L\tau D_1'}}{\mu\sqrt{\varepsilon}}\right) \quad iterations.$$

Applying Lemma D.3 we get the complexity result in the case when $\mu = 0$.

**Corollary K.14.** *Let the assumptions of Theorem K.7 hold and $\mu = 0$. Then after $K$ iterations of `D-QLSVRG` with the stepsize*

$$\gamma_0 = \min\left\{\frac{3}{56L(1 + 2\omega/n)}, \frac{1}{8L\sqrt{\tau\left(\tau + 2\left(1 + 2\omega/n\right)\left(1 + 2/(1-p)\right)\right)}}\right\}, \quad R_0 = \|x^0 - x^*\|,$$

$$\gamma = \min\left\{\gamma_0, \sqrt{\frac{R_0^2}{M_2\sigma_0^2}}, \sqrt[3]{\frac{R_0^2 p}{12L\tau\left(1 + \frac{2\omega}{n}\right)(2+p)}}, \sqrt{\frac{R_0^2}{D_1'K}}, \sqrt[3]{\frac{R_0^2}{3L\tau D_1'K}}\right\}$$

*and $p = \frac{1}{m}$, $m \geq 2$ we have $\mathbf{E}\left[f(\bar{x}^K) - f(x^*)\right]$ of order*

$$\mathcal{O}\left(\frac{LR_0^2\left(1 + \frac{\omega}{n} + \sqrt{\tau\left(\tau + \frac{\omega}{n}\right)}\right)}{K} + \frac{\sqrt{R_0^2 m\left(1 + \frac{\omega}{n}\right)\sigma_0^2}}{K} + \frac{\sqrt[3]{R_0^4 L\tau m\left(1 + \frac{\omega}{n}\right)}}{K}\right)$$

$$+\mathcal{O}\left(\sqrt{\frac{R_0^2 D_1'}{K}} + \frac{\sqrt[3]{LR_0^4\tau D_1'}}{K^{2/3}}\right).$$

*That is, to achive $\mathbf{E}\left[f(\bar{x}^K) - f(x^*)\right] \leq \varepsilon$ `D-QLSVRG` requires*

$$\mathcal{O}\left(\frac{LR_0^2\left(1 + \frac{\omega}{n} + \sqrt{\tau\left(\tau + \frac{\omega}{n}\right)}\right)}{\varepsilon} + \frac{\sqrt{R_0^2 m\left(1 + \frac{\omega}{n}\right)\sigma_0^2}}{\varepsilon} + \frac{\sqrt[3]{R_0^4 L\tau m\left(1 + \frac{\omega}{n}\right)}}{\varepsilon}\right)$$

$$+\mathcal{O}\left(\frac{R_0^2 D_1'}{\varepsilon^2} + \frac{R_0^2\sqrt{L\tau D_1'}}{\varepsilon^{3/2}}\right).$$

*iterations.*

**Algorithm 18** D-QLSVRGstar

**Input:** learning rate $\gamma > 0$, initial vector $x^0 \in \mathbb{R}^d$
 1: Set $e_i^0 = 0$ for all $i = 1, \ldots, n$
 2: **for** $k = 0, 1, \ldots$ **do**
 3:      Broadcast $x^{k-\tau}$ to all workers
 4:      **for** $i = 1, \ldots, n$ in parallel **do**
 5:          Pick $l$ uniformly at random from $[m]$
 6:          Set $\hat{g}_i^{k-\tau} = \nabla f_{il}(x^{k-\tau}) - \nabla f_{il}(w_i^{k-\tau}) + \nabla f_i(w_i^{k-\tau})$
 7:          Set $g_i^{k-\tau} = Q(\hat{g}_i^{k-\tau} - \nabla f_i(x^*))$ (quantization is performed independently from
     other nodes)
 8:          $v_i^k = \gamma g_i^{k-\tau}$
 9:          $e_i^{k+1} = e_i^k + \gamma g_i^k - v_i^k$
10:          $w_i^{k-\tau+1} = \begin{cases} x^{k-\tau}, & \text{with probability } p, \\ w_i^{k-\tau}, & \text{with probability } 1-p \end{cases}$
11:      **end for**
12:      $e^k = \frac{1}{n} \sum_{i=1}^n e_i^k$, $g^k = \frac{1}{n} \sum_{i=1}^n g_i^k$, $v^k = \frac{1}{n} \sum_{i=1}^n v_i^k$
13:      $x^{k+1} = x^k - v^k$
14: **end for**

### K.8   D-QLSVRGstar

Now we assume that $i$-th node has an access to $\nabla f_i(x^*)$ and modify D-QLSVRG in order to get convergence asymptotically to the exact optimum.

**Lemma K.10.** *For all $k \geq 0$, $i \in [n]$ we have*

$$\mathbf{E}\left[g^k \mid x^k\right] = \nabla f(x^k) \tag{141}$$

*and*

$$\mathbf{E}\left[\|g^k\|^2 \mid x^k\right] \leq 2L\left(1 + \frac{\omega}{n}\right)\left(f(x^k) - f(x^*)\right) + 2\left(1 + \frac{\omega}{n}\right)\sigma_k^2, \tag{142}$$

*where $\sigma_k^2 = \frac{1}{nm} \sum_{i=1}^n \sum_{j=1}^n \|\nabla f_{ij}(w_i^k) - \nabla f_{ij}(x^*)\|^2$.*

*Proof.* First of all, we derive unbiasedness of $g_i^k$:

$$\mathbf{E}\left[g^k \mid x^k\right] \overset{(35)}{=} \mathbf{E}\left[\mathbf{E}_Q\left[\frac{1}{n}\sum_{i=1}^n Q(\hat{g}_i^k - \nabla f_i(x^*))\right] \mid x^k\right] \overset{(26)}{=} \mathbf{E}\left[\frac{1}{n}\sum_{i=1}^n \left(\hat{g}_i^k - \nabla f_i(x^*)\right) \mid x^k\right]$$

$$= \frac{1}{nm}\sum_{i=1}^n \sum_{j=1}^m \left(\nabla f_{ij}(x^k) - \nabla f_{ij}(w_i^k) + \nabla f_i(w_i^k)\right) = \nabla f(x^k).$$

Next, we estimate the second moment of $g^k$:

$$\mathbf{E}_Q\left[\|g^k\|^2\right] = \mathbf{E}_Q\left[\left\|\frac{1}{n}\sum_{i=1}^n Q(\hat{g}_i^k - \nabla f_i(x^*))\right\|^2\right]$$

$$\overset{(34)}{=} \mathbf{E}_Q\left[\left\|\frac{1}{n}\sum_{i=1}^n \left(Q(\hat{g}_i^k - \nabla f_i(x^*)) - (\hat{g}_i^k - \nabla f_i(x^*))\right)\right\|^2\right] + \left\|\frac{1}{n}\sum_{i=1}^n \hat{g}_i^k - \nabla f_i(x^*)\right\|^2.$$

Since quantization on nodes is performed independently we can decompose the first term from the last row of the previous inequality into the sum of variances:

$$
\mathbf{E}_Q \left[\|g^k\|^2\right] \quad = \quad \frac{1}{n^2} \sum_{i=1}^{n} \mathbf{E}_Q \left\|Q(\hat{g}_i^k - \nabla f_i(x^*)) - (\hat{g}_i^k - \nabla f_i(x^*))\right\|^2 + \left\|\frac{1}{n}\sum_{i=1}^{n} \hat{g}_i^k - \nabla f_i(x^*)\right\|^2
$$

$$
\overset{(26)}{\leq} \quad \frac{\omega}{n^2} \sum_{i=1}^{n} \|\hat{g}_i^k - \nabla f_i(x^*)\|^2 + \left\|\frac{1}{n}\sum_{i=1}^{n} \left(\hat{g}_i^k - \nabla f_i(x^*)\right)\right\|^2
$$

$$
\overset{(31)}{\leq} \quad \left(1 + \frac{\omega}{n}\right)\frac{1}{n}\sum_{i=1}^{n} \|\hat{g}_i^k - \nabla f_i(x^*)\|^2.
$$

Taking conditional mathematical expectation $\mathbf{E}\left[\cdot \mid x^k\right]$ from the both sides of previous inequality and using the bound

$$
\frac{1}{n}\sum_{i=1}^{n}\mathbf{E}\left[\|\hat{g}_i^k - \nabla f_i(x^*)\|^2 \mid x^k\right] \leq 4L\left(f(x^k) - f(x^*)\right) + 2\sigma_k^2
$$

implicitly obtained in the proof of Lemma K.8 we get (142). $\qquad\square$

**Lemma K.11.** *For all $k \geq 0$, $i \in [n]$ we have*

$$
\mathbf{E}\left[\sigma_{k+1}^2 \mid x^k\right] \leq (1-p)\sigma_k^2 + 2Lp\left(f(x^k) - f(x^*)\right), \tag{143}
$$

*where $\sigma_k^2 = \frac{1}{nm}\sum_{i=1}^{n}\sum_{j=1}^{n}\|\nabla f_{ij}(w_i^k) - \nabla f_{ij}(x^*)\|^2$.*

*Proof.* The proof is identical to the proof of Lemma J.8. $\qquad\square$

**Theorem K.8.** *Assume that $f(x)$ is $\mu$-quasi strongly convex and functions $f_{ij}$ are convex and $L$-smooth for all $i \in [n], j \in [m]$. Then* D-QLSVRGstar *satisfies Assumption 3.4 with*

$$
A' = 2L\left(1 + \frac{2\omega}{n}\right), \quad B_1' = 0, \quad B_2' = 2\left(1 + \frac{2\omega}{n}\right), \quad D_1' = 0, \quad \sigma_{1,0}^2 \equiv 0,
$$

$$
\sigma_{2,k}^2 = \sigma_k^2 = \frac{1}{nm}\sum_{i=1}^{n}\sum_{j=1}^{m}\|\nabla f_{ij}(w_i^k) - \nabla f_{ij}(x^*)\|^2, \quad \rho_1 = 1, \quad \rho_2 = p, \quad C_2 = Lp, \quad D_2 = 0,
$$

$$
C_1 = 0, \quad G = 0, \quad F_1 = 0, \quad F_2 = \frac{12\gamma^2 L\tau\left(1 + \frac{2\omega}{n}\right)(2+p)}{p}, \quad D_3 = 0
$$

*with $\gamma$ satisfying*

$$
\gamma \leq \min\left\{\frac{3}{56L(1 + 2\omega/n)}, \frac{1}{8L\sqrt{\tau\left(\tau + 2\left(1 + 2\omega/n\right)\left(1 + 2/(1-p)\right)\right)}}\right\}, \quad M_2 = \frac{8\left(1 + \frac{2\omega}{n}\right)}{3p}
$$

*and for all $K \geq 0$*

$$
\mathbf{E}\left[f(\bar{x}^K) - f(x^*)\right] \leq \left(1 - \min\left\{\frac{\gamma\mu}{2}, \frac{p}{4}\right\}\right)^K \frac{4(T^0 + \gamma F_2\sigma_0^2)}{\gamma}
$$

*when $\mu > 0$ and*

$$
\mathbf{E}\left[f(\bar{x}^K) - f(x^*)\right] \leq \frac{4(T^0 + \gamma F_2\sigma_0^2)}{\gamma K}
$$

*when $\mu = 0$, where $T^k \overset{def}{=} \|\tilde{x}^k - x^*\|^2 + M_2\gamma^2\sigma_k^2$.*

In other words, D-QLSVRGstar converges with linear rate

$$
\mathcal{O}\left(\left(\frac{1}{p} + \kappa\left(1 + \frac{\omega}{n}\right) + \kappa\sqrt{\tau\left(\tau + \left(1 + \frac{\omega}{n}\right)\left(1 + \frac{1}{(1-p)}\right)\right)}\right)\ln\frac{1}{\varepsilon}\right)
$$

to the exact solution when $\mu > 0$. If $m \geq 2$ then taking $p = \frac{1}{m}$ we get that in expectation the sample complexity of one iteration of `D-QLSVRGstar` is $\mathcal{O}(1)$ gradients calculations per node as for `D-QSGDsr` with standard sampling and the rate of convergence to the exact solution becomes

$$\mathcal{O}\left(\left(m + \kappa\left(1 + \frac{\omega}{n}\right) + \kappa\sqrt{\tau\left(\tau + \frac{\omega}{n}\right)}\right)\ln\frac{1}{\varepsilon}\right).$$

Applying Lemma D.3 we get the complexity result in the case when $\mu = 0$.

**Corollary K.15.** *Let the assumptions of Theorem K.8 hold and $\mu = 0$. Then after $K$ iterations of* `D-QLSVRGstar` *with the stepsize*

$$\gamma_0 \;\; = \;\; \min\left\{\frac{3}{56L(1 + 2\omega/n)}, \frac{1}{8L\sqrt{\tau\left(\tau + 2\left(1 + 2\omega/n\right)\left(1 + 2/(1-p)\right)\right)}}\right\}, \quad R_0 = \|x^0 - x^*\|,$$

$$\gamma \;\; = \;\; \min\left\{\gamma_0, \sqrt{\frac{R_0^2}{M_2\sigma_0^2}}, \sqrt[3]{\frac{R_0^2 p}{12L\tau\left(1 + \frac{2\omega}{n}\right)(2 + p)}}\right\}$$

*and $p = \frac{1}{m}$, $m \geq 2$ we have $\mathbf{E}\left[f(\bar{x}^K) - f(x^*)\right]$ of order*

$$\mathcal{O}\left(\frac{LR_0^2\left(1 + \frac{\omega}{n} + \sqrt{\tau\left(\tau + \frac{\omega}{n}\right)}\right)}{K} + \frac{\sqrt{R_0^2 m\left(1 + \frac{\omega}{n}\right)\sigma_0^2}}{K} + \frac{\sqrt[3]{R_0^4 L\tau m\left(1 + \frac{\omega}{n}\right)}}{K}\right).$$

*That is, to achive $\mathbf{E}\left[f(\bar{x}^K) - f(x^*)\right] \leq \varepsilon$* `D-QLSVRGstar` *requires*

$$\mathcal{O}\left(\frac{LR_0^2\left(1 + \frac{\omega}{n} + \sqrt{\tau\left(\tau + \frac{\omega}{n}\right)}\right)}{\varepsilon} + \frac{\sqrt{R_0^2 m\left(1 + \frac{\omega}{n}\right)\sigma_0^2}}{\varepsilon} + \frac{\sqrt[3]{R_0^4 L\tau m\left(1 + \frac{\omega}{n}\right)}}{\varepsilon}\right)$$

*iterations.*

However, such convergence guarantees are obtained under very restrictive assumption: the method requires to know vectors $\nabla f_i(x^*)$.

### K.9 D-LSVRG-DIANA

In the setup of Section K.6 we construct a new method with delayed updates and quantization called `D-LSVRG-DIANA` which does not require to know $\nabla f_i(x^*)$ and has linear convergence to the exact solution.

**Lemma K.12.** *Assume that $f_{ij}(x)$ is convex and $L$-smooth for all $i = 1, \ldots, n$, $j = 1, \ldots, m$. Then, for all $k \geq 0$ we have*

$$\mathbf{E}\left[g^k \mid x^k\right] \;\; = \;\; \nabla f(x^k), \tag{144}$$

$$\mathbf{E}\left[\|g^k\|^2 \mid x^k\right] \;\; \leq \;\; 4L\left(1 + \frac{2\omega}{n}\right)\left(f(x^k) - f(x^*)\right) + \frac{2\omega}{n}\sigma_{1,k}^2 + 2\left(1 + \frac{2\omega}{n}\right)\sigma_{2,k}^2 \tag{145}$$

*where $\sigma_{1,k}^2 = \frac{1}{n}\sum_{i=1}^n \|h_i^k - \nabla f(x^*)\|^2$ and $\sigma_{2,k}^2 = \frac{1}{nm}\sum_{i=1}^n \sum_{j=1}^m \|\nabla f_{ij}(w_i^k) - \nabla f_{ij}(x^*)\|^2$.*

*Proof.* First of all, we show unbiasedness of $g^k$:

$$\mathbf{E}\left[g^k \mid x^k\right] \;\; \overset{(35)}{=} \;\; h^k + \frac{1}{n}\sum_{i=1}^n \mathbf{E}\left[\mathbf{E}_Q\left[\hat{\Delta}_i^k\right] \mid x^k\right] \overset{(26)}{=} h^k + \frac{1}{n}\sum_{i=1}^n \mathbf{E}\left[\hat{g}_i^k - h_i^k \mid x^k\right]$$

$$= \;\; \frac{1}{nm}\sum_{i=1}^n \sum_{j=1}^m \left(\nabla f_{ij}(x^k) - \nabla f_{ij}(w_i^k) + \nabla f_i(w_i^k)\right) = \nabla f(x^k).$$

---

**Algorithm 19** D-LSVRG-DIANA

---

**Input:** learning rates $\gamma > 0$, $\alpha \in (0,1]$, initial vectors $x^0, h_1^0, \ldots, h_n^0 \in \mathbb{R}^d$

1: Set $e_i^0 = 0$ for all $i = 1, \ldots, n$
2: Set $h^0 = \frac{1}{n} \sum_{i=1}^n h_i^0$
3: **for** $k = 0, 1, \ldots$ **do**
4:      Broadcast $x^{k-\tau}$ to all workers
5:      **for** $i = 1, \ldots, n$ in parallel **do**
6:          Pick $l$ uniformly at random from $[m]$
7:          Set $\hat{g}_i^{k-\tau} = \nabla f_{il}(x^{k-\tau}) - \nabla f_{il}(w_i^{k-\tau}) + \nabla f_i(w_i^{k-\tau})$
8:          $\hat{\Delta}_i^{k-\tau} = Q(\hat{g}_i^{k-\tau} - h_i^{k-\tau})$ (quantization is performed independently from other nodes)
9:          $g_i^{k-\tau} = h_i^{k-\tau} + \hat{\Delta}_i^{k-\tau}$
10:         $v_i^k = \gamma g_i^{k-\tau}$
11:         $e_i^{k+1} = e_i^k + \gamma g_i^k - v_i^k$
12:         $h_i^{k-\tau+1} = h_i^{k-\tau} + \alpha \hat{\Delta}_i^{k-\tau}$
13:      **end for**
14:      $e^k = \frac{1}{n} \sum_{i=1}^n e_i^k$, $g^k = \frac{1}{n} \sum_{i=1}^n g_i^k = h^k + \frac{1}{n} \sum_{i=1}^n \hat{\Delta}_i^k$, $v^k = \frac{1}{n} \sum_{i=1}^n v_i^k = \gamma h^{k-\tau} + \frac{\gamma}{n} \sum_{i=1}^n \hat{\Delta}_i^{k-\tau}$
15:      $h^{k-\tau+1} = \frac{1}{n} \sum_{i=1}^n h_i^{k-\tau+1} = h^{k-\tau} + \alpha \frac{1}{n} \sum_{i=1}^n \hat{\Delta}_i^{k-\tau}$
16:      $x^{k+1} = x^k - v^k$
17: **end for**

---

Next, we derive the upper bound for the second moment of $g^k$:

$$
\begin{aligned}
\mathbf{E}_Q\left[\|g^k\|^2\right] &= \mathbf{E}_Q\left[\left\|h^k + \frac{1}{n}\sum_{i=1}^n \hat{\Delta}_i^k\right\|^2\right] \\
&\overset{(34)}{=} \mathbf{E}_Q\left[\left\|\frac{1}{n}\sum_{i=1}^n\left(\hat{\Delta}_i^k - \hat{g}_i^k + h_i^k\right)\right\|^2\right] + \left\|\frac{1}{n}\sum_{i=1}^n \hat{g}_i^k\right\|^2.
\end{aligned}
$$

Since quantization on nodes is performed independently we can decompose the first term from the last row of the previous inequality into the sum of variances:

$$
\begin{aligned}
\mathbf{E}_Q\left[\|g^k\|^2\right] &\leq \frac{1}{n^2}\sum_{i=1}^n \mathbf{E}_Q\left[\|\hat{\Delta}_i^k - \hat{g}_i^k + h_i^k\|^2\right] + \left\|\frac{1}{n}\sum_{i=1}^n\left(\hat{g}_i^k - \nabla f_i(x^*)\right)\right\|^2 \\
&\overset{(26),(31)}{\leq} \frac{\omega}{n^2}\sum_{i=1}^n \|\hat{g}_i^k - h_i^k\|^2 + \frac{1}{n}\sum_{i=1}^n \|\hat{g}_i^k - \nabla f_i(x^*)\|^2 \\
&\overset{(31)}{\leq} \left(1 + \frac{2\omega}{n}\right)\frac{1}{n}\sum_{i=1}^n \|\hat{g}_i^k - \nabla f_i(x^*)\|^2 + \frac{2\omega}{n^2}\sum_{i=1}^n \|h_i^k - f_i(x^*)\|^2.
\end{aligned}
$$

Taking mathematical expectation $\mathbf{E}\left[\cdot \mid x^k\right]$ from the both sides of the previous inequality and using the bound

$$
\frac{1}{n}\sum_{i=1}^n \mathbf{E}\left[\|\hat{g}_i^k - \nabla f_i(x^*)\|^2 \mid x^k\right] \leq 4L\left(f(x^k) - f(x^*)\right) + \frac{2}{nm}\sum_{i=1}^n\sum_{j=1}^m \|\nabla f_{ij}(w_i^k) - \nabla f_{ij}(x^*)\|^2
$$

implicitly obtained in the proof of Lemma K.8 we get (145). $\qquad\square$

**Lemma K.13.** *Assume that* $\alpha \leq 1/(\omega+1)$. *Then, for all* $k \geq 0$ *we have*

$$
\mathbf{E}\left[\sigma_{1,k+1}^2 \mid x^k\right] \leq (1-\alpha)\sigma_{1,k}^2 + 6L\alpha(f(x^k) - f(x^*)) + 2\alpha\sigma_{2,k}^2,
$$

$$\mathbf{E}\left[\sigma_{2,k+1}^2 \mid x^k\right] \leq (1-p)\sigma_{k,2}^2 + 2Lp\left(f(x^k) - f(x^*)\right)$$

*where $\sigma_{1,k}^2 = \frac{1}{n}\sum_{i=1}^n \|h_i^k - \nabla f_i(x^*)\|^2$ and $\sigma_{2,k}^2 = \frac{1}{nm}\sum_{i=1}^n \sum_{j=1}^m \|\nabla f_{ij}(w_i^k) - \nabla f_{ij}(x^*)\|^2$.*

*Proof.* The proof is identical to the proof of Lemma J.12. □

**Theorem K.9.** *Assume that $f_{ij}(x)$ is convex and $L$-smooth for all $i = 1, \ldots, n$, $j = 1, \ldots, m$ and $f(x)$ is $\mu$-quasi strongly convex. Then* `D-LSVRG-DIANA` *satisfies Assumption 3.4 with*

$$A' = 2L\left(1 + \frac{2\omega}{n}\right), \quad B_1' = \frac{2\omega}{n}, \quad B_2' = 2\left(1 + \frac{2\omega}{n}\right), \quad D_1' = 0,$$

$$\sigma_{1,k}^2 = \frac{1}{n}\sum_{i=1}^n \|h_i^k - \nabla f_i(x^*)\|^2, \quad \sigma_{2,k}^2 = \frac{1}{nm}\sum_{i=1}^n \sum_{j=1}^m \|\nabla f_{ij}(w_i^k) - \nabla f_{ij}(x^*)\|^2,$$

$$\rho_1 = \alpha, \quad \rho_2 = p, \quad C_1 = 3L\alpha, \quad C_2 = Lp, \quad D_2 = 0, \quad G = 2,$$

$$F_1 = \frac{12\gamma^2 L\omega\tau(2+\alpha)}{n\alpha}, \quad F_2 = \frac{12\gamma^2\tau L(2+p)}{p}\left(\frac{4\omega}{n(1-\alpha)} + 1 + \frac{2\omega}{n}\right), \quad D_3 = 0$$

*with $\gamma$ and $\alpha$ satisfying*

$$\gamma \leq \min\left\{\frac{1}{8L\left(\frac{37}{9} + \frac{24\omega}{3n}\right)}, \frac{1}{8L\sqrt{\tau\left(2 + \tau + \frac{4}{1-p} + \frac{4\omega}{n}\left(1 + \frac{3}{1-\alpha} + \frac{2}{1-p} + \frac{4}{(1-\alpha)(1-p)}\right)\right)}}\right\},$$

$$\alpha \leq \frac{1}{\omega + 1}, \quad M_1 = \frac{8\omega}{3n\alpha}, \quad M_2 = \frac{8\left(7 + \frac{6\omega}{n}\right)}{9p}.$$

*and for all $K \geq 0$*

$$\mathbf{E}\left[f(\bar{x}^K) - f(x^*)\right] \leq \left(1 - \min\left\{\frac{\gamma\mu}{2}, \frac{\alpha}{4}, \frac{p}{4}\right\}\right)^K \frac{4(T^0 + \gamma F_1\sigma_{1,0}^2 + \gamma F_2\sigma_{2,0}^2)}{\gamma}$$

*when $\mu > 0$ and*

$$\mathbf{E}\left[f(\bar{x}^K) - f(x^*)\right] \leq \frac{4(T^0 + \gamma F_1\sigma_{1,0}^2 + \gamma F_2\sigma_{2,0}^2)}{\gamma K}$$

*when $\mu = 0$, where $T^k \stackrel{def}{=} \|\tilde{x}^k - x^*\|^2 + M_1\gamma^2\sigma_{1,k}^2 + M_2\gamma^2\sigma_{2,k}^2$.*

In other words, if $m \geq 2$, $p = 1/m$, $\alpha = \min\left\{\frac{1}{\omega+1}, \frac{1}{2}\right\}$ and

$$\gamma \leq \min\left\{\frac{1}{8L\left(\frac{37}{9} + \frac{24\omega}{3n}\right)}, \frac{1}{8L\sqrt{\tau\left(2 + \tau + \frac{4}{1-p} + \frac{4\omega}{n}\left(1 + \frac{3}{1-\alpha} + \frac{2}{1-p} + \frac{4}{(1-\alpha)(1-p)}\right)\right)}}\right\},$$

`D-LSVRG-DIANA` converges with the linear rate

$$\mathcal{O}\left(\left(\omega + m + \kappa\left(1 + \frac{\omega}{n}\right) + \kappa\sqrt{\tau\left(\tau + \frac{\omega}{n}\right)}\right)\ln\frac{1}{\varepsilon}\right)$$

to the exact solution when $\mu > 0$.

Applying Lemma D.3 we get the complexity result in the case when $\mu = 0$.

**Corollary K.16.** *Let the assumptions of Theorem K.9 hold and $\mu = 0$. Then after $K$ iterations of* `D-LSVRG-DIANA` *with the stepsize*

$$\gamma_0 = \min\left\{\frac{1}{8L\left(\frac{37}{9} + \frac{24\omega}{3n}\right)}, \frac{1}{8L\sqrt{\tau\left(2 + \tau + \frac{4}{1-p} + \frac{4\omega}{n}\left(1 + \frac{3}{1-\alpha} + \frac{2}{1-p} + \frac{4}{(1-\alpha)(1-p)}\right)\right)}}\right\},$$

$$\gamma = \min\left\{\gamma_0, \sqrt{\frac{R_0^2}{M_1\sigma_{1,0}^2 + M_2\sigma_{2,0}^2}}, \sqrt[3]{\frac{R_0^2}{12\tau L\left(\frac{\omega(2+\alpha)}{n\alpha} + \frac{2+p}{p}\left(1 + \frac{2\omega}{n} + \frac{4\omega}{n(1-\alpha)}\right)\right)}}\right\},$$

where $R_0 = \|x^0 - x^*\|$, $\alpha = \min\left\{\frac{1}{\omega+1}, \frac{1}{2}\right\}$ and $p = \frac{1}{m}$, $m \geq 2$ we have $\mathbf{E}\left[f(\bar{x}^K) - f(x^*)\right]$ of order

$$\mathcal{O}\left(\frac{LR_0^2\left(1 + \frac{\omega}{n} + \sqrt{\tau\left(\tau + \frac{\omega}{n}\right)}\right)}{K} + \frac{\sqrt{R_0^2\omega(\omega+1)\sigma_{1,0}^2}}{\sqrt{n}K} + \frac{\sqrt{R_0^2 m\left(1 + \frac{\omega}{n}\right)\sigma_{2,0}^2}}{K}\right)$$

$$+\mathcal{O}\left(\frac{\sqrt[3]{R_0^4\tau L\omega(\omega+1)\sigma_{1,0}^2}}{\sqrt[3]{n}K} + \frac{\sqrt[3]{R_0^4\tau Lm\left(1 + \frac{\omega}{n}\right)\sigma_{2,0}^2}}{K}\right)$$

That is, to achive $\mathbf{E}\left[f(\bar{x}^K) - f(x^*)\right] \leq \varepsilon$ $\mathtt{D\text{-}LSVRG\text{-}DIANA}$ requires

$$\mathcal{O}\left(\frac{LR_0^2\left(1 + \frac{\omega}{n} + \sqrt{\tau\left(\tau + \frac{\omega}{n}\right)}\right)}{\varepsilon} + \frac{\sqrt{R_0^2\omega(\omega+1)\sigma_{1,0}^2}}{\sqrt{n}\varepsilon} + \frac{\sqrt{R_0^2 m\left(1 + \frac{\omega}{n}\right)\sigma_{2,0}^2}}{\varepsilon}\right)$$

$$+\mathcal{O}\left(\frac{\sqrt[3]{R_0^4\tau L\omega(\omega+1)\sigma_{1,0}^2}}{\sqrt[3]{n}\varepsilon} + \frac{\sqrt[3]{R_0^4\tau Lm\left(1 + \frac{\omega}{n}\right)\sigma_{2,0}^2}}{\varepsilon}\right)$$

iterations.

Table 5: The parameters for which the methods from Tables 1 and 4 satisfy Assumption 3.4. The meaning of the expressions appearing in the table, as well as their justification is defined in details in the Sections J and K. Symbols: $\varepsilon$ = error tolerance; $\delta$ = contraction factor of compressor $\mathcal{C}$; $\omega$ = variance parameter of compressor $\mathcal{Q}$; $\kappa = L/\mu$; $\mathcal{L}$ = expected smoothness constant; $\sigma_*^2$ = variance of the stochastic gradients in the solution; $\zeta_*^2$ = average of $\|\nabla f_i(x^*)\|^2$; $\sigma^2$ = average of the uniform bounds for the variances of stochastic gradients of workers.

| Method | $A'$ | $B'_1$ | $B'_2$ | $\rho_1$ | $\rho_2$ | $C_1$ | $C_2$ | $F_1,\ F_2$ | $G$ | $D'_1,\ D_2,\ D_3$ |
|---|---|---|---|---|---|---|---|---|---|---|
| EC-SGDsr | $2\mathcal{L}$ | $0$ | $0$ | $1$ | $1$ | $0$ | $0$ | $0,\ 0$ | $0$ | $\frac{2\sigma_*^2}{n},\ 0,\ \frac{6L\gamma}{\delta}\left(\frac{4\zeta_*^2}{\delta}+3\sigma_*^2\right)$ |
| EC-SGD | $2L$ | $0$ | $0$ | $1$ | $1$ | $0$ | $0$ | $0,\ 0$ | $0$ | $\frac{2\sigma_*^2}{n},\ 0,\ \frac{12L\gamma}{\delta}\left(\frac{2\zeta_*^2}{\delta}+\sigma_*^2\right)$ |
| EC-GDstar | $L$ | $0$ | $0$ | $1$ | $1$ | $0$ | $0$ | $0,\ 0$ | $0$ | $0,\ 0,\ 0$ |
| EC-SGD-DIANA | $L$ | $0$ | $0$ | $\alpha$ | $1$ | $L\alpha$ | $0$ | $\frac{96L\gamma^2}{\delta^2\alpha(1-\eta)},\ 0$ | $0$ | $\frac{\sigma_*^2}{n},\ \frac{6L\gamma}{\delta}\left(\frac{\alpha^2(\omega+1)}{4\alpha(\omega+1)}+1\right)\sigma^2,\ \sigma^2$ |
| EC-SGDsr-DIANA | $2\mathcal{L}$ | $0$ | $0$ | $\alpha$ | $1$ | $2\alpha(3\mathcal{L}+4L)$ | $0$ | $\frac{96L\gamma^2}{\delta^2\alpha(1-\eta)},\ 0$ | $0$ | $\frac{2\sigma_*^2}{n},\ \frac{18L\gamma}{\delta}\left(\frac{\alpha^2(\omega+1)}{4\alpha(\omega+1)}+1\right)\sigma^2,\ \sigma_*^2$ |
| EC-LSVRG | $2L$ | $0$ | $2$ | $1$ | $p$ | $0$ | $Lp$ | $0,\ \frac{72L\gamma^2}{\delta p(1-\eta)}$ | $0$ | $0,\ 0,\ \frac{24L\gamma}{\delta^2}\zeta_*^2$ |
| EC-LSVRGstar | $2L$ | $0$ | $2$ | $1$ | $p$ | $0$ | $Lp$ | $0,\ \frac{48L\gamma^2}{\delta p}$ | $0$ | $0,\ 0,\ 0$ |
| EC-LSVRG-DIANA | $2L$ | $0$ | $2$ | $\alpha$ | $p$ | $3L\alpha$ | $Lp$ | $24L\gamma^2\left(\frac{4}{\delta}+3\right),\ \frac{24L\gamma^2\left(\frac{\delta\alpha(1-\eta)}{1-\alpha}\left(\frac{4}{\delta}+3\right)+3\right)}{\delta p(1-\eta)}$ | $2$ | $0,\ 0,\ 0$ |
| D-SGDsr | $2\mathcal{L}$ | $0$ | $0$ | $1$ | $1$ | $0$ | $0$ | $0,\ 0$ | $0$ | $\frac{2\sigma_*^2}{n},\ 0,\ \frac{6L\tau\gamma\sigma_*^2}{n}$ |
| D-SGD | $2L$ | $0$ | $0$ | $1$ | $1$ | $0$ | $0$ | $0,\ 0$ | $0$ | $\frac{2\sigma_*^2}{n},\ 0,\ \frac{6L\tau\gamma\sigma_*^2}{n}$ |
| D-QSGD | $L\left(1+\frac{2\omega}{n}\right)$ | $0$ | $0$ | $1$ | $1$ | $0$ | $0$ | $0,\ 0$ | $0$ | $\frac{(\omega+1)\sigma^2}{n}+\frac{2\omega\zeta_*^2}{n},\ 0,\ \frac{3\gamma\tau L}{n}\left((\omega+1)\sigma^2+2\omega\zeta_*^2\right)$ |
| D-QSGDstar | $L\left(1+\frac{\omega}{n}\right)$ | $0$ | $0$ | $1$ | $1$ | $0$ | $0$ | $0,\ 0$ | $0$ | $\frac{(\omega+1)\sigma^2}{n},\ 0,\ \frac{3\gamma\tau L(\omega+1)\sigma^2}{n}$ |
| D-QGDstar | $L\left(1+\frac{\omega}{n}\right)$ | $0$ | $0$ | $1$ | $1$ | $0$ | $0$ | $0,\ 0$ | $0$ | $0,\ 0,\ 0$ |
| D-SGD-DIANA | $L\left(1+\frac{2\omega}{n}\right)$ | $\frac{2\omega}{n}$ | $0$ | $\alpha$ | $1$ | $L\alpha$ | $0$ | $\frac{12\gamma^2 L\omega\tau(2+\alpha)}{n\alpha},\ 0$ | $0$ | $\frac{(\omega+1)\sigma^2}{n},\ 3\gamma\tau L\left(1+\frac{4\omega}{n}\right),\ \frac{\alpha(\omega+1)\sigma^2}{n},\ \frac{(\omega+1)\sigma^2}{n}$ |
| D-LSVRG | $2L$ | $0$ | $2$ | $1$ | $p$ | $0$ | $Lp$ | $0,\ \frac{12\gamma^2 L\tau(2+p)}{np}$ | $0$ | $0,\ 0,\ 0$ |
| D-QLSVRG | $2L\left(1+\frac{2\omega}{n}\right)$ | $0$ | $2\left(1+\frac{2\omega}{n}\right)$ | $1$ | $p$ | $0$ | $Lp$ | $0,\ \frac{12\gamma^2 L\tau\left(1+\frac{2\omega}{n}\right)\tau(2+p)}{p}$ | $0$ | $\frac{2\omega\zeta_*^2}{n},\ 0,\ \frac{6\gamma\tau L\omega\zeta_*^2}{n}$ |
| D-QLSVRGstar | $2L\left(1+\frac{2\omega}{n}\right)$ | $0$ | $2\left(1+\frac{2\omega}{n}\right)$ | $1$ | $p$ | $0$ | $Lp$ | $0,\ \frac{12\gamma^2 L\omega\tau\left(1+\frac{2\omega}{n}\right)\tau(2+\alpha)}{p}$ | $0$ | $0,\ 0,\ 0$ |
| D-LSVRG-DIANA | $2L\left(1+\frac{2\omega}{n}\right)$ | $\frac{2\omega}{n}$ | $2\left(1+\frac{2\omega}{n}\right)$ | $\alpha$ | $p$ | $3L\alpha$ | $Lp$ | $\frac{12\gamma^2\tau L(2+p)}{p},\ \frac{12\gamma^2 L\omega\tau(2+\alpha)}{n\alpha}\left(1+\frac{2\omega(3-\alpha)}{n(1-\alpha)}\right)$ | $0$ | $0,\ 0,\ 0$ |