[Reviews · NeurIPS 2020]

Review 1

Summary and Contributions: Distributed optimization with communication based errors are a fundamental problem in optimization. Here, n agents have to communicate gradients of the component functions each of them store, but to make communication efficient, they are compressed with some possibly lossy map, which removes the unbiased nature of the stochastic gradients computed. Therefore, Error Compensated SGD (EC SGD) is used in literature to overcome this problem. This work considers various variants of EC SBD and achieves a unified analyses which obtains state-of-the-art bounds in the popular variants and introduces new algorithms with variance reduction techniques which also achieve linear rate.

Strengths: This work is a compendium on the stochastic gradient methods used in EC SGD literature and unifies many of the present analyses while discovering new, powerful variance reduction based algorithms (example: EC LSVRGstar) which achieve linear rates of convergence. I went through the main body of the paper and skimmed through the supplementary material and the unified analysis makes sense. This is a powerful work which adds significant value to the field. ======== The authors have addressed most of my concerns with the rebuttal. I maintain my score.

Weaknesses: The paper is too large to sufficiently describe in the main body of the paper. The main novelty is that it accumulates a lot of methods in the literature to present a unified analysis whereas it does not seem to introduce many new ideas.

Correctness: I went through the main body of the paper and skimmed through the supplementary material and the unified analysis makes sense. I have not checked the supplementary material thoroughly.

Clarity: The paper is very well written and explained, especially given the fact that it contains multitude of results. I found the intellectual honesty very refreshing.

Relation to Prior Work: As per the best of my knowledge it clearly discusses previous contributions and locates itself in the literature.

Reproducibility: Yes

Additional Feedback:


Review 2

Summary and Contributions: The authors are proposing a unified analysis of different variants of distributed SGD-based algorithms with arbitrary compressions and delayed updates. In particular, the result of their theoretical analysis mainly consists in a single theorem that is general enough to be adapted to all the methods that fit to the considered framework. In addition, taking inspiration from their problem description and analysis, the authors are proposing 6 different variants of distributed SGD-based methods, which vary based on the compression method considered and/or the adopted variance reduction techniques. In the section dedicated to the numerical examples, the authors consider the problem of logistic regression with L2 regularization and show the performance of the proposed methods across different datasets, drawing also some conclusions on the base of the empirical results on the characteristics of each method. The problem framework analyzed is \min_{x\in\mathbb{R}^d} \frac{1}{n}\sum_{i=1}^n f_i(x) where n is the number of workers and the information of each function f_i are assumed to be available only locally on the i-th worker. In addition, each i-th function is assumed to have a particular form (either finite-sum form or expected-value form). The theoretical analysis is conducted under assumptions of quasi-convexity and smoothness, in addition to an extra parametric assumption. The key for the general theoretical analysis are: 1) the formulation of the general method's updates given by equations (4), (5) and (7) 2) parametric assumptions 3.3 and 3.4, where the parameters are used to describe the assumed relations between gradient estimates, iterates, function values and errors.

Strengths: The paper is really well-written and rich in content. The structure of the paper is good and balanced across the different sections. The theoretical analysis is novel given its generality and also given the assumptions under which it is conducted. The proposed methods are not groundbreaking but overall the paper's content is strong.

Weaknesses: the paper presents minor weaknesses: 1) it is too rich in content probably given the page limitations: there is a lot (too much) material that is just relegated to the appendix...an appendix of 90 pages is excessive..maybe I would suggest to submit the paper to a journal rather than a conference in order to allow the reviewers to have the time to go through the appendix material and to include more of the material in the actual paper. 2) I find the title not so in line with the main content of the paper. The title should probably be focused on the uniform analysis of distributed SGD-based methods which is the main topic of the paper. 3) the authors are not discussing the following methods which seem to be highly related to their problem framework: https://arxiv.org/abs/1611.02189 https://www.research-collection.ethz.ch/handle/20.500.11850/183454 http://proceedings.mlr.press/v80/duenner18a.html maybe they should point to those and cite those papers. 4) it is a bit disappointing that for the experiments the distributed setting is only 'simulated' on a single machine with a for loop... 5) improve the readability of the plots by increasing their size (they are currently a bit too small) ========== After Authors' Rebuttal ========== I would like to thank the authors for providing their comments/reply to each potential weakness point that I had underlined in this section. Regarding the first point, although there are no-strict limits on the appendix length enforced, I still find the current appendix's length (and content) to be excessive wrt the paper's length (and content) and still think that the content should have been re-organized between paper and appendix and submitted to a journal rather than a conference. At the same time, I agree with the authors regarding the fact that details on derivations for special cases are important and can be exploited in the future by other researchers (indeed I have never said to remove those details/derivations but just pointed out that most of the content is unfortunately and in my opinion wrongly relegated to the appendix....while the appendix's role in my opinion should be different). Regarding the second point, I suggest to switch back to the title of the 'working version' or find a new one that incorporates both these aspects without sacrificing the unified theory aspect. In the light of the authors's reply and taking into account also some of the other reviewers'votes, I confirm my (positive) vote for this paper.

Correctness: The paper's claims and empirical methodology is correct.

Clarity: The paper is really well-written and has a solid and clear structure.

Relation to Prior Work: Relation to prior work is discussed and the authors are clearly exposing what are the main differences between their analysis and the previous contributions. At the same time, the authors are not pointing to some existing methods that though seem to be quite related with the analyzed framework (see point 3 of the section on the weaknesses).

Reproducibility: Yes

Additional Feedback:


Review 3

Summary and Contributions: This paper proposed a framework to analyze a class of error-compensated SGD. This framework can cover many existing methods, and also inspires some new methods. In some special cases, the derived bounds can match the state-of-the-art results.

Strengths: The framework this paper proposed it quite general that can be useful in analyzing a wide range of error-compensated methods.

Weaknesses: Though technically interesting, the proposed framework is too general so that the authors didn't find a good way to present it in an eight-page paper.

Correctness: I doubt there is something missing in the main theorem: I cannot find constant \delta in the main theorem or its proceeding assumptions. This is not normal: \delta should play a critical role in the convergence.

Clarity: I personally dislike the writing of this paper. The authors try to combine too many things in this paper (16 new methods!), and used more nearly 3 pages to demonstrate the contributions, leaving very few spaces for the technical part. Too many things are defined in the appendix, which makes the main paper not easy to read.

Relation to Prior Work: Yes

Reproducibility: Yes

Additional Feedback: ========== After Authors' Rebuttal ========== After reading authors' rebuttal and other reviewers' comments, I decided to raise my score. The results are interesting, though I still dislike the writing of this paper.


Review 4

Summary and Contributions: The paper provides a unified analysis on distributed SGD described in (4) and (5). This framework covers a number of variants of SGD such as SGD with compression. The main contribution (Theorem 3.1) shows the convergence rate of these algorithms. The authors also propose many new error compensated SGD's and presents its complexity. Numerical experiments supports the effectiveness of error compensated schemes.

Strengths: Error compensation for distributed SGD seems necessary when using gradient compression. The paper provides a general theory for error compensated distributed SGD. The main contribution, Theorem 3.1 clarifies the convergence rate and dependency on several parameters. This can be considered as a great contribution to the field of distribution optimization. Theorem 3.1 is the key to show the existence of linearly converging EC method. The authors also propose several EC-SGD's (with complexity analysis ) )and among them, several EC-SGD have linear convergence rate. In summary, these algorithm shown in this paper are also good contributions for the field of distributed optimization and distributed learning. Furthermore, the experimental results confirm the theoretical discussion.

Weaknesses: In numerical experiments, it would be nice to include a comparison between "non-EC SGD" and "EC-SGD" to show effectiveness of error compensation. ========== After Authors' Rebuttal ========== I read the authors' response and fully understand their explanation.

Correctness: The discussion seems technically sound although I could not check whole technical arguments.

Clarity: The paper is well written and the scope of the paper is also clear.

Relation to Prior Work: The paper properly discusses related works.

Reproducibility: Yes

Additional Feedback:

[Author Response · NeurIPS 2020]

We thank the reviewers for their feedback and time! We are encouraged they found our work "powerful" and "adding a significant value to the field" of distributed optimization (R1; score 8), "novel" and "rich in content" (R2; score 7), "technically interesting" and "can be useful in analyzing a wide range of error-compensated methods" (R3, score 4), "a great contribution to the field" (R4; score 9) and emphasized that our paper is well-written (R1, R2, R4).

R1: ***The paper is too large to sufficiently describe in the main body of the paper. The main novelty is that it accumulates a lot of methods in the literature to present a unified analysis whereas it does not seem to introduce many new ideas.*** Indeed, our work contains many new results. However, we do not merely "accumulate" existing methods in the literature: our new general approach allows us to design many new and innovative EC methods, with SOTA convergence guarantees (e.g., we design first EC methods with linear rates - an open problem since Seide et al in 2014; see Sec 2 for more)! In order to achieve this, we had to innovate substantially. So, our approach is both very general, as evidenced by the many concrete methods that it supports, and leads to SOTA theory.

R2: ***1) It is too rich in content probably given the page limitations: there is a lot (too much) material that is just relegated to the appendix...an appendix of 90 pages is excessive..maybe I would suggest to submit the paper to a journal rather than a conference...*** The size of the appendix is substantial because we did not want to hide important details of the proofs/convergence results for the different *special cases* and tried to be as precise as possible. As the result, we have many theorems requiring many proofs. These extra details will be also useful for other researchers who can build on our work. Moreover, there are no strict limits on the appendix enforced by the rules of NeurIPS. The key insights and results are explained in the main body of the paper. ***2) I find the title not so in line with the main content of the paper. The title should probably be focused on the uniform analysis of distributed SGD-based methods which is the main topic of the paper.*** The working version of our title was different and highlighted the unified analysis, as you suggest. We later decided it was better to change it by emphasizing the *new algorithms and SOTA convergence theory* presented in the paper. ***3) The authors are not discussing the following methods...*** These works are broadly relevant, but not particularly so: while they address distributed learning, they are not about compressed communication, nor about EC; both being the key aspects of our work. However, we are happy to mention papers in a larger neigborhood of our work in final version of the paper as an extra page is allowed there. ***4) It is a bit disappointing that for the experiments the distributed setting is only 'simulated' on a single machine with a for loop...*** One of the main goals of the paper is to develop a unifying theory for distributed methods with EC. The simulated experiments showing the number of bits sent by workers and the number of oracle calls are enough to justify the theoretical results, independently from a particular implementation. They correlate well with the performance of the methods in real distributed systems. We can easily add a few genuine distributed experiments, but this will not affect any of the findings of our work. ***5) Improve the readability of the plots by increasing their size (they are currently a bit too small).*** Thanks for the suggestion. We will increase the size of the plots in the final version of the paper.

R3: ***1) The proposed framework is too general so that the authors didn't find a good way to present it in an eight-page paper.*** We politely disagree. Our framework is general, but we reflected the key parts/results and insights in the main body: we have *6 pages* devoted to the presentation of our results including *a high-level explanation* of our framework and new methods, *formal statement of the main convergence result* and *12 plots* in the numerical part with a detailed description of our experimental setup. Furthermore, the appendix consists mostly of proofs and other technical details related to special cases that we believe should be in the appendix. ***2) I doubt there is something missing in the main theorem: I cannot find constant $\delta$ in the main theorem or its proceeding assumptions. This is not normal: $\delta$ should play a critical role in the convergence.*** All is OK with the result. First, the def. of $\delta$ is given at the end of page 2. Second, $\delta$ of course plays a critical role in the convergence results for *EC* methods. However, our framework and, in particular, Theorem 3.1 works even for the methods *without error-compensation*, see Sections E and G in the appendix. Third, the explicit dependencies on $\delta$ of the convergence results for EC methods are presented in Table 1, Thm F.1, and Section I (see also Table 5 for the dependency of the parameters from Assumption 3.4 on $\delta$). ***3) I think the writing of this paper is awful.*** It is against reviewer guidelines and common courtesy to be impolite in a review. We respectfully ask the AC to ignore this comment. We also kindly ask the other reviewers who found our writing clear to defend us; thanks!! We did our best to present our many results in as simple and understandable way as possible, and we believe we achieved a very good result. No suggestions for improvement were proposed by the reviewer. ***4) Authors try to combine too many things in this paper (16 new methods!), and used more nearly 3 pages to demonstrate the contributions, leaving very few spaces for the technical part. Too many things are defined in the appendix, which makes the main paper really hard to read. Though NIPS allows supplementary material, the paper itself still needs to be as self-contained as possible.*** We politely disagree with the reviewer's criticism. In fact, we used *less than 2 pages* to demonstrate the contributions since Tables 1 and 2 contain a lot of technical details.

R4: ***In numerical experiments, it would be nice to include a comparison between "non-EC SGD" and "EC-SGD" to show effectiveness of error compensation.*** You are right: one can add this kind of comparison to justify the needing for EC for biased compressors like TopK. However, this question was theoretically addressed in [7], where authors proposed an example of distributed optimization problems for which "non-EC SGD" with TopK compression diverges exponentially fast. However, we will have extra page if our paper gets accepted, and can add a couple plots of this type.

[Meta-Review · NeurIPS 2020]

This is a very complete paper that all reviewers feel make a substantial contribution. The paper analyzes many different algorithms and proposes new algorithms.